# The Cost of Robustness: Tighter Bounds on Parameter Complexity for Robust Memorization in ReLU Nets

**Yujun Kim** *   **Chaewon Moon** *   **Chulhee Yun**
KAIST
{kyujun02, chaewon.moon, chulhee.yun}@kaist.ac.kr

## Abstract

We study the parameter complexity of *robust memorization* for $\mathrm{ReLU}$ networks: the number of parameters required to interpolate any given dataset with $\epsilon$-separation between differently labeled points, while ensuring predictions remain consistent within a $\mu$-ball around each training sample. We establish upper and lower bounds on the parameter count as a function of the robustness ratio $\rho = \mu/\epsilon$. Unlike prior work, we provide a fine-grained analysis across the entire range $\rho \in (0, 1)$ and obtain tighter upper and lower bounds that improve upon existing results. Our findings reveal that the parameter complexity of robust memorization *matches* that of *non-robust* memorization when $\rho$ is small, but grows with increasing $\rho$.

## 1 Introduction

The topic of memorization investigates the expressive power of neural networks required to fit any given dataset exactly. This line of inquiry seeks to determine the minimal network size—measured in the number of parameters, or equivalently, parameter complexity—needed to interpolate any finite collection of $N$ labeled examples. A number of works study both upper and lower bounds on the parameter complexity [Baum, 1988, Yun et al., 2019, Bubeck et al., 2020, Park et al., 2021]. The VC-dimension implies a lower bound of $\Omega(\sqrt{N})$ [Chervonenkis, 2015, Goldberg and Jerrum, 1995, Bartlett et al., 2019], while Vardi et al. [2021] show that $\tilde{\Theta}(\sqrt{N})$ parameters suffice for $\mathrm{ReLU}$ networks. Together, these results establish that memorizing any $N$ distinct samples with $\mathrm{ReLU}$ networks can be done with $\tilde{\Theta}(\sqrt{N})$ parameters, tight up to logarithmic factors.

We now turn to a more challenging task beyond mere interpolation of data: **robust memorization**. We aim to quantify the additional parameter complexity required for a network to remain *robust* against adversarial attacks, going beyond standard non-robust memorization. To address the sensitivity of neural networks to small adversarial perturbations [Szegedy et al., 2014, Goodfellow et al., 2015, Ding et al., 2019, Gowal et al., 2021, Zhang et al., 2021, Bastounis et al., 2025], we consider the setting in which not only the data points but all points within a distance $\mu$—referred to as the *robustness radius*—from each data point must be mapped to the corresponding label. More concretely, for any dataset with $\epsilon$-separation between differently labeled data points, the network must memorize the dataset and the prediction must remain consistent within a $\mu$-ball centered at each training sample. As will be seen shortly, the parameter complexity for robust memorization is governed by the *robustness ratio* $\rho = \mu/\epsilon \in (0, 1)$ rather than the individual values of $\mu$ and $\epsilon$. However, a precise understanding of how this complexity scales with $\rho$ remains limited.

---

*Authors contributed equally to this paper.

## 1.1 What is Known So Far?

**Existing Lower Bounds.** Since classical memorization requires $\Omega(\sqrt{N})$ parameters, it follows that robust memorization must also satisfy a lower bound of at least $\Omega(\sqrt{N})$ parameters for any $\rho \in (0, 1)$. A lower bound specific to robust memorization is established by the work of Li et al. [2022], which shows that for input dimension $d$, $\Omega(\sqrt{Nd})$ parameters are necessary for robust memorization under $\ell_2$-norm for sufficiently large $\rho$. However, the authors do not characterize the range of $\rho$ over which this lower bound remains valid. Our Proposition 3.3 presented later shows that the $\Omega(\sqrt{Nd})$ lower bound can be extended to the range $\rho \in \left(\sqrt{1 - 1/d}, 1\right)$. Combining these observations, we obtain the following unified lower bound: suppose that for any dataset $\mathcal{D}$ with input dimension $d$ and size $N$, there exists a neural network with at most $P$ parameters that robustly memorizes $\mathcal{D}$ with robustness ratio $\rho$ under $\ell_2$-norm. Then, the number of parameters $P$ must satisfy

$$P = \Omega\left(\left(1 + \sqrt{d} \cdot \mathbf{1}_{\rho \geq \sqrt{1 - \frac{1}{d}}}\right)\sqrt{N} + d\right), \tag{1}$$

where the $d$ term accounts for the parameters connected to the input neurons. In the setting $d = O(\sqrt{N})$, the lower bounds increase discontinuously from $\sqrt{N}$ to $\sqrt{Nd}$.

While our main analysis focuses on the $\ell_2$-norm, there also exist results under the $\ell_\infty$-norm. In particular, Yu et al. [2024] show that under the $\ell_\infty$-norm and certain assumptions, $\rho$-robust memorization requires the first hidden layer to have width at least $d$. Our analysis not only strengthens but also generalizes this $\ell_\infty$-norm result by removing the assumption on the dataset—made in prior work—that the number of data points must be greater than $d$.

**Existing Upper Bounds.** From the work of Yu et al. [2024], it is proven that $O(Nd^2)$ parameters suffice for any $\rho \in (0, 1)$. See Appendix D.2 for an analysis of the parameter complexity of their construction. Furthermore, Egosi et al. [2025] show that for $\rho \in \left(0, \frac{1}{\sqrt{d}}\right)$, a network of width $\log N$ suffices for $\rho$-robust memorization. Although they do not explicitly quantify the total number of parameters, their construction with a width $\log N$ network requires $\tilde{O}(N)$ parameters, as we verify in Appendix D.3. Additionally, we state that their construction implicitly yields a smooth interpolation between $\tilde{O}(N)$ and $\tilde{O}(Nd^2)$ as $\rho$ varies within the intermediate range $(1/\sqrt{d}, 1/\sqrt[6]{d})$.

To sum up, the existing upper bound states that for any dataset $\mathcal{D}$ with input dimension $d$ and size $N$, there exists a neural network that achieves robust memorization on $\mathcal{D}$ with the robustness ratio $\rho$ under $\ell_2$-norm, with the number of parameters $P$ bounded as follows:

$$P = \begin{cases} \tilde{O}(N + d) & \text{if } \rho \in (0, 1/\sqrt{d}]. \\ \tilde{O}(Nd^3\rho^6 + d) & \text{if } \rho \in (1/\sqrt{d}, 1/\sqrt[6]{d}]. \\ \tilde{O}(Nd^2) & \text{if } \rho \in (1/\sqrt[6]{d}, 1). \end{cases} \tag{2}$$

When $d = O(N)$, the upper bound transitions continuously from $\tilde{O}(N)$ to $\tilde{O}(Nd^2)$.

## 1.2 Summary of Contribution

We investigate how the number of parameters required for robust memorization in ReLU networks varies with the robustness ratio $\rho$. We improve both upper and lower bounds on the minimal number of parameters over all possible $\rho \in (0, 1)$, which are tight in some regimes and substantially reduce the existing gap elsewhere. The improvement across different regimes of $\rho$ is visualized in Figure 1.

- **Necessary Conditions for Robust Memorization.** We show that the first hidden layer must have a width of at least $\rho^2 \min\{N, d\}$, by constructing a dataset that cannot be robustly memorized using a smaller width. Consequently, the network must have at least $\Omega(\rho^2 \min\{N, d\}d)$ parameters. Moreover, we prove that at least $\Omega(\sqrt{N/(1 - \rho^2)})$ parameters are necessary for $\rho \leq \sqrt{1 - 1/d}$ by analyzing the VC-dimension. Combining these two results, we obtain a tighter lower bound on the parameter complexity of robust memorization of the form

$$P = \Omega\left((\rho^2 \min\{N, d\} + 1)d + \min\left\{\frac{1}{\sqrt{1 - \rho^2}}, \sqrt{d}\right\}\sqrt{N}\right).$$

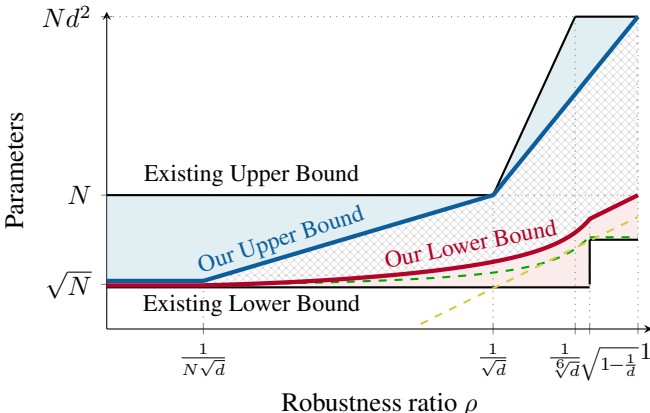

Figure 1: Summary of parameter bounds on a log-log scale when $d = \Theta(\sqrt{N})$. We omit constant factors in both axes. Solid blue and red curves show the sufficient (Theorem 4.2) and necessary (Theorem 3.1) numbers of parameters, respectively; the solid black curves are the best prior bounds. Light-blue shading highlights our improvement in the upper bound, and light-red shading highlights our improvement in the lower bound. The cross-hatched area marks the remaining gap. Notably, this gap disappears in the smallest $\rho$ regime. The yellow and green dashed line denotes the first term (Proposition 3.2) and the second term (Proposition 3.3) in Theorem 3.1, respectively.

- **Sufficient Conditions for Robust Memorization.** We establish improved upper bounds on the parameter count by analyzing three distinct regimes of $\rho$, tightening the bound in each case. For $\rho \in \left(0, \frac{1}{5N\sqrt{d}}\right]$, we achieve robust memorization using $\tilde{O}(\sqrt{N})$ parameters, matching the existing lower bound. For $\rho \in \left(\frac{1}{5N\sqrt{d}}, \frac{1}{5\sqrt{d}}\right]$, we obtain robust memorization with $\tilde{O}(Nd^{1/4}\rho^{1/2})$ parameters up to an arbitrarily small error, which interpolates between the existing lower bound $\Omega(\sqrt{N})$ and the existing upper bound $\tilde{O}(N)$. Finally, for larger values of $\rho$, where $\rho \in \left(\frac{1}{5\sqrt{d}}, 1\right)$, robust memorization is achieved with $\tilde{O}(Nd^2\rho^4)$ parameters, which interpolates between the existing upper bound $\tilde{O}(N)$ and $\tilde{O}(Nd^2)$.

All together, we provide, to the best of our knowledge, the first theoretical analysis of parameter complexity for robust memorization that characterizes its dependence on the robustness ratio $\rho$ over the entire range $\rho \in (0, 1)$. Notably, when $\rho < \frac{1}{5N\sqrt{d}}$, the same number of parameters as in classical (non-robust) memorization suffices for robust memorization. These results suggest that, in terms of parameter count, achieving robustness against adversarial attacks is relatively inexpensive when the robustness radius is small. As the radius grows, however, the number of required parameters increases, reflecting the rising cost of achieving stronger robustness.

## 2  Preliminaries

### 2.1  Notation

Throughout the paper, we use $d$ to denote the input dimension of the data, $N$ to denote the number of data points in a dataset, and $C$ to denote the number of classes for a classification task. For a natural number $n \in \mathbb{N}$, $[n]$ denotes the set $\{1, 2, \ldots, n\}$.

For two sets $A, B \subseteq \mathbb{R}^d$, we denote the $\ell_2$-norm distance between $A$ and $B$ as $\mathrm{dist}_2(A, B) := \inf\{\|\boldsymbol{a} - \boldsymbol{b}\|_2 \mid \boldsymbol{a} \in A, \boldsymbol{b} \in B\}$, where $\|\cdot\|_2$ denotes the Euclidean norm. When either $A$ or $B$ is a singleton set, such as $\{\boldsymbol{a}\}$ or $\{\boldsymbol{b}\}$, we identify the set with the element and write $\boldsymbol{a}$ or $\boldsymbol{b}$ in place of $A$ or $B$, respectively; for example, $\mathrm{dist}_2(\boldsymbol{a}, B)$. In the case $d = 1$, we omit the subscript 2 and write $\mathrm{dist}(\cdot, \cdot)$ to denote the standard absolute distance on $\mathbb{R}$. We use $\mathcal{B}_2(\boldsymbol{x}, \mu) = \{\boldsymbol{x}' \mid \|\boldsymbol{x}' - \boldsymbol{x}\|_2 < \mu\}$ to denote an open Euclidean ball centered at $\boldsymbol{x}$ with a radius $\mu$.

We use $\tilde{O}(\cdot)$ to hide the poly-logarithmic dependencies in problem parameters such as $N$, $d$, and $\rho$.

## 2.2 Dataset and Robust Memorization

For $d \geq 1$ and $N \geq C \geq 2$, let $\boldsymbol{D}_{d,N,C}$ be the collection of all datasets of the form $\mathcal{D} = \{(\boldsymbol{x}_i, y_i)\}_{i=1}^{N} \subset \mathbb{R}^d \times [C]$, such that $\boldsymbol{x}_i \neq \boldsymbol{x}_j$ for all $i \neq j$ and has at least one data point per each class label. Hence, any $\mathcal{D} \in \boldsymbol{D}_{d,N,C}$ is a pairwise distinct $d$-dimensional dataset of size $N$ with labels in $[C]$.

**Definition 2.1.** For $\mathcal{D} \in \boldsymbol{D}_{d,N,C}$, the separation constant $\epsilon_{\mathcal{D}}$ is defined as

$$\epsilon_{\mathcal{D}} := \frac{1}{2} \min \left\{ \|\boldsymbol{x}_i - \boldsymbol{x}_j\|_2 \mid (\boldsymbol{x}_i, y_i), (\boldsymbol{x}_j, y_j) \in \mathcal{D}, \ y_i \neq y_j \right\}.$$

Since the datasets we consider have at least one data point for each class label, the set we minimize over is nonempty. Moreover, since we consider $\mathcal{D}$ with $\boldsymbol{x}_i \neq \boldsymbol{x}_j$ for all $i \neq j$, we have $\epsilon_{\mathcal{D}} > 0$. Next, we define robust memorization of the given dataset.

**Definition 2.2.** For $\mathcal{D} \in \boldsymbol{D}_{d,N,C}$ and a given *robustness ratio* $\rho \in (0, 1)$, define the *robustness radius* as $\mu := \rho \epsilon_{\mathcal{D}}$. We say that a function $f : \mathbb{R}^d \to \mathbb{R}$ $\rho$-*robustly memorizes* $\mathcal{D}$ if

$$f(\boldsymbol{x}') = y_i, \quad \text{for all } (\boldsymbol{x}_i, y_i) \in \mathcal{D} \text{ and } \boldsymbol{x}' \in \mathcal{B}_2(\boldsymbol{x}_i, \mu),$$

and $\mathcal{B}_2(\boldsymbol{x}_i, \mu)$ is referred to as the *robustness ball* of $\boldsymbol{x}_i$.

When $\rho = 0$, robust memorization reduces to classical memorization, which requires $f(\boldsymbol{x}_i) = y_i$ for all $(\boldsymbol{x}_i, y_i) \in \mathcal{D}$. We emphasize that the range $\rho \in (0, 1)$ covers the entire regime in which robust memorization is possible. Specifically, for $\rho > 1$, requiring memorization of $\rho \epsilon_{\mathcal{D}}$-radius neighbor of each data point leads to a contradiction as $\mathcal{B}_2(\boldsymbol{x}_i, \rho \epsilon_{\mathcal{D}}) \cap \mathcal{B}_2(\boldsymbol{x}_j, \rho \epsilon_{\mathcal{D}}) \neq \emptyset$ for some $y_i \neq y_j$. Moreover, if $\rho = 1$, any continuous function $f$ cannot $\rho$-robustly memorize $\mathcal{D}$. If $f$ is continuous and 1-robustly memorizes $\mathcal{D}$, we have $f(\overline{\mathcal{B}_2(\boldsymbol{x}_i, \epsilon_{\mathcal{D}})}) = \{y_i\}$ for all $i \in [N]$, where $\overline{\mathcal{B}_2(\boldsymbol{x}_i, \epsilon_{\mathcal{D}})}$ is the closed ball with center $\boldsymbol{x}_i$ and radius $\epsilon_{\mathcal{D}}$. Since $\overline{\mathcal{B}_2(\boldsymbol{x}_i, \epsilon_{\mathcal{D}})} \cap \overline{\mathcal{B}_2(\boldsymbol{x}_j, \epsilon_{\mathcal{D}})} \neq \emptyset$ for some $y_i \neq y_j$, this leads to a contradiction.

## 2.3 ReLU Neural Network

We define the neural network $f$ recursively over $L$ layers:

$$\begin{aligned}
\boldsymbol{a}_0(\boldsymbol{x}) &= \boldsymbol{x}, \\
\boldsymbol{a}_\ell(\boldsymbol{x}) &= \sigma(\boldsymbol{W}_\ell \boldsymbol{a}_{\ell-1}(\boldsymbol{x}) + \boldsymbol{b}_\ell) \ \text{ for } \ell = 1, 2, \ldots, L-1, \\
f(\boldsymbol{x}) &= \boldsymbol{W}_L \boldsymbol{a}_{L-1}(\boldsymbol{x}) + \boldsymbol{b}_L,
\end{aligned}$$

where the activation $\sigma(\boldsymbol{u}) := \max\{\boldsymbol{0}, \boldsymbol{u}\}$ is the element-wise ReLU. We use $d_1, \ldots, d_{L-1}$ to denote the widths of the $L-1$ hidden layers. We define the width of the network to be the maximum hidden layer width, $\max_{\ell \in [L-1]} d_\ell$. For $\ell \in [L]$, the symbols $\boldsymbol{W}_\ell \in \mathbb{R}^{d_\ell \times d_{\ell-1}}$ and $\boldsymbol{b}_\ell \in \mathbb{R}^{d_\ell}$ denote the weight matrix and the bias vector for the $\ell$-th layer, respectively; here, we use the convention $d_0 = d$ and $d_L = 1$.

We count the number of parameters $P$ of $f$ as the count of all entries in the weight matrices and biases $\{\boldsymbol{W}_\ell, \boldsymbol{b}_\ell\}_{\ell=1}^{L}$ (including entries set to zero), as

$$P = \sum_{\ell=1}^{L} (d_{\ell-1} + 1) \cdot d_\ell. \tag{3}$$

This reflects the common convention of parameter counting in practice. The set of neural networks with input dimension $d$ and at most $P$ parameters is denoted as

$$\mathcal{F}_{d,P} = \left\{ f : \mathbb{R}^d \to \mathbb{R} \mid f \text{ is a neural network with at most } P \text{ parameters} \right\}. \tag{4}$$

Although less relevant in practice, some prior work counts only nonzero entries when reporting the number of parameters. Appendix E adopts this alternative counting scheme and explains how our results translate under it, enabling comparisons with prior studies from a different perspective. Even then, the key findings of this paper remain true: for small $\rho$, robustness incurs no additional parameter cost, whereas as $\rho$ grows, the number of required parameters increases.

## 2.4 Why Only $\rho = \mu/\epsilon_{\mathcal{D}}$ Matters

We describe both necessary and sufficient conditions for robust memorization in terms of the ratio $\rho = \mu/\epsilon_{\mathcal{D}}$, rather than describing it in terms of individual values $\mu$ and $\epsilon_{\mathcal{D}}$. This is because the results remain invariant under scaling of the dataset.

Specifically regarding the sufficient condition, suppose $f$ $\rho$-robustly memorizes $\mathcal{D}$ with robustness radius $\mu = \rho\epsilon_{\mathcal{D}}$. Then for any $c > 0$, the scaled dataset $c\mathcal{D} := \{(c\boldsymbol{x}_i, y_i)\}_{i=1}^N$, whose separation $\epsilon_{c\mathcal{D}} = c\epsilon_{\mathcal{D}}$, can be $\rho$-robustly memorized with robustness radius $c\mu$ by the scaled function $\boldsymbol{x} \mapsto f(\frac{1}{c}\boldsymbol{x})$. Moreover, the scaled function can be implemented through a network with the same number of parameters as the neural network $f$ via scaling the first hidden layer weight matrix by $1/c$.

On the other hand, this implies that the necessary condition can also be characterized in terms of $\rho$. Suppose we have a dataset $\mathcal{D}$ with a fixed $\epsilon_{\mathcal{D}}$ for which $\rho$-robustly memorizing it requires a certain number of parameters $P$. Then, the scaled dataset $c\mathcal{D}$ with a separation $\epsilon_{c\mathcal{D}} = c\epsilon_{\mathcal{D}}$ also requires the same number of parameters for $\rho$-robust memorization. If $c\mathcal{D}$ can be $\rho$-robustly memorized with less than $P$ parameters, then by parameter rescaling from the previous paragraph, $\mathcal{D}$ can also be $\rho$-robustly memorized with less than $P$ parameters, leading to a contradiction.

Hence, the robustness ratio $\rho = \mu/\epsilon_{\mathcal{D}}$ captures the essential difficulty of robust memorization, independent of scaling. We henceforth state our upper and lower bounds in terms of $\rho$.

## 3 Necessary Number of Parameters for Robust Memorization

In this section, we establish necessity conditions on the number of parameters and the width of neural networks for robust memorization, expressed in terms of the robustness ratio $\rho \in (0, 1)$. The following theorem presents our main lower bound result on the parameter complexity of robust memorization.

**Theorem 3.1.** *Let $\rho \in (0, 1)$. Suppose for any $\mathcal{D} \in \boldsymbol{D}_{d,N,2}$, there exists a neural network $f \in \mathcal{F}_{d,P}$ that can $\rho$-robustly memorize $\mathcal{D}$. Then, the number of parameters $P$ must satisfy*

$$P = \Omega\left((\rho^2 \min\{N, d\} + 1)d + \min\left\{\frac{1}{\sqrt{1-\rho^2}}, \sqrt{d}\right\}\sqrt{N}\right).$$

The proof of Theorem 3.1 is provided in Appendix A.1. The theorem states a necessary condition on the number of parameters for binary classification ($C = 2$). The same bound applies to $C > 2$: any classifier that robustly memorizes a multiclass dataset can be converted into a one-vs-rest binary classifier by appending a final two-parameter layer (one weight and one bias) that separates a designated label from the others. Therefore, a multiclass task requires at least the parameter scale needed for the binary case. Hence, Theorem 3.1 extends to $C > 2$. Moreover, while Theorem 3.1 focuses on $\ell_2$-norm, we extend the necessity results to general $\ell_p$-norm in Theorem C.5. The lower bound on the number of parameters consists of two parts: one derived from the requirement on the first hidden layer width and the other from the VC-dimension.

**First Term: Necessary Condition by the First Hidden Layer Width.** The first term $\Omega((\rho^2 \min\{N, d\} + 1)d)$ comes from the following proposition on the first hidden layer width.

**Proposition 3.2.** *There exists $\mathcal{D} \in \boldsymbol{D}_{d,N,2}$ such that, for any $\rho \in (0, 1)$, any neural network $f : \mathbb{R}^d \to \mathbb{R}$ that $\rho$-robustly memorizes $\mathcal{D}$ must have the first hidden layer width at least $\rho^2 \min\{N - 1, d\}$.*

For any fixed $N, d$, we can choose a single dataset $\mathcal{D}$ that enforces the bound simultaneously for all $\rho \in (0, 1)$: every $\rho$-robust memorizer of $\mathcal{D}$ must have the first hidden layer width at least $\rho^2 \min\{N - 1, d\}$. Section 5.1 treats the simple case $N - 1 = d$ to illustrate the construction and provide a sketch of proof, while Appendix A.2 provides the full proof for the general case.

Proposition 3.2 for the $\ell_2$-norm extends to the general $\ell_p$-norm in Proposition C.6. For every $p \geq 2$, the same lower bound on the first hidden layer width, $\rho^2 \min\{N - 1, d\}$, holds. For $1 \leq p < 2$, a nontrivial lower bound still holds. Furthermore, for the $\ell_\infty$-norm, we strengthen the result of Yu et al. [2024]—while they show that width at least $d$ is necessary when $N > d$ and $\rho \geq 0.8$, we obtain the stronger width requirement $\min\{N - 1, d\}$ for any $\rho \in (1/2, 1)$, *without* the assumption $N > d$, as formalized in Proposition C.7.

We now discuss the implications of Proposition 3.2 on the parameter complexity in Theorem 3.1. Since the input dimension is $d$, any neural network $f : \mathbb{R}^d \to \mathbb{R}$ with the first hidden layer width $m$ must have at least $md$ parameters. Moreover, we have a trivial lower bound $m \geq 1$. Hence, the lower bound of width $m$ becomes $\max\{\rho^2 \min\{N-1, d\}, 1\} \geq \frac{1}{2}(\rho^2 \min\{N-1, d\} + 1)$, yielding a necessity of $\Omega((\rho^2 \min\{N, d\} + 1)d)$ parameters in Theorem 3.1. The width from Proposition 3.2 dominates over the trivial lower bound of 1 whenever $\rho \geq 1/\sqrt{\min\{N-1, d\}}$.

Let us compare the result with Egosi et al. [2025], where they show logarithmic width in $N$ is sufficient under the restricted condition of $\rho \leq 1/\sqrt{d}$ for robust memorization. Our necessary condition on width does not conflict with their logarithmic sufficiency, as their sufficiency holds only under $\rho \leq 1/\sqrt{d}$, in which our lower bound becomes trivial.

On the other hand, the necessary condition on width by Egosi et al. [2025] given as $2\log N / \log(4832\rho^{-1})$ exceeds the trivial lower bound 1 only when $\rho \geq 4832/N$. Even in the case where their lower bound becomes nontrivial, their bound is still at the $\tilde{O}(1)$ scale, so that our lower bound either becomes tighter or matches their bound up to a polylogarithmic factor over all $\rho \in (0, 1)$. As a side note, although we generally ignore polylogarithmic factors, we may also consider logarithmic terms for completeness. Under this consideration, the lower bound of Egosi et al. [2025] remains logarithmically nontrivial while ours remains trivial for $4832/N < \rho < 1/\sqrt{\min\{N-1, d\}}$, provided that such $\rho$ exists.

**Second Term: Necessary Condition by the VC-Dimension.** Now, let us look at the necessary number of parameters given by the VC-dimension of the function class.

**Proposition 3.3.** *Let $\rho \in \left(0, \sqrt{1 - \frac{1}{d}}\right]$. Suppose for any $\mathcal{D} \in \mathbf{D}_{d,N,2}$, there exists $f \in \mathcal{F}_{d,P}$ that $\rho$-robustly memorizes $\mathcal{D}$. Then, the number of parameters $P$ must satisfy*

$$P = \Omega\left(\sqrt{\frac{N}{1-\rho^2}}\right).$$

The detailed proof of Proposition 3.3 is in Appendix A.3 and its extension to the $\ell_p$-norm appears in Proposition C.8. Before presenting our approach, we briefly review how the existing bound is obtained using VC-dimension arguments. Gao et al. [2019], Li et al. [2022] prove that for sufficiently large $\rho$, whenever $\mathcal{F}_{d,P}$ contains $\rho$-robust memorizer of any $\mathcal{D} \in \mathbf{D}_{d,N,2}$, then VC-dim$(\mathcal{F}_{d,P}) = \Omega(Nd)$. Combining this with a known upper bound VC-dim$(\mathcal{F}_{d,P}) = O(P^2)$ [Goldberg and Jerrum, 1995], they obtain $P = \Omega(\sqrt{Nd})$.

However, the prior lower bound $\Omega(\sqrt{Nd})$ is only known to apply for sufficiently large $\rho$, without specifying the precise range. Before our result, the only lower bound applicable to all $\rho$—including small $\rho$ regime—was the one that trivially comes from non-robust memorization: $\Omega(\sqrt{N})$. A wide range of $\rho$ lacks a VC-dimension-based lower bound tailored to robust memorization.

In Proposition 3.3, we carefully characterize how the VC-dimension scales over the range $\rho \in (0, \sqrt{1 - 1/d}]$. In this range of $\rho$, we show whenever $\mathcal{F}_{d,P}$ contains $\rho$-robust memorizer of any $\mathcal{D} \in \mathbf{D}_{d,N,2}$, then VC-dim$(\mathcal{F}_{d,P}) = \Omega(N/1-\rho^2)$; this thus gives the tighter bound $P = \Omega(\sqrt{N/1-\rho^2})$. At the endpoint $\rho = \sqrt{1 - 1/d}$, Proposition 3.3 implies that $\Omega(\sqrt{Nd})$ parameters are required. Therefore, the same lower bound applies for all $\rho \geq \sqrt{1 - 1/d}$, characterizing the regime in which the existing bound of $\sqrt{Nd}$ holds. By combining Proposition 3.3 over $\rho \in (0, \sqrt{1 - 1/d}]$ and the $\Omega(\sqrt{Nd})$ bound over $\rho \in (\sqrt{1 - 1/d}, 1)$, we obtain the second term $\Omega(\min\{1/\sqrt{1-\rho^2}, \sqrt{d}\}\sqrt{N})$ in Theorem 3.1.

Finally, we clarify why Proposition 3.3 is stated for $\rho \leq \sqrt{1 - 1/d}$ and why, for $\rho > \sqrt{1 - 1/d}$, this approach cannot improve upon the $\sqrt{Nd}$ scale. Any such improvement via VC-dimension would require showing that VC-dim$(\mathcal{F}_{d,P})$ strictly exceeds $Nd$, i.e., that a $\rho$-robust memorizer in $\mathbb{R}^d$ shatters more than $Nd$ points. Our shattering argument shows that robustly memorizing two arbitrary points forces shattering of (a subset of) the standard basis directions in $\mathbb{R}^d$; iterating over $N/2$ disjoint pairs can yield $Nd/2$ shattered points. Consequently, our current construction neither establishes that a robust memorizer of $N$ points can shatter beyond the $Nd$ scale, nor that a robust memorizer of two points can shatter beyond the $d$ scale. Thus, within this framework, the VC-dimension cannot

be pushed beyond $Nd$ scale, and the induced parameter lower bound does not improve beyond the $\sqrt{Nd}$ scale for $\rho > \sqrt{1 - 1/d}$.

# 4 Sufficient Number of Parameters for Robust Memorization

In this section, we establish sufficient conditions on the number of parameters for robust memorization, thereby complementing the lower bounds presented in the previous section. In fact, one of our upper bound results is derived under a relaxed definition of robust memorization. For this, we define $\rho$-robust memorization error of a neural network.

**Definition 4.1.** For any $\mathcal{D} \in \boldsymbol{D}_{d,N,C}$, we define the $\rho$-robust memorization error of a network $f : \mathbb{R}^d \to \mathbb{R}$ on $\mathcal{D}$ as

$$\mathcal{L}_\rho(f, \mathcal{D}) := \max_{(\boldsymbol{x}_i, y_i) \in \mathcal{D}} \mathbb{P}_{\boldsymbol{x}' \sim \text{Unif}(\mathcal{B}(\boldsymbol{x}_i, \mu))}[f(\boldsymbol{x}') \neq y_i],$$

where $\mu = \rho \epsilon_{\mathcal{D}}$. When $\mathcal{L}_\rho(f, \mathcal{D}) < \eta$, we say $f$ can $\rho$-robustly memorize $\mathcal{D}$ with error at most $\eta$.

Note that if a network $f$ $\rho$-robustly memorizes $\mathcal{D}$ (as in Definition 2.2), then the error is zero; that is, by definition $\mathcal{L}_\rho(f, \mathcal{D}) = 0$.

We now state our main upper bounds, showing that any given dataset in $\boldsymbol{D}_{d,N,C}$ can be $\rho$-robustly memorized by a network with $\rho$-dependent number of parameters.

**Theorem 4.2.** *For any dataset $\mathcal{D} \in \boldsymbol{D}_{d,N,C}$ and $\eta \in (0, 1)$, the following statements hold:*

(i) *If $\rho \in \left(0, \frac{1}{5N\sqrt{d}}\right]$, there exists $f \in \mathcal{F}_{d,P}$ with $P = \tilde{O}(\sqrt{N})$ that $\rho$-robustly memorizes $\mathcal{D}$.*

(ii) *If $\rho \in \left(\frac{1}{5N\sqrt{d}}, \frac{1}{5\sqrt{d}}\right]$, there exists $f \in \mathcal{F}_{d,P}$ with $P = \tilde{O}(Nd^{\frac{1}{4}}\rho^{\frac{1}{2}})$ that $\rho$-robustly memorizes $\mathcal{D}$ with error at most $\eta$.*

(iii) *If $\rho \in \left(\frac{1}{5\sqrt{d}}, 1\right)$, there exists $f \in \mathcal{F}_{d,P}$ with $P = \tilde{O}(Nd^2\rho^4)$ that $\rho$-robustly memorizes $\mathcal{D}$.*

We note that we omitted the trivial additive factor $d$ that accounts for parameters connected to input neurons. The three regimes in Theorem 4.2—each referred to as small, moderate, and large $\rho$ regime respectively—collectively cover all values of $\rho \in (0, 1)$ and provide explicit upper bound complexity for robust memorization. Moreover, the constructions behind Theorem 4.2 use a single network architecture that depends only on the problem parameters $N, d, C, \rho$ and not on the dataset: for every $\mathcal{D} \in \boldsymbol{D}_{d,N,C}$ and given $\rho$, choosing appropriate weights and biases on this same architecture achieves the stated guarantee.

We present a proof sketch in Section 5.2 and the detailed proof in Appendix B. The extended version of Theorem 4.2, which additionally states the explicit bounds on depth, width, and bit complexity is presented as Theorem B.1. Importantly, the upper bound on the number of parameters in Theorem 4.2 does not come at the cost of implausible bit complexity. In fact, Remark B.2 shows that the constructions in Theorem 4.2(i) and 4.2(ii) can be implemented with bit complexities that match the necessary bit complexity required for networks with the stated parameter counts. The extension of Theorem 4.2 to the $\ell_p$-norm setting is given in Theorem C.11.

In contrast to prior results, Theorems 4.2(i) and 4.2(ii) provide the first upper bounds for robust memorization that are sublinear in $N$. Notably, our construction reveals a continuous interpolation—driven by the robustness ratio $\rho$—from the classical memorization complexity of $\Theta(\sqrt{N})$ to the existing upper bound of $\tilde{O}(N)$ in Theorem 4.2(ii), and further from $\tilde{O}(N)$ to $\tilde{O}(Nd^2)$ as shown in Theorem 4.2(iii). This demonstrates how the sufficient parameter complexity increases gradually with $\rho$, capturing the full spectrum of the robustness ratio.

**Tight Bounds for Robust Memorization with Small $\rho$.** Theorem 4.2(i) establishes a tight upper bound $\tilde{O}(\sqrt{N})$ on the number of parameters required for robust memorization when the robustness ratio satisfies $\rho < \frac{1}{5N\sqrt{d}}$. Since VC-dimension theory [Goldberg and Jerrum, 1995] implies that any network exactly memorizing given $N$ arbitrary samples must use at least $\Omega(\sqrt{N})$ parameters, our construction is optimal up to logarithmic factors. This shows that, for sufficiently small $\rho$, robust memorization requires the same parameter complexity $\tilde{\Theta}(\sqrt{N})$ as classical (non-robust) memorization.

**Perfect Robust Memorization with Threshold Activation Function.** Theorem 4.2(ii) builds upon the techniques in Theorem 4.2(i), extending the applicability from small values of $\rho$ to moderate ones. However, the extension requires the allowance of an arbitrarily small robust memorization error. As discussed in Section 5.2 and shown Figure 4, the error arises because ReLU-only networks can represent only continuous functions. Near discontinuous transition regions, they incur small errors—though these can be made arbitrarily small. In contrast, if we are allowed to use discontinuous threshold activation in combination with ReLU network, we can achieve $\rho$-robust memorization—and therefore zero robust memorization error—even in the moderate regime using $\tilde{O}(Nd^{1/4}\rho^{1/2})$ parameters, the same rate as Theorem 4.2(ii).

**Tight Bounds of Width.** For small and moderate $\rho$, our construction shows width $\tilde{O}(1)$ is sufficient, recovering the logarithmic width sufficiency of Egosi et al. [2025]. For large $\rho$, our construction shows width of $\tilde{O}(\rho^2 d)$ is sufficient for $\rho$-robust memorization. A complementary lower bound (Proposition 3.2) requires width at least $\rho^2 \min\{N-1, d\}$ is also necessary, which matches with our upper bound when $N > d$. As a result, when the number of data points exceeds the data dimension, our results tightly characterize the required width up to polylogarithmic factors across the entire range $\rho \in (0, 1)$.

# 5 Key Proof Ideas

In this section, we outline the sketch of proof for some of the results from Sections 3 and 4.

## 5.1 Proof Sketch for Proposition 3.2

We briefly overview the sketch of the proof for Proposition 3.2. For simplicity, we sketch the case $N = d + 1$, where Proposition 3.2 reduces to showing that the first hidden layer must have width at least $\rho^2 d$. To this end, we construct the dataset $\mathcal{D} = \{(\boldsymbol{e}_j, 1)\}_{j \in [d]} \cup \{(\boldsymbol{0}, 2)\}$, assigning label 1 to the standard basis points and label 2 to the origin, as shown in Figure 2a.

Let $f$ be an $\rho$-robust memorizer of $\mathcal{D}$ with the first hidden layer width $m$, and let $\boldsymbol{W} \in \mathbb{R}^{m \times d}$ denote the weight matrix of the first hidden layer. Since $\epsilon_{\mathcal{D}} = 1/2$, the robustness radius is $\mu = \rho\epsilon_{\mathcal{D}} = \rho/2$. For any $j \in [d]$, take any $\boldsymbol{x} \in \mathcal{B}_2(\boldsymbol{e}_j, \mu)$ and $\boldsymbol{x}' \in \mathcal{B}_2(\boldsymbol{0}, \mu)$. Then, $f(\boldsymbol{x}) = 1$ and $f(\boldsymbol{x}') = 2$ must hold, implying $\boldsymbol{W}\boldsymbol{x} \neq \boldsymbol{W}\boldsymbol{x}'$. Therefore, $\boldsymbol{x} - \boldsymbol{x}'$ should not lie in the null space of $\boldsymbol{W}$. All such possible differences $\boldsymbol{x} - \boldsymbol{x}'$ form a ball of radius $2\mu$ around each standard basis point, illustrated as the gray ball in Figure 2b. Thus, the distance between each standard basis point and the null space of $\boldsymbol{W}$ must be at least $2\mu$; otherwise, some gray balls intersect with the null space.

The null space of $\boldsymbol{W}$ is a $d - m$ dimensional space, assuming that $\boldsymbol{W}$ has full row rank. (The full proof generalizes even without this assumption.) By Lemma A.1, the distance between the set of standard basis points and any subspace of dimension $d - m$ is at most $\sqrt{m/d}$. Therefore, we have $\rho = 2\mu \leq \text{dist}_2\left(\{\boldsymbol{e}_j\}_{j \in [d]}, \text{Null}(\boldsymbol{W})\right) \leq \sqrt{m/d}$ and thus the first hidden layer width satisfies $m \geq \rho^2 d$.

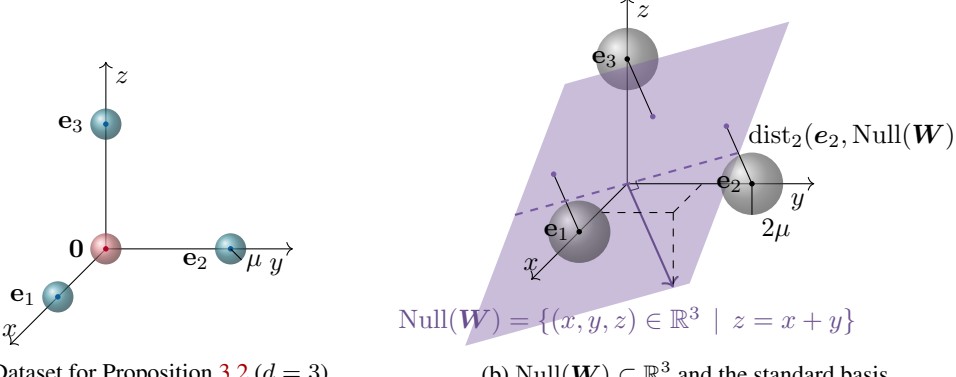

(a) Dataset for Proposition 3.2 ($d = 3$).
(b) $\text{Null}(\boldsymbol{W}) \subset \mathbb{R}^3$ and the standard basis

Figure 2: In (a), blue balls have label 1; the red ball has label 2. (b) illustrates the distance between $\text{Null}(\boldsymbol{W}) \subset \mathbb{R}^3$ and the standard basis for $\boldsymbol{W} = \begin{bmatrix} 1 & 1 & -1 \end{bmatrix}$ with the first hidden layer width 1.

## 5.2 Proof Sketch for Theorem 4.2

We now highlight the key construction techniques used to prove Theorem 4.2.

**Separation-Preserving Dimensionality Reduction.**
All three results in Theorem 4.2 leverage a strengthened version of the Johnson-Lindenstrauss (JL) lemma (Lemma B.18) to project data from a high-dimensional space $\mathbb{R}^d$ (left in Figure 3) to a lower-dimensional space $\mathbb{R}^m$ (right), while preserving pairwise distances up to a multiplicative factor. Specifically, any pair of points that are $2\epsilon_{\mathcal{D}}$-separated in $\mathbb{R}^d$ can remain at least $\frac{4}{5}\sqrt{\frac{m}{d}}\epsilon_{\mathcal{D}}$-separated after the projection. Meanwhile, each robustness ball of radius $\mu$ is preserved under the

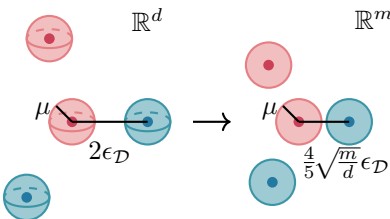

Figure 3: Separation-Preserving Projection

projection because our strengthened JL lemma uses randomized orthonormal projections [Matousek, 2013]. Since the geometry is preserved—specifically, the separation remains at least $\frac{4}{5}\sqrt{\frac{m}{d}}$ times its original value and the robustness radius is unchanged under projection—we can $\rho$-robustly memorize data points in $\mathbb{R}^d$ by projecting them to $\mathbb{R}^m$ and memorizing the projected points, provided that projected robustness balls do not overlap, i.e., as long as $\rho \leq \frac{2}{5}\sqrt{\frac{m}{d}}$.

In Theorems 4.2(i) and 4.2(ii), we project to $\mathbb{R}^m$ with $m = O(\log N)$ in the first hidden layer. The remaining layers have width $O(m)$, so the network width is $O(m) = O(\log N)$, i.e., constant up to polylogarithmic factors. This logarithmic projection is valid only for $\rho = O(1/\sqrt{d})$: projected $\rho$-balls remain disjoint as long as $\rho \leq \frac{2}{5}\sqrt{m/d} = \tilde{O}(1/\sqrt{d})$. If $\rho$ exceeds this scale, the projected balls overlap. For the larger-$\rho$ regime, Theorem 4.2(iii) increases the projection dimension. As long as $\rho \leq \frac{2}{5}\sqrt{m/d}$, the projected robustness balls remain disjoint; accordingly, taking $m \propto \rho^2 d$ maintains disjointness. Consequently, the width is proportional to $\rho^2 d$, and the parameter count is proportional to $\rho^4 d^2$.

The idea of separation-preserving dimension reduction and deriving conditions under which robustness balls remain disjoint after projection is concurrently proposed by Egosi et al. [2025]. However, their approach to ensuring the separability of robustness balls is substantially different from ours. Since the classical JL lemma does not inherently guarantee the preservation of ball separability, the authors do not rely on the JL lemma directly. Instead, they establish a probabilistic analogue through a technically involved analysis that bounds the probability that a random projection satisfies the required separation property. In contrast, we employ a strengthened version of the JL lemma and give a straightforward proof that there exists a projection preserving separability; see Appendix B.5.

**Mapping to Lattices from Grid.** For Theorem 4.2(i) and 4.2(ii), we utilize the $\tilde{O}(\sqrt{N})$-parameter memorization devised by Vardi et al. [2021]. In order to adopt the technique, it is necessary to assign a scalar value in $\mathbb{R}$ to each data point. This is because the construction memorizes the data after projecting them onto $\mathbb{R}$. Furthermore, this scalar assignment must meaningfully reflect the spatial structure of the data—preserving relative distances and neighborhood relationships of robustness ball.

We achieve this using grid-based lattice mapping. Specifically, we first reduce the dimension to $m = O(\log N)$. Then we partition $\mathbb{R}^m$ into a regular grid, and assign an integer index to each grid cell. Through this grid indexing, we map each unit cube $\prod_{j \in [m]}[z_j, z_j + 1)$ to an index $z_1 R^{m-1} + z_2 R^{m-2} + \cdots + z_m$ for each $\boldsymbol{z} = (z_1, \cdots, z_m) \in \mathbb{Z}^m$ and some sufficiently large integer $R$. Finally, we associate each index with the label of the projected robustness ball contained in that cell. The network then memorizes the mapping from each grid index to its corresponding label.

Under the condition on $\rho$ in Theorem 4.2(i), after an appropriate translation of the projected data, every projected robustness ball can be contained in a single grid cell in a way that no cell contains balls of two different labels; see Figure 4a. Hence, the label is constant on each cell that contains a ball, and all points in the ball can be associated with the cell's grid index.

What remains is implementability with ReLU networks. The grid-indexing map is discontinuous, while ReLU networks are continuous and can only approximate it. Consequently, approximation errors can occur only in thin neighborhoods of cell boundaries (the purple bands in Figure 4a).

Theorem 4.2(i) guarantees a translation that places every (projected) robustness ball strictly inside a cell and sufficiently far from all cell boundaries so that the ReLU-based indexing is accurate on the entire ball. Hence, each ball is disjoint from the purple error-tolerant regions, every point in the ball is mapped to the same grid index, and this yields $\rho$-robust memorization using only $\tilde{O}(\sqrt{N})$ parameters.

However, in Theorem 4.2(ii), we consider larger $\rho$, where projected robustness balls can overlap more than one grid cell and may intersect the error-tolerant regions where the ReLU-based indexing is inaccurate. As $\rho$ grows, the number of such balls increases. To cope with this regime, we use a sequential memorization strategy. We robustly memorize only the subset whose robustness balls are disjoint from the error-tolerant regions. The remaining balls may intersect those regions, but any resulting error is confined to those error-tolerant regions and can be made arbitrarily small by narrowing the error-tolerant regions.

In particular, we partition the $N$ points into multiple groups of approximately equal size and, at each stage, we robustly memorize one group, which we call the active group of this stage and we call the remaining groups of data points as inactive groups. We apply a translation so that the robustness balls of the active group lie strictly inside grid cells and away from the error-tolerant regions, while inactive balls may cross cell boundaries, provided they do not interfere with the cells occupied by the active group of this stage; see Figure 4b. The grid indexing is then implemented by a ReLU approximator whose error-tolerant regions are chosen sufficiently thin—by increasing the slope as in Lemma B.16—so that indexing is exact on the active balls. Any error for the inactive balls is confined to those thin error-tolerant regions. By Lemma B.11, the portion of a robustness ball covered by the error region scales with the region's width, and this width decreases as the ReLU slope grows; hence, the error can be driven arbitrarily small. The active group is robustly memorized using the construction of Theorem 4.2(i), and inactive balls do not interfere with the labels assigned in this stage. Iterating the stages and composing the resulting subnetworks yields memorization of all $N$ points with arbitrarily small error.

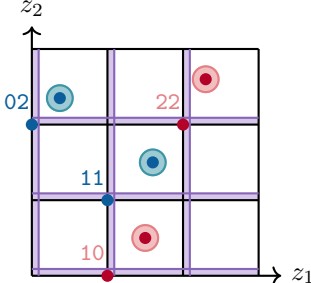

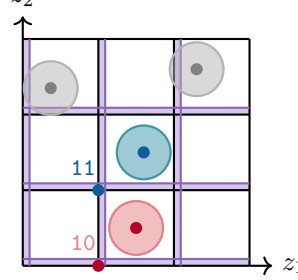

(a) The setting for Theorem 4.2(i), where each robust ball is entirely contained within a single grid cell, and no two balls with different labels occupy the same cell. This guarantees well-defined indexing without ambiguity.

(b) The relaxed setting in Theorem 4.2(ii) allows some balls to extend across adjacent grid cell boundaries, as long as they do not interfere with the specific cells being memorized at that step.

Figure 4: Grid-based Lattice Mapping.

## 6 Conclusion

We present a tighter characterization of the parameter complexity necessary and sufficient for robust memorization across the full range of robustness ratio $\rho \in (0, 1)$. Our results establish matching upper and lower bounds for small $\rho$, and show that robustness demands significantly more parameters than classical memorization as $\rho$ grows. These findings highlight how robustness fundamentally increases memorization difficulty under adversarial attacks.

We establish tight complexity bounds in the regime where $\rho < \frac{1}{5N\sqrt{d}}$. However, in the remaining cases, a gap between the upper and lower bounds persists. A precise characterization of the parameter complexity for some $\rho$ remains open and is essential for a complete understanding of the trade-off between robustness and network complexity.

## Acknowledgement

This work was supported by three Institute of Information & communications Technology Planning & Evaluation (IITP) grants (No. RS-2019-II190075, Artificial Intelligence Graduate School Program (KAIST); No. RS-2022-II220184, Development and Study of AI Technologies to Inexpensively Conform to Evolving Policy on Ethics; No. RS-2024-00457882, National AI Research Lab Project) funded by the Korean government (MSIT) and the InnoCORE program of the Ministry of Science and ICT (No. N10250156).

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

# Contents

# A Proofs for Section 3

## A.1 Explicit Proof of Theorem 3.1

**Theorem 3.1.** *Let $\rho \in (0,1)$. Suppose for any $\mathcal{D} \in \boldsymbol{D}_{d,N,2}$, there exists a neural network $f \in \mathcal{F}_{d,P}$ that can $\rho$-robustly memorize $\mathcal{D}$. Then, the number of parameters $P$ must satisfy*

$$P = \Omega\left((\rho^2 \min\{N,d\}+1)d + \min\left\{\frac{1}{\sqrt{1-\rho^2}}, \sqrt{d}\right\}\sqrt{N}\right).$$

*Proof.* From Proposition 3.2, we obtain $\mathcal{D} \in \boldsymbol{D}_{d,N,2}$ such that any $f : \mathbb{R}^d \to \mathbb{R}$ that $\rho$-robustly memorizes $\mathcal{D}$ must have the first hidden layer width at least $\rho^2 \min\{N-1,d\}$. By the assumption of Theorem 3.1, there exists $f \in \mathcal{F}_{d,P}$ that $\rho$-robustly memorizes $\mathcal{D}$ with the first hidden layer width $m \geq \rho^2 \min\{N-1,d\}$. With the trivial lower bound that $m \geq 1$, we have

$$m \geq \max\{\rho^2 \min\{N-1,d\},1\} \geq \frac{1}{2}(\rho^2 \min\{N-1,d\}+1).$$

Since we count all parameters according to Equation (3), the number of parameters in the first layer is $(d+1)m$. Therefore,

$$P \geq (d+1)\cdot m \geq (d+1)\cdot\frac{1}{2}(\rho^2 \min\{N-1,d\}+1) = \Omega(d(\rho^2 \min\{N,d\}+1)).$$

In addition, for $\rho \in \left(0, \sqrt{1-\frac{1}{d}}\right]$, using Proposition 3.3 gives the lower bound of parameters

$$P = \Omega\left(\sqrt{\frac{N}{1-\rho^2}}\right).$$

For $\rho \in \left(0, \sqrt{1-\frac{1}{d}}\right]$, we have $\frac{1}{\sqrt{1-\rho^2}} \leq \sqrt{d}$ so that the following relation holds:

$$\min\left\{\frac{1}{\sqrt{1-\rho^2}}, \sqrt{d}\right\} \cdot \sqrt{N} = \sqrt{\frac{N}{1-\rho^2}}.$$

For $\rho \in \left(\sqrt{1-\frac{1}{d}}, 1\right)$, the lower bound $P = \Omega(\sqrt{Nd})$ obtained by the case $\rho = \sqrt{1-\frac{1}{d}}$ also can be applied. In this case, $\frac{1}{\sqrt{1-\rho^2}} > \sqrt{d}$ so that the following relation holds:

$$\min\left\{\frac{1}{\sqrt{1-\rho^2}}, \sqrt{d}\right\} \cdot \sqrt{N} = \sqrt{Nd}.$$

Hence, in both $\rho$ regimes,

$$P = \Omega\left(\min\{\frac{1}{\sqrt{1-\rho^2}}, \sqrt{d}\}\sqrt{N}\right),$$

serves as the lower bound on the number of parameters.

By combining the bounds from Proposition 3.2 and Proposition 3.3, we conclude:

$$\begin{aligned} P =&\Omega\left(\max\left\{(\rho^2 \min\{N,d\}+1)d, \min\{\frac{1}{\sqrt{1-\rho^2}}, \sqrt{d}\}\sqrt{N}\right\}\right)\\ =&\Omega\left((\rho^2 \min\{N,d\}+1)d + \min\{\frac{1}{\sqrt{1-\rho^2}}, \sqrt{d}\}\sqrt{N}\right). \end{aligned}$$

$\square$

## A.2   Necessary Condition on Width for Robust Memorization

**Proposition 3.2.** *There exists $\mathcal{D} \in \boldsymbol{D}_{d,N,2}$ such that, for any $\rho \in (0,1)$, any neural network $f : \mathbb{R}^d \to \mathbb{R}$ that $\rho$-robustly memorizes $\mathcal{D}$ must have the first hidden layer width at least $\rho^2 \min\{N-1, d\}$.*

*Proof.* To prove Proposition 3.2, we consider two cases based on the relationship between $N-1$ and $d$. In the first case, where $N-1 \leq d$, establishing the proposition requires that the first hidden layer has width at least $\rho^2(N-1)$. In the second case, where $N-1 > d$, the required width is at least $\rho^2 d$. For each case, we construct a dataset $\mathcal{D} \in \boldsymbol{D}_{d,N,2}$ such that any network that $\rho$-robustly memorizes $\mathcal{D}$ must have a first hidden layer of width no smaller than the corresponding bound.

**Case I :** $N-1 \leq d$**.**   Let $\mathcal{D} = \{(\boldsymbol{e}_j, 2)\}_{j \in [N-1]} \cup \{(\boldsymbol{0}, 1)\}$. Then, $\mathcal{D}$ has separation constant $\epsilon_{\mathcal{D}} = 1/2$. Let $f$ be a neural network that $\rho$-robust memorizes $\mathcal{D}$, and denote the width of its first hidden layer as $m$. Denote by $\boldsymbol{W} \in \mathbb{R}^{m \times d}$ the weight matrix of the first hidden layer of $f$. Assume for contradiction that $m < \rho^2(N-1)$.

Let $\mu = \rho\epsilon_{\mathcal{D}}$ denote the robustness radius. Then, the network $f$ must distinguish every point in $\mathcal{B}_2(\boldsymbol{e}_j, \mu)$ from every point in $\mathcal{B}_2(\boldsymbol{0}, \mu)$, for all $j \in [N-1]$. Therefore, for any $\boldsymbol{x} \in \mathcal{B}_2(\boldsymbol{e}_j, \mu)$ and $\boldsymbol{x}' \in \mathcal{B}_2(\boldsymbol{0}, \mu)$, we must have

$$\boldsymbol{W}\boldsymbol{x} \neq \boldsymbol{W}\boldsymbol{x}',$$

or equivalently, $\boldsymbol{x} - \boldsymbol{x}' \notin \mathrm{Null}(\boldsymbol{W})$, where $\mathrm{Null}(\cdot)$ denotes the null space of a given matrix. Note that

$$\mathcal{B}_2(\boldsymbol{e}_j, \mu) - \mathcal{B}_2(\boldsymbol{0}, \mu) := \{\boldsymbol{x} - \boldsymbol{x}' \mid \boldsymbol{x} \in \mathcal{B}_2(\boldsymbol{e}_j, \mu) \text{ and } \boldsymbol{x}' \in \mathcal{B}_2(\boldsymbol{0}, \mu)\} = \mathcal{B}_2(\boldsymbol{e}_j, 2\mu).$$

Hence, it is necessary that $\mathcal{B}_2(\boldsymbol{e}_j, 2\mu) \cap \mathrm{Null}(\boldsymbol{W}) = \emptyset$ for all $j \in [N-1]$, or equivalently,

$$\mathrm{dist}_2(\boldsymbol{e}_j, \mathrm{Null}(\boldsymbol{W})) \geq 2\mu \quad \text{for all } j \in [N-1]. \tag{5}$$

Since $\dim(\mathrm{Col}(\boldsymbol{W}^\top)) \leq m$, where $\mathrm{Col}(\cdot)$ denotes the column space of the given matrix, it follows that $\dim(\mathrm{Null}(\boldsymbol{W})) \geq d - m$. Using Lemma A.2, we can upper bound the distance between the set $\{\boldsymbol{e}_j\}_{j \in [N-1]} \subseteq \mathbb{R}^d$ and any subspace of dimension $d - m$.

Let $Z \subseteq \mathrm{Null}(\boldsymbol{W})$ be a subspace such that $\dim(Z) = d - m$, and apply Lemma A.2 with substitutions $d = d$, $t = N - 1$, $k = d - m$ and $Z = Z$. The conditions of lemma, namely $t \leq d$ and $k \geq d - t$, are satisfied since $N - 1 \leq d$ and $m < \rho^2(N-1) \leq N - 1$. Therefore, we obtain the bound

$$\min_{j \in [N-1]} \mathrm{dist}_2(\boldsymbol{e}_j, Z) \leq \sqrt{\frac{m}{N-1}}.$$

By combining the above inequality with Equation (5), we obtain

$$2\mu \leq \min_{j \in [N-1]} \mathrm{dist}_2(\boldsymbol{e}_j, \mathrm{Null}(\boldsymbol{W})) \overset{(a)}{\leq} \min_{j \in [N-1]} \mathrm{dist}_2(\boldsymbol{e}_j, Z) \leq \sqrt{\frac{m}{N-1}}, \tag{6}$$

where (a) follows from that $Z \subseteq \mathrm{Null}(\boldsymbol{W})$. Since $\epsilon_{\mathcal{D}} = 1/2$, we have $2\mu = 2\rho\epsilon_{\mathcal{D}} = \rho$, so Equation (6) becomes

$$\rho \leq \sqrt{\frac{m}{N-1}}.$$

This implies that $m \geq \rho^2(N-1)$, contradicting the assumption $m < \rho^2(N-1)$. Therefore, the width requirement $m \geq \rho^2(N-1)$ is necessary. This concludes the statement for the case $N - 1 \leq d$.

**Case II :** $N - 1 > d$**.**   We construct the first $d + 1$ data points in the same manner as in Case I, using the construction for $N = d + 1$. For the remaining $N - d - 1$ data points, we set them sufficiently distant from the first $d + 1$ data points to ensure that the separation constant remains $\epsilon_{\mathcal{D}} = 1/2$.

In particular, we set $\boldsymbol{x}_{d+2} = 2\boldsymbol{e}_1$, $\boldsymbol{x}_{d+3} = 3\boldsymbol{e}_1, \cdots, \boldsymbol{x}_N = (N-d)\boldsymbol{e}_1$ and assign $y_{d+2} = y_{d+3} = \cdots = y_N = 2$. Compared to the case $N = d + 1$, this construction preserves $\epsilon_{\mathcal{D}}$ while adding more data points to memorize. Since the first $d + 1$ data points are constructed as in the case $N = d + 1$, the same lower bound applies. Specifically, by the result of Case I, any network that $\rho$-robustly

memorizes this dataset must have a first hidden layer of width at least $\rho^2((d+1)-1) = \rho^2 d$. This concludes the argument for the case $N - 1 > d$.

Combining the results from the two cases $N - 1 \leq d$ and $N - 1 > d$ completes the proof of the proposition. □

### A.3 Necessary Condition on Parameters for Robust Memorization

For sufficiently large $\rho$, Gao et al. [2019] and Li et al. [2022] prove that, for any $\mathcal{D} \in \boldsymbol{D}_{d,N,C}$, if there exists $f \in \mathcal{F}_{d,P}$ that $\rho$-robustly memorizes $\mathcal{D}$, the number of parameters $P$ should satisfy $P = \Omega(\sqrt{Nd})$. However, the authors do not characterize the range of $\rho$ over which this lower bound remains valid.

Motivated from Gao et al. [2019] and Li et al. [2022], we establish a lower bound that depends on $\rho$ in the regime $\rho \leq \sqrt{1 - 1/d}$, which becomes $\sqrt{Nd}$ when $\rho = \sqrt{1 - 1/d}$. This implies that the existing lower bound $\sqrt{Nd}$ remains valid for $\rho \in [\sqrt{1 - 1/d}, 1)$. As a result, we obtain a lower bound that holds continuously from $\rho \approx 0$ up to $\rho \approx 1$, and thus interpolates between the lower bound $\sqrt{N}$ for memorization to the lower bound $\sqrt{Nd}$ for robust memorization.

**Proposition 3.3.** *Let* $\rho \in \left(0, \sqrt{1 - \frac{1}{d}}\right]$. *Suppose for any* $\mathcal{D} \in \boldsymbol{D}_{d,N,2}$, *there exists* $f \in \mathcal{F}_{d,P}$ *that* $\rho$-*robustly memorizes* $\mathcal{D}$. *Then, the number of parameters* $P$ *must satisfy*

$$P = \Omega\left(\sqrt{\frac{N}{1 - \rho^2}}\right).$$

*Proof.* To prove the statement, we show that for any $\mathcal{D} \in \boldsymbol{D}_{d,N,2}$, if there exists a network $f \in \mathcal{F}_{d,P}$ that $\rho$-robustly memorizes $\mathcal{D}$, then

$$\text{VC-dim}(\mathcal{F}_{d,P}) = \Omega\left(\frac{N}{1 - \rho^2}\right).^2 \tag{7}$$

If the above bound holds, then as $\text{VC-dim}(\mathcal{F}_{d,P}) = O(P^2)$, it follows that $P = \Omega(\sqrt{N/(1 - \rho^2)})$.

Let $k := \lfloor \frac{1}{1-\rho^2} \rfloor$. To establish the desired VC-dimension lower bound, it suffices to show that

$$\text{VC-dim}(\mathcal{F}_{d,P}) \geq k \cdot \lfloor \frac{N}{2} \rfloor.$$

This implies Equation (7), as desired. To this end, it suffices to construct $k \cdot \lfloor \frac{N}{2} \rfloor$ points in $\mathbb{R}^d$ that can be shattered by $\mathcal{F}_{d,P}$. These points are organized as an union of $\lfloor \frac{N}{2} \rfloor$ groups, each group consisting of $k$ points.

**Step 1** (Constructing $\Omega(N/(1 - \rho^2))$ points $\mathcal{X}$ to be shattered by $\mathcal{F}_{d,P}$).

We begin by constructing the first group. Since $\rho \in \left(0, \sqrt{\frac{d-1}{d}}\right]$, we have $k = \lfloor \frac{1}{1-\rho^2} \rfloor \in (1, d]$. Define the first group $\mathcal{X}_1 := \{\boldsymbol{e}_j\}_{j=1}^k \subseteq \mathbb{R}^d$, consisting of the first $k$ standard basis vectors in $\mathbb{R}^d$. The remaining $\lfloor \frac{N}{2} \rfloor - 1$ groups are constructed by translating $\mathcal{X}_1$. For each $l = 1, \cdots \lfloor \frac{N}{2} \rfloor$, define

$$\mathcal{X}_l := \boldsymbol{c}_l + \mathcal{X}_1 = \{\boldsymbol{c}_l + \boldsymbol{x} \mid \boldsymbol{x} \in \mathcal{X}_1\},$$

where $\boldsymbol{c}_l := 2d^2(l - 1) \cdot \boldsymbol{e}_1$ ensures that the groups are sufficiently distant from one another. Note that $\boldsymbol{c}_1 = \boldsymbol{0}$, so that $\mathcal{X}_1$ is consistent with the definition above. Now, define $\mathcal{X} := \cup_{l \in [\lfloor N/2 \rfloor]} \mathcal{X}_l$ as the union of all groups, comprising $k \times \lfloor \frac{N}{2} \rfloor$ points in total.

**Step 2** (Showing $\mathcal{F}_{d,P}$ shatter $\mathcal{X}$).

---

[2] We follow the definition of VC-dimension by Bartlett et al. [2019]. Note that the VC-dimension of a real-valued function class is defined as the VC-dimension of $\text{sign}(\mathcal{F}) := \{\text{sign} \circ f \mid f \in \mathcal{F}\}$. Since we consider the label set $[2] = \{1, 2\}$ for robust memorization while the VC-dimension requires the label set $\{+1, -1\}$, we take an additional step of an affine transformation in the last step of the proof.

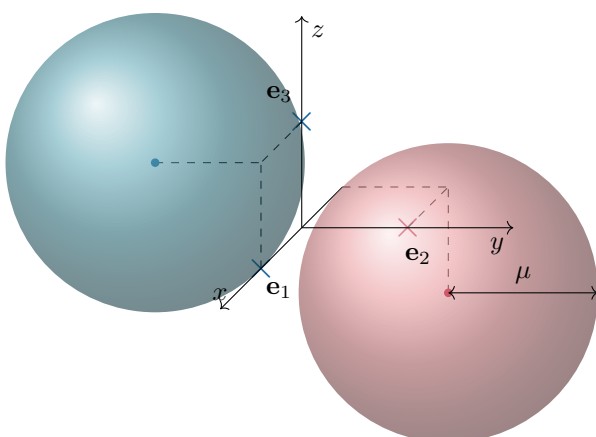

Figure 5: Reduction of Shattering to Robust Memorization. The cross marks refer to the points to be shattered, and the circular dots refer to the points for robust memorization. The centers of robustness balls change with respect to the labels of the points to be shattered.

We claim that for any $\mathcal{D} \in \boldsymbol{D}_{d,N,2}$, if there exists a network $f \in \mathcal{F}_{d,P}$ that $\rho$-robustly memorizes $\mathcal{D}$, then the point set $\mathcal{X}$ is shattered by $\mathcal{F}_{d,P}$. To prove the claim, consider an arbitrary labeling $\mathcal{Y} = \{y_{l,j}\}_{l \in [\lfloor N/2 \rfloor], j \in [k]}$ of the points in $\mathcal{X}$, where each label $y_{l,j} \in \{\pm 1\}$ corresponds to the point $\boldsymbol{x}_{l,j} := \boldsymbol{c}_l + \boldsymbol{e}_j \in \mathcal{X}$.

Given the labeling $\mathcal{Y}$, we construct $\mathcal{D} \in \boldsymbol{D}_{d,N,2}$ with labels in $\{1, 2\}$ such that any function $f \in \mathcal{F}_{d,P}$ that $\rho$-robustly memorizes $\mathcal{D}$ can be affinely transformed to $f' = 2f - 3 \in \mathcal{F}_{d,P}$, which satisfies $f'(\boldsymbol{x}_{l,j}) = y_{l,j} \in \{\pm 1\}$ for all $\boldsymbol{x}_{l,j} \in \mathcal{X}$. In other words, $f'$ exactly memorizes the given labeling $\mathcal{Y}$ over $\mathcal{X}$, thereby showing that $\mathcal{X}$ is shatterd by $\mathcal{F}_{d,P}$. The affine transformation is necessary to match the $\{1, 2\}$-valued outputs of $f$ with the $\{\pm 1\}$ labeling required for the shattering argument.

For each $l \in [\lfloor N/2 \rfloor]$, define the index sets

$$J_l^+ = \{j \in [k] \mid y_{l,j} = +1\}, \quad J_l^- = \{j \in [k] \mid y_{l,j} = -1\},$$

which partition the group-wise labeling $\{y_{l,j}\}_{j \in [k]} \subset \mathcal{Y}$ into positive and negative indices. We then define

$$\boldsymbol{x}_{2l-1} = \boldsymbol{c}_l + \sum_{j \in J_l^+} \boldsymbol{e}_j - \sum_{j \in J_l^-} \boldsymbol{e}_j,$$

$$\boldsymbol{x}_{2l} = \boldsymbol{c}_l + \sum_{j \in J_l^-} \boldsymbol{e}_j - \sum_{j \in J_l^+} \boldsymbol{e}_j.$$

Let $y_{2l-1} = 2, y_{2l} = 1$, and define the dataset $\mathcal{D} = \{(\boldsymbol{x}_i, y_i)\}_{i \in [N]} \in \boldsymbol{D}_{d,N,2}$. Figure 5 illustrates the first group $l = 1$ with $k = 3$ where the labels gives the index sets $J_1^+ = \{1, 3\}$ and $J_1^- = \{2\}$. The blue and red dots denote the points $\boldsymbol{x}_1$ and $\boldsymbol{x}_2$, respectively.

To analyze the separation constant $\epsilon_{\mathcal{D}}$, we consider the distance between pairs of points with different labels. Specifically, for each $l$, the two points $\boldsymbol{x}_{2l-1}$ and $\boldsymbol{x}_{2l}$ have opposite labels by construction. Consider their distance:

$$\|\boldsymbol{x}_{2l-1} - \boldsymbol{x}_{2l}\|_2 = \left\| 2 \left( \sum_{j \in J_l^+} \boldsymbol{e}_j - \sum_{j \in J_l^-} \boldsymbol{e}_j \right) \right\|_2 \overset{(a)}{=} 2\sqrt{k},$$

where (a) holds since $J_l^+ \cap J_l^- = \emptyset$ and $J_l^+ \cup J_l^- = [k]$. Now, for $l \neq l'$, consider the distance between $\boldsymbol{x}_{2l-1}$ and $\boldsymbol{x}_{2l'}$, which again correspond to different labels. We have:

$$\mathrm{dist}_2(\boldsymbol{x}_{2l-1}, \boldsymbol{x}_{2l'}) \overset{(a)}{\geq} \mathrm{dist}_2(\boldsymbol{c}_l, \boldsymbol{c}_{l'}) - \mathrm{dist}_2(\boldsymbol{c}_l, \boldsymbol{x}_{2l-1}) - \mathrm{dist}_2(\boldsymbol{c}_{l'}, \boldsymbol{x}_{2l'})$$

$$\overset{(b)}{\geq} 2d^2 - \sqrt{k} - \sqrt{k}$$

$$\overset{(c)}{\geq} 2d^2 - 2\sqrt{d}$$

$$\overset{(d)}{\geq} 2\sqrt{d}$$

$$\overset{(e)}{\geq} 2\sqrt{k},$$

where (a) follows from the triangle inequality, (b) uses $\mathrm{dist}_2(\boldsymbol{c}_l, \boldsymbol{x}_{2l-1}) = \mathrm{dist}_2(\boldsymbol{c}_{l'}, \boldsymbol{x}_{2l'}) = \sqrt{k}$, (c) and (e) use $k \leq d$, and (d) holds for all $d \geq 2$. Thus, we conclude that $\epsilon_{\mathcal{D}} \geq \sqrt{k}$.

Let $f \in \mathcal{F}_{d,P}$ be a function that $\rho$-robustly memorizes $\mathcal{D}$. We begin by deriving a lower bound on the robustness radius $\mu$ in order to verify that $f' = 2f - 3$ correctly memorizes the given labeling $\mathcal{Y}$ over $\mathcal{X}$. Define $\phi(t) := \sqrt{\frac{t-1}{t}}$. The function $\phi$ is strictly increasing for $t \geq 1$, and maps $[1, \infty)$ onto $[0, 1)$. Hence, it admits an inverse $\phi^{-1} : [0, 1) \to [1, \infty)$, defined as $\phi^{-1}(\rho) = \frac{1}{1-\rho^2}$. Therefore, we have

$$\rho = \phi(\phi^{-1}(\rho)) = \phi\left(\frac{1}{1-\rho^2}\right) \geq \phi\left(\lfloor \frac{1}{1-\rho^2} \rfloor\right) = \phi(k) = \sqrt{\frac{k-1}{k}}.$$

Given $\epsilon_{\mathcal{D}} \geq \sqrt{k}$ and $\rho \geq \sqrt{\frac{k-1}{k}}$, it follows that $\mu = \rho \epsilon_{\mathcal{D}} \geq \sqrt{k-1}$. Thus, any function $f$ that $\rho$-robustly memorizes $\mathcal{D}$ must also memorize all points within an $\ell_2$-ball of radius $\sqrt{k-1}$ centered at each point in $\mathcal{D}$.

Next, for $\boldsymbol{x}_{l,j} \in \mathcal{X}$ with positive label $y_{l,j} = +1$, we have

$$\|\boldsymbol{x}_{l,j} - \boldsymbol{x}_{2l-1}\|_2 = \left\| (\boldsymbol{c}_l + \boldsymbol{e}_j) - (\boldsymbol{c}_l + \sum_{j' \in J_l^+} \boldsymbol{e}_{j'} - \sum_{j' \in J_l^-} \boldsymbol{e}_{j'}) \right\|_2$$

$$= \left\| \sum_{\substack{j' \in J_l^+ \\ j' \neq j}} \boldsymbol{e}_{j'} - \sum_{j' \in J_l^-} \boldsymbol{e}_{j'} \right\|_2$$

$$= \sqrt{k-1}.$$

Now consider a sequence $\{\boldsymbol{z}_n\}_{n \in \mathbb{N}}$ such that $\boldsymbol{z}_n \to \boldsymbol{x}_{l,j}$ as $n \to \infty$ and

$$\|\boldsymbol{z}_n - \boldsymbol{x}_{2l-1}\|_2 < \sqrt{k-1} \quad \text{for all } n \in \mathbb{N}.$$

In particular, we can take

$$\boldsymbol{z}_n := \frac{n-1}{n}\boldsymbol{x}_{l,j} + \frac{1}{n}\boldsymbol{x}_{2l-1},$$

which satisfies such properties. Then, $\boldsymbol{z}_n \in \mathcal{B}(\boldsymbol{x}_{2l-1}, \mu)$ for all $n$, and by robustness of $f$, $f(\boldsymbol{z}_n) = f(\boldsymbol{x}_{2l-1}) = 2$. By continuity of $f$, we have

$$f(\boldsymbol{x}_{l,j}) = f(\lim_{n\to\infty} \boldsymbol{z}_n) = \lim_{n\to\infty} f(\boldsymbol{z}_n) = \lim_{n\to\infty} 2 = 2.$$

Similarly, for $\boldsymbol{x}_{l,j} \in \mathcal{X}$ with negative label $y_{l,j} = -1$, we have $\|\boldsymbol{x}_{l,j} - \boldsymbol{x}_{2l}\|_2 = \sqrt{k-1}$, so that $f(\boldsymbol{x}_{l,j}) = 1$.

Since we can adjust the weight and the bias of the last hidden layer, $\mathcal{F}_{d,P}$ is closed under affine transformation; that is, $af + b \in \mathcal{F}_{d,P}$ whenever $f \in \mathcal{F}_{d,P}$. In particular, $f' := 2f - 3 \in \mathcal{F}_{d,P}$. This $f'$ satisfies $f'(\boldsymbol{x}_{l,j}) = 2f(\boldsymbol{x}_{l,j}) - 3 = 2 \cdot 2 - 3 = +1$ whenever $y_{l,j} = +1$ and $f'(\boldsymbol{x}_{l,j}) = 2f(\boldsymbol{x}_{l,j}) - 3 = 2 \cdot 1 - 3 = -1$ whenever $y_{l,j} = -1$. Thus, $\mathrm{sign} \circ f'$ perfectly classifies $\mathcal{X}$ according to the given labeling $\mathcal{Y}$. Since such $f' \in \mathcal{F}_{d,P}$ exists for an arbitrary labeling $\mathcal{Y}$, it follows that $\mathcal{F}_{d,P}$ shatters $\mathcal{X}$, completing the proof of the theorem.

$\square$

### A.4 Lemmas for Appendix A

The following lemma upper bounds the $\ell_2$-distance between the standard basis and any subspace of a given dimension.

**Lemma A.1.** *Let $\{e_j\}_{j\in[d]} \subseteq \mathbb{R}^d$ denote the standard basis of $\mathbb{R}^d$. Then, for any $k$-dimensional subspace $Z \subseteq \mathbb{R}^d$,*

$$\max_{j\in[d]} \|\mathrm{Proj}_Z(e_j)\|_2 \geq \sqrt{\frac{k}{d}}.$$

*In particular,*

$$\min_{j\in[d]} \mathrm{dist}_2(e_j, Z) \leq \sqrt{\frac{d-k}{d}}.$$

*Proof.* Let $\{u_1, u_2, \cdots, u_k\} \subseteq \mathbb{R}^d$ be an orthonormal basis of $Z$, and denote each $u_l = (u_{l1}, u_{l2}, \cdots, u_{ld})^\top$. Let $U \in \mathbb{R}^{d\times k}$ be the matrix whose columns are $u_1, \cdots, u_k$, so that

$$U = \begin{bmatrix} | & | & & | \\ u_1 & u_2 & \cdots & u_k \\ | & | & & | \end{bmatrix}.$$

Then the projection matrix $P$ onto $Z$ is given by

$$P = U(U^\top U)^{-1} U^\top = UU^\top = \sum_{l=1}^k u_l u_l^\top \in \mathbb{R}^{d\times d}.$$

Now, for each standard basis vector $e_j$, the squared norm of its projection onto $Z$ is:

$$\|Pe_j\|_2^2 = \left\| \sum_{l=1}^k u_l u_l^\top e_j \right\|_2^2 = \left\| \sum_{l=1}^k u_{lj} u_l \right\|_2^2 = \sum_{l=1}^k (u_{lj})^2,$$

where the last equality holds as $u_l$ are orthonormal. Moreover,

$$\max_{j\in[d]} \|Pe_j\|_2^2 \geq \frac{1}{d} \sum_{j\in[d]} \|Pe_j\|_2^2 = \frac{1}{d} \sum_{j\in[d]} \sum_{l=1}^k (u_{lj})^2 = \frac{1}{d} \sum_{l=1}^k \sum_{j\in[d]} (u_{lj})^2 = \frac{1}{d} \sum_{l=1}^k 1 = \frac{k}{d}.$$

This proves the first statement of the lemma. To prove the second statement, observe that for any $v \in \mathbb{R}^d$, we can write

$$v = \mathrm{Proj}_Z(v) + \mathrm{Proj}_{Z^\perp}(v),$$

so that $\|v\|_2^2 = \|\mathrm{Proj}_Z(v)\|_2^2 + \|\mathrm{Proj}_{Z^\perp}(v)\|_2^2$. Noticing $\mathrm{dist}_2(v, Z) = \|\mathrm{Proj}_{Z^\perp}(v)\|_2$ together with the first statement, we have

$$\begin{aligned}
\min_{j\in[d]} \mathrm{dist}_2(e_j, Z) &= \min_{j\in[d]} \|\mathrm{Proj}_{Z^\perp}(e_j)\|_2 \\
&= \min_{j\in[d]} \sqrt{1 - \|\mathrm{Proj}_Z(e_j)\|_2^2} \\
&= \sqrt{1 - \max_{j\in[d]} \|\mathrm{Proj}_Z(e_j)\|_2^2} \\
&\leq \sqrt{1 - \frac{k}{d}} \\
&= \sqrt{\frac{d-k}{d}},
\end{aligned}$$

which concludes the second statement. $\qquad\square$

The next lemma generalizes Lemma A.1 to the case where we consider only the distance to a subset of the standard basis, instead of the whole standard basis.

**Lemma A.2.** *Let $1 \leq t \leq d$, and let $\{e_j\}_{j \in [t]} \subseteq \mathbb{R}^d$ denote the first $t$ standard basis vectors. Then, for any $k$-dimensional subspace $Z \subseteq \mathbb{R}^d$ with $k \geq d - t$, we have*

$$\max_{j \in [t]} \|\mathrm{Proj}_Z(e_j)\|_2 \geq \sqrt{\frac{k - (d - t)}{t}}.$$

*In particular,*

$$\min_{j \in [t]} \mathrm{dist}_2(e_j, Z) \leq \sqrt{\frac{d - k}{t}}.$$

*Proof.* Let $Q = [e_1 e_2 \cdots e_t]^\top \in \mathbb{R}^{t \times d}$. Then, we have the orthogonal decomposition:

$$\mathbb{R}^d = \mathrm{Col}(Q^\top) \oplus \mathrm{Null}(Q) = (Z \cap \mathrm{Col}(Q^\top)) \oplus (Z^\perp \cap \mathrm{Col}(Q^\top)) \oplus \mathrm{Null}(Q).$$

By taking dimensions,

$$
\begin{aligned}
\dim(Z \cap \mathrm{Col}(Q^\top)) &= \dim(\mathbb{R}^d) - \dim(Z^\perp \cap \mathrm{Col}(Q^\top)) - \dim(\mathrm{Null}(Q)) \\
&\geq \dim(\mathbb{R}^d) - \dim(Z^\perp) - \dim(\mathrm{Null}(Q)) \\
&= d - (d - k) - (d - t) \\
&= k - (d - t).
\end{aligned}
$$

Now, consider the restriction of $\mathbb{R}^d$ to $\mathbb{R}^t$ by the linear map

$$\phi : \mathrm{span}\{e_1, \ldots, e_t\} \subset \mathbb{R}^d \to \mathbb{R}^t, \quad \phi\left(\sum_{i=1}^t a_i e_i\right) = \begin{bmatrix} a_1 \\ \vdots \\ a_t \end{bmatrix}.$$

Since $\mathrm{Col}(Q^\top) = \mathrm{span}\{e_1, \ldots, e_t\}$, the projection satisfies:

$$\max_{j \in [t]} \left\|\mathrm{Proj}_{Z \cap \mathrm{Col}(Q^\top)}(e_j)\right\|_2 = \max_{j \in [t]} \left\|\mathrm{Proj}_{\phi(Z \cap \mathrm{Col}(Q^\top))}(\phi(e_j))\right\|_2.$$

By applying Lemma A.1 with the restricted space $\mathbb{R}^t$, we obtain

$$\max_{j \in [t]} \left\|\mathrm{Proj}_{Z \cap \mathrm{Col}(Q^\top)}(e_j)\right\|_2 \geq \sqrt{\frac{k - (d - t)}{t}}.$$

Since $Z \supseteq Z \cap \mathrm{Col}(Q^\top)$, it follows that

$$\max_{j \in [t]} \|\mathrm{Proj}_Z(e_j)\|_2 \geq \max_{j \in [t]} \left\|\mathrm{Proj}_{Z \cap \mathrm{Col}(Q^\top)}(e_j)\right\|_2 \geq \sqrt{\frac{k - (d - t)}{t}}.$$

This proves the first statement. To prove the second statement, for any $v \in \mathbb{R}^d$, decompose $v$ as

$$v = \mathrm{Proj}_Z(v) + \mathrm{Proj}_{Z^\perp}(v),$$

and note that $\|v\|_2^2 = \|\mathrm{Proj}_Z(v)\|_2^2 + \|\mathrm{Proj}_{Z^\perp}(v)\|_2^2$. Using $\mathrm{dist}_2(v, Z) = \|\mathrm{Proj}_{Z^\perp}(v)\|_2$ together with the first statement, we have

$$
\begin{aligned}
\min_{j \in [t]} \mathrm{dist}_2(e_j, Z) &= \min_{j \in [t]} \|\mathrm{Proj}_{Z^\perp}(e_j)\|_2 \\
&= \min_{j \in [t]} \sqrt{1 - \|\mathrm{Proj}_Z(e_j)\|_2^2} \\
&= \sqrt{1 - \max_{j \in [t]} \|\mathrm{Proj}_Z(e_j)\|_2^2} \\
&\leq \sqrt{1 - \frac{k - (d - t)}{t}} \\
&= \sqrt{\frac{d - k}{t}},
\end{aligned}
$$

which concludes the second statement. $\square$

# B  Proofs for Section 4

In this section, we prove an extended version of Theorem 4.2, which additionally states the explicit bounds on depth, width, and bit complexity, in addition to the sufficient number of parameters. We present the $\ell_p$-norm version of Theorem 4.2 in Theorem C.11.

**Theorem B.1.** *For any dataset $\mathcal{D} \in \boldsymbol{D}_{d,N,C}$ and $\eta \in (0,1)$, the following statements hold:*

(i) *If $\rho \in \left(0, \frac{1}{5N\sqrt{d}}\right]$, there exists $f$ with $\tilde{O}(\sqrt{N})$ parameters, depth $\tilde{O}(\sqrt{N})$, width $\tilde{O}(1)$ and bit complexity $\tilde{O}(\sqrt{N})$ that $\rho$-robustly memorizes $\mathcal{D}$.*

(ii) *If $\rho \in \left(\frac{1}{5N\sqrt{d}}, \frac{1}{5\sqrt{d}}\right]$, there exists $f$ with $\tilde{O}(Nd^{\frac{1}{4}}\rho^{\frac{1}{2}})$ parameters, depth $\tilde{O}(Nd^{\frac{1}{4}}\rho^{\frac{1}{2}})$, width $\tilde{O}(1)$ and bit complexity $\tilde{O}\left(1/d^{\frac{1}{4}}\rho^{\frac{1}{2}}\right)$ that $\rho$-robustly memorizes $\mathcal{D}$ with error at most $\eta$.*

(iii) *If $\rho \in \left(\frac{1}{5\sqrt{d}}, 1\right)$, there exists $f$ with $\tilde{O}(Nd^2\rho^4)$ parameters, depth $\tilde{O}(N)$, width $\tilde{O}(\rho^2 d)$ and bit complexity $\tilde{O}(N)$ that $\rho$-robustly memorizes $\mathcal{D}$.*

Here, the bit complexity is defined as a bit needed per parameter under a fixed point precision. To prove Theorem B.1, we decompose it into three theorems (Theorems B.3, B.5 and B.14), each corresponding to one of the cases in the statement. Their proofs are provided in Appendices B.1 to B.3, respectively.

**Remark B.2** (Tight Bit Complexity). The bit complexities in Theorems B.1(i) and B.1(ii) are essentially tight within our construction framework. Vardi et al. [2021] provide a lower bound on bit complexity using upper and lower bounds on VC-dimension. In particular, for a network with $P$ *nonzero* parameters (refer Appendix E for detailed analysis on nonzero parameters) and bit complexity $B$, the VC-dimension is upper bounded as

$$\text{VC-Dim} = O(PB + P\log P).$$

Since VC-dimension is lower bounded by $N$ by the robust memorization, combining these two bounds suggests the necessary bit complexity required under our constructions in Theorem 4.2. For simplicity, assume the case where the omitted $d$ in the upper bound is not dominant. In Theorem E.2, we show that under our constructions, the number of nonzero parameters satisfies $P = \tilde{O}(\sqrt{N})$ for small $\rho$ and $P = \tilde{O}(Nd^{1/4}\rho^{1/2})$ for moderate $\rho$. Consequently, the bit complexity becomes

$$B = \tilde{\Omega}(\sqrt{N}) \text{ and } B = \tilde{\Omega}(\frac{1}{d^{1/4}\rho^{1/2}}),$$

respectively, which matches the upper bounds.

## B.1  Sufficient Condition for Robust Memorization with Small Robustness Radius

**Theorem B.3.** *Let $\rho \in \left(0, \frac{1}{5N\sqrt{d}}\right]$. For any dataset $\mathcal{D} \in \boldsymbol{D}_{d,N,C}$, there exists $f$ with $\tilde{O}(\sqrt{N})$ parameters, depth $\tilde{O}(\sqrt{N})$, width $\tilde{O}(1)$ and bit complexity $\tilde{O}(\sqrt{N})$ that $\rho$-robustly memorizes $\mathcal{D}$.*

*Proof.* For given $\rho$ and $\mathcal{D} = \{(\boldsymbol{x}_i, y_i)\}_{i\in[N]} \in \boldsymbol{D}_{d,N,C}$, we construct a network $f \in \mathcal{F}_{d,P}$ that $\rho$-robustly memorizes $\mathcal{D}$ with $\tilde{O}(\sqrt{N})$ parameters. The construction proceeds in four stages. In each stage, we define a function implementable by a neural network, such that their composition yields a $\rho$-robust memorizer for $\mathcal{D}$.

**Stage I** (Projection onto $\log$-scale Dimension and Scaling via the First Hidden Layer Weight Matrix). By Lemma B.20, we obtain an integer $m = \tilde{O}(\log N)$ and a 1-Lipschitz linear map $\phi : \mathbb{R}^d \to \mathbb{R}^m$ such that the projected dataset $\mathcal{D}' := \{(\phi(\boldsymbol{x}_i), y_i)\}_{i\in[N]} \in \boldsymbol{D}_{m,N,C}$ satisfies the separation bound

$$\epsilon'_{\mathcal{D}} \geq \frac{5}{12}\sqrt{\frac{m}{d}}\epsilon_{\mathcal{D}}. \tag{8}$$

We define $f_{\text{proj}} : \mathbb{R}^d \to \mathbb{R}^m$ as $f_{\text{proj}}(\boldsymbol{x}) = \frac{11}{9} \cdot \frac{\sqrt{d}}{\epsilon_{\mathcal{D}}}\phi(\boldsymbol{x})$, which is $\frac{11}{9} \cdot \frac{\sqrt{d}}{\epsilon_{\mathcal{D}}}$-Lipschitz.

We apply Lemma B.23 with $f_{\text{proj}}$ whose depth is 1, $\nu = \min\left\{\frac{109}{11880}\sqrt{m}, \frac{1}{88}\mu, \frac{1}{360N}, 1\right\}$ and $\bar{R} := \max\{\|\boldsymbol{x}\|_2 \mid \boldsymbol{x} \in \mathcal{B}_2(\boldsymbol{x}_i, \mu) \quad \text{for some } i \in [N]\}$ to obtain $\bar{f}_{\text{proj}}$ with the same number of parameters, depth and width and $\tilde{O}(1)$ bit complexity such that

$$\max_{\|\boldsymbol{x}\|_2 \leq \bar{R}} \|\bar{f} - f\|_2 \leq \nu. \tag{9}$$

We set the first hidden layer bias $\boldsymbol{b} \in \mathbb{R}^m$ so that

$$\bar{f}_{\text{proj}}(\boldsymbol{x}) + \boldsymbol{b} \geq \boldsymbol{0} \text{ for all } i \in [N] \text{ and all } \boldsymbol{x} \in \mathcal{B}_2(\boldsymbol{x}_i, \mu), \tag{10}$$

where the comparison between two vectors is element-wise.

We claim that for $\mathcal{D}'' := \{(\sigma(\bar{f}_{\text{proj}}(\boldsymbol{x}_i) + \boldsymbol{b}), y_i)\}_{i\in[N]}$, we have (i) $\epsilon_{\mathcal{D}''} \geq \sqrt{m}/2$ and (ii) for $\rho'' := \frac{1}{4N\epsilon_{\mathcal{D}''}}$, if $g(\boldsymbol{x}) \in \mathcal{F}_{m,P}$ can $\rho''$-robustly memorize $\mathcal{D}''$, then $g \circ \sigma \circ (\bar{f}_{\text{proj}}(\boldsymbol{x}) + \boldsymbol{b})$ can $\rho$-robustly memorize $\mathcal{D}$. For any $i \neq j$ with $y_i \neq y_j$, we have

$$\left\|\sigma(\bar{f}_{\text{proj}}(\boldsymbol{x}_i) + \boldsymbol{b}) - \sigma(\bar{f}_{\text{proj}}(\boldsymbol{x}_j) + \boldsymbol{b})\right\|_2$$
$$\stackrel{(a)}{=} \left\|(\bar{f}_{\text{proj}}(\boldsymbol{x}_i) + \boldsymbol{b}) - (\bar{f}_{\text{proj}}(\boldsymbol{x}_j) + \boldsymbol{b})\right\|_2$$
$$= \left\|\bar{f}_{\text{proj}}(\boldsymbol{x}_i) - \bar{f}_{\text{proj}}(\boldsymbol{x}_j)\right\|_2$$
$$= \left\|(\bar{f}_{\text{proj}}(\boldsymbol{x}_i) - f_{\text{proj}}(\boldsymbol{x}_i)) - (\bar{f}_{\text{proj}}(\boldsymbol{x}_j) - f_{\text{proj}}(\boldsymbol{x}_j)) + (f_{\text{proj}}(\boldsymbol{x}_i) - f_{\text{proj}}(\boldsymbol{x}_j))\right\|_2.$$

where (a) holds by the construction of $\boldsymbol{b}$ (Equation (10)). For simplicity, we denote

$$\Delta(\boldsymbol{x}_i, \boldsymbol{x}_j) := (\bar{f}_{\text{proj}}(\boldsymbol{x}_i) - f_{\text{proj}}(\boldsymbol{x}_i)) - (\bar{f}_{\text{proj}}(\boldsymbol{x}_j) - f_{\text{proj}}(\boldsymbol{x}_j)).$$

Then, we have

$$\left\|\sigma(\bar{f}_{\text{proj}}(\boldsymbol{x}_i) + \boldsymbol{b}) - \sigma(\bar{f}_{\text{proj}}(\boldsymbol{x}_j) + \boldsymbol{b})\right\|_2^2$$
$$= \|\Delta(\boldsymbol{x}_i, \boldsymbol{x}_j) + (f_{\text{proj}}(\boldsymbol{x}_i) - f_{\text{proj}}(\boldsymbol{x}_j))\|_2^2$$
$$\stackrel{(a)}{\geq} (\|f_{\text{proj}}(\boldsymbol{x}_i) - f_{\text{proj}}(\boldsymbol{x}_j)\|_2 - \|\Delta(\boldsymbol{x}_i, \boldsymbol{x}_j)\|_2)^2$$
$$= \|f_{\text{proj}}(\boldsymbol{x}_i) - f_{\text{proj}}(\boldsymbol{x}_j)\|_2^2 - 2\|f_{\text{proj}}(\boldsymbol{x}_i) - f_{\text{proj}}(\boldsymbol{x}_j)\|_2\|\Delta(\boldsymbol{x}_i, \boldsymbol{x}_j)\|_2 + \|\Delta(\boldsymbol{x}_i, \boldsymbol{x}_j)\|_2^2$$
$$\geq \|f_{\text{proj}}(\boldsymbol{x}_i) - f_{\text{proj}}(\boldsymbol{x}_j)\|_2^2 - 2\|f_{\text{proj}}(\boldsymbol{x}_i) - f_{\text{proj}}(\boldsymbol{x}_j)\|_2\|\Delta(\boldsymbol{x}_i, \boldsymbol{x}_j)\|_2,$$

where (a) holds from $\|\boldsymbol{a} + \boldsymbol{b}\|_2^2 \geq (\|\boldsymbol{a}\|_2 - \|\boldsymbol{b}\|_2)^2$. By the construction of $\bar{f}$ (Equation (9)),

$$\|\Delta(\boldsymbol{x}_i, \boldsymbol{x}_j)\|_2 \leq \|\bar{f}_{\text{proj}}(\boldsymbol{x}_i) - f_{\text{proj}}(\boldsymbol{x}_i)\|_2 + \|\bar{f}_{\text{proj}}(\boldsymbol{x}_j) - f_{\text{proj}}(\boldsymbol{x}_j)\|_2 \leq 2\nu,$$

so we have

$$\left\|\sigma(\bar{f}_{\text{proj}}(\boldsymbol{x}_i) + \boldsymbol{b}) - \sigma(\bar{f}_{\text{proj}}(\boldsymbol{x}_j) + \boldsymbol{b})\right\|_2^2$$
$$\geq \|f_{\text{proj}}(\boldsymbol{x}_i) - f_{\text{proj}}(\boldsymbol{x}_j)\|_2^2 - 4\nu\|f_{\text{proj}}(\boldsymbol{x}_i) - f_{\text{proj}}(\boldsymbol{x}_j)\|_2. \tag{11}$$

Now we derive

$$\|f_{\text{proj}}(\boldsymbol{x}_i) - f_{\text{proj}}(\boldsymbol{x}_j)\|_2 \stackrel{(a)}{=} \frac{11}{9} \cdot \frac{\sqrt{d}}{\epsilon_{\mathcal{D}}} \|\phi(\boldsymbol{x}_i) - \phi(\boldsymbol{x}_j)\|_2$$
$$\stackrel{(b)}{\geq} \frac{11}{9} \cdot \frac{\sqrt{d}}{\epsilon_{\mathcal{D}}} \cdot 2\epsilon_{\mathcal{D}'}$$
$$\stackrel{(c)}{\geq} \frac{11}{9} \cdot \frac{\sqrt{d}}{\epsilon_{\mathcal{D}}} \times 2 \cdot \frac{5}{12}\sqrt{\frac{m}{d}}\epsilon_{\mathcal{D}}$$
$$= \frac{55}{54}\sqrt{m}, \tag{12}$$

where (a) is by the definition of $f_{\text{proj}}$, (b) is by the definition of $\mathcal{D}'$ and its separation constant, and (c) follows from Equation (8).

Plugging this inequality to Equation (11) gives

$$\left\|\sigma(\bar{f}_{\text{proj}}(\boldsymbol{x}_i) + \boldsymbol{b}) - \sigma(\bar{f}_{\text{proj}}(\boldsymbol{x}_j) + \boldsymbol{b})\right\|_2^2 \geq \|f_{\text{proj}}(\boldsymbol{x}_i) - f_{\text{proj}}(\boldsymbol{x}_j)\|_2(\frac{55}{54}\sqrt{m} - 4\nu)$$

$$\overset{(a)}{\geq} \|f_{\text{proj}}(\boldsymbol{x}_i) - f_{\text{proj}}(\boldsymbol{x}_j)\|_2 \big(\frac{55}{54}\sqrt{m} - \frac{109}{2970}\sqrt{m}\big)$$

$$= \|f_{\text{proj}}(\boldsymbol{x}_i) - f_{\text{proj}}(\boldsymbol{x}_j)\|_2 \cdot \frac{54}{55}\sqrt{m}$$

$$\overset{(b)}{\geq} \frac{55}{54}\sqrt{m} \cdot \frac{54}{55}\sqrt{m}$$

$$= m$$

where (a) holds from $\nu \leq \frac{109}{11880}\sqrt{m}$, and (b) holds from Equation (12). This proves the first claim $\epsilon_{\mathcal{D}''} \geq \sqrt{m}/2$. To prove the second claim, let $\mu := \rho\epsilon_{\mathcal{D}}$ and $\mu'' := \rho''\epsilon_{\mathcal{D}''}$. Then,

$$\sigma(\bar{f}_{\text{proj}}(\mathcal{B}_2(\boldsymbol{x}_i, \mu)) + \boldsymbol{b}) \overset{(a)}{=} \bar{f}_{\text{proj}}(\mathcal{B}_2(\boldsymbol{x}_i, \mu)) + \boldsymbol{b}$$

$$\overset{(b)}{\subseteq} f_{\text{proj}}(\mathcal{B}_2(\boldsymbol{x}_i, \mu + \nu)) + \boldsymbol{b}$$

$$\overset{(c)}{\subseteq} f_{\text{proj}}(\mathcal{B}_2(\boldsymbol{x}_i, \frac{89}{88}\mu)) + \boldsymbol{b}$$

$$\overset{(d)}{\subseteq} \mathcal{B}_2(f_{\text{proj}}(\boldsymbol{x}_i), \frac{89}{72} \cdot \frac{\sqrt{d}}{\epsilon_{\mathcal{D}}} \times \mu) + \boldsymbol{b}$$

$$\overset{(e)}{=} \mathcal{B}_2(f_{\text{proj}}(\boldsymbol{x}_i), \frac{89}{72} \cdot \sqrt{d}\rho) + \boldsymbol{b}$$

$$\overset{(f)}{\subseteq} \mathcal{B}_2(f_{\text{proj}}(\boldsymbol{x}_i), \frac{89}{360N}) + \boldsymbol{b}$$

$$\overset{(g)}{\subseteq} \mathcal{B}_2(\bar{f}_{\text{proj}}(\boldsymbol{x}_i), \frac{89}{360N} + \nu) + \boldsymbol{b}$$

$$\overset{(h)}{\subseteq} \mathcal{B}_2(\bar{f}_{\text{proj}}(\boldsymbol{x}_i), \frac{1}{4N}) + \boldsymbol{b}$$

$$\overset{(i)}{=} \mathcal{B}_2(\bar{f}_{\text{proj}}(\boldsymbol{x}_i), \rho''\epsilon_{\mathcal{D}''}) + \boldsymbol{b}$$

$$= \mathcal{B}_2(\bar{f}_{\text{proj}}(\boldsymbol{x}_i) + \boldsymbol{b}, \rho''\epsilon_{\mathcal{D}''})$$

$$\overset{(j)}{=} \mathcal{B}_2(\sigma(\bar{f}_{\text{proj}}(\boldsymbol{x}_i) + \boldsymbol{b}), \rho''\epsilon_{\mathcal{D}''}),$$

where (a) and (j) are by Equation (10), (b) and (g) are by the construction of $\bar{f}$ (Equation (9)), (c) is because $\nu \leq \frac{1}{88}\mu$, (d) is because $f_{\text{proj}}$ is $\frac{11}{9} \cdot \frac{\sqrt{d}}{\epsilon_{\mathcal{D}}}$-Lipschitz, (e) uses $\mu = \rho\epsilon_{\mathcal{D}}$, (f) uses $\rho \leq \frac{1}{5N\sqrt{d}}$, (h) is because $\nu \leq \frac{1}{360N}$ and (i) is because $\rho''\epsilon_{\mathcal{D}''} = \frac{1}{4N\epsilon_{\mathcal{D}''}}\epsilon_{\mathcal{D}''} = \frac{1}{4N}$.

Hence, $g(\boldsymbol{x})$ memorizing the robustness ball $\mathcal{B}_2(\sigma(\bar{f}_{\text{proj}}(\boldsymbol{x}_i) + \boldsymbol{b}), \rho''\epsilon_{\mathcal{D}''})$ on projected space leads to $g \circ \sigma \circ (\bar{f}_{\text{proj}}(\boldsymbol{x}) + \boldsymbol{b})$ memorizing the robustness ball for $\mathcal{D}$. In other words, if $g(\boldsymbol{x}) \in \mathcal{F}_{m,P}$ can $\rho''$-robustly memorize $\mathcal{D}''$, then $g \circ \sigma \circ (\bar{f}_{\text{proj}}(\boldsymbol{x}) + \boldsymbol{b})$ can $\rho$-robustly memorize $\mathcal{D}$. With $\rho'' = \frac{1}{4N\epsilon_{\mathcal{D}''}}$, Stage II to IV aims to find a $\rho''$-robust memorizer $g$ of $\mathcal{D}''$.

**Stage II** (Translation for Distancing from Lattice via the Bias) For simplicity of the notation, let $\boldsymbol{z}_i := \sigma(\bar{f}_{\text{proj}}(\boldsymbol{x}_i) + \boldsymbol{b})$ for each $i \in [N]$, so that $\mathcal{D}'' = \{(\boldsymbol{z}_i, y_i)\}_{i \in [N]}$. Recall that $\rho'' = \frac{1}{4N\epsilon_{\mathcal{D}''}}$ gives the robustness radius is $\mu'' = \rho''\epsilon_{\mathcal{D}''} = \frac{1}{4N}$.

By applying Lemma B.15 to $\boldsymbol{z}_1, \cdots, \boldsymbol{z}_N$, we obtain a translation vector $\boldsymbol{b}_2 = (b_{21}, \cdots, b_{2m}) \in \mathbb{R}^m$ with bit complexity $\lceil \log(6N) \rceil$ such that

$$\text{dist}(z_{i,j} - b_{2j}, \mathbb{Z}) \geq \frac{1}{3N}, \quad \forall i \in [N], j \in [d], \tag{13}$$

i.e., the translated points $\{\boldsymbol{z}_i - \boldsymbol{b}_2\}_{i \in [N]}$ are coordinate-wise far from the integer lattice. Moreover, by additional translation to $\{\boldsymbol{z}_i - \boldsymbol{b}_2\}_{i \in [N]}$ (by adding some natural number, coordinate-wise), we can ensure all coordinates are positive while keeping the property Equation (13). Hence, we may assume without loss of generality $\boldsymbol{b}_2$ also has the property

$$\boldsymbol{z}_i - \boldsymbol{b}_2 \geq \boldsymbol{0} \text{ for all } i \in [N]. \tag{14}$$

Let us denote $\mathcal{D}''' = \{(\boldsymbol{z}_i', y_i)\}_{i \in [N]}$, where $\boldsymbol{z}_i' := \boldsymbol{z}_i - \boldsymbol{b}_2$. Then $\epsilon_{\mathcal{D}'''} = \epsilon_{\mathcal{D}''}(\geq \sqrt{m}/2)$. For $\rho''' := \rho'' = \frac{1}{4N\epsilon_{\mathcal{D}''}}$, we have the robustness radius $\mu''' := \rho''' \epsilon_{\mathcal{D}'''} = \rho'' \epsilon_{\mathcal{D}''} = \mu'' = \frac{1}{4N}$. Define $f_{\text{trans}}$ as $f_{\text{trans}}(\boldsymbol{z}) := \boldsymbol{z} - \boldsymbol{b}_2$. Then, $f_{\text{trans}}$ can be implemented via one hidden layer in a neural network, with $O(m^2)$ parameters.

Upon the two layers constructed from stage I and II, it suffices to construct a network that $\rho'''$-robustly memorizes $\mathcal{D}'''$ since the translation preserves separation and ball containment properties. Note that the robustness balls after stage II are not affected when passing the $\sigma$, by Equations (13) and (14).

**Stage III** (Grid Indexing) From Equation (13), each $\boldsymbol{z}_i' \in \mathbb{R}^m$ is at least $\frac{4}{3}\mu'''$ distant away from any lattice hyperplane $H_{z,j} := \{\boldsymbol{z} \in \mathbb{R}^m \mid z_j = z\}$ with any $j \in [m]$ and $z \in \mathbb{Z}$. Thus, each robustness ball of $\mathcal{D}'''$ lies completely within a single integer lattice (or unit grid) of the form $\prod_{j=1}^m [n_j, n_j + 1)$, for some $(n_1, \cdots, n_m) \in \mathbb{Z}^m$.

Moreover, as $\epsilon_{\mathcal{D}'''} \geq \sqrt{m}/2$, for any $i \neq i'$ with $y_i \neq y_{i'}$, we have $\|\boldsymbol{z}_i' - \boldsymbol{z}_{i'}'\|_2 \geq \sqrt{m}$ . Since $\sup\{\|\boldsymbol{z} - \boldsymbol{z}'\|_2 \mid \boldsymbol{z}, \boldsymbol{z}' \in \prod_{j=1}^m [n_j, n_j + 1)\} = \sqrt{m}$, two such points $\boldsymbol{z}_i'$ and $\boldsymbol{z}_{i'}'$ that corresponds to distinct labels cannot lie in the same grid. Since each $\mu'''$-ball lies within a single grid, we conclude that no two $\mu'''$-ball with different labels lie within the same grid.

We define $R := \lceil \max_{i \in [N]} \|\boldsymbol{z}_i'\|_\infty (= \max_{i \in [N], j \in [m]}(z_{i,j}')) \rceil \in \mathbb{N}$. Our goal in this stage is to construct Flatten mapping defined as

$$\text{Flatten}(\boldsymbol{z}) := R^{m-1} \lfloor z_1 \rfloor + R^{m-2} \lfloor z_2 \rfloor + \cdots + \lfloor z_m \rfloor.$$

This maps each grid $\prod_{j=1}^m [n_j, n_{j+1})$ onto the point $\sum_{j=1}^m R^{j-1} n_j$.

However, since Flatten is discontinuous due to the use of floor functions, we construct $\overline{\text{Flatten}}$, which is a continuous approximation that exactly matches Flatten in the region of our interest. By applying Lemma B.16 to $\gamma = \frac{1}{4N}$ and $n = \lceil \log_2 R \rceil$, we obtain the network $\overline{\text{Floor}} := \overline{\text{Floor}}_{\lceil \log_2 R \rceil}$ with $O(\log_2 R)$ parameters such that

$$\overline{\text{Floor}}(z) = \lfloor z \rfloor \quad \forall z \in [0, R] \text{ with } z - \lfloor z \rfloor > \frac{1}{4N}. \tag{15}$$

Moreover, since we apply $\gamma = 1/4N$ to Lemma B.16, the lemma guarantees that $\overline{\text{Floor}}_n$ can be exactly implemented with $O(n + \log N) = O(\log R + \log N)$ bit complexity. In particular, we can define our network $\overline{\text{Flatten}}$ with $O(\log R + \log N + \log R^{m-1}) = O(\log R + \log N + \log N \log R) = \tilde{O}(1)$ bit complexity as

$$\overline{\text{Flatten}}(\boldsymbol{z}) = R^{m-1}\overline{\text{Floor}}(z_1) + \cdots + \overline{\text{Floor}}(z_m). \tag{16}$$

This implementation is valid—i.e. $\overline{\text{Flatten}}(\boldsymbol{z}) = \text{Flatten}(\boldsymbol{z})$—in the region of interest ($\{\boldsymbol{z} \in [0, R]^m \mid \text{dist}_2(z_j, \mathbb{Z}) \geq \frac{1}{2N} \text{ for all } j \in [m]\}$) characterized by the margin guaranteed by Equation (13).

As $\overline{\text{Floor}} : \mathbb{R} \to \mathbb{R}$ can be implemented with width 5 and depth $O(\log_2 R)$ network (Lemma B.16), $\overline{\text{Flatten}}$ can be implemented with width $5m$ and depth $O(\log_2 R)$ network. Thus, we can construct $\overline{\text{Flatten}}$ with $O(m^2 \log_2 R) = \tilde{O}(m^2)$ parameters.

By Equations (13) and (15), we guarantee that each robustness ball lies in the region where the Flatten is properly approximated by $\overline{\text{Flatten}}$. i.e.

$$\overline{\text{Flatten}}(\boldsymbol{z}) = \text{Flatten}(\boldsymbol{z}) \text{ for all } \boldsymbol{z} \in \mathcal{B}_2(\boldsymbol{z}_i', \mu''').$$

Since Flatten maps each unit grid into a point and each robustness ball of $\mathcal{D}'''$ lies on a single unit grid, we conclude

$$\overline{\text{Flatten}}(\boldsymbol{z}) = \text{Flatten}(\boldsymbol{z}) = \text{Flatten}(\boldsymbol{z}_i') \text{ for all } \boldsymbol{z} \in \mathcal{B}_2(\boldsymbol{z}_i', \mu''').$$

Let $m_i := \text{Flatten}(\boldsymbol{z}_i')$. Then each robustness ball around $\boldsymbol{x}_i$ is mapped to $m_i$. We have $m_i \in \mathbb{Z} \cap [0, R^{m+1}]$ for all $i \in [N]$, since

$$\begin{aligned}
m_i &= \text{Flatten}(\boldsymbol{z}_i') \\
&= R^{m-1} \lfloor z_{i1}' \rfloor + R^{m-2} \lfloor z_{i2}' \rfloor + \cdots \lfloor z_{im}' \rfloor
\end{aligned}$$

$$\overset{(a)}{\leq} R^{m-1} R + R^{m-2} R + \cdots + R$$
$$\leq R^{m+1},$$

where (a) is by $\|\boldsymbol{z}_i'\|_\infty \leq R$.

**Stage IV** (Memorization) Finally, it remains to memorize $N$ points $\{(m_i, y_i)\}_{i=1}^N \subset \mathbb{Z}_{\geq 0} \times [C]$. Since multiple robustness balls for $\mathcal{D}'''$ with the same label may correspond to the same grid index in Stage III, it is possible that for some $i \neq j$ with $y_i = y_j$, we have $m_i = m_j$. Let $N' \leq N$ denote the number of distinct pairs $(m_i, y_i)$. It remains to memorize these $N'$ distinct data points in $\mathbb{R}$.

Since $m_i = \text{Flatten}(\boldsymbol{x}_i) \leq R^{m+1}$, we apply Theorem B.4 from Vardi et al. [2021] with $r = R^{m+1}$ to construct $f_{mem} : \mathbb{R} \to \mathbb{R}$ with width 12 and depth

$$\tilde{O}(\sqrt{N'} \cdot \log(5R^m N^2 \epsilon^{-1} \sqrt{\pi m})) = \tilde{O}(m\sqrt{N'}) = \tilde{O}(\log N \sqrt{N'}) = \tilde{O}(\sqrt{N'}) = \tilde{O}(\sqrt{N})$$

such that $f_{mem}(m_i) = y_i$.

The final network $f : \mathbb{R}^d \to \mathbb{R}$ is defined as

$$f(\boldsymbol{x}) = f_{\text{mem}} \circ \sigma \circ \overline{\text{Flatten}} \circ \sigma \circ f_{\text{trans}} \circ \sigma \circ (\bar{f}_{\text{proj}}(\boldsymbol{x}) + \boldsymbol{b}).$$

The depth 1 network $\bar{f}_{\text{proj}}(\boldsymbol{x}) + \boldsymbol{b}$ has width $m$, and also the depth 1 network $f_{\text{trans}}$ has width $m$. $\overline{\text{Flatten}}$ has width $5m$ and depth $O(\log_2 R)$ and $f_{\text{mem}}$ has width 12 and depth $\tilde{O}(\sqrt{N})$. The total construction requires $\tilde{O}(md + m^2 + m^2 + \sqrt{N}) = \tilde{O}(d + \sqrt{N})$ parameters, where each term $md, m^2, m^2$, and $\sqrt{N}$ comes from $f_{\text{proj}}, f_{\text{trans}}, \overline{\text{Flatten}}$, and $f_{\text{mem}}$ respectively. The width of the final network is $\tilde{O}(1)$ and the depth is $\tilde{O}(\sqrt{N})$.

The bit complexity of $\bar{f}_{\text{proj}}$ is $\tilde{O}(1)$ and that of $\boldsymbol{b}$ is

$$O(\lceil \log(6N) \rceil, \log(\max\{\|\bar{f}_{\text{proj}}(\boldsymbol{x})\|_\infty \mid \boldsymbol{x} \in \mathcal{B}_2(\boldsymbol{x}_i, \mu) \quad \text{for some } i \in [N]\})) = \tilde{O}(1).$$

The network $f_{\text{trans}}$ has the bit complexity $\log(\max\{\|\boldsymbol{z}_i\|_\infty \mid i \in [N]\}) = \tilde{O}(1)$. $\overline{\text{Flatten}}$ has the bit complexity $\tilde{O}(1)$, and $f_{\text{mem}}$ needs at most $\tilde{O}(\sqrt{N})$. Hence, the bit complexity of the final network is $\tilde{O}(\sqrt{N})$. $\qquad\square$

The following is the classical memorization upper bound of parameters used in the proof of Theorem B.3

**Theorem B.4** (Classical Memorization, Theorem 3.1 from Vardi et al. [2021]). *Let $N, d, C \in \mathbb{N}$, and $r, \epsilon > 0$, and let $(\boldsymbol{x}_1, y_1), \ldots, (\boldsymbol{x}_N, y_N) \in \mathbb{R}^d \times [C]$ be a set of $N$ labeled samples with $\|\boldsymbol{x}_i\| \leq r$ for every $i$ and $\|\boldsymbol{x}_i - \boldsymbol{x}_j\| \geq 2\epsilon$ for every $i \neq j$. Denote $R := 5rN^2\epsilon^{-1}\sqrt{\pi d}$. Then, there exists a neural network $F : \mathbb{R}^d \to \mathbb{R}$ with width 12 and depth*

$$O\left(\sqrt{N \log N} + \sqrt{\frac{N}{\log N}} \cdot \max\{\log R, \log C\}\right),$$

*and bit complexity bounded by $O(\log d + \sqrt{\frac{N}{\log N}} \cdot \max\{\log R, \log C\})$ such that $F(\boldsymbol{x}_i) = y_i$ for every $i \in [N]$.*

## B.2 Sufficient Condition for Near-Perfect Robust Memorization with Moderate Robustness Radius

**Theorem B.5.** *Let $\rho \in \left(0, \frac{1}{5\sqrt{d}}\right]$, and $\eta \in (0, 1)$. For any dataset $\mathcal{D} \in \boldsymbol{D}_{d,N,C}$, there exists $f$ with $\tilde{O}(Nd^{\frac{1}{4}}\rho^{\frac{1}{2}})$ parameters, depth $\tilde{O}(Nd^{\frac{1}{4}}\rho^{\frac{1}{2}})$, width $\tilde{O}(1)$ and bit complexity $\tilde{O}\left(1/d^{\frac{1}{4}}\rho^{\frac{1}{2}}\right)$ that $\rho$-robustly memorizes $\mathcal{D}$ with error at most $\eta$.*

*Proof.* For given $\rho$, any desired error $\eta$, and $\mathcal{D} = \{(\boldsymbol{x}_i, y_i)\}_{i \in [N]} \in \boldsymbol{D}_{d,N,C}$, we construct a network $f$ that $\rho$-robustly memorizes $\mathcal{D}$ with $\tilde{O}(Nd^{\frac{1}{4}}\rho^{\frac{1}{2}})$ parameters.

**Stage I** (Projection onto log-scale Dimension and Scaling via the First Hidden Layer Weight Matrix).

The first stage closely follows that of Theorem B.3. By Lemma B.20, we obtain an integer $m = \tilde{O}(\log N)$ and a 1-Lipschitz linear map $\phi : \mathbb{R}^d \to \mathbb{R}^m$ such that the projected dataset $\mathcal{D}' := \{(\phi(\boldsymbol{x}_i), y_i)\}_{i \in [N]} \in \boldsymbol{D}_{m,N,C}$ satisfies the separation bound

$$\epsilon'_{\mathcal{D}} \geq \frac{5}{12}\sqrt{\frac{m}{d}}\epsilon_{\mathcal{D}}. \tag{17}$$

We define $f_{\text{proj}} : \mathbb{R}^d \to \mathbb{R}^m$ as $f_{\text{proj}}(\boldsymbol{x}) = \frac{11}{9} \cdot \frac{\sqrt{d}}{\epsilon_{\mathcal{D}}}\phi(\boldsymbol{x})$, which is $\frac{11}{9} \cdot \frac{\sqrt{d}}{\epsilon_{\mathcal{D}}}$-Lipschitz.

We apply Lemma B.23 with $f_{\text{proj}}$ whose depth is 1, $\nu = \min\left\{\frac{109}{11880}\sqrt{m}, \frac{1}{88}\mu, \frac{1}{360N}, 1\right\}$ and $\bar{R} := \max\{\|\boldsymbol{x}\|_2 \mid \boldsymbol{x} \in \mathcal{B}_2(\boldsymbol{x}_i, \mu) \quad \text{for some } i \in [N]\}$ to obtain $\bar{f}_{\text{proj}}$ with the same number of parameters, depth and width and $\tilde{O}(1)$ bit complexity such that

$$\max_{\|\boldsymbol{x}\|_2 \leq \bar{R}}\|\bar{f} - f\|_2 \leq \nu. \tag{18}$$

We set the first hidden layer bias $\boldsymbol{b} \in \mathbb{R}^m$ so that

$$\bar{f}_{\text{proj}}(\boldsymbol{x}) + \boldsymbol{b} \geq \boldsymbol{0} \text{ for all } i \in [N] \text{ and all } \boldsymbol{x} \in \mathcal{B}_2(\boldsymbol{x}_i, \mu), \tag{19}$$

where the comparison between two vectors is element-wise.

We obtain the grouping scale $\alpha \in [0,1]$ here for Stage II—we call $\alpha$ the grouping scale, as we group the points by approximately $N^\alpha$ points per group in Stage II. From the $\rho$ condition, we have $\frac{1}{5\rho\sqrt{d}} \geq 1$. Thus, there exists $\alpha \in [0,1]$ such that satisfies $\lceil N^\alpha \rceil = \lfloor\frac{1}{5\rho\sqrt{d}}\rfloor$. Let us bound the $\rho$ in terms of $\alpha$. Since $\lceil N^\alpha \rceil = \lfloor\frac{1}{5\rho\sqrt{d}}\rfloor \leq \frac{1}{5\rho\sqrt{d}}$, we have

$$\rho \leq \frac{1}{5\lceil N^\alpha \rceil\sqrt{d}} \leq \frac{1}{5\lfloor N^\alpha \rfloor\sqrt{d}}. \tag{20}$$

We claim that for $\mathcal{D}'' := \{(\sigma(\bar{f}_{\text{proj}}(\boldsymbol{x}_i) + \boldsymbol{b}), y_i)\}_{i \in [N]} \in \boldsymbol{D}_{m,N,C}$, we have (i) $\epsilon_{\mathcal{D}''} \geq \sqrt{m}/2$ and (ii) for $\rho'' := \frac{1}{4\lfloor N^\alpha \rfloor\epsilon_{\mathcal{D}''}}$, if $g(\boldsymbol{x}) \in \mathcal{F}_{m,P}$ can $\rho''$-robustly memorize $\mathcal{D}''$, then $g \circ \sigma \circ (\bar{f}_{\text{proj}}(\boldsymbol{x}_i) + \boldsymbol{b})$ can $\rho$-robustly memorize $\mathcal{D}$. For any $i \neq j$ with $y_i \neq y_j$, we have

$$\left\|\sigma(\bar{f}_{\text{proj}}(\boldsymbol{x}_i) + \boldsymbol{b}) - \sigma(\bar{f}_{\text{proj}}(\boldsymbol{x}_j) + \boldsymbol{b})\right\|_2$$
$$\overset{(a)}{=} \left\|(\bar{f}_{\text{proj}}(\boldsymbol{x}_i) + \boldsymbol{b}) - (\bar{f}_{\text{proj}}(\boldsymbol{x}_j) + \boldsymbol{b})\right\|_2$$
$$= \left\|\bar{f}_{\text{proj}}(\boldsymbol{x}_i) - \bar{f}_{\text{proj}}(\boldsymbol{x}_j)\right\|_2$$
$$= \left\|(\bar{f}_{\text{proj}}(\boldsymbol{x}_i) - f_{\text{proj}}(\boldsymbol{x}_i)) - (\bar{f}_{\text{proj}}(\boldsymbol{x}_j) - f_{\text{proj}}(\boldsymbol{x}_j)) + (f_{\text{proj}}(\boldsymbol{x}_i) - f_{\text{proj}}(\boldsymbol{x}_j))\right\|_2.$$

where (a) holds by the construction of $\boldsymbol{b}$ (Equation (19)). For simplicity, we denote

$$\Delta(\boldsymbol{x}_i, \boldsymbol{x}_j) := (\bar{f}_{\text{proj}}(\boldsymbol{x}_i) - f_{\text{proj}}(\boldsymbol{x}_i)) - (\bar{f}_{\text{proj}}(\boldsymbol{x}_j) - f_{\text{proj}}(\boldsymbol{x}_j)).$$

Then, we have

$$\left\|\sigma(\bar{f}_{\text{proj}}(\boldsymbol{x}_i) + \boldsymbol{b}) - \sigma(\bar{f}_{\text{proj}}(\boldsymbol{x}_j) + \boldsymbol{b})\right\|_2^2$$
$$= \|\Delta(\boldsymbol{x}_i, \boldsymbol{x}_j) + (f_{\text{proj}}(\boldsymbol{x}_i) - f_{\text{proj}}(\boldsymbol{x}_j))\|_2^2$$
$$\overset{(a)}{\geq} (\|f_{\text{proj}}(\boldsymbol{x}_i) - f_{\text{proj}}(\boldsymbol{x}_j)\|_2 - \|\Delta(\boldsymbol{x}_i, \boldsymbol{x}_j)\|_2)^2$$
$$= \|f_{\text{proj}}(\boldsymbol{x}_i) - f_{\text{proj}}(\boldsymbol{x}_j)\|_2^2 - 2\|f_{\text{proj}}(\boldsymbol{x}_i) - f_{\text{proj}}(\boldsymbol{x}_j)\|_2\|\Delta(\boldsymbol{x}_i, \boldsymbol{x}_j)\|_2 + \|\Delta(\boldsymbol{x}_i, \boldsymbol{x}_j)\|_2^2$$
$$\geq \|f_{\text{proj}}(\boldsymbol{x}_i) - f_{\text{proj}}(\boldsymbol{x}_j)\|_2^2 - 2\|f_{\text{proj}}(\boldsymbol{x}_i) - f_{\text{proj}}(\boldsymbol{x}_j)\|_2\|\Delta(\boldsymbol{x}_i, \boldsymbol{x}_j)\|_2,$$

where (a) holds from $\|\boldsymbol{a} + \boldsymbol{b}\|_2^2 \geq (\|\boldsymbol{a}\|_2 - \|\boldsymbol{b}\|_2)^2$. By the construction of $\bar{f}$ (Equation (18)),

$$\|\Delta(\boldsymbol{x}_i, \boldsymbol{x}_j)\|_2 \leq \|\bar{f}_{\text{proj}}(\boldsymbol{x}_i) - f_{\text{proj}}(\boldsymbol{x}_i)\|_2 + \|\bar{f}_{\text{proj}}(\boldsymbol{x}_j) - f_{\text{proj}}(\boldsymbol{x}_j)\|_2 \leq 2\nu,$$

so we have

$$\left\|\sigma(\bar{f}_{\text{proj}}(\boldsymbol{x}_i) + \boldsymbol{b}) - \sigma(\bar{f}_{\text{proj}}(\boldsymbol{x}_j) + \boldsymbol{b})\right\|_2^2$$

$$\geq \|f_{\text{proj}}(\boldsymbol{x}_i) - f_{\text{proj}}(\boldsymbol{x}_j)\|_2^2 - 4\nu\|f_{\text{proj}}(\boldsymbol{x}_i) - f_{\text{proj}}(\boldsymbol{x}_j)\|_2. \tag{21}$$

Now we derive

$$
\begin{aligned}
\|f_{\text{proj}}(\boldsymbol{x}_i) - f_{\text{proj}}(\boldsymbol{x}_j)\|_2 &\overset{(a)}{=} \frac{11}{9} \cdot \frac{\sqrt{d}}{\epsilon_{\mathcal{D}}} \|\phi(\boldsymbol{x}_i) - \phi(\boldsymbol{x}_j)\|_2 \\
&\overset{(b)}{\geq} \frac{11}{9} \cdot \frac{\sqrt{d}}{\epsilon_{\mathcal{D}}} \cdot 2\epsilon_{\mathcal{D}'} \\
&\overset{(c)}{\geq} \frac{11}{9} \cdot \frac{\sqrt{d}}{\epsilon_{\mathcal{D}}} \times 2 \cdot \frac{5}{12}\sqrt{\frac{m}{d}}\epsilon_{\mathcal{D}} \\
&= \frac{55}{54}\sqrt{m}, \tag{22}
\end{aligned}
$$

where (a) is by the definition of $f_{\text{proj}}$, (b) is by the definition of $\mathcal{D}'$ and its separation constant, and (c) follows from Equation (17).

Plugging this inequality to Equation (21) gives

$$
\begin{aligned}
\left\|\sigma(\bar{f}_{\text{proj}}(\boldsymbol{x}_i) + \boldsymbol{b}) - \sigma(\bar{f}_{\text{proj}}(\boldsymbol{x}_j) + \boldsymbol{b})\right\|_2^2 &\geq \|f_{\text{proj}}(\boldsymbol{x}_i) - f_{\text{proj}}(\boldsymbol{x}_j)\|_2(\frac{55}{54}\sqrt{m} - 4\nu) \\
&\overset{(a)}{\geq} \|f_{\text{proj}}(\boldsymbol{x}_i) - f_{\text{proj}}(\boldsymbol{x}_j)\|_2(\frac{55}{54}\sqrt{m} - \frac{109}{2970}\sqrt{m}) \\
&= \|f_{\text{proj}}(\boldsymbol{x}_i) - f_{\text{proj}}(\boldsymbol{x}_j)\|_2 \cdot \frac{54}{55}\sqrt{m} \\
&\overset{(b)}{\geq} \frac{55}{54}\sqrt{m} \cdot \frac{54}{55}\sqrt{m} \\
&= m
\end{aligned}
$$

where (a) holds from $\nu \leq \frac{109}{11880}\sqrt{m}$, and (b) holds from Equation (22). This proves the first claim $\epsilon_{\mathcal{D}''} \geq \sqrt{m}/2$. To prove the second claim, let $\mu := \rho\epsilon_{\mathcal{D}}$ and $\mu'' := \rho''\epsilon_{\mathcal{D}''}$. Then,

$$
\begin{aligned}
\sigma(\bar{f}_{\text{proj}}(\mathcal{B}_2(\boldsymbol{x}_i, \mu)) + \boldsymbol{b}) &\overset{(a)}{=} \bar{f}_{\text{proj}}(\mathcal{B}_2(\boldsymbol{x}_i, \mu)) + \boldsymbol{b} \\
&\overset{(b)}{\subseteq} f_{\text{proj}}(\mathcal{B}_2(\boldsymbol{x}_i, \mu + \nu)) + \boldsymbol{b} \\
&\overset{(c)}{\subseteq} f_{\text{proj}}(\mathcal{B}_2(\boldsymbol{x}_i, \frac{89}{88}\mu)) + \boldsymbol{b} \\
&\overset{(d)}{\subseteq} \mathcal{B}_2(f_{\text{proj}}(\boldsymbol{x}_i), \frac{89}{72} \cdot \frac{\sqrt{d}}{\epsilon_{\mathcal{D}}} \times \mu) + \boldsymbol{b} \\
&\overset{(e)}{=} \mathcal{B}_2(f_{\text{proj}}(\boldsymbol{x}_i), \frac{89}{72} \cdot \sqrt{d}\rho) + \boldsymbol{b} \\
&\overset{(f)}{\subseteq} \mathcal{B}_2(f_{\text{proj}}(\boldsymbol{x}_i), \frac{89}{360N}) + \boldsymbol{b} \\
&\overset{(g)}{\subseteq} \mathcal{B}_2(\bar{f}_{\text{proj}}(\boldsymbol{x}_i), \frac{89}{360N} + \nu) + \boldsymbol{b} \\
&\overset{(h)}{\subseteq} \mathcal{B}_2(\bar{f}_{\text{proj}}(\boldsymbol{x}_i), \frac{1}{4N}) + \boldsymbol{b} \\
&\overset{(i)}{=} \mathcal{B}_2(\bar{f}_{\text{proj}}(\boldsymbol{x}_i), \rho''\epsilon_{\mathcal{D}''}) + \boldsymbol{b} \\
&= \mathcal{B}_2(\bar{f}_{\text{proj}}(\boldsymbol{x}_i) + \boldsymbol{b}, \rho''\epsilon_{\mathcal{D}''}) \\
&\overset{(j)}{=} \mathcal{B}_2(\sigma(\bar{f}_{\text{proj}}(\boldsymbol{x}_i) + \boldsymbol{b}), \rho''\epsilon_{\mathcal{D}''}),
\end{aligned}
$$

where (a) and (j) are by Equation (19), (b) and (g) are by the construction of $\bar{f}$ (Equation (18)), (c) is because $\nu \leq \frac{1}{88}\mu$, (d) is because $f_{\text{proj}}$ is $\frac{11}{9} \cdot \frac{\sqrt{d}}{\epsilon_{\mathcal{D}}}$-Lipschitz, (e) uses $\mu = \rho\epsilon_{\mathcal{D}}$, (f) uses $\rho \leq \frac{1}{5N\sqrt{d}}$, (h) is because $\nu \leq \frac{1}{360N}$ and (i) is because $\rho''\epsilon_{\mathcal{D}''} = \frac{1}{4N\epsilon_{\mathcal{D}''}}\epsilon_{\mathcal{D}''} = \frac{1}{4N}$.

Hence, $g(\boldsymbol{x})$ memorizing the robustness ball $\mathcal{B}_2(\sigma(\bar{f}_{\mathrm{proj}}(\boldsymbol{x}_i) + \boldsymbol{b}), \rho''\epsilon_{\mathcal{D}''})$ on projected space leads to $g \circ \sigma \circ (\bar{f}_{\mathrm{proj}}(\boldsymbol{x}) + \boldsymbol{b})$ memorizing the robustness ball for $\mathcal{D}$. In other words, if $g(\boldsymbol{x}) \in \mathcal{F}_{m,P}$ can $\rho''$-robustly memorize $\mathcal{D}''$, then $g \circ \sigma \circ (\bar{f}_{\mathrm{proj}}(\boldsymbol{x}) + \boldsymbol{b})$ can $\rho$-robustly memorize $\mathcal{D}$. With $\rho'' = \frac{1}{4\lfloor N^\alpha \rfloor \epsilon_{\mathcal{D}''}}$, Stage II aims to find a $\rho''$-robust memorizer $g$ of $\mathcal{D}''$. For simplicity of the notation, let $\boldsymbol{z}_i := \sigma(\bar{f}_{\mathrm{proj}}(\boldsymbol{x}_i) + \boldsymbol{b})$ for each $i \in [N]$, so that $\mathcal{D}'' = \{(\boldsymbol{z}_i, y_i)\}_{i\in[N]}$.

**Stage II** (Memorizing $N^\alpha$ Points at Each Layer): Using the grouping scale $\alpha$ obtain in Stage I, we group $N$ data points to $\lceil N^{1-\alpha} \rceil$ groups with index $\{I_j\}_{j=1}^{N^{1-\alpha}}$, each with $|I_j| \leq \lfloor N^\alpha \rfloor + 1$. Then, we construct $\tilde{f}_j$ that memorizes data points and their robustness balls with index $I_j$, and the error rate remains small for other data points and their robustness balls.

For each $j \in [\lceil N^{1-\alpha} \rceil]$, we apply Lemma B.13 with error rate $\eta \leftarrow \frac{\eta}{N^{1-\alpha}}$, $\alpha \leftarrow \alpha$, $\mathcal{D} \leftarrow \mathcal{D}'' \in \boldsymbol{D}_{m,N,C}$, $\rho \leftarrow \rho''$ and $I \leftarrow I_j$. Then it satisfies that $\epsilon_{\mathcal{D}''} \geq \sqrt{m}/2$, $\rho'' = \frac{1}{4\lfloor N^\alpha \rfloor \epsilon_{\mathcal{D}''}}$, and $|I| \leq \lfloor N^\alpha \rfloor + 1$. Thus, we obtain a neural network $\tilde{f}_j$ with width $O(m) = \tilde{O}(1)$, depth $\tilde{O}(N^{\frac{\alpha}{2}})$ and $\tilde{O}\left(N^{\frac{\alpha}{2}} + m^2\right) = \tilde{O}\left(N^{\frac{\alpha}{2}}\right)$ parameters and bit complexity $\tilde{O}\left(N^{\frac{\alpha}{2}} + m\right) = \tilde{O}\left(N^{\frac{\alpha}{2}}\right)$ such that:

$$\tilde{f}_j(\boldsymbol{z}) = y_i \qquad\qquad \forall i \in I_j, \boldsymbol{z} \in \mathcal{B}(\boldsymbol{z}_i, \rho''\epsilon_{\mathcal{D}''}),$$

$$\mathbb{P}_{\boldsymbol{z}\in\mathrm{Unif}(\mathcal{B}(\boldsymbol{z}_i,\rho''\epsilon_{\mathcal{D}''}))}\left[\tilde{f}_j(\boldsymbol{z}) \in \{0, y_i\}\right] \geq 1 - \frac{\eta}{N^{1-\alpha}} \qquad \forall i \in [N]\backslash I_j.$$

Thus, we have

$$\mathbb{P}_{\boldsymbol{z}\in\mathrm{Unif}(\mathcal{B}(\boldsymbol{z}_i,\rho''\epsilon_{\mathcal{D}''}))}\left[\tilde{f}_j(\boldsymbol{z}) \in \{0, y_i\}\right] \geq 1 - \frac{\eta}{N^{1-\alpha}} \quad \forall i \in [N], j \in [N^{1-\alpha}]. \qquad (23)$$

We define for each $j$:

$$f_j\left(\begin{pmatrix} \boldsymbol{z} \\ y \end{pmatrix}\right) = \begin{pmatrix} \boldsymbol{z} \\ y + \sigma\left(\tilde{f}_j(\boldsymbol{z}) - y\right) \end{pmatrix},$$

so that the last coordinate is given as $y + \sigma\left(\tilde{f}_j(\boldsymbol{z}) - y\right) = \max\{\tilde{f}_j(\boldsymbol{z}), y\}$. Finally, we define the full robust memorizing network as

$$f(\boldsymbol{x}) := \begin{pmatrix} \boldsymbol{0} \\ 1 \end{pmatrix}^\top f_{\lceil N^{1-\alpha} \rceil} \circ \cdots \circ f_2 \circ f_1\left(\begin{matrix} \bar{f}_{\mathrm{proj}}(\boldsymbol{x}) + \boldsymbol{b} \\ 0 \end{matrix}\right).$$

We now verify the correctness of the construction. For any $\boldsymbol{x} \in \mathcal{B}(\boldsymbol{x}_i, \rho\epsilon_{\mathcal{D}})$ and $\boldsymbol{z} = f_{\mathrm{proj}}(\boldsymbol{x}) \in \mathcal{B}(\boldsymbol{z}_i, \rho''\epsilon_{\mathcal{D}''})$, since we partition $[N]$ into disjoint groups $\{I_j\}_{j\in[N^{1-\alpha}]}$, there exists a unique index $j_i$ such that $i \in I_{j_i}$ and thus $\tilde{f}_{j_i}(\boldsymbol{z}) = y_i$ holds. For all $j \neq j_i$, the networks satisfy $\tilde{f}_j(\boldsymbol{z}) \in \{0, y_i\}$ with high probability, so none of them can exceed $y_i$. Since the final network outputs the maximum among $y$ and all $\tilde{f}_j(\boldsymbol{z})$, we have $f(\boldsymbol{x}) = y_i$ as long as each $\tilde{f}_j(\boldsymbol{z}) \in \{0, y_i\}$. Therefore, it suffices to show that $\tilde{f}_j(\boldsymbol{z}) \in \{0, y_i\}$ holds for $\forall j$, namely,

$$\left[\tilde{f}_j(\boldsymbol{z}) \in \{0, y_i\} \quad \forall j \in [N^{1-\alpha}]\right] \Rightarrow f(\boldsymbol{x}) = y_i.$$

Considering the contrapositive, we have

$$f(\boldsymbol{x}) \neq y_i \Rightarrow \left[\tilde{f}_j(\boldsymbol{z}) \notin \{0, y_i\} \quad \text{for some } j \in [N^{1-\alpha}]\right]$$

Since each $\tilde{f}_j$ satisfies $\mathbb{P}_{\boldsymbol{z}\sim\mathrm{Unif}(\mathcal{B}(\boldsymbol{z}_i,\rho''\epsilon_{\mathcal{D}''}))}[\tilde{f}_j(\boldsymbol{z}) \in \{0, y_i\}] \geq 1 - \frac{\eta}{N^{1-\alpha}}$ for all $j \in [N^{1-\alpha}]$, we upper bound the error probability using the union bound:

$$\mathbb{P}_{\boldsymbol{x}\in\mathrm{Unif}(\mathcal{B}(\boldsymbol{x}_i,\rho\epsilon_{\mathcal{D}}))}\left[f(\boldsymbol{x}) \neq y_i\right] \leq \mathbb{P}_{\boldsymbol{z}\in\mathrm{Unif}(\mathcal{B}(\boldsymbol{z}_i,\rho''\epsilon_{\mathcal{D}''}))}\left[\tilde{f}_j(\boldsymbol{z}) \notin \{0, y_i\} \quad \text{for some } j \in [N^{1-\alpha}]\right]$$

$$\leq \sum_{j\in[N^{1-\alpha}]} \mathbb{P}_{\boldsymbol{z}\in\mathrm{Unif}(\mathcal{B}(\boldsymbol{z}_i,\rho''\epsilon_{\mathcal{D}''}))}\left[\tilde{f}_j(\boldsymbol{z}) \notin \{0, y_i\}\right]$$

$$\overset{(a)}{\leq} N^{1-\alpha} \times \left(\frac{\eta}{N^{1-\alpha}}\right)$$

$$\leq \eta,$$

where (a) holds by Equation (23). Hence,

$$\mathbb{P}_{\boldsymbol{x} \in \mathrm{Unif}(\mathcal{B}(\boldsymbol{x}_i, \rho \epsilon_{\mathcal{D}}))} [f(\boldsymbol{x}) = y_i] = 1 - \mathbb{P}_{\boldsymbol{x} \in \mathrm{Unif}(\mathcal{B}(\boldsymbol{x}_i, \rho \epsilon_{\mathcal{D}}))} [f(\boldsymbol{x}) \neq y_i]$$
$$\geq 1 - \eta.$$

We verify width, depth and the number of parameters of $f$. Recall that the final network is

$$f(\boldsymbol{x}) := \begin{pmatrix} \boldsymbol{0} \\ 1 \end{pmatrix}^\top f_{\lceil N^{1-\alpha} \rceil} \circ \cdots \circ f_2 \circ f_1 \left( \begin{matrix} \bar{f}_{\mathrm{proj}}(\boldsymbol{x}) + \boldsymbol{b} \\ 0 \end{matrix} \right).$$

The depth 1 network $\bar{f}_{\mathrm{proj}}(\boldsymbol{x}) + \boldsymbol{b}$ has width $m = \tilde{O}(1)$. For $j \in [\lceil N^{1-\alpha} \rceil]$, each $\tilde{f}_j$ has width $\tilde{O}(1)$, depth $\tilde{O}(N^{\frac{\alpha}{2}})$ and $\tilde{O}\left(N^{\frac{\alpha}{2}}\right)$ parameters.

The network $\bar{f}_{\mathrm{proj}}$ needs $dm = \tilde{O}(d)$ parameters. Hence, the number of parameters of $f$ is

$$\tilde{O}\left(N^{1-\alpha} \times N^{\frac{\alpha}{2}}\right) = \tilde{O}\left(N^{1-\frac{\alpha}{2}}\right) \overset{(a)}{=} \tilde{O}\left(Nd^{\frac{1}{4}}\rho^{\frac{1}{2}}\right),$$

where (a) holds by $\lceil N^\alpha \rceil = \lfloor \frac{1}{5\rho\sqrt{d}} \rfloor$. The width of $f$ is $\tilde{O}(1)$ and the depth of $f$ is $\tilde{O}\left(N^{1-\alpha} \times N^{\frac{\alpha}{2}}\right) = \tilde{O}\left(Nd^{\frac{1}{4}}\rho^{\frac{1}{2}}\right)$.

The bit complexity of $\bar{f}_{\mathrm{proj}}$ is $\tilde{O}(1)$ and that of $\boldsymbol{b}$ is $\log(\max\{\|\bar{f}_{\mathrm{proj}}(\boldsymbol{x})\|_\infty \mid \boldsymbol{x} \in \mathcal{B}_2(\boldsymbol{x}_i, \mu) \text{ for some } i \in [N]\}) = \tilde{O}(1)$. The network $f_j$ has the same bit complexity as $\tilde{f}_j$, which is $\tilde{O}\left(N^{\frac{\alpha}{2}}\right) = \tilde{O}\left(1/d^{\frac{1}{4}}\rho^{\frac{1}{2}}\right)$. Hence, the bit complexity of the final network is $\tilde{O}\left(1/d^{\frac{1}{4}}\rho^{\frac{1}{2}}\right)$.

$\square$

The above construction is motivated by the need to handle overlapped robustness balls with the same label. We transform the construction of classical memorization in Vardi et al. [2021] in two key directions: first, from memorizing isolated data points $\boldsymbol{x}_i$ to memorizing entire robustness neighborhoods $\mathcal{B}_p(\boldsymbol{x}_i, \mu)$; and second, to ensuring correct classification even within regions where multiple robustness balls with the same label overlap. To accomplish this, we introduce disjoint, integer-aligned interval encodings and carefully control the error propagation caused by dimension reduction, as addressed in Lemma B.11.

### B.2.1  Memorization of Integers with Sublinear Parameters in $N$

Lemmas in this subsection are a slight extension of those in Vardi et al. [2021], adapted to our integer-based encoding scheme.

From here, $\mathrm{BIN}_{i:j}(n)$ denotes the bit string from position $i$ to $j$ (inclusive) in the binary representation of $n$. For example, $\mathrm{BIN}_{1:3}(37) = 4$, since $(37)_{10} = (100101)_2$ so that $\mathrm{BIN}_{1:3}(37) = (100)_2 = (4)_{10}$.

**Lemma B.6.** *Let $\eta > 0$ and $m, n \in \mathbb{N}$ with $m < n$. Then, there exists a neural network $F : \mathbb{R} \to \mathbb{R}$ with width 2, depth 2 and bit complexity $\tilde{O}(1)$ such that*

$$F(x) = \begin{cases} 1 & \text{for } x \in [m, n - \eta], \\ 0 & \text{for } x \in (-\infty, m - \eta] \cup [n, \infty). \end{cases}$$

*Proof.* We construct a network $F$:

$$F(x) = \sigma\left(1 - \sigma\left(-\frac{1}{\eta}(x - m)\right)\right) + \sigma\left(1 - \sigma\left(\frac{1}{\eta}(x - (n - \eta))\right)\right) - 1.$$

It satisfies the requirements with depth 2 and width 2. The bit complexity is $O(\log m + \log n + \log(1/\eta)) = \tilde{O}(1)$.

$\square$

**Lemma B.7.** *Let $\eta \in (0,1)$, and let $m_1 < \cdots < m_N$ be natural numbers. Let $N_1, N_2 \in \mathbb{N}$ satisfy $N_1 \cdot N_2 \geq N$, and let $w_1, \ldots, w_{N_1} \in \mathbb{N}$. Then, there exists a neural network $F : \mathbb{R} \to \mathbb{R}$ with width 4, depth $3N_1 + 2$ and bit complexity $\tilde{O}(1)$ such that,*

$$F(x) = \begin{cases} w_{\lceil \frac{i}{N_2} \rceil} & \text{if } x \in [m_i, m_i + 1 - \eta] \text{ for some } i \in [N], \\ 0 & \text{if } x \notin \bigcup_{j \in [N_1]} (m_{(j-1)N_2+1} - \eta, \ m_{jN_2} + 1). \end{cases}$$

*where we define $m_{N+1} = \cdots = m_{N_1 N_2} = m_N$.*

*Proof.* Let $j \in [N_1]$. We define network blocks $\tilde{F}_j : \mathbb{R} \to \mathbb{R}$ and $F_j : \mathbb{R}^2 \to \mathbb{R}^2$ as follows. By applying Lemma B.6, we construct $\tilde{F}_j$ such that:

$$\tilde{F}_j(x) = \begin{cases} 1 & \text{if } x \in \left[ m_{(j-1)N_2+1}, \ m_{jN_2} + 1 - \eta \right], \\ 0 & \text{if } x \leq m_{(j-1)N_2+1} - \eta \text{ or } x \geq m_{jN_2} + 1. \end{cases}$$

As a result, for any $i \in [N]$, any $x \in [m_i, m_i + 1 - \eta]$ satisfies

$$\tilde{F}_i(x) = \begin{cases} 1 & \text{if } i \in [(j-1) \cdot N_2 + 1, j \cdot N_2], \\ 0 & \text{otherwise.} \end{cases}$$

Next, we define:

$$F_j \left( \begin{pmatrix} x \\ y \end{pmatrix} \right) = \begin{pmatrix} x \\ y + w_j \cdot \tilde{F}_j(x) \end{pmatrix}.$$

Finally, we define the network $F(x) = \begin{pmatrix} 0 \\ 1 \end{pmatrix}^{\top} F_{N_1} \circ \cdots \circ F_1 \left( \begin{pmatrix} x \\ 0 \end{pmatrix} \right).$

We now verify the correctness of the construction. For $i \in [N]$, let $x \in [m_i, m_i + 1 - \eta]$. For $j = \lceil \frac{i}{N_2} \rceil$, we have $\tilde{F}_j(x) = 1$, and for all $j' \neq j$, $\tilde{F}_{j'}(x) = 0$. Therefore, the output of $F$ satisfies $F(x) = w_j = w_{\lceil \frac{i}{N_2} \rceil}$.

The width of each $F_j$ is at most the width required to implement $\tilde{F}_j$, plus two additional units to carry the values of $x$ and $y$. Since the width of $\tilde{F}_j$ is 2, the width of $F$ is at most 4. Each block $F_j$ has depth 3, and $F$ is a composition of $N_1$ blocks. Additionally, one layer is used for the input to get $x \mapsto \begin{pmatrix} x \\ 0 \end{pmatrix}$, and another to extract the last coordinate of the final input. Thus, the total depth of $F$ is $3N_1 + 2$. The bit complexity is $\tilde{O}(1)$. $\qquad\qquad\square$

**Lemma B.8** (Lemma A.7, Vardi et al. [2021])**.** *Let $n \in \mathbb{N}$ and let $i, j \in \mathbb{N}$ with $i < j \leq n$. Denote Telgarsky's triangle function by $\varphi(z) := \sigma(\sigma(2z) - \sigma(4z - 2))$. Then, there exists a neural network $F : \mathbb{R}^2 \to \mathbb{R}^3$ with width 5 and depth $3(j - i + 1)$, and bit complexity $n + 2$, such that for any $x \in \mathbb{N}$ with $x \leq 2^n$, if the input of $F$ is $\begin{pmatrix} \varphi^{(i-1)} \left( \frac{x}{2^n} + \frac{1}{2^{n+1}} \right) \\ \varphi^{(i-1)} \left( \frac{x}{2^n} + \frac{1}{2^{n+2}} \right) \end{pmatrix}$, then it outputs: $\begin{pmatrix} \varphi^{(j)} \left( \frac{x}{2^n} + \frac{1}{2^{n+1}} \right) \\ \varphi^{(j)} \left( \frac{x}{2^n} + \frac{1}{2^{n+2}} \right) \\ \mathrm{BIN}_{i:j}(x) \end{pmatrix}.$*

In the following lemma, note that $\rho$ does not refer to the robustness ratio.

**Lemma B.9** (Extension of Lemma A.5, Vardi et al. [2021])**.** *Let $\eta > 0$, and let $n, \rho, c \in \mathbb{N}$ and $u, w \in \mathbb{N}$. Assume that for all $\ell, k \in \{0, 1, \ldots, n - 1\}$ with $\ell \neq k$, the bit segments of $u$ satisfy*

$$\mathrm{BIN}_{\rho \cdot \ell + 1 : \rho \cdot (\ell + 1)}(u) \neq \mathrm{BIN}_{\rho \cdot k + 1 : \rho \cdot (k + 1)}(u).$$

*Then, there exists a neural network $F : \mathbb{R}^3 \to \mathbb{R}$ with width 12, depth $3n \cdot \max\{\rho, c\} + 2n + 2$ and bit complexity $n \max\{\rho, c\} + 2$, such that the following holds:*

*For every $x > 0$, if there exist $j \in \{0, 1, \ldots, n - 1\}$ such that*

$$x \in [\mathrm{BIN}_{\rho \cdot j + 1 : \rho \cdot (j + 1)}(u), \mathrm{BIN}_{\rho \cdot j + 1 : \rho \cdot (j + 1)}(u) + 1 - \eta],$$

*then the network satisfies*

$$F \left( \begin{pmatrix} x \\ w \\ u \end{pmatrix} \right) = \mathrm{BIN}_{c \cdot j + 1 : c \cdot (j + 1)}(w).$$

*Moreover,* $F\left(\begin{pmatrix} x \\ w \\ u \end{pmatrix}\right) = 0$ *for*

$$x \in \mathbb{R} \setminus \bigcup_{j \in \{0, \cdots, n-1\}} (\text{BIN}_{\rho \cdot j + 1 : \rho \cdot (j+1)}(u) - \eta, \text{BIN}_{\rho \cdot j + 1 : \rho \cdot (j+1)}(u) + 1).$$

*Proof.* We define the triangle function $\varphi(z) := \sigma(\sigma(2z) - \sigma(4z - 2))$ as introduced by Telgarsky [2016]. For $i \in \{0, 1, \ldots, n-1\}$, we construct a network block $F_i$:

$$F_i : \begin{pmatrix} x \\ \varphi^{(i \cdot \rho)}\left(\frac{u}{2^{n \cdot \rho}} + \frac{1}{2^{n \cdot \rho + 1}}\right) \\ \varphi^{(i \cdot \rho)}\left(\frac{u}{2^{n \cdot \rho}} + \frac{1}{2^{n \cdot \rho + 2}}\right) \\ \varphi^{(i \cdot c)}\left(\frac{w}{2^{n \cdot c}} + \frac{1}{2^{n \cdot c + 1}}\right) \\ \varphi^{(i \cdot c)}\left(\frac{w}{2^{n \cdot c}} + \frac{1}{2^{n \cdot c + 2}}\right) \\ y \end{pmatrix} \mapsto \begin{pmatrix} x \\ \varphi^{((i+1) \cdot \rho)}\left(\frac{u}{2^{n \cdot \rho}} + \frac{1}{2^{n \cdot \rho + 1}}\right) \\ \varphi^{((i+1) \cdot \rho)}\left(\frac{u}{2^{n \cdot \rho}} + \frac{1}{2^{n \cdot \rho + 2}}\right) \\ \varphi^{((i+1) \cdot c)}\left(\frac{w}{2^{n \cdot c}} + \frac{1}{2^{n \cdot c + 1}}\right) \\ \varphi^{((i+1) \cdot c)}\left(\frac{w}{2^{n \cdot c}} + \frac{1}{2^{n \cdot c + 2}}\right) \\ y + y_i \end{pmatrix},$$

where

$$y_i = \begin{cases} \text{BIN}_{i \cdot c + 1 : (i+1) \cdot c}(w) & \text{if } x \in [\text{BIN}_{i \cdot \rho + 1 : (i+1) \cdot \rho}(u), \text{BIN}_{i \cdot \rho + 1 : (i+1) \cdot \rho}(u) + 1 - \eta], \\ 0 & \text{if } x \leq \text{BIN}_{i \cdot \rho + 1 : (i+1) \cdot \rho}(u) - \eta \text{ or } x \geq \text{BIN}_{i \cdot \rho + 1 : (i+1) \cdot \rho}(u) + 1. \end{cases}$$

To compute $y_i$, we first extract the relevant bit segments from $u$ and $w$ using Lemma B.8. We define two subnetworks $F_i^w, F_i^u$:

$$F_i^u : \begin{pmatrix} \varphi^{(i \cdot \rho)}\left(\frac{u}{2^{n \cdot \rho}} + \frac{1}{2^{n \cdot \rho + 1}}\right) \\ \varphi^{(i \cdot \rho)}\left(\frac{u}{2^{n \cdot \rho}} + \frac{1}{2^{n \cdot \rho + 2}}\right) \end{pmatrix} \mapsto \begin{pmatrix} \varphi^{((i+1) \cdot \rho)}\left(\frac{u}{2^{n \cdot \rho}} + \frac{1}{2^{n \cdot \rho + 1}}\right) \\ \varphi^{((i+1) \cdot \rho)}\left(\frac{u}{2^{n \cdot \rho}} + \frac{1}{2^{n \cdot \rho + 2}}\right) \\ \text{BIN}_{i \cdot \rho + 1 : (i+1) \cdot \rho}(u) \end{pmatrix}$$

$$F_i^w : \begin{pmatrix} \varphi^{(i \cdot c)}\left(\frac{w}{2^{n \cdot c}} + \frac{1}{2^{n \cdot c + 1}}\right) \\ \varphi^{(i \cdot c)}\left(\frac{w}{2^{n \cdot c}} + \frac{1}{2^{n \cdot c + 2}}\right) \end{pmatrix} \mapsto \begin{pmatrix} \varphi^{((i+1) \cdot c)}\left(\frac{w}{2^{n \cdot c}} + \frac{1}{2^{n \cdot c + 1}}\right) \\ \varphi^{((i+1) \cdot c)}\left(\frac{w}{2^{n \cdot c}} + \frac{1}{2^{n \cdot c + 2}}\right) \\ \text{BIN}_{i \cdot c + 1 : (i+1) \cdot c}(w) \end{pmatrix}.$$

A subnetwork $F_i^u$ maps the pair of triangle encodings of $u$ to the updated encodings for $i + 1$, along with the extracted bits $\text{BIN}_{i \cdot \rho + 1 : (i+1) \cdot \rho}(u)$. A subnetwork $F_i^w$ does the same for $w$, yielding $\text{BIN}_{i \cdot c + 1 : (i+1) \cdot c}(w)$.

We then construct a network with width 2 and depth 2 to obtain $y_i$ from inputs $\text{BIN}_{i \cdot \rho + 1 : (i+1) \cdot \rho}(u)$ and $x$. Firstly, we use Lemma B.6 to construct a network that output $\tilde{y}_i$:

$$\tilde{y}_i = \begin{cases} 1 & \text{if } x \in [\text{BIN}_{i \cdot \rho + 1 : (i+1) \cdot \rho}(u), \text{BIN}_{i \cdot \rho + 1 : (i+1) \cdot \rho}(u) + 1 - \eta], \\ 0 & \text{if } x \leq \text{BIN}_{i \cdot \rho + 1 : (i+1) \cdot \rho}(u) - \eta \text{ or } x \geq \text{BIN}_{i \cdot \rho + 1 : (i+1) \cdot \rho}(u) + 1. \end{cases}$$

Secondly, we construct the following 1-layer network that use $\tilde{y}_i$ as input:

$$\begin{pmatrix} \tilde{y}_i \\ \text{BIN}_{i \cdot c + 1 : (i+1) \cdot c}(w) \end{pmatrix} \mapsto \sigma\left(\tilde{y}_i \cdot 2^{c+1} - 2^{c+1} + \text{BIN}_{i \cdot c + 1 : (i+1) \cdot c}(w)\right).$$

This ensures that the output is $\text{BIN}_{i \cdot c + 1 : (i+1) \cdot c}(w)$ if $\tilde{y}_i = 1$, and the output is 0 if $\tilde{y}_i = 0$ since $\text{BIN}_{i \cdot c + 1 : (i+1) \cdot c}(w) \leq 2^c$.

Finally, the full network $F$ is constructed as a composition:

$$F := G \circ F_{n-1} \circ \cdots \circ F_0 \circ H,$$

where for $x, w, u > 0$:

(1) $H : \mathbb{R}^3 \to \mathbb{R}^6$ is a 1-layer network that maps $(x, w, u)$ to the required initial encoding inputs, namely:

$$H : \begin{pmatrix} x \\ w \\ u \end{pmatrix} \mapsto \begin{pmatrix} x \\ \frac{u}{2^{n \cdot \rho}} + \frac{1}{2^{n \cdot \rho + 1}} \\ \frac{u}{2^{n \cdot \rho}} + \frac{1}{2^{n \cdot \rho + 2}} \\ \frac{w}{2^{n \cdot c}} + \frac{1}{2^{n \cdot c + 1}} \\ \frac{w}{2^{n \cdot c}} + \frac{1}{2^{n \cdot c + 2}} \\ 0 \end{pmatrix},$$

(2) $G : \mathbb{R}^6 \to \mathbb{R}$ is a 1-layer network that outputs the last coordinate.

We verify the correctness of the construction. The output of the full network is given by:

$$F\left(\begin{pmatrix} x \\ w \\ u \end{pmatrix}\right) = \sum_{i=0}^{n-1} y_i.$$

If there exists $j \in \{0, 1 \ldots, n-1\}$ such that $x \in [\text{BIN}_{\rho \cdot j+1:\rho \cdot (j+1)}(u), \text{BIN}_{\rho \cdot j+1:\rho \cdot (j+1)}(u) + 1 - \eta]$, then by the construction we obtain $y_j = \text{BIN}_{c \cdot j+1:c \cdot (j+1)}(w)$, while $y_\ell = 0$ for all $\ell \neq j$. This is because the bit-encoded intervals are disjoint as $\text{BIN}_{\rho \cdot \ell+1:\rho \cdot (\ell+1)}(u) \neq \text{BIN}_{\rho \cdot k+1:\rho \cdot (k+1)}(u)$. Hence, the final output of $F$ is:

$$\sum_{i=0}^{n-1} y_i = y_j = \text{BIN}_{c \cdot j+1:c \cdot (j+1)}(w).$$

We now analyze the width and depth of the constructed network $F$. Each block $F_i$ comprises $F_i^w$ and $F_i^u$, each of width 5. In addition, two neurons are used to process $x$ and $y$, resulting in a total width of 12. The outputs $\tilde{y}_i$ and $y_i$ are produced by additional layers with width 2 and 1, respectively, both of which are smaller than 12. We also compose the networks $H$ and $G$, with width 6 and 1, respectively, again remaining within 12.

Each of the networks $F_i^u$ and $F_i^w$ has depth at most $3 \max\{\rho, c\}$. The layers obtaining $\tilde{y}_i$ and $y_i$ contribute an additional 2 layers, resulting in a total depth of $3 \max\{\rho, c\} + 2$ for each block $F_i$. Composing all $n$ such blocks, and including one additional layer each for $H$ and $G$, the total depth of the network $F$ is $3n \cdot \max\{\rho, c\} + 2n + 2$.

The bit complexity of $F_i^u$, $F_i^w$ and $H$ is bounded by $n \max\{\rho, c\} + 2$, and all other parts of the network require less bit complexity. Hence, the bit complexity of $F$ is bounded by $n \max\{\rho, c\} + 2$. □

### B.2.2 Precise Control of Robust Memorization Error

Lemma B.13 constructs the network for Stage II in Theorem B.5, while the robust memorization error is controlled in Lemma B.11.

**Lemma B.10.** *Let $N, C \in \mathbb{N}$, and let $(m_1, y_1), \ldots, (m_N, y_N) \in \boldsymbol{D}_{1,N,C} \subset \mathbb{N} \times [C]$ be a set of $N$ labeled samples with $m_i \neq m_j$ for every $i \neq j$. Then, there exists a neural network $F : \mathbb{R} \to \mathbb{R}$ with width 12, depth $\tilde{O}(\sqrt{N})$, $\tilde{O}(\sqrt{N})$ parameters and bit complexity $\tilde{O}(\sqrt{N})$ such that*

$$F(m) = \begin{cases} y_i & \text{for every } m = m_i \text{ with } i \in [N], \\ 0 & \text{for every } m \in \mathbb{N} \setminus \{m_i\}_{i \in [N]}. \end{cases}$$

*Proof.* Let $\mathcal{M} = \{m_i\}_{i \in [N]}$. We group the elements in $\mathcal{M}$ to $\lceil \sqrt{N} \rceil$ groups, each containing at most $\lfloor \sqrt{N} \rfloor + 1$ natural numbers inside. For each interval indexed by $j \in \{1, \ldots, \lceil \sqrt{N} \rceil\}$, we define two integers $w_j, u_j \in \mathbb{N}$ to encode the integer $m_i \in \mathcal{M}$ and the corresponding labels $y_i$ as follows.

For each $i \in [N]$, letting $j := \left\lceil \frac{i}{\lfloor \sqrt{N} \rfloor + 1} \right\rceil$, $k := i \mod (\lfloor \sqrt{N} \rfloor + 1)$ and $R := max_{i \in [N]} m_i$, we define:

$$\text{BIN}_{k \cdot \log_2 R + 1:(k+1) \cdot \log_2 R}(u_j) = m_i$$
$$\text{BIN}_{k \cdot \log_2 C + 1:(k+1) \cdot \log_2 C}(w_j) = y_i.$$

Thus, in each group $j$, the integer $u_j$ contains $\log_2 R$ bits per integer, which represent the $k$-th integer in this group. In the same manner, $w_j$ contains $\log_2 C$ bits per integer, which represent the label of the $k$-th integer in this group.

By applying Lemma B.7 to $\eta = \frac{1}{2}$, we construct a neural network $F_1$ that maps $m \in \mathcal{M}$ to their corresponding groups, and maps $m \in \mathbb{N} \setminus \bigcup_{j \in [\lceil \sqrt{N} \rceil]} [m_{(j-1)(\lfloor \sqrt{N} \rfloor+1)+1}, \ m_{j(\lfloor \sqrt{N} \rfloor+1)} + 1)$ to 0. Thus, all natural numbers are assigned to their corresponding group or 0.

For each $i \in [N]$, we define the group index

$$j_i := \left\lceil \frac{i}{\lfloor \sqrt{N} \rfloor + 1} \right\rceil.$$

Then, the network $F_1$ maps any input $m \in \mathcal{M}$ to the representation

$$F_1(m) = \begin{pmatrix} m \\ w_{j_i} \\ u_{j_i} \end{pmatrix},$$

and $F_1(m) = \begin{pmatrix} m \\ 0 \\ 0 \end{pmatrix}$ for $m \in \mathbb{N} \backslash \bigcup_{j \in [\lceil \sqrt{N} \rceil]} [m_{(j-1)(\lfloor \sqrt{N} \rfloor + 1) + 1},\ m_{j(\lfloor \sqrt{N} \rfloor + 1)} + 1)$. The network $F_1$ has width 9, depth $\tilde{O}(\sqrt{N})$ and bit complexity $\tilde{O}(1)$.

Now, we apply Lemma B.9 to construct a network $F_2 : \mathbb{R}^3 \to \mathbb{R}$ with the following property. For each $i \in [N], j \in \left[ \lceil \sqrt{N} \rceil \right]$, and $k \in \left\{ 0, \ldots, \lfloor \sqrt{N} \rfloor \right\}$, suppose that $m_i$ is the $k$-th integer in the $j$-th group. Then, the network satisfies :

$$F_2\left( \begin{pmatrix} m_i \\ w_j \\ u_j \end{pmatrix} \right) = \mathrm{BIN}_{k \cdot \log_2 C + 1 : (k+1) \cdot \log_2 C}(w_j) = y_i.$$

Moreover, for $m \in \mathbb{N} \setminus \mathcal{M}$, $F_2\left( \begin{pmatrix} m \\ w_j \\ u_j \end{pmatrix} \right) = 0$ or $F_2\left( \begin{pmatrix} m \\ 0 \\ 0 \end{pmatrix} \right) = 0$. Thus, the network $F_2$ extracts the label corresponding to each data point from the encoded label set of the group to which the interval belongs or outputs 0. The network $F_2$ has width 12, depth $\tilde{O}(\sqrt{N})$ and bit complexity $\tilde{O}(\sqrt{N})$.

Finally, we define the classifier network $F : \mathbb{R}^d \to \mathbb{R}$ as

$$F(\boldsymbol{x}) = F_2 \circ F_1(\boldsymbol{x}).$$

The overall network $F$ has width 12 and depth $\tilde{O}(\sqrt{N})$, which corresponds to the maximum width and total depth of its component networks. The bit complexity of $F$ is $\tilde{O}(\sqrt{N})$. $\qquad\square$

**Lemma B.11.** *Let $\mathcal{B}_2(\boldsymbol{x}_0, \mu)$ be a Euclidean ball with center $\boldsymbol{x}_0 \in \mathbb{R}^d$ and radius $\mu > 0$. Let $\boldsymbol{u} \in \mathbb{R}^d$ be a unit vector, and define the affine function $f(\boldsymbol{x}) := \frac{1}{2\mu}(\boldsymbol{u}^\top \boldsymbol{x} + b)$ for some $b \in \mathbb{R}$. Then for any interval $I \subset \mathbb{R}$ of length $\eta$, the volume fraction of the ball mapped into $I$ satisfies:*

$$\frac{\mathrm{Vol}\left( \{ x \in \mathcal{B}_2(x_0, \mu) \mid f(x) \in I \} \right)}{\mathrm{Vol}\left( \mathcal{B}_2(x_0, \mu) \right)} \leq 2\eta \frac{V_{d-1}}{V_d},$$

*where $V_d = \frac{\pi^{d/2}}{\Gamma(\frac{d}{2}+1)}$ denotes the volume of the $d$-dimensional unit ball.*

*Proof.* Let $\boldsymbol{x} = \boldsymbol{x}_0 + \mu \boldsymbol{y}$, so that $\boldsymbol{y} \in \mathcal{B}_d(\boldsymbol{0}, 1)$. Under this change of variables,

$$f(\boldsymbol{x}) = \frac{1}{2\mu}(\boldsymbol{u}^\top(\boldsymbol{x}_0 + \mu \boldsymbol{y}) + b) = \frac{1}{2}(\boldsymbol{u}^\top \boldsymbol{y}) + \frac{1}{2\mu}(\boldsymbol{u}^\top \boldsymbol{x}_0 + b).$$

Thus, $f(\boldsymbol{x}) \in I$ if and only if $\boldsymbol{u}^\top \boldsymbol{y} \in J$, where

$$J := 2I - \frac{1}{\mu}(\boldsymbol{u}^\top \boldsymbol{x}_0 + b) \subset \mathbb{R}$$

is an interval of length $2\eta$. We define the preimage of $I$ under $f$ with the intersection of $\mathcal{B}_2(\boldsymbol{x}_0, \mu)$ as

$$A := \{ x \in \mathcal{B}_2(\boldsymbol{x}_0, \mu) \mid f(\boldsymbol{x}) \in I \}.$$

Then,

$$\mathrm{Vol}(A) = \mu^d \cdot \mathrm{Vol}\left( \{ y \in \mathcal{B}_d(\boldsymbol{0}, 1) \mid \boldsymbol{u}^\top \boldsymbol{y} \in J \} \right).$$

The distribution of $\boldsymbol{u}^\top \boldsymbol{y}$, where $\boldsymbol{y} \sim \mathrm{Unif}(\mathcal{B}_2(\boldsymbol{0}, 1))$, has density

$$p(t) = \frac{V_{d-1}}{V_d}(1 - t^2)^{\frac{d-1}{2}} \text{ for } t \in [-1, 1], \quad \text{where } V_d = \frac{\pi^{d/2}}{\Gamma(\frac{d}{2}+1)}.$$

Thus,

$$\mathrm{Vol}(A) = \mu^d \cdot V_d \int_J p(t)dt \leq \mu^d \int_J V_{d-1}\, dt = 2\eta \cdot \mu^d V_{d-1},$$

$$\mathrm{Vol}(\mathcal{B}_2(\boldsymbol{x}_0, \mu)) = \mu^d V_d.$$

Hence,

$$\frac{\mathrm{Vol}\left(\boldsymbol{x} \in \mathcal{B}_2(\boldsymbol{x}_0, \mu) : f(\boldsymbol{x}) \in I\right)}{\mathrm{Vol}\left(\mathcal{B}_2(\boldsymbol{x}_0, \mu)\right)} = \frac{\mathrm{Vol}(A)}{\mathrm{Vol}(\mathcal{B}_2(\boldsymbol{x}_0, \mu))} \leq 2\eta \frac{V_{d-1}}{V_d}.$$

$\square$

**Lemma B.12.** *For all integers $d \geq 1$,*

$$\frac{V_d}{V_{d-1}} \leq 2,$$

*where $V_d = \dfrac{\pi^{d/2}}{\Gamma(\frac{d}{2}+1)}$ is the volume of the d-dimensional unit ball.*

*Proof.* Set

$$R_d := \frac{V_d}{V_{d-1}} = \sqrt{\pi}\, \frac{\Gamma\left(\frac{d+1}{2}\right)}{\Gamma\left(\frac{d}{2}+1\right)}.$$

Let $x = \frac{d}{2}$ and define

$$g(x) := \log R_{2x} = \tfrac{1}{2}\log \pi + \log \Gamma\left(x + \tfrac{1}{2}\right) - \log \Gamma(x+1).$$

Differentiating and using the digamma function $\psi = \Gamma'/\Gamma$, we get

$$g'(x) = \psi\left(x + \tfrac{1}{2}\right) - \psi(x+1) < 0,$$

since $\psi$ is strictly increasing. Hence $R_d$ is strictly decreasing in $d$. Therefore $\max_{d \geq 1} R_d = R_1 = V_1/V_0 = 2$, which proves $R_d \leq 2$ with equality only at $d = 1$. $\square$

**Lemma B.13.** *Let $\eta \in (0,1)$, $\alpha \in [0,1]$ and $\mathcal{D} = \{(\boldsymbol{x}_i, y_i)\}_{i \in [N]} \in \boldsymbol{D}_{d,N,C}$ be a dataset with separation $\epsilon_\mathcal{D} \geq \sqrt{d}/2$, and let the robustness ratio be $\rho = \frac{1}{4\lfloor N^\alpha \rfloor \epsilon_\mathcal{D}}$. Then, for any index set $I \subseteq [N]$ with $|I| \leq \lfloor N^\alpha \rfloor + 1$, there exists a neural network $f$ with width $O(d)$, depth $\tilde{O}(N^{\frac{\alpha}{2}})$, $\tilde{O}\left(N^{\frac{\alpha}{2}} + d^2\right)$ parameters and $\tilde{O}(N^{\frac{\alpha}{2}} + d)$ bit complexity such that:*

$$f(\boldsymbol{x}) = y_i \qquad\qquad \forall i \in I, \boldsymbol{x} \in \mathcal{B}(\boldsymbol{x}_i, \rho\epsilon_\mathcal{D}),$$
$$\mathbb{P}_{\boldsymbol{x} \in \mathrm{Unif}(\mathcal{B}(\boldsymbol{x}_i, \rho\epsilon_\mathcal{D}))}\left[f(\boldsymbol{x}) \in \{0, y_i\}\right] \geq 1 - \eta \qquad\qquad \forall i \in [N] \backslash I.$$

A network $f$, obtained from this lemma, memorizes each data point and its robustness ball for all indices $i \in I$. $f$ maps every other data point and its robustness ball to either its correct label or $0$ with high probability $1 - \eta$.

*Proof.* We construct a network proceeding in three stages. In each stage, we define subnetworks such that their composition satisfies the requirements.

**Stage I** (Translation for Distancing from Lattice via the Bias) We first translate the data points so that for $i \in I$, the robustness ball centered at $\boldsymbol{x}_i$ lies far from integer lattice boundaries. This ensures that each ball lies entirely within a single unit grid cell. By applying Lemma B.15 to the points $\{\boldsymbol{x}_i\}_{i \in I}$, we obtain a translation vector $\boldsymbol{b} = (b_1, \cdots, b_d) \in \mathbb{R}^d$ with bit complexity $\lceil \log(6|I|) \rceil$ such that

$$\mathrm{dist}(x_{i,j} - b_j, \mathbb{Z}) \geq \frac{1}{3\lfloor N^\alpha \rfloor}, \quad \forall i \in I, j \in [d], \tag{24}$$

i.e., the translated points $\{\boldsymbol{x}_i - \boldsymbol{b}\}_{i \in I}$ are coordinate-wise far from the integer lattice. Additionally, we apply an integer-valued translation (coordinate-wise) so that all coordinates of the points $\{\boldsymbol{x}_i - \boldsymbol{b}\}_{i \in [N]}$ become positive, while preserving the distance property in Equation (24). Hence, without loss of generality, we can assume $\boldsymbol{b}$ also has the property

$$\boldsymbol{x}_i - \boldsymbol{b} \geq \boldsymbol{0} \text{ for all } i \in [N]. \tag{25}$$

Let $\mathcal{D}' := \{(\boldsymbol{x}'_i, y_i)\}_{i\in[N]}$, where $\boldsymbol{x}'_i := \boldsymbol{x}_i - \boldsymbol{b}$. Then $\epsilon_{\mathcal{D}'} = \epsilon_{\mathcal{D}}$. For $\rho' := \rho = \frac{1}{4\lfloor N^\alpha\rfloor\epsilon_{\mathcal{D}}}$, we have the robustness radius $\mu' := \rho'\epsilon_{\mathcal{D}'} = \rho\epsilon_{\mathcal{D}} = \frac{1}{4\lfloor N^\alpha\rfloor}$. Define $f_{\text{trans}}$ as $f_{\text{trans}}(\boldsymbol{x}) := \boldsymbol{x} - \boldsymbol{b}$. Then, $f_{\text{trans}}$ can be implemented via one hidden layer with $O(d^2)$ parameters in a neural network.

Since the translation preserves separation ($\epsilon_{\mathcal{D}} = \epsilon_{\mathcal{D}'}$) and ball containment properties (robustness ball of $\mathcal{D}$ is mapped to the robustness ball of $\mathcal{D}'$ through the translation), it suffices to construct a network that satisfies the requirements with $\rho \leftarrow \rho'$ and $\mathcal{D} \leftarrow \mathcal{D}'$. Observe that the robustness balls after Stage I are not affected when passing the $\sigma$, by Equations (24) and (25).

**Stage II** (Grid Indexing) From Equation (24), each $\boldsymbol{x}'_i \in \mathbb{R}^d$ (for $i \in I$) is at least $\frac{4}{3}\mu'$ distant from any lattice hyperplane $H_{z,j} := \{\boldsymbol{x} \in \mathbb{R}^d \mid x_j = z\}$ for each $j \in [d]$ and $z \in \mathbb{Z}$. Hence, each robustness ball centered at $\boldsymbol{x}'_i$ (for $i \in I$) lies completely within a single integer lattice (or unit grid) $\prod_{j=1}^{d}[n_j, n_j + 1)$, for some $(n_1, \cdots, n_d) \in \mathbb{Z}^d$. Moreover, for any $\boldsymbol{x} \in \mathcal{B}_2(\boldsymbol{x}'_i, \mu')$, the distance from the integer lattice remains at least $\mu'$.

Furthermore, by the separation condition $\epsilon_{\mathcal{D}'} = \epsilon_{\mathcal{D}} \geq \sqrt{d}/2$, for any $i \neq i'$ with $y_i \neq y_{i'}$, we have $\|\boldsymbol{x}'_i - \boldsymbol{x}'_{i'}\|_2 \geq \sqrt{d}$. Since $\sup\{\|\boldsymbol{x} - \boldsymbol{x}'\|_2 \mid \boldsymbol{x}, \boldsymbol{x}' \in \prod_{j=1}^{d}[n_j, n_j + 1)\} = \sqrt{d}$, two such points cannot lie in the same grid. Recall the separation condition holds for all data points $\boldsymbol{x}'_i$ for $i \in [N]$ and each ball $\mathcal{B}_2(\boldsymbol{x}'_i, \mu')$ (for $i \in I$) lies within a single grid. We conclude that for each $i \in I$, the robustness ball $\mathcal{B}_2(\boldsymbol{x}'_i, \mu')$ is not intersected by any other robustness ball $\mathcal{B}_2(\boldsymbol{x}'_j, \mu')$ with a different label, for any $j \in [N]$, i.e., no ball with a different label overlaps the grid cell containing $\mathcal{B}_2(\boldsymbol{x}'_i, \mu')$.

We define $R := \lceil \max_{i\in I}\|\boldsymbol{x}'_i\|_\infty \ (= \max_{i\in I, j\in[d]}(x'_{i,j})) \rceil \in \mathbb{N}$. Our goal in this stage is to construct Flatten mapping defined as

$$\text{Flatten}(\boldsymbol{x}) := R^{d-1}\lfloor x_1\rfloor + R^{d-2}\lfloor x_2\rfloor + \cdots + \lfloor x_d\rfloor.$$

This maps each grid $\prod_{j=1}^{d}[n_j, n_{j+1})$ onto the point $\sum_{j=1}^{d} R^{j-1}n_j$.

However, since Flatten is discontinuous due to the use of floor functions, we construct $\overline{\text{Flatten}}$ which is a continuous approximation that exactly matches Flatten on the region $\bigcup_{i\in I}\mathcal{B}_2(\boldsymbol{x}'_i, \mu')$, and incurs only a small error on the remaining region $\bigcup_{i\in[N]\setminus I}\mathcal{B}_2(\boldsymbol{x}'_i, \mu')$. We choose large enough $t \in \mathbb{N}$ so that for $\eta' := 1/t$, we have $\eta' \leq \frac{V_d}{2dV_{d-1}}\mu'\eta$ where $V_d = \frac{\pi^{d/2}}{\Gamma(\frac{d}{2}+1)}$ denotes the volume of the $d$-dimensional unit ball. Moreover, we can take such $t \in \mathbb{N}$ which at the same time satisfies

$$t \leq \frac{2dV_{d-1}}{V_d\mu'\eta} + 1 = \frac{2d\Gamma(\frac{d}{2}+1)}{\pi^{1/2}\Gamma(\frac{d-1}{2}+1)\mu'\eta} + 1 = O(d^2/(\mu'\eta)) = O(d^2\lfloor N^\alpha\rfloor/\eta).$$

By Lemma B.16, for $\gamma := 1/t = \eta'$ and $n := \lceil\log_2 R\rceil$, we obtain the network $\overline{\text{Floor}} := \overline{\text{Floor}}_{\lceil\log_2 R\rceil}$ with $O(\log_2 R)$ parameters such that

$$\overline{\text{Floor}}(x) = \lfloor x\rfloor \quad \forall x \in [0, R] \text{ with } x - \lfloor x\rfloor > \eta'. \tag{26}$$

Since we apply $\gamma = 1/t$ to Lemma B.16, $\overline{\text{Floor}}$ can be implemented with $O(n + \log t) = O(\log R + \log(d^2\lfloor N^\alpha\rfloor/\eta)) = O(\log(dRN/\eta))$ bit complexity. In particular, we can define our network $\overline{\text{Flatten}}$ with $O(\log(dRN/\eta) + \log R^{d-1}) = O(\log(dRN/\eta) + d\log R) = \tilde{O}(d)$ bit complexity as

$$\overline{\text{Flatten}}(\boldsymbol{x}) = R^{d-1}\overline{\text{Floor}}(x_1) + \cdots + \overline{\text{Floor}}(x_d). \tag{27}$$

As $\overline{\text{Floor}} : \mathbb{R} \to \mathbb{R}$ can be implemented with width 5 and depth $O(\log_2 R)$ network (Lemma B.16), $\overline{\text{Flatten}}$ can be implemented with width $5d$ and depth $O(\log_2 R)$ network. Thus, we can construct $\overline{\text{Flatten}}$ with $O(d^2\log_2 R) = \tilde{O}(d^2)$ parameters.

We first observe that this implementation is valid on the region $\bigcup_{i\in I}\mathcal{B}_2(\boldsymbol{x}'_i, \mu')$. For $i \in I$ and $\boldsymbol{x} \in \mathcal{B}_2(\boldsymbol{x}'_i, \mu')$, we have

$$x_j - \lfloor x_j\rfloor \overset{(a)}{>} \mu' \overset{(b)}{>} \mu'\eta \overset{(c)}{\geq} \frac{V_d}{2V_{d-1}}\mu'\eta \overset{}{>} \frac{V_d}{2dV_{d-1}}\mu'\eta \overset{(d)}{\geq} \eta',$$

where (a) holds by Equation (24), (b) holds since $\eta < 1$, (c) holds since $\frac{V_d}{V_{d-1}} \leq 2$ by Lemma B.12, and (d) holds from the choice of $\eta'$. Thus, for any $i \in I$ and any $\boldsymbol{x} \in \mathcal{B}_2(\boldsymbol{x}'_i, \mu')$, $x_j$ satisfies the

requirement in Equation (26). Therefore, we guarantee that each robustness ball centered at $\boldsymbol{x}_i'$ for $i \in I$ lies in the region where the Flatten is properly approximated by $\overline{\text{Flatten}}$. i.e.

$$\overline{\text{Flatten}}(\boldsymbol{x}) = \text{Flatten}(\boldsymbol{x}) \text{ for all } i \in I \text{ and } \boldsymbol{x} \in \mathcal{B}_2(\boldsymbol{x}_i', \mu').$$

Since Flatten maps each unit grid into a point and each robustness ball centered at $\boldsymbol{x}_i'$ for $i \in I$ lies on a single unit grid, we conclude

$$\overline{\text{Flatten}}(\boldsymbol{x}) = \text{Flatten}(\boldsymbol{x}) = \text{Flatten}(\boldsymbol{x}_i') \text{ for all } i \in I \text{ and } \boldsymbol{x} \in \mathcal{B}_2(\boldsymbol{x}_i', \mu').$$

Let $m_i := \text{Flatten}(\boldsymbol{x}_i')$ for $i \in I$. Then for $i \in I$, each robustness ball centered at $\boldsymbol{x}_i'$ is mapped to $m_i$. We have $m_i \in \mathbb{Z} \cap [0, R^{d+1}]$ for all $i \in I$, since

$$\begin{aligned}
m_i &= \text{Flatten}(\boldsymbol{x}_i') \\
&= R^{d-1}\lfloor x_{i1}' \rfloor + R^{d-2}\lfloor x_{i2}' \rfloor + \cdots \lfloor x_{id}' \rfloor \\
&\overset{(a)}{\leq} R^{d-1}R + R^{d-2}R + \cdots + R \\
&\leq R^{d+1},
\end{aligned}$$

where (a) is by $\|\boldsymbol{x}_i'\|_\infty \leq R$.

Next, we consider the case $i \in [N] \setminus I$. Note that the lattice distance condition in Equation (24) applies only to the subset $\{(\boldsymbol{x}_i, y_i)\}_{i \in I}$, rather than the entire dataset. As a result, for indices $i \in [N] \setminus I$, the distance from the lattice is not guaranteed. Thus, it can lie across the lattice.

For $i \in [N] \setminus I$, we analyze the error of $\overline{\text{Flatten}}$ on the remaining region $\bigcup_{i \in [N] \setminus I} \mathcal{B}_2(\boldsymbol{x}_i', \mu')$. For $i \in [N] \setminus I$, we have

$$\begin{aligned}
&\mathbb{P}_{\boldsymbol{x} \in \text{Unif}(\mathcal{B}_2(\boldsymbol{x}_i', \mu'))} \left[ \overline{\text{Flatten}}(\boldsymbol{x}) \neq \text{Flatten}(\boldsymbol{x}) \right] \\
&\overset{(a)}{\leq} \mathbb{P}_{\boldsymbol{x} \in \text{Unif}(\mathcal{B}_2(\boldsymbol{x}_i', \mu'))} \left[ \exists j \in [d], \quad x_j - \lfloor x_j \rfloor \leq \eta' \right] \\
&\leq \sum_{j \in [d]} \mathbb{P}_{\boldsymbol{x} \in \text{Unif}(\mathcal{B}(\boldsymbol{x}_i', \mu'))} \left[ x_j - \lfloor x_j \rfloor \leq \eta' \right] \\
&\leq \sum_{j \in [d]} \max_{\substack{\mathcal{I} \subseteq \mathbb{R} \\ \text{s.t. } \text{Len}(\mathcal{I}) = \eta'}} \mathbb{P}_{\boldsymbol{x} \in \text{Unif}(\mathcal{B}(\boldsymbol{x}_i', \mu'))} \left[ x_j \in \mathcal{I} \right] \\
&\overset{(b)}{\leq} \sum_{j \in [d]} \frac{2\eta' V_{d-1}}{\mu' V_d} \\
&= \frac{2\eta' d V_{d-1}}{\mu' V_d} \\
&\overset{(c)}{\leq} \eta,
\end{aligned}$$

where (a) follows from Equation (26) and the fact that $\overline{\text{Flatten}}(\boldsymbol{x}) = \text{Flatten}(\boldsymbol{x})$ whenever $x_j - \lfloor x_j \rfloor > \eta'$ for all $j \in [d]$, (b) follows from Lemma B.11 applied to a unit vector $u = \mathbf{e_j}$, $b = 0$, and an interval $\frac{\mathcal{I}_j}{\mu'}$, and (c) holds by the choice of $\eta'$. Hence, we have

$$\mathbb{P}_{\boldsymbol{x} \in \text{Unif}(\mathcal{B}_2(\boldsymbol{x}_i', \mu'))} \left[ \overline{\text{Flatten}}(\boldsymbol{x}) \neq \text{Flatten}(\boldsymbol{x}) \right] \leq \eta. \tag{28}$$

We observe at what happens if $\overline{\text{Flatten}}(\boldsymbol{x}) = \text{Flatten}(\boldsymbol{x})$ for $i \in [N] \setminus I$ and $\boldsymbol{x} \in \mathcal{B}_2(\boldsymbol{x}_i', \mu')$. To ensure that no robustness ball centered at $\boldsymbol{x}_i$ for $i \in [N] \setminus I$ is mapped to grid index $m_j$ with a different label, namely, satisfying $j \in I$ with $y_i \neq y_j$, we define label-specific grid index sets. For each class $c \in [C]$, define the set

$$G_c := \bigcup_{\substack{i \in I \\ \text{s.t. } y_i = c}} \{m_i\}, \text{ and } G := \bigcup_{c \in [C]} G_c = \{m_i\}_{i \in [N]},$$

where $G_c$ is the collection of all grid indices $m_i$ assigned to data points in $I$ that have label $y_i = c$. In other words, $G_c$ contains all grid cells that are claimed by class $c$. The set $G$ represents all valid grid indices.

Recall that for each $i \in I$, the robustness ball $\mathcal{B}_2(\boldsymbol{x}'_i, \mu')$ is not intersected by any other robustness ball $\mathcal{B}_2(\boldsymbol{x}'_j, \mu')$ with a different label. Specifically, consider $i \in [N] \setminus I$. For $j \in I$ with $y_i = y_j$, the robustness ball $\mathcal{B}_2(\boldsymbol{x}'_i, \mu')$ can have a portion that intersects the grid containing $\mathcal{B}_2(\boldsymbol{x}'_j, \mu')$, then the portion is mapped to the corresponding grid index $m_j$. However, for $j \in I$ with $y_i \neq y_j$, the robustness ball never intersects the grid, and is never mapped to $m_j$. Formally, if $\mathrm{Flatten}(\boldsymbol{x}) \in G$, it must be $\mathrm{Flatten}(\boldsymbol{x}) \in G_{y_i}$. Otherwise, $\mathrm{Flatten}(\boldsymbol{x}) \notin G$, i.e., the robustness ball does not intersect any selected grid. Thus, combining the probabilities,

$$\mathbb{P}_{\boldsymbol{x} \in \mathrm{Unif}(\mathcal{B}_2(\boldsymbol{x}'_i, \mu'))} \left[\overline{\mathrm{Flatten}}(\boldsymbol{x}) \in G_{y_i} \text{ or } \overline{\mathrm{Flatten}}(\boldsymbol{x}) \notin G\right]$$

$$\overset{(a)}{\geq} \mathbb{P}_{\boldsymbol{x} \in \mathrm{Unif}(\mathcal{B}_2(\boldsymbol{x}'_i, \mu'))} \left[\overline{\mathrm{Flatten}}(\boldsymbol{x}) = \mathrm{Flatten}(\boldsymbol{x})\right]$$

$$\overset{(b)}{\geq} 1 - \eta, \tag{29}$$

where (a) holds since $\mathbb{P}_{\boldsymbol{x} \in \mathrm{Unif}(\mathcal{B}_2(\boldsymbol{x}'_i, \mu'))} \left[\mathrm{Flatten}(\boldsymbol{x}) \in G_{y_i} \text{ or } \mathrm{Flatten}(\boldsymbol{x}) \notin G\right] = 1$, and (b) follows by Equation (28). Hence, if we memorize $\{(m_i, y_i)\}_{i \in I}$ and map other integer $\mathbb{N} \setminus \{m_i\}_{i \in I}$ to zero, $\mathcal{B}_2(\boldsymbol{x}'_i, \mu')$ for $i \in I$ is exactly mapped to $y_i$, and with high probability $1 - \eta$, $\mathcal{B}_2(\boldsymbol{x}'_i, \mu')$ for $i \in [N] \setminus I$ is mapped to either $y_i$ or 0.

**Stage III** (Memorization) Finally, we construct the network to memorize $\lfloor N^\alpha \rfloor$ points $\{(m_i, y_i)\}_{i=1}^I \subset \mathbb{Z}_{\geq 0} \times [C]$. Since multiple robustness balls for $\mathcal{D}'$ with the same label may correspond to the same grid index in Stage II, it is possible that for some $i \neq j$ with $y_i = y_j$, we have $m_i = m_j$. Let $N' \leq |I|$ denote the number of distinct pairs $(m_i, y_i)$. It remains to memorize these $N'$ distinct data points in $\mathbb{R}$.

Applying Lemma B.10, we obtain a neural network $f_{\mathrm{mem}}$ with width 12, depth $\tilde{O}(N)$, $\tilde{O}(N^{\frac{\alpha}{2}})$ parameters and bit complexity $\tilde{O}(N^{\frac{\alpha}{2}})$ satisfying:

$$f_{\mathrm{mem}}(m) = \begin{cases} y_i & \text{for every } m = m_i \text{ with } i \in I, \\ 0 & \text{for every } m \in \mathbb{N} \setminus G. \end{cases}$$

For $m \in \mathbb{N}$, $f_{\mathrm{mem}}(m) = c$ for some $c \in [C]$ if and only if $m \in G_c$.

The final network $f : \mathbb{R}^d \to \mathbb{R}$ is defined as

$$f = f_{\mathrm{mem}} \circ \sigma \circ \overline{\mathrm{Flatten}} \circ \sigma \circ f_{\mathrm{trans}}.$$

Let us verify the correctness of the construction.

For $i \in I$ and any $\boldsymbol{x} \in \mathcal{B}(\boldsymbol{x}_i, \rho \epsilon_{\mathcal{D}})$, we have

$$f(\boldsymbol{x}) = f_{\mathrm{mem}} \circ \sigma \circ \overline{\mathrm{Flatten}} \circ \sigma \circ f_{\mathrm{trans}}(\boldsymbol{x}) \overset{(a)}{=} f_{\mathrm{mem}} \circ \sigma(m_i) \overset{(b)}{=} f_{\mathrm{mem}}(m_i) = y_i,$$

where (a) holds since $\overline{\mathrm{Flatten}} \circ \sigma \circ f_{\mathrm{trans}}(\boldsymbol{x}) = m_i$, (b) holds since $m_i \in \mathbb{N}$, and (c) follows that $f$ is constructed to memorize $\{(m_i, y_i)\}_{i=1}^I$.

Next, consider $i \in [N] \setminus I$. We observe

$$\mathbb{P}_{\boldsymbol{x} \in \mathrm{Unif}(\mathcal{B}(\boldsymbol{x}_i, \rho \epsilon_{\mathcal{D}}))} \left[f(\boldsymbol{x}) \in \{0, y_i\}\right]$$

$$\overset{(a)}{=} \mathbb{P}_{\boldsymbol{x} \in \mathrm{Unif}(\mathcal{B}_2(\boldsymbol{x}_i, \mu))} \left[\sigma \circ \overline{\mathrm{Flatten}} \circ \sigma \circ f_{\mathrm{trans}}(\boldsymbol{x}) \in G_{y_i} \text{ or } \sigma \circ \overline{\mathrm{Flatten}} \circ \sigma \circ f_{\mathrm{trans}}(\boldsymbol{x}) \notin G\right]$$

$$\overset{(b)}{=} \mathbb{P}_{\boldsymbol{x}' \in \mathrm{Unif}(\mathcal{B}_2(\boldsymbol{x}'_i, \mu'))} \left[\sigma \circ \overline{\mathrm{Flatten}}(\boldsymbol{x}') \in G_{y_i} \text{ or } \sigma \circ \overline{\mathrm{Flatten}}(\boldsymbol{x}') \notin G\right]$$

$$\overset{(c)}{=} \mathbb{P}_{\boldsymbol{x}' \in \mathrm{Unif}(\mathcal{B}_2(\boldsymbol{x}'_i, \mu'))} \left[\overline{\mathrm{Flatten}}(\boldsymbol{x}') \in G_{y_i} \text{ or } \overline{\mathrm{Flatten}}(\boldsymbol{x}') \notin G\right]$$

$$\overset{(d)}{\geq} 1 - \eta,$$

where (a) holds from the construction of $f_{\mathrm{mem}}$, (b) holds using $\boldsymbol{x}' := \sigma \circ f_{\mathrm{trans}}(\boldsymbol{x})$, (c) holds by Equation (25) and (d) holds by Equation (29). This concludes the proof.

The depth 1 network $f_{\mathrm{trans}}$ has width $d$. $\overline{\mathrm{Flatten}}$ has width $5d$ and depth $O(\log_2 R)$ and $f_{\mathrm{mem}}$ has width 12 and depth $\tilde{O}(N^{\frac{\alpha}{2}})$. The total construction requires $\tilde{O}(d^2 + d^2 + N^{\frac{\alpha}{2}}) = \tilde{O}(d^2 + N^{\frac{\alpha}{2}})$

parameters, where each term $d^2, d^2$, and $N^{\frac{\alpha}{2}}$ comes from $f_{\text{trans}}, \overline{\text{Flatten}}$, and $f_{\text{mem}}$ respectively. The width of the final network is $O(d)$ and the depth is $\tilde{O}(N^{\frac{\alpha}{2}})$.

The bit complexity of $f_{\text{trans}}$ is $O(\log N^\alpha, \log(\max\{\|\boldsymbol{x}_i\|_\infty \mid i \in [N]\})) = \tilde{O}(1)$. $\overline{\text{Flatten}}$ has the bit complexity $\tilde{O}(d)$, and $f_{\text{mem}}$ needs at most $\tilde{O}(N^{\frac{\alpha}{2}})$. Hence, the bit complexity of the final network is $\tilde{O}(N^{\frac{\alpha}{2}} + d)$. $\qquad\square$

## B.3 Sufficient Condition for Robust Memorization with Large Robustness Radius

**Theorem B.14.** *Let $\rho \in \left(\frac{1}{5\sqrt{d}}, 1\right)$. For any dataset $\mathcal{D} \in \boldsymbol{D}_{d,N,C}$, there exists $f$ with $\tilde{O}(Nd^2\rho^4)$ parameters, depth $\tilde{O}(N)$, width $\tilde{O}(\rho^2 d)$ and bit complexity $\tilde{O}(N)$ that $\rho$-robustly memorizes $\mathcal{D}$.*

*Proof.* Let $\mathcal{D} = \{(\boldsymbol{x}_i, y_i)\}_{i\in[N]} \in \boldsymbol{D}_{d,N,C}$ be given. We divide the proof into five cases, the first case under $\rho \in [1/3, 1)$, the second case under $\rho \in (1/5\sqrt{d}, 1/3)$ and $d < 600\log N$, the third case under $\rho \in (1/5\sqrt{d}, 1/3)$ and $N < 600\log N \leq d$, the fourth case under $\rho \in (1/5\sqrt{d}, 1/3)$, $N \geq d \geq 600\log N$, and finally the fifth case under $\rho \in (1/5\sqrt{d}, 1/3)$ and $d > N \geq 600\log N$. To check that these cases cover all the cases, refer to Figure 6.

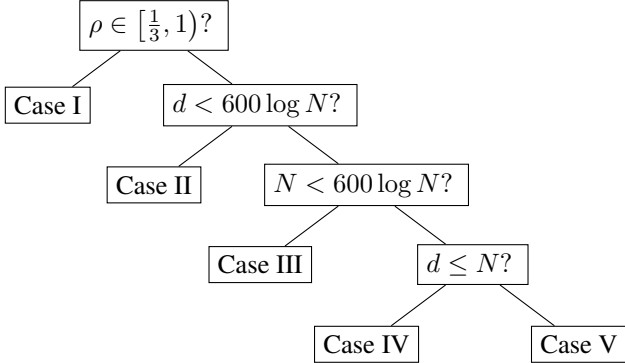

Figure 6: Different cases for Theorem B.14. The left child is for the answer "Yes", and the right child is for the answer "No"

The first two cases follow easily from prior works, while the remaining cases require careful analysis using dimension reduction techniques. The most interesting cases are cases IV and V. While we track the width, depth, and parameter complexity for each case, we initially implement them using infinite precision. We address the bit complexity by approximating the infinite precision network using a finite precision network at the very last part of the proof. As a spoiler, the bit complexity of all cases is handled within a unified framework using Lemma B.23. Let us deal with each case one by one.

**Case I:** $\rho \in [1/3, 1)$. In the first case, where $\rho \in [1/3, 1)$, the result directly follows from the prior result by Yu et al. [2024]. In particular, we apply Lemma D.2. Let us denote $R := \max_{i\in[N]} \|\boldsymbol{x}_i\|_2$ and $\gamma := (1-\rho)\epsilon_\mathcal{D}$. Note that $R \geq \|\boldsymbol{x}_i\|_\infty$ for all $i \in [N]$ as $\|\boldsymbol{x}\|_2 \geq \|\boldsymbol{x}\|_\infty$ for all $\boldsymbol{x} \in \mathbb{R}^d$. By applying Lemma D.2, there exists $f \in \mathcal{F}_{d,P}$ with $P = O(Nd^2(\log(\frac{d}{\gamma^2}) + \log R))$ parameters that $\rho$-robustly memorize $\mathcal{D}$. The number of parameters can be further bounded as follows:

$$O(Nd^2(\log(\frac{d}{\gamma^2}) + \log R)) \overset{(a)}{=} O(Nd^2\rho^4 \cdot (\log(\frac{d}{\gamma^2}) + \log R)) \overset{(b)}{=} \tilde{O}(Nd^2\rho^4),$$

where (a) is due to $\rho = \Omega(1)$, (b) hides the logarithmic factors. Moreover, by Lemma D.2, the network has width $O(d) = O(\rho^2 d)$ and depth $\tilde{O}(N)$.

**Case II:** $\rho \in (1/5\sqrt{d}, 1/3)$ **and** $d < 600\log N$. In the second case, where $d < 600\log N$ and $(1/5\sqrt{d}, 1/3)$, the result also directly follows from the prior result by Yu et al. [2024]. In particular, we apply Lemma D.2. Let us denote $R := \max_{i\in[N]} \|\boldsymbol{x}_i\|_2$ and $\gamma := (1-\rho)\epsilon_\mathcal{D}$. Note that $R \geq \|\boldsymbol{x}_i\|_\infty$ for all $i \in [N]$ as $\|\boldsymbol{x}\|_2 \geq \|\boldsymbol{x}\|_\infty$ for all $\boldsymbol{x} \in \mathbb{R}^d$. By Lemma D.2, there exists $f \in \mathcal{F}_{d,P}$ with

$P = O(Nd^2(\log(\frac{d}{\gamma^2}) + \log R))$ parameters that $\rho$-robustly memorize $\mathcal{D}$. The number of parameters can be further bounded as follows:

$$O(Nd^2(\log(\frac{d}{\gamma^2}) + \log R)) \overset{(a)}{=} O(N(\log N)^2 \cdot (\log(\frac{d}{\gamma^2}) + \log R)) \overset{(b)}{=} \tilde{O}(N) \overset{(c)}{=} \tilde{O}(Nd^2\rho^4),$$

where (a) is due to $d \leq 600 \log N$, (b) hides the logarithmic factors, and (c) is because $N \leq 625Nd^2\rho^4$ for all $\rho \in \left(\frac{1}{5\sqrt{d}}, \frac{1}{3}\right)$. Moreover, by Lemma D.2, the network has width $O(d) = O(\log N) = \tilde{O}(1) = \tilde{O}(\rho^2 d)$ and depth $\tilde{O}(N)$.

**Case III:** $\rho \in (1/5\sqrt{d}, 1/3)$ **and** $N < 600 \log N \leq d$. In the third case, where $N < 600 \log N \leq d$ and $(1/5\sqrt{d}, 1/3)$, we first apply Proposition B.21 to $\mathcal{D}$ to obtain 1-Lipschitz linear $\varphi : \mathbb{R}^d \to \mathbb{R}^N$ such that $\mathcal{D}' := \{(\varphi(\boldsymbol{x}_i), y_i)\}_{i \in [N]}$ has $\epsilon_{\mathcal{D}'} = \epsilon_{\mathcal{D}}$. This is possible as $d \geq N$.

We apply Lemma D.2 by Yu et al. [2024] to $\mathcal{D}'$. Let us denote $R := \max_{i \in [N]} \|\varphi(\boldsymbol{z}_i)\|_2$ and $\gamma := (1 - \rho)\epsilon_{\mathcal{D}'}$. Note that $R \geq \|\varphi(\boldsymbol{z}_i)\|_\infty$ for all $i \in [N]$ as $\|\boldsymbol{z}\|_2 \geq \|\boldsymbol{z}\|_\infty$ for all $\boldsymbol{z} \in \mathbb{R}^N$. By Lemma D.2, there exists $f_1 \in \mathcal{F}_{N,P}$ with $P = O(N \cdot N^2(\log(\frac{N}{\gamma^2}) + \log R))$ parameters that $\rho$-robustly memorize $\mathcal{D}'$. $f_1$ has width $O(N)$ and depth $\tilde{O}(N)$.

Let $f = f_1 \circ \varphi$. This can be implemented by changing the first hidden layer matrix of $f_1$ by composing $\varphi$. This is possible because $\varphi$ is linear. $f$ has at most $dN$ additional parameters compared to $f_1$, and has same width and depth as $f_1$. Since $f_1$ is 1-Lipschitz and $\epsilon_{\mathcal{D}'} = \epsilon_{\mathcal{D}}$, every robustness ball of $\mathcal{D}$ is mapped to the robustness ball of $\mathcal{D}'$ via $f_1$. As $f_1$ $\rho$-robustly memorizes $\mathcal{D}'$, the composed $f$ satisfies the desired property.

The number of parameters can be further bounded as follows:

$$O(Nd + N \cdot N^2(\log(\frac{d}{\gamma^2}) + \log R)) \overset{(a)}{=} O(d \log N + (\log N)^3 \cdot (\log(\frac{d}{\gamma^2}) + \log R)) \overset{(b)}{=} \tilde{O}(d),$$

where (a) is due to $N \leq 600 \log N$, and (b) hides the logarithmic factors. The width of $f$ is $O(N) = O(\log(N)) = \tilde{O}(1) = \tilde{O}(\rho^2 d)$. The depth of $f$ is $\tilde{O}(N)$.

**Case IV:** $\rho \in (1/5\sqrt{d}, 1/3)$**, and** $N \geq d \geq 600 \log N$. In the fourth case, where $d \geq 600 \log N$, we utilize the dimension reduction technique by Proposition B.19. We apply Proposition B.19 to $\mathcal{D}$ with $m = \max\{\lceil 9d\rho^2 \rceil, \lceil 600 \log N \rceil, \lceil 10 \log d \rceil\}$ and $\alpha = 1/5$. Let us first check that the specified $m$ satisfies the condition $24\alpha^{-2} \log N \leq m \leq d$ for the proposition to be applied. $\alpha = 1/5$ and $m \geq 600 \log N$ ensure the first inequality $24\alpha^{-2} \log N \leq m$. The second inequality $m \leq d$ is decomposed into three parts. Since $\rho \leq \frac{1}{3}$, we have $9d\rho^2 \leq d$ so that

$$\lceil 9d\rho^2 \rceil \leq d. \tag{30}$$

Moreover, $600 \log N \leq d$ implies

$$\lceil 600 \log N \rceil \leq d. \tag{31}$$

Additionally, as $N \geq 2$, we have $d \geq 600 \log N \geq 600 \log 2 \geq 400$. By Lemma B.22, this implies $10 \log d \leq d$ and therefore

$$\lceil 10 \log d \rceil \leq d. \tag{32}$$

Gathering Equations (30) to (32) proves $m \leq d$.

By the Proposition B.19, there exists 1-Lipschitz linear mapping $\phi : \mathbb{R}^d \to \mathbb{R}^m$ and $\beta > 0$ such that $\mathcal{D}' := \{(\phi(\boldsymbol{x}_i), y_i)\}_{i \in [N]} \in \boldsymbol{D}_{m,N,C}$ satisfies

$$\epsilon_{\mathcal{D}'} \geq \frac{4}{5}\beta\epsilon_{\mathcal{D}}. \tag{33}$$

As $m \geq 10 \log d$, the inequality $\beta \geq \frac{1}{2}\sqrt{\frac{m}{d}}$ is also satisfied by Proposition B.19. Therefore, we have

$$\beta \geq \frac{1}{2}\sqrt{\frac{m}{d}} \overset{(a)}{\geq} \frac{1}{2}\sqrt{\frac{\lceil 9d\rho^2 \rceil}{d}} \geq \frac{1}{2}\sqrt{\frac{9d\rho^2}{d}} = \frac{3}{2}\rho, \tag{34}$$

where (a) is by the definition of $m$. Moreover, since $\phi$ is 1-Lipschitz linear,

$$\|\phi(\boldsymbol{x}_i)\|_2 = \|\phi(\boldsymbol{x}_i - \mathbf{0})\|_2 = \|\phi(\boldsymbol{x}_i) - \phi(\mathbf{0})\|_2 \leq \|\boldsymbol{x}_i - \mathbf{0}\|_2 = \|\boldsymbol{x}_i\|_2, \tag{35}$$

for all $i \in [N]$. Hence, by letting $R := \max_{i \in [N]}\{\|\boldsymbol{x}_i\|_2\}$, we have $\|\phi(\boldsymbol{x}_i)\|_2 \leq R$ for all $i \in [N]$.

Now, we set the first layer hidden matrix as the matrix $\boldsymbol{W} \in \mathbb{R}^{m \times d}$ corresponding to $\phi$ under the standard basis of $\mathbb{R}^d$ and $\mathbb{R}^m$. Moreover, set the first hidden layer bias as $\boldsymbol{b} := 2R\mathbf{1} = 2R(1, 1, \cdots, 1) \in \mathbb{R}^m$. Then, we have

$$\boldsymbol{W}\boldsymbol{x} + \boldsymbol{b} \geq \mathbf{0}, \tag{36}$$

for all $\boldsymbol{x} \in \mathcal{B}_2(\boldsymbol{x}_i, \epsilon_{\mathcal{D}})$ for all $i \in [N]$, where the comparison between two vectors are element-wise. This is because for all $i \in [N], j \in [m]$ and $\boldsymbol{x} \in \mathcal{B}_2(\boldsymbol{x}, \epsilon_{\mathcal{D}})$, we have

$$(\boldsymbol{W}\boldsymbol{x} + \boldsymbol{b})_j = (\boldsymbol{W}\boldsymbol{x})_j + 2R \geq 2R - \|\boldsymbol{W}\boldsymbol{x}\|_2 \overset{(a)}{\geq} 2R - \|\boldsymbol{x}\|_2 \overset{(b)}{\geq} 2R - (R + \epsilon_{\mathcal{D}}) \overset{(c)}{\geq} 0,$$

where (a) is by Equation (35), (b) is by the triangle inequality, and (c) is due to $R > \epsilon_{\mathcal{D}}$.

We construct the first layer of the neural network as $f_1(\boldsymbol{x}) := \sigma(\boldsymbol{W}\boldsymbol{x} + \boldsymbol{b})$ which includes the activation $\sigma$. Then, by above properties, $\mathcal{D}'' := \{(f_1(\boldsymbol{x}_i), y_i)\}_{i \in [N]}$ satisfies

$$\epsilon_{\mathcal{D}''} \geq \frac{6}{5}\rho\epsilon_{\mathcal{D}}. \tag{37}$$

This is because for $i \neq j$ with $y_i \neq y_j$ we have

$$\begin{aligned}
\|f_1(\boldsymbol{x}_i) - f_1(\boldsymbol{x}_j)\|_2 &= \|\sigma(\boldsymbol{W}\boldsymbol{x}_i + \boldsymbol{b}) - \sigma(\boldsymbol{W}\boldsymbol{x}_j + \boldsymbol{b})\|_2 \\
&\overset{(a)}{=} \|(\boldsymbol{W}\boldsymbol{x}_i + \boldsymbol{b}) - (\boldsymbol{W}\boldsymbol{x}_j + \boldsymbol{b})\|_2 \\
&= \|\phi(\boldsymbol{x}_i) - \phi(\boldsymbol{x}_k)\|_2 \\
&\overset{(b)}{\geq} 2\epsilon_{\mathcal{D}'} \\
&\overset{(c)}{\geq} 2 \times \frac{4}{5}\beta\epsilon_{\mathcal{D}} \\
&\overset{(d)}{\geq} 2 \times \frac{4}{5} \times \frac{3}{2}\rho\epsilon_{\mathcal{D}} \\
&= \frac{12}{5}\rho\epsilon_{\mathcal{D}},
\end{aligned}$$

where (a) is by Equation (36), (b) is by the definition of the $\epsilon_{\mathcal{D}'}$, (c) is by Equation (33), and (d) is by Equation (34). By Lemma D.2 applied to $\mathcal{D}'' \in \boldsymbol{D}_{m,N,C}$, there exists $f_2 \in \mathcal{F}_{m,P}$ with $P = O(Nm^2(\log(\frac{m}{(\gamma'')^2}) + \log R''))$ number of parameters that $\frac{5}{6}$-robustly memorize $\mathcal{D}''$, where

$$\begin{aligned}
\gamma'' &:= (1 - \frac{5}{6})\epsilon_{\mathcal{D}''} \overset{(a)}{\geq} \frac{1}{6} \times \frac{12}{5}\rho\epsilon_{\mathcal{D}} = \frac{2}{5}\rho\epsilon_{\mathcal{D}}, \\
R'' &:= \max_{i \in [N]} \|f_1(\boldsymbol{x}_i)\|_2 = \max_{i \in [N]} \|\sigma(\boldsymbol{W}\boldsymbol{x}_i + \boldsymbol{b})\|_2 = \max_{i \in [N]} \|\boldsymbol{W}\boldsymbol{x}_i + \boldsymbol{b}\|_2 \\
&\leq \max_{i \in [N]} \|\boldsymbol{W}\boldsymbol{x}_i\|_2 + \|\boldsymbol{b}\|_2 \leq 3R.
\end{aligned}$$

Here (a) is by Equation (37). Moreover $f_2$ has width $O(m)$ and depth $\tilde{O}(N)$ by Lemma D.2.

Now, we claim that $f := f_2 \circ f_1$ $\rho$-robustly memorize $\mathcal{D}$. For any $i \in [N]$, take $\boldsymbol{x} \in \mathcal{B}_2(\boldsymbol{x}_i, \rho\epsilon_{\mathcal{D}})$. Then, by Equation (36), we have $f_1(\boldsymbol{x}) = \boldsymbol{W}\boldsymbol{x} + \boldsymbol{b}$ and $f_1(\boldsymbol{x}_i) = \boldsymbol{W}\boldsymbol{x}_i + \boldsymbol{b}$ so that

$$\|f_1(\boldsymbol{x}) - f_1(\boldsymbol{x}_i)\|_2 = \|\boldsymbol{W}\boldsymbol{x} - \boldsymbol{W}\boldsymbol{x}_i\|_2 \leq \|\boldsymbol{x} - \boldsymbol{x}_i\|_2 \leq \rho\epsilon_{\mathcal{D}}. \tag{38}$$

Moreover, combining Equations (37) and (38) results $\|f_1(\boldsymbol{x}) - f_1(\boldsymbol{x}_i)\|_2 \leq \frac{5}{6}\epsilon_{\mathcal{D}''}$. Since $f_2$ $\frac{5}{6}$-robustly memorize $\mathcal{D}''$, we have

$$f(\boldsymbol{x}) = f_2(f_1(\boldsymbol{x})) = f_2(f_1(\boldsymbol{x}_i)) = y_i.$$

In particular, $f(\boldsymbol{x}) = y_i$ for any $\boldsymbol{x} \in \mathcal{B}_2(\boldsymbol{x}_i, \rho\epsilon_{\mathcal{D}})$, concluding that $f$ is a $\rho$-robust memorizer $\mathcal{D}$. Regarding the number of parameters to construct $f$, notice that $f_1$ consists of $(d+1)m = \tilde{O}(d^2\rho^2)$

parameters as $m = \tilde{O}(d\rho^2)$. $f_2$ consists of $\tilde{O}(Nm^2) = \tilde{O}(Nd^2\rho^4)$ parameters. Since the case IV assumes $N \geq d$ and large $\rho$ regime deals with $\rho \geq \frac{1}{5\sqrt{d}}$, we have

$$d^2\rho^2 \leq Nd\rho^2 \leq 25Nd^2\rho^4$$

Therefore, $f$ in total consists of $\tilde{O}(d^2\rho^2 + Nd^2\rho^4) = \tilde{O}(Nd^2\rho^4)$ number of parameters. Moreover, since $f$ has the same width as $f_2$ and depth one larger than the depth of $f_2$, it follows that $f$ has width $O(m) = \tilde{O}(\rho^2 d)$ and depth $\tilde{O}(N)$. This proves the theorem for the fourth case.

**Case V:** $\rho \in (1/5\sqrt{d}, 1/3)$**, and** $d > N \geq 600 \log N$**.** The last case combines the two techniques used in Cases III and IV. We first apply Proposition B.21 to $\mathcal{D}$ to obtain 1-Lipschitz linear $\varphi : \mathbb{R}^d \to \mathbb{R}^N$ such that $\mathcal{D}' := \{(\varphi(\boldsymbol{x}_i), y_i)\}_{i\in[N]} \in \boldsymbol{D}_{N,N,C}$ has $\epsilon_{\mathcal{D}'} = \epsilon_{\mathcal{D}}$. Note that we can apply the proposition since $d \geq N$.

Next, we apply Proposition B.19 to $\mathcal{D}' \in \boldsymbol{D}_{N,N,C}$ with $m = \max\{\lceil 9N\rho^2 \rceil, \lceil 600 \log N \rceil\}$ and $\alpha = 1/5$. Let us first check that the specified $m$ satisfies the condition $24\alpha^{-2} \log N \leq m \leq N$ for the proposition to be applied. $\alpha = 1/5$ and $m \geq 600 \log N$ ensure the first inequality $24\alpha^{-2} \log N \leq m$. The second inequality $m \leq N$ is decomposed into two parts. Since $\rho \leq \frac{1}{3}$, we have $9N\rho^2 \leq N$ so that

$$\lceil 9N\rho^2 \rceil \leq N. \tag{39}$$

Moreover, $600 \log N \leq N$ implies

$$\lceil 600 \log N \rceil \leq N. \tag{40}$$

Gathering Equations (30) and (31) proves $m \leq N$. Additionally, as $N \geq 2$, we have $N \geq 600 \log N \geq 600 \log 2 \geq 400$. By Lemma B.22, this implies $10 \log N \leq N$.

By the Proposition B.19, there exists 1-Lipschitz linear mapping $\phi : \mathbb{R}^N \to \mathbb{R}^m$ and $\beta > 0$ such that $\mathcal{D}'' := \{(\phi(\varphi(\boldsymbol{x}_i)), y_i)\}_{i\in[N]} \in \boldsymbol{D}_{m,N,C}$ satisfies

$$\epsilon_{\mathcal{D}''} \geq \frac{4}{5}\beta\epsilon_{\mathcal{D}}'. \tag{41}$$

As $m \geq 600 \log N \geq 10 \log N$, the inequality $\beta \geq \frac{1}{2}\sqrt{\frac{m}{N}}$ is also satisfied by Proposition B.19. Therefore, we have

$$\beta \geq \frac{1}{2}\sqrt{\frac{m}{N}} \overset{(a)}{\geq} \frac{1}{2}\sqrt{\frac{\lceil 9N\rho^2 \rceil}{N}} \geq \frac{1}{2}\sqrt{\frac{9N\rho^2}{N}} = \frac{3}{2}\rho, \tag{42}$$

where (a) is by the definition of $m$. Moreover, since $\varphi$ and $\phi$ are both 1-Lipschitz linear, $\phi \circ \varphi : \mathbb{R}^d \to \mathbb{R}^m$ is also 1-Lipschitz linear. Therefore,

$$\|\phi(\varphi(\boldsymbol{x}_i))\|_2 = \|\phi(\varphi(\boldsymbol{x}_i - \boldsymbol{0}))\|_2 = \|\phi(\varphi(\boldsymbol{x}_i)) - \phi(\varphi(\boldsymbol{0}))\|_2 \leq \|\boldsymbol{x}_i - \boldsymbol{0}\|_2 = \|\boldsymbol{x}_i\|_2, \tag{43}$$

for all $i \in [N]$. Hence, by letting $R := \max_{i\in[N]}\{\|\boldsymbol{x}_i\|_2\}$, we have $\|\phi(\varphi(\boldsymbol{x}_i))\|_2 \leq R$ for all $i \in [N]$.

Now, we set the first layer hidden matrix as the matrix $\boldsymbol{W} \in \mathbb{R}^{m \times d}$ corresponding to $\phi \circ \varphi$ under the standard basis of $\mathbb{R}^d$ and $\mathbb{R}^m$. Moreover, set the first hidden layer bias as $\boldsymbol{b} := 2R\boldsymbol{1} = 2R(1, 1, \cdots, 1) \in \mathbb{R}^m$. Then, we have

$$\boldsymbol{W}\boldsymbol{x} + \boldsymbol{b} \geq \boldsymbol{0}, \tag{44}$$

for all $\boldsymbol{x} \in \mathcal{B}_2(\boldsymbol{x}_i, \epsilon_{\mathcal{D}})$ for all $i \in [N]$, where the comparison between two vectors are element-wise. This is because for all $i \in [N], j \in [m]$ and $\boldsymbol{x} \in \mathcal{B}_2(\boldsymbol{x}, \epsilon_{\mathcal{D}})$, we have

$$(\boldsymbol{W}\boldsymbol{x} + \boldsymbol{b})_j = (\boldsymbol{W}\boldsymbol{x})_j + 2R \geq 2R - \|\boldsymbol{W}\boldsymbol{x}\|_2 \overset{(a)}{\geq} 2R - \|\boldsymbol{x}\|_2 \overset{(b)}{\geq} 2R - (R + \epsilon_{\mathcal{D}}) \overset{(c)}{\geq} 0,$$

where (a) is by Equation (43), (b) is by the triangle inequality, and (c) is due to $R \geq \epsilon_{\mathcal{D}}$.

We construct the first layer of the neural network as $f_1(\boldsymbol{x}) := \sigma(\boldsymbol{W}\boldsymbol{x} + \boldsymbol{b})$ which includes the activation $\sigma$. Next, we show that, $\mathcal{D}'' := \{(f_1(\boldsymbol{x}_i), y_i)\}_{i\in[N]}$ satisfies

$$\epsilon_{\mathcal{D}''} \geq \frac{6}{5}\rho\epsilon_{\mathcal{D}}, \tag{45}$$

by the above properties. This is because for $i \neq j$ with $y_i \neq y_j$ we have

$$\|f_1(\boldsymbol{x}_i) - f_1(\boldsymbol{x}_j)\|_2 = \|\sigma(\boldsymbol{W}\boldsymbol{x}_i + \boldsymbol{b}) - \sigma(\boldsymbol{W}\boldsymbol{x}_j + \boldsymbol{b})\|_2$$

$$\stackrel{(a)}{=} \|(\boldsymbol{W}\boldsymbol{x}_i + \boldsymbol{b}) - (\boldsymbol{W}\boldsymbol{x}_j + \boldsymbol{b})\|_2$$

$$= \|\phi(\varphi(\boldsymbol{x}_i)) - \phi(\varphi(\boldsymbol{x}_k))\|_2$$

$$\stackrel{(b)}{\geq} 2\epsilon_{\mathcal{D}''}$$

$$\stackrel{(c)}{\geq} 2 \times \frac{4}{5}\beta\epsilon'_{\mathcal{D}}$$

$$\stackrel{(d)}{\geq} 2 \times \frac{4}{5} \times \frac{3}{2}\rho\epsilon'_{\mathcal{D}}$$

$$= \frac{12}{5}\rho\epsilon'_{\mathcal{D}}$$

$$\stackrel{(e)}{=} \frac{12}{5}\rho\epsilon_{\mathcal{D}},$$

where (a) is by Equation (44), (b) is by the definition of the $\epsilon_{\mathcal{D}''}$, (c) is by Equation (41), (d) is by Equation (42), and (e) is because $\epsilon_{\mathcal{D}'} = \epsilon_{\mathcal{D}}$.

By Lemma D.2 applied to $\mathcal{D}'' \in \boldsymbol{D}_{m,N,C}$, there exists $f_2 \in \mathcal{F}_{m,P}$ with $P = O(Nm^2(\log(\frac{m}{(\gamma'')^2}) + \log R''))$ number of parameters that $\frac{5}{6}$-robustly memorize $\mathcal{D}''$, where

$$\gamma'' := (1 - \frac{5}{6})\epsilon_{\mathcal{D}''} \stackrel{(a)}{\geq} \frac{1}{6} \times \frac{12}{5}\rho\epsilon_{\mathcal{D}} = \frac{2}{5}\rho\epsilon_{\mathcal{D}},$$

$$R'' := \max_{i \in [N]} \|f_1(\boldsymbol{x}_i)\|_2 = \max_{i \in [N]} \|\sigma(\boldsymbol{W}\boldsymbol{x}_i + \boldsymbol{b})\|_2 = \max_{i \in [N]} \|\boldsymbol{W}\boldsymbol{x}_i + \boldsymbol{b}\|_2$$

$$\leq \max_{i \in [N]} \|\boldsymbol{W}\boldsymbol{x}_i\|_2 + \|\boldsymbol{b}\|_2 \leq 3R.$$

Here, (a) is by Equation (45). Moreover, $f_2$ has width $O(m)$ and depth $\tilde{O}(N)$ by Lemma D.2.

Now, we claim that $f := f_2 \circ f_1$ $\rho$-robustly memorize $\mathcal{D}$. For any $i \in [N]$, take $\boldsymbol{x} \in \mathcal{B}_2(\boldsymbol{x}_i, \rho\epsilon_{\mathcal{D}})$. Then, by Equation (44), we have $f_1(\boldsymbol{x}) = \boldsymbol{W}\boldsymbol{x} + \boldsymbol{b}$ and $f_1(\boldsymbol{x}_i) = \boldsymbol{W}\boldsymbol{x}_i + \boldsymbol{b}$ so that

$$\|f_1(\boldsymbol{x}) - f_1(\boldsymbol{x}_i)\|_2 = \|\boldsymbol{W}\boldsymbol{x} - \boldsymbol{W}\boldsymbol{x}_i\|_2 \leq \|\boldsymbol{x} - \boldsymbol{x}_i\|_2 \leq \rho\epsilon_{\mathcal{D}}. \tag{46}$$

Moreover, putting Equation (45) to Equation (46) results $\|f_1(\boldsymbol{x}) - f_1(\boldsymbol{x}_i)\|_2 \leq \frac{5}{6}\epsilon_{\mathcal{D}''}$. Since $f_2$ $\frac{5}{6}$-robustly memorize $\mathcal{D}''$, we have

$$f(\boldsymbol{x}) = f_2(f_1(\boldsymbol{x})) = f_2(f_1(\boldsymbol{x}_i)) = y_i.$$

In particular, $f(\boldsymbol{x}) = y_i$ for any $\boldsymbol{x} \in \mathcal{B}_2(\boldsymbol{x}_i, \rho\epsilon_{\mathcal{D}})$, concluding that $f$ is a $\rho$-robust memorizer $\mathcal{D}$.

Regarding the number of parameters to construct $f$, notice that $f_1$ consists of $(d+1)m = \tilde{O}(Nd\rho^2)$ parameters as $m = \tilde{O}(N\rho^2)$. $f_2$ consists of $\tilde{O}(Nm^2) = \tilde{O}(N^3\rho^4)$ parameters. Since the case V assumes $N < d$ and large $\rho$ regime deals with $\rho \geq \frac{1}{5\sqrt{d}}$, we have

$$Nd\rho^2 \leq 25Nd^2\rho^4,$$

$$N^3\rho^4 \leq Nd^2\rho^4.$$

Therefore, $f$ in total consists of $\tilde{O}(N^3\rho^4 + Nd\rho^2) = \tilde{O}(Nd^2\rho^4)$ number of parameters. Moreover, since $f$ has the same width as $f_2$ and depth one larger than the depth of $f_2$, it follows that width of $f$ is $O(m) = \tilde{O}(\rho^2 N) = \tilde{O}(\rho^2 d)$ and the depth of $f$ is $\tilde{O}(N)$. This proves the theorem for the last case.

**Bounding the Bit Complexity.** Now, let us analyze how we can implement the above network under a finite precision. We have demonstrated that for every five cases, the depth $\tilde{O}(N)$ and width $\rho^2 d$ suffice for constructing $f$ that robustly memorizes $\mathcal{D}$.

Let $R := \max_{i \in [N]} \|\boldsymbol{x}_i\|_2 + \mu$. Let $D = \tilde{O}(\rho^2 d)$ and $L = \tilde{O}(N)$ denote the width and the depth of the constructed network. Let $M$ be the maximum absolute value of the parameter used for constructing $f$. Finally let $\nu = 0.1$. By Lemma B.23, there exists $\bar{f}$ with $\tilde{O}(N)$ bit complexity, that approximates $f$ uniformly over $\mathcal{B}_2(\mathbf{0}, R)$ with error at most $\nu$, where $\tilde{O}(\cdot)$ here hides polylogarithmic terms in $D, M, L$ and $R$. i.e.

$$\max_{\|\boldsymbol{x}\|_2 \leq R} \left|\bar{f} - f\right| \leq \nu.$$

Finally, to handle the error $\nu$, we use the floor function approximation from Lemma B.16. By Lemma B.16 with $\gamma = 1/10$, there exists $\overline{\text{Floor}} : \mathbb{R} \to \mathbb{R}$ with depth $n := \lceil \log_2(C+1) \rceil$ and width $5$ such that $\overline{\text{Floor}}(x) = \lfloor x \rfloor$ for all $x \in [0, C+1)$ with $x - \lfloor x \rfloor > \gamma = 0.1$. Moreover, the lemma guarantees that $\overline{\text{Floor}}$ can be exactly implemented with $O(n + \log 10) = O(\log C) = \tilde{O}(1)$ bit complexity.

Thus, if $y' \in \mathbb{R}$ satisfies $|y' - y| \leq \nu = 0.1$ for some $y \in [C]$, then

$$y' + 0.5 \in [y - 0.1 + 0.5, y + 0.1 + 0.5] \subseteq (y + 0.1, y + 1).$$

In particular, $\lfloor y' + 0.5 \rfloor = y$ and $\lfloor y' + 0.5 \rfloor - (y' + 0.5) \in (0.1, 1)$ so that $\text{Floor}(y') = \lfloor y' \rfloor = y$. For $\boldsymbol{x} \in \mathcal{B}_2(\boldsymbol{x}_i, \mu)$, we have $f(\boldsymbol{x}) = y_i$ so that the approximation $\bar{f}$ outputs $y' = \bar{f}(\boldsymbol{x})$ such that

$$\left|\bar{f}(\boldsymbol{x}) - f(\boldsymbol{x})\right| = |y' - y_i| \leq \nu.$$

This shows $\overline{\text{Floor}}(\bar{f}(\boldsymbol{x})) = \overline{\text{Floor}}(y') = y_i$. Moreover, $\overline{\text{Floor}} \circ \bar{f}$ can be implemented with parameters, width, and depth of the same scale as $\bar{f}$, and bit complexity $\tilde{O}(N)$. This finishes the proof for the bit complexity. $\qquad \square$

## B.4 Lemmas for Lattice Mapping

**Lemma B.15** (Avoiding Being Near Grid). *Let $N, d \in \mathbb{N}$ and $\boldsymbol{x}_1, \cdots, \boldsymbol{x}_N \in \mathbb{R}^d$. Then, there exists a translation vector $\boldsymbol{b} \in \mathbb{R}^d$ such that:*

$$\text{dist}(x_{i,j} - b_j, \mathbb{Z}) \geq \frac{1}{2N}, \quad \forall i \in [N], j \in [d],$$

*i.e., the translated points $\{\boldsymbol{x}_i - \boldsymbol{b}\}_{i \in [N]}$ are coordinate-wise $\frac{1}{2N}$-far from the integer lattice.*

*Moreover, there exists $\bar{\boldsymbol{b}} \in \mathbb{R}^d$ which has bit complexity $\lceil \log(6N) \rceil$ and satisfies*

$$\text{dist}(x_{i,j} - b_j, \mathbb{Z}) \geq \frac{1}{3N}, \quad \forall i \in [N], j \in [d].$$

*Proof.* For each coordinate $j \in [d]$, consider the set $\{x_{i,j}\}_{i \in [N]}$ of all $j$-th coordinate values. For $x \in \mathbb{R}$, let $\{x\} := x - \lfloor x \rfloor$ denote the fractional part of $x$. We consider the collection of fractional parts $\{\{x_{i,j}\}\}_{i \in [N]}$. Without loss of generality, we may assume $0 \leq \{x_{1,j}\} < \{x_{2,j}\} < \cdots < \{x_{N,j}\} < 1$.

For each $j \in [d]$, define the maximum fractional gap $g_j \in [0, 1)$ as

$$g_j := \max\left( \max_{i \in [N-1]} (\{x_{i+1,j}\} - \{x_{i,j}\}), \ 1 - \{x_{N,j}\} + \{x_{1,j}\} \right).$$

We claim:

$$g_j \geq \frac{1}{N}, \text{ for all } j \in [d].$$

Suppose for a contradiction that $g_j < \frac{1}{N}$ for some $j \in [d]$. Then, we have by the definition of $g_j$,

$$\{x_{N,j}\} - \{x_{1,j}\} = \sum_{i=1}^{N-1} (\{x_{i+1,j}\} - \{x_{i,j}\}) \leq (N-1)g_j < \frac{N-1}{N}, \tag{47}$$

$$1 - \{x_{N,j}\} + \{x_{1,j}\} \leq g_j < \frac{1}{N} \iff \{x_{N,j}\} - \{x_{1,j}\} > \frac{N-1}{N}. \tag{48}$$

Equations (47) and (48) lead to a contradiction, proving the claim.

Now, we define the translation of the $j$-th coordinate, $b_j \in \mathbb{R}$, based on the location where the maximum $g_j$ is attained. If the maximum in the definition of $g_j$ occurs by the difference of some consecutive pair $(\{x_{i',j}\}, \{x_{i'+1,j}\})$ satisfying $\{x_{i'+1,j}\} - \{x_{i',j}\} = g_j$, we set

$$b_j = \frac{\{x_{i',j}\} + \{x_{i'+1,j}\}}{2}.$$

In this case, $\mathrm{dist}(x_{i',j} - b_j, \mathbb{Z}) = \mathrm{dist}(x_{i'+1,j} - b_j, \mathbb{Z}) = \frac{1}{2}g_j \geq \frac{1}{2N}$. For other $i$, $\mathrm{dist}(x_{i,j} - b_j, \mathbb{Z})$ is even larger by the order relation within $\{\{x_{i,j}\}\}_{i \in [N]}$.

Otherwise, if the maximum in the definition of $g_j$ is attained as $1 - \{x_{N,j}\} + \{x_{1,j}\} = g_j$, we define

$$b_j = \frac{(\{x_{N,j}\} - 1) + \{x_{1,j}\}}{2}.$$

In this case, $\mathrm{dist}(x_{1,j} - b_j, \mathbb{Z}) = \mathrm{dist}(x_{N,j} - b_j, \mathbb{Z}) = \frac{1}{2}g_j \geq \frac{1}{2N}$. For other $i$, $\mathrm{dist}(x_{i,j} - b_j, \mathbb{Z})$ is even larger by the order relation within $\{\{x_{i,j}\}\}_{i \in [N]}$.

We define the full translation vector $\boldsymbol{b} = (b_1, \ldots, b_d) \in \mathbb{R}^d$. Then the translated points $\{\boldsymbol{x}_i - \boldsymbol{b}\}_{i \in [N]}$ satisfy:

$$\mathrm{dist}(x_{i,j} - b_j, \mathbb{Z}) \geq \frac{1}{2N}, \quad \text{for all } i \in [N], j \in [d].$$

Intuitively, $b_j$ is chosen as the midpoint of the widest gap between fractional values, ensuring that all fractional parts after the translation are at least $\frac{g_j}{2}$ away from the nearest integer. Therefore, the translated points are coordinate-wise $\frac{1}{2N}$-far from lattice points.

We define $\bar{\boldsymbol{b}}$ such that each of its coordinates is equal to the first $\lceil \log(6N) \rceil$ bits of the corresponding coordinate of $\boldsymbol{b}$. Then, for all $j \in [d]$, we have $|b_j - \bar{b}_j| \leq \frac{1}{2^{\log(6N)}} \leq \frac{1}{6N}$. Using $\bar{\boldsymbol{b}}$ with bit complexity $\lceil \log(6N) \rceil$, we can still ensure the distance $\frac{1}{3N}$ from the lattice points.

$$\mathrm{dist}(x_{i,j} - \bar{b}_j, \mathbb{Z}) \geq \mathrm{dist}(x_{i,j} - b_j, \mathbb{Z}) - |b_j - \bar{b}_j| \geq \frac{1}{2N} - \frac{1}{6N} = \frac{1}{3N}, \quad \text{for all } i \in [N], j \in [d].$$

$\square$

The following lemma shows that we can approximate the floor function using a logarithmic number of ReLU units with respect to the length of the interval of interest.

**Lemma B.16** (Floor Function Approximation). *For any $n \in \mathbb{N}$ and any $\gamma \in (0,1)$, there exists an $n$-layer network $\overline{\mathrm{Floor}}_n$ with width 5 and $5n$ ReLU units such that*

$$\overline{\mathrm{Floor}}_n(x) = \lfloor x \rfloor \text{ for all } x \in [0, 2^n) \text{ such that } x - \lfloor x \rfloor > \gamma.$$

*Moreover, if $\gamma = \frac{1}{t}$ for some $t \in \mathbb{N}$, then $\overline{\mathrm{Floor}}_n$ can be exactly implemented with $2n + \log_2 t$ bit complexity under a fixed point precision.*

*Proof.* To reconcile the discontinuity of the floor function with the continuity of ReLU networks, we first define a discontinuous ideal building block that exactly replicates the floor function on the target interval $[0, 2^n)$. We then approximate this building block using a continuous neural network with ReLU activations.

The ideal building block $\Delta$ is defined as:

$$\Delta(x) := \begin{cases} 2x & \text{if } x \in (0, \frac{1}{2}], \\ 2x - 1 & \text{if } x \in (\frac{1}{2}, 1], \\ 0 & \text{otherwise.} \end{cases}$$

For $n \in \mathbb{N}$, define the function $\mathrm{Floor}_n$ by:

$$\mathrm{Floor}_n(x) = \Delta^n(-\frac{x}{2^n} + 1) + x - 1.$$

We will show by induction that $\mathrm{Floor}_n = \lfloor x \rfloor$ for all $x \in [0, 2^n)$.

For the base case $n = 1$,

$$\mathrm{Floor}_1(x) = \Delta(-\frac{x}{2} + 1) + x - 1 = \begin{cases} 2(-\frac{x}{2} + 1) - 1 + x - 1 = 0 & \text{if } x \in [0, 1), \\ 2(-\frac{x}{2} + 1) + x - 1 = 1 & \text{if } x \in [1, 2), \\ 0 + x - 1 = x - 1 & \text{otherwise.} \end{cases}$$

This proves the base case: for all $x \in [0, 2)$, we have $\mathrm{Floor}_1(x) = \lfloor x \rfloor$.

For the inductive step, assume that $\mathrm{Floor}_n(x) = \lfloor x \rfloor$ holds for all $x \in [0, \ 2^n)$. We aim to prove that $\mathrm{Floor}_{n+1}(x) = \lfloor x \rfloor$ for all $x \in [0, \ 2^{n+1})$.

Recall that:

$$\Delta(-\frac{x}{2^{n+1}} + 1) = \begin{cases} -\dfrac{x}{2^n} + 1 & \text{if } x \in [0, 2^n) & (\Leftrightarrow -\dfrac{x}{2^{n+1}} + 1 \in (\tfrac{1}{2}, 1]), \\ -\dfrac{x}{2^n} + 2 = -\dfrac{x - 2^n}{2^n} + 1 & \text{if } x \in [2^n, 2^{n+1}) & (\Leftrightarrow -\dfrac{x}{2^{n+1}} + 1 \in (0, \tfrac{1}{2}]), \\ 0 & \text{otherwise.} \end{cases}$$

Thus, we have

$$\mathrm{Floor}_{n+1}(x) = \Delta^{n+1}(-\frac{x}{2^{n+1}} + 1) + x - 1$$

$$= \Delta^n(\Delta(-\frac{x}{2^{n+1}} + 1)) + x - 1$$

$$= \begin{cases} \Delta^n(-\frac{x}{2^n} + 1) + x - 1 = \mathrm{Floor}_n(x) = \lfloor x \rfloor & \text{if } x \in [0, 2^n), \\ \Delta^n(-\frac{x - 2^n}{2^n} + 1) + x - 1 = \mathrm{Floor}_n(x - 2^n) + 2^n = \lfloor x \rfloor & \text{if } x \in [2^n, 2^{n+1}), \\ \Delta^n(0) + x - 1 = x - 1 & \text{otherwise.} \end{cases}$$

Therefore, by induction,
$$\mathrm{Floor}_n = \lfloor x \rfloor \text{ for all } x \in [0, 2^n).$$

Next, we define the $\sigma$ approximation $\overline{\Delta_{\gamma,n}}$ of the discontinuous block $\Delta$ as:

$$\overline{\Delta_{\gamma,n}}(x) := 2\sigma(x) - \frac{1}{\gamma_n}\sigma\left(x - \frac{1}{2} + \gamma_n\right) + \frac{1}{\gamma_n}\sigma\left(x - \frac{1}{2}\right)$$

$$- \frac{1}{\gamma_n}\sigma(x - 1 + \gamma_n) + \left(\frac{1}{\gamma_n} - 2\right)\sigma(x - 1), \tag{49}$$

where $\gamma_n = \frac{\gamma}{2^n}$. Check Figure 7 for an illustration of how $\overline{\Delta_{\gamma,n}}$ looks like on $[0, 1]$. It is straightforward to check that

$$\overline{\Delta_{\gamma,n}}(x) = \Delta(x) \text{ for all } x \in [0, \frac{1}{2} - \gamma_n] \cup [\frac{1}{2}, 1 - \gamma_n].$$

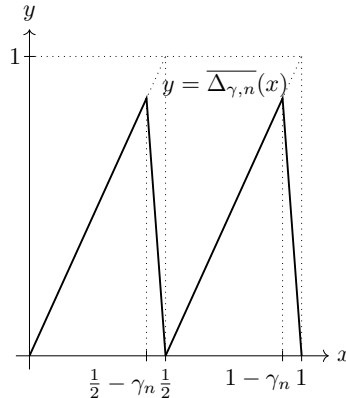

Figure 7: Plot of the ReLU-based approximation $\overline{\Delta_{\gamma,n}}(x)$ of the ideal discontinuous building block $\Delta(x)$ on $[0, 1]$.

We now explain why this approximation remains valid under recursive composition up to depth $n$.

Let us define the variable $x' := -\frac{x}{2^n} + 1$, so that $x = 2^n(1 - x')$ and $x' \in (0, 1]$. Our target function is:
$$\text{Floor}_n(x) = \Delta^n\left(-\frac{x}{2^n} + 1\right) + x - 1 = \Delta^n(x') + x - 1.$$

We are given the assumption $x - \lfloor x \rfloor > \gamma$, and we aim to express this in terms of $x'$ to ensure $\overline{\Delta_{\gamma,n}}^n(x') = \Delta^n(x')$. We proceed step-by-step:

$$x - \lfloor x \rfloor > \gamma$$
$$\Longleftrightarrow 2^n(1 - x') - \lfloor 2^n(1 - x') \rfloor > \gamma$$
$$\Longleftrightarrow -2^n x' - \lfloor -2^n x' \rfloor > \gamma$$
$$\Longleftrightarrow -2^n x' + \lceil 2^n x' \rceil > \gamma$$
$$\Longleftrightarrow 2^n x' < \lceil 2^n x' \rceil - \gamma$$
$$\Longleftrightarrow 2^n x' \in (\lceil 2^n x' \rceil - 1, \lceil 2^n x' \rceil - \gamma)$$
$$\Longleftrightarrow 2^n x' \in \bigcup_{k \in \mathbb{Z}} (k - 1, k - \gamma)$$
$$\Longleftrightarrow x' \in \bigcup_{k \in \mathbb{Z}} \left(\frac{k - 1}{2^n}, \frac{k - \gamma}{2^n}\right).$$

Since $x' \in (0, 1]$, we only need to consider $k \in [2^n]$, i.e.,
$$x' \in \bigcup_{k \in [2^n]} \left(\frac{k - 1}{2^n}, \frac{k - \gamma}{2^n}\right).$$

We will now prove by induction on $n$ the following statement:

For any $\gamma \in (0, 1)$, $\overline{\Delta_{\gamma,n}}^n(x) = \Delta^n(x)$ for $x \in \bigcup_{k \in [2^n]} \left(\frac{k - 1}{2^n}, \frac{k - \gamma}{2^n}\right)$.

For the base case $n = 1$, by construction of $\overline{\Delta_{\gamma,1}}(x)$, we know $\overline{\Delta_{\gamma,1}}(x) = \Delta(x)$ for all $x \in [0, \frac{1}{2} - \frac{\gamma}{2}] \cup [\frac{1}{2}, 1 - \frac{\gamma}{2}]$, which contains the union $\bigcup_{k \in [2]} \left(\frac{k-1}{2}, \frac{k-\gamma}{2}\right)$. Hence the base case holds.

For the inductive step, assume the claim holds for $n$. We show it holds for $n + 1$. By using $\gamma/2$ in place of $\gamma$ for the inductive hypothesis, we have

$$\overline{\Delta_{\gamma/2,n}}^n(x) = \overline{\Delta_{\gamma,n+1}}^n(x) = \Delta^n(x) \text{ for all } x \in \bigcup_{k \in [2^n]} \left(\frac{k - 1}{2^n}, \frac{k - \gamma/2}{2^n}\right). \tag{50}$$

Let $x \in \bigcup_{k \in [2^{n+1}]} \left(\frac{k-1}{2^{n+1}}, \frac{k-\gamma}{2^{n+1}}\right)$. We analyze two cases based on $x \in [0, \frac{1}{2})$ or $x \in [\frac{1}{2}, 1)$.

First, consider the case $x \in \bigcup_{k \in [2^n]} \left(\frac{k-1}{2^{n+1}}, \frac{k-\gamma}{2^{n+1}}\right) \subset [0, \frac{1}{2})$. Then $x < \frac{k-\gamma}{2^{n+1}} \leq \frac{1}{2} - \gamma_{n+1}$, so $\overline{\Delta_{\gamma,n+1}}(x) = 2x$. Let $y := 2x$. Then:

$$y \in \bigcup_{k \in [2^n]} \left(\frac{k - 1}{2^n}, \frac{k - \gamma}{2^n}\right) \subseteq \bigcup_{k \in [2^n]} \left(\frac{k - 1}{2^n}, \frac{k - \gamma/2}{2^n}\right).$$

Thus, we have

$$\overline{\Delta_{\gamma,n+1}}^{n+1}(x) = \overline{\Delta_{\gamma,n+1}}^n(\overline{\Delta_{\gamma,n+1}}(x))$$
$$= \overline{\Delta_{\gamma,n+1}}^n(2x)$$
$$= \overline{\Delta_{\gamma,n+1}}^n(y)$$
$$\overset{(a)}{=} \Delta^n(y)$$

$$=\Delta^n(2x)$$
$$=\Delta^{n+1}(x),$$

where the equation (a) follows by Equation (50).

Second, consider the case $x \in \bigcup_{k \in [2^{n+1}] \setminus [2^n]} \left( \frac{k-1}{2^{n+1}}, \frac{k-\gamma}{2^{n+1}} \right) \subset [\frac{1}{2}, 1)$. Then $\frac{1}{2} \le x < \frac{k-\gamma}{2^{n+1}} \le 1 - \gamma_{n+1}$, so $\overline{\Delta_{\gamma,n+1}}(x) = 2x - 1$. Let $y := 2x - 1$. Then:

$$y \in \bigcup_{k \in [2^{n+1}] \setminus [2^n]} \left( \frac{k - 2^n - 1}{2^n}, \frac{k - 2^n - \gamma}{2^n} \right) = \bigcup_{k \in [2^n]} \left( \frac{k-1}{2^n}, \frac{k-\gamma}{2^n} \right) \subseteq \bigcup_{k \in [2^n]} \left( \frac{k-1}{2^n}, \frac{k-\gamma/2}{2^n} \right).$$

Thus, we have

$$\overline{\Delta_{\gamma,n+1}}^{n+1}(x) = \overline{\Delta_{\gamma,n+1}}^{n}(\overline{\Delta_{\gamma,n+1}}(x))$$
$$= \overline{\Delta_{\gamma,n+1}}^{n}(2x - 1)$$
$$= \overline{\Delta_{\gamma,n+1}}^{n}(y)$$
$$\overset{(a)}{=} \Delta^n(y)$$
$$= \Delta^n(2x - 1)$$
$$= \Delta^{n+1}(x),$$

where the equation (a) follows by Equation (50).

Therefore, by induction, we have shown that for any $\gamma \in (0, 1)$ and any $n \in \mathbb{N}$,

$$\overline{\Delta_{\gamma,n}}^{n}(x') = \Delta^n(x') \quad \text{for all } x' \in \bigcup_{k \in [2^n]} \left( \frac{k-1}{2^n}, \frac{k-\gamma}{2^n} \right).$$

We now define the ReLU-based floor approximation by

$$\overline{\text{Floor}_n}(x) := \overline{\Delta_{\gamma,n}}^{n} \left( -\frac{x}{2^n} + 1 \right) + x - 1. \tag{51}$$

Recall that the ideal target function is given by

$$\text{Floor}_n(x) = \Delta^n \left( -\frac{x}{2^n} + 1 \right) + x - 1.$$

Let us denote $x' := -\frac{x}{2^n} + 1$. When $x - \lfloor x \rfloor > \gamma$, the value $x'$ satisfies

$$x' \in \bigcup_{k \in [2^n]} \left( \frac{k-1}{2^n}, \frac{k-\gamma}{2^n} \right),$$

so that $\overline{\Delta_{\gamma,n}}^{n}(x') = \Delta^n(x')$ by the result above. Therefore, we conclude:

$$\overline{\text{Floor}_n}(x) = \text{Floor}_n(x) = \lfloor x \rfloor \quad \text{for all } x \in [0, 2^n) \text{ such that } x - \lfloor x \rfloor > \gamma.$$

Finally to prove the additional statement regarding the bit complexity, consider the case $\gamma = \frac{1}{t}$ for some $t \in \mathbb{N}$. By Equation (51), the bit complexity to implement $\overline{\text{Floor}_n}$ is upper bounded by $n$ plus the bit complexity to implement $\overline{\Delta_{\gamma,n}}$. Now, it suffices to consider the bit complexity required to implement $\overline{\Delta_{\gamma,n}}$. From Equation (49), observe that $1/\gamma_n = 2^n/\gamma = 2^n \times t$ for $\gamma = 1/t$. Since $t \in \mathbb{N}$, this can be exactly implemented with $\log(2^n \times t) = n + \log_2 t$ bit complexity. Thus, $\overline{\text{Floor}_n}$ can be implemented exactly with $2n + \log_2 t$ bit complexity. $\qquad \square$

## B.5 Dimension Reduction via Careful Analysis of the Johnson-Lindenstrauss Lemma

We begin with a lemma that states a concentration of the length of the projection.

**Lemma B.17** (Lemma 15.2.2, Matousek [2013]). *For a unit vector $\boldsymbol{x} \in S^{d-1}$, let*

$$\phi(\boldsymbol{x}) = (x_1, x_2, \cdots, x_m)$$

*be the mapping of $\boldsymbol{x}$ onto the subspace spanned by the first $m$ coordinates. Consider $\boldsymbol{x} \in S^{d-1}$ chosen uniformly at random. Then, there exists $\beta$ such that $\|\phi(\boldsymbol{x})\|_2$ is sharply concentrated around $\beta$,*

$$\mathbb{P}[\|\phi(\boldsymbol{x})\|_2 \geq \beta + t] \leq 2e^{-t^2 d/2} \text{ and } \mathbb{P}[\|\phi(\boldsymbol{x})\|_2 \leq \beta - t] \leq 2e^{-t^2 d/2},$$

*where for $m \geq 10 \log d$, we have $\beta \geq \frac{1}{2}\sqrt{\frac{m}{d}}$.*

Based on the above concentration inequality, we state the Johnson-Lindenstrauss lemma, in a version which reflects the benefit on the ratio of the norm preserved when the projecting dimension increases. The proof follows that of Theorem 15.2.1 in Matousek [2013] with a slight modification.

**Lemma B.18** (Strengthened Version of the Johnson-Lindenstrauss Lemma). *For $N \geq 2$, let $X \subseteq \mathbb{R}^d$ be an $N$ point set. Then, for any $\alpha \in (0,1)$ and $24\alpha^{-2} \log N \leq m \leq d$, there exists a 1-Lipschitz linear mapping $\phi : \mathbb{R}^d \to \mathbb{R}^m$ and $\beta > 0$ such that*

$$(1-\alpha)\beta \|\boldsymbol{x} - \boldsymbol{x}'\|_2 \leq \|\phi(\boldsymbol{x}) - \phi(\boldsymbol{x}')\|_2 \leq (1+\alpha)\beta \|\boldsymbol{x} - \boldsymbol{x}'\|_2, \tag{52}$$

*for all $\boldsymbol{x}, \boldsymbol{x}' \in X$. Moreover, $\beta \geq \frac{1}{2}\sqrt{\frac{m}{d}}$ whenever $m \geq 10 \log d$.*

*Proof.* If $\boldsymbol{x} = \boldsymbol{x}'$, the inequality trivially holds for any $\phi$. Hence, it suffices to find $\phi$ that satisfies Equation (52) for all $\boldsymbol{x}, \boldsymbol{x}' \in X$ with $\boldsymbol{x} \neq \boldsymbol{x}'$. Consider a random $m$-dimensional subspace $L$, and $\phi$ be a projection onto $L$. For any fixed $\boldsymbol{x} \neq \boldsymbol{x}' \in X$, Lemma B.17 implies that $\left\|\phi(\frac{\boldsymbol{x}-\boldsymbol{x}'}{\|\boldsymbol{x}-\boldsymbol{x}'\|_2})\right\|_2$ is concentrated around some constant $\beta$. i.e.

$$\mathbb{P}\left[\left\|\phi\left(\frac{\boldsymbol{x}-\boldsymbol{x}'}{\|\boldsymbol{x}-\boldsymbol{x}'\|_2}\right)\right\|_2 \geq (1+\alpha)\beta\right] \leq 2e^{-\alpha^2\beta^2 d/2} \overset{(a)}{\leq} 2e^{-\alpha^2 m/8} \overset{(b)}{\leq} 2e^{-3\log N} = \frac{2}{N^3} \overset{(c)}{\leq} \frac{1}{N^2},$$

where we use $\beta \geq \frac{1}{2}\sqrt{\frac{m}{d}}$ at (a), $m \geq 24\alpha^{-2} \log N$ at (b), and $N \geq 2$ at (c). Similarly,

$$\mathbb{P}\left[\left\|\phi\left(\frac{\boldsymbol{x}-\boldsymbol{x}'}{\|\boldsymbol{x}-\boldsymbol{x}'\|_2}\right)\right\|_2 \leq (1-\alpha)\beta\right] \leq \frac{1}{N^2}.$$

By linearity of $\phi$, we have $\phi(\boldsymbol{x} - \boldsymbol{x}') = \phi(\boldsymbol{x}) - \phi(\boldsymbol{x}')$. Taking the union bound over the two probability bounds above, the following event happens with probability at most $2/N^2$:

$$\|\phi(\boldsymbol{x}) - \phi(\boldsymbol{x}')\|_2 \geq (1+\alpha)\beta \|\boldsymbol{x} - \boldsymbol{x}'\|_2 \text{ or } \|\phi(\boldsymbol{x}) - \phi(\boldsymbol{x}')\|_2 \leq (1-\alpha)\beta \|\boldsymbol{x} - \boldsymbol{x}'\|_2. \tag{53}$$

Next, we take a union bound over all $\frac{N(N-1)}{2}$ pairs $\boldsymbol{x}, \boldsymbol{x}' \in X$ with $\boldsymbol{x} \neq \boldsymbol{x}'$. Then, the probability that Equation (53) happens for any $\boldsymbol{x}, \boldsymbol{x}' \in X$ with $\boldsymbol{x} \neq \boldsymbol{x}'$ is at most $\frac{2}{N^2} \times \frac{N(N-1)}{2} = 1 - \frac{1}{N} < 1$. Hence, there exists a $m$-dimensional subspace $L$ such that Equation (53) does not hold for any pair of $\boldsymbol{x}, \boldsymbol{x}' \in X$. In other words, there exists a $m$-dimensional subspace $L$ such that

$$(1-\alpha)\beta \|\boldsymbol{x} - \boldsymbol{x}'\|_2 \leq \|\phi(\boldsymbol{x}) - \phi(\boldsymbol{x}')\|_2 \leq (1+\alpha)\beta \|\boldsymbol{x} - \boldsymbol{x}'\|_2,$$

for all $\boldsymbol{x} \neq \boldsymbol{x}'$. By Lemma B.17, $\beta \geq \frac{1}{2}\sqrt{\frac{m}{d}}$ whenever $m \geq 10 \log d$. This concludes the lemma. $\square$

**Proposition B.19** (Lipschitz Projection with Separation). *For $N \geq 2$, let $\mathcal{D} = \{(\boldsymbol{x}_i, y_i)\}_{i=1}^N \in \boldsymbol{D}_{d,N,C}$. For any $\alpha \in (0,1)$ and $24\alpha^{-2} \log N \leq m \leq d$, there exists 1-Lipschitz linear mapping $\phi : \mathbb{R}^d \to \mathbb{R}^m$ and $\beta > 0$ such that $\mathcal{D}' := \{(\phi(\boldsymbol{x}_i), y_i)\}_{i=1}^N \in \boldsymbol{D}_{m,N,C}$ satisfies*

$$\epsilon'_{\mathcal{D}} \geq (1-\alpha)\beta\epsilon_{\mathcal{D}}.$$

*In particular, $\mathcal{D}' \in \boldsymbol{D}_{m,N,C}$ whenever $\mathcal{D} \in \boldsymbol{D}_{d,N,C}$. Moreover, $\beta \geq \frac{1}{2}\sqrt{\frac{m}{d}}$ whenever $m \geq 10 \log d$.*

*Proof.* Let $X = \{\boldsymbol{x}_i\}_{i=1}^N$. By Lemma B.18, there exists 1-Lipschitz linear mapping $\phi : \mathbb{R}^d \to \mathbb{R}^m$ and $\beta > 0$ such that

$$(1-\alpha)\beta \|\boldsymbol{x}_i - \boldsymbol{x}_j\|_2 \leq \|\phi(\boldsymbol{x}_i) - \phi(\boldsymbol{x}_j)\|_2 \leq (1+\alpha)\beta \|\boldsymbol{x}_i - \boldsymbol{x}_j\|_2 \tag{54}$$

for all $i, j \in [N]$.

The inequality $\epsilon_{\mathcal{D}'} \geq (1-\alpha)\beta\epsilon_{\mathcal{D}}$ follows from the inequality from Lemma B.18. In particular,

$$\begin{aligned}
\epsilon_{\mathcal{D}'} &= \frac{1}{2}\min\{\|\phi(\boldsymbol{x}_i) - \phi(\boldsymbol{x}_j)\|_2 \mid i,j \in [N] \text{ and } y_i \neq y_j\} \\
&\overset{(a)}{\geq} \frac{1}{2}\min\{(1-\alpha)\beta\|\boldsymbol{x}_i - \boldsymbol{x}_j\|_2 \mid i,j \in [N] \text{ and } y_i \neq y_j\} \\
&= (1-\alpha)\beta \times \frac{1}{2}\min\{\|\boldsymbol{x}_i - \boldsymbol{x}_j\|_2 \mid i,j \in [N] \text{ and } y_i \neq y_j\} \\
&= (1-\alpha)\beta\epsilon_{\mathcal{D}},
\end{aligned}$$

where we use Equation (54) at (a).

We next show $\mathcal{D}' \in \boldsymbol{D}_{m,N,C}$ whenever $\mathcal{D} \in \boldsymbol{D}_{d,N,C}$. To show this, we need to prove $\phi(\boldsymbol{x}_i) \neq \phi(\boldsymbol{x}_j)$ for all $i \neq j$. Since $1 - \alpha > 0$ and $\beta > 0$, we have $\|\phi(\boldsymbol{x}_i) - \phi(\boldsymbol{x}_j)\|_2 \geq (1-\alpha)\beta\|\boldsymbol{x}_i - \boldsymbol{x}_j\|_2 > 0$ whenever $\boldsymbol{x}_i \neq \boldsymbol{x}_j$. Moreover, $\mathcal{D} \in \boldsymbol{D}_{d,N,C}$ indicates that $\boldsymbol{x}_i \neq \boldsymbol{x}_j$ whenever $i \neq j$. All together, we have $\phi(\boldsymbol{x}_i) \neq \phi(\boldsymbol{x}_j)$ for all $i \neq j$ so that $\mathcal{D}' \in \boldsymbol{D}_{m,N,C}$. $\square$

**Lemma B.20** (Projection onto log-scale Dimension). *Let $\mathcal{D} \in \boldsymbol{D}_{d,N,C}$. Then, there exist an integer $m = \tilde{O}(\log N)$ and a 1-Lipschitz linear map $\phi : \mathbb{R}^d \to \mathbb{R}^m$ such that the projected dataset $\mathcal{D}' = \{(\phi(\boldsymbol{x}_i), y_i)\}_{i \in [N]} \in \boldsymbol{D}_{m,N,C}$ satisfies the separation bound*

$$\epsilon_{\mathcal{D}}' \geq \frac{5}{12}\sqrt{\frac{m}{d}}\epsilon_{\mathcal{D}}.$$

*Proof.* Let $\alpha = 1/6$ and $m := \min\{d, \max\{\lceil 24\alpha^{-2}\log N\rceil, \lceil 10\log d\rceil\}\}$, then $m = \tilde{O}(\log N)$. We construct the linear mapping into $m$ dimension by dividing the cases into $d < 24\alpha^{-2}\log N$ or $d \geq 24\alpha^{-2}\log N$.

For the case $d < 24\alpha^{-2}\log N$, we have $d < \max\{\lceil 24\alpha^{-2}\log N\rceil, \lceil 10\log d\rceil\}$, and therefore $m = d$. We consider the identity map $\phi : \mathbb{R}^d \to \mathbb{R}^d(= \mathbb{R}^m)$, which is 1-Lipschitz. We have $\mathcal{D}' := \{(\phi(\boldsymbol{x}_i), y_i)\}_{i \in [N]} = \{(\boldsymbol{x}_i, y_i)\}_{i \in [N]} = \mathcal{D}$, so that $\epsilon_{\mathcal{D}'} = \epsilon_{\mathcal{D}} > \frac{5}{12}\epsilon_{\mathcal{D}} = \frac{5}{12}\sqrt{\frac{m}{d}}\epsilon_{\mathcal{D}}$.

Otherwise, for the case $d \geq 24\alpha^{-2}\log N$, we first observe that $m \leq d$. Since $24\alpha^{-2}\log N \leq d$, we have

$$\lceil 24\alpha^{-2}\log N\rceil \leq d. \tag{55}$$

Additionally, as $N \geq 2$, we have $d \geq 24\alpha^{-2}\log N \geq 864\log 2 \geq e^4$. By Lemma B.22, this implies $10\log d \leq d$ and therefore

$$\lceil 10\log d\rceil \leq d. \tag{56}$$

By Equations (55) and (56), we have $\max\{\lceil 24\alpha^{-2}\log N\rceil, \lceil 10\log d\rceil\} \leq d$. Thus, it follows $m = \max\{\lceil 24\alpha^{-2}\log N\rceil, \lceil 10\log d\rceil\} \leq d$. By Proposition B.19 with $\alpha = \frac{1}{6}$, there exists 1-Lipschitz linear mapping $\phi : \mathbb{R}^d \to \mathbb{R}^m$ and $\beta > 0$ such that $\mathcal{D}' = \{(\phi(\boldsymbol{x}_i), y_i)\}_{i \in [N]}$ satisfies $\epsilon_{\mathcal{D}'} \geq \frac{5}{6}\beta\epsilon_{\mathcal{D}}$. Since $m = \max\{\lceil 24\alpha^{-2}\log N\rceil, \lceil 10\log d\rceil\} \geq 10\log d$, the inequality $\beta \geq \frac{1}{2}\sqrt{\frac{m}{d}}$ is also satisfied by Proposition B.19. Therefore, $\epsilon_{\mathcal{D}'} \geq \frac{5}{6}\beta\epsilon_{\mathcal{D}} \geq \frac{5}{12}\sqrt{\frac{m}{d}}\epsilon_{\mathcal{D}}$.

In both cases, we have 1-Lipschitz linear map $\phi$ such that $\mathcal{D}' = \{(\phi(\boldsymbol{x}_i), y_i)\}_{i \in [N]}$ has separation

$$\epsilon_{\mathcal{D}'} \geq \frac{5}{12}\sqrt{\frac{m}{d}}\epsilon_{\mathcal{D}}.$$

$\square$

**Proposition B.21** (Natural Projection of High Dimensional Data). *For $d \geq N$, let $\mathcal{D} = \{(\boldsymbol{x}_i, y_i)\}_{i \in [N]} \in \boldsymbol{D}_{d,N,C}$. Then, there exists 1-Lipschitz linear mapping $\varphi : \mathbb{R}^d \to \mathbb{R}^N$ such that $\mathcal{D}' = \{(\varphi(\boldsymbol{x}_i), y_i)\}_{i \in [N]} \in \boldsymbol{D}_{N,N,C}$ satisfies*

$$\epsilon_{\mathcal{D}'} = \epsilon_{\mathcal{D}}$$

*Proof.* Consider the tall matrix $\boldsymbol{X} \in \mathbb{R}^{d \times N}$ defined as

$$\boldsymbol{X} = \begin{bmatrix} | & | & \cdots & | \\ \boldsymbol{x}_1 & \boldsymbol{x}_2 & \cdots & \boldsymbol{x}_N \\ | & | & \cdots & | \end{bmatrix}. \tag{57}$$

Then $\dim \mathrm{Col}(\boldsymbol{X}) \leq N \leq d$. Take any subspace $V$ such that $\mathrm{Col}(\boldsymbol{X}) \subseteq V \subseteq \mathbb{R}^d$ and $\dim V = N$, and let $\mathcal{B} = \{\boldsymbol{v}_1, \cdots, \boldsymbol{v}_N\}$ be an orthonormal basis of $V$. Let $\boldsymbol{V} \in \mathbb{R}^{d \times N}$ be the matrix whose columns consist of vectors in $\mathcal{B}$:

$$\boldsymbol{V} = \begin{bmatrix} | & | & \cdots & | \\ \boldsymbol{v}_1 & \boldsymbol{v}_2 & \cdots & \boldsymbol{v}_N \\ | & | & \cdots & | \end{bmatrix}.$$

Define $\varphi : \mathbb{R}^d \to \mathbb{R}^N$ as $\varphi(\boldsymbol{x}) = \boldsymbol{V}^\top \boldsymbol{x}$. We first verify that $\varphi$ is 1-Lipschitz. For any $\boldsymbol{x} \in \mathbb{R}^d$, let $\boldsymbol{x} = \boldsymbol{x}_V + \boldsymbol{x}_{V^\perp}$ where $\boldsymbol{x}_V \in V$ and $\boldsymbol{x}_{V^\perp} \in V^\perp$. Then, $\boldsymbol{x}_V = \boldsymbol{V}\boldsymbol{z}$ for some $\boldsymbol{z} \in \mathbb{R}^N$, as $\boldsymbol{x}_V \in \mathrm{Col}(V)$. Moreover,

$$\begin{aligned} \boldsymbol{V}^\top \boldsymbol{x} &= \boldsymbol{V}^\top (\boldsymbol{x}_V + \boldsymbol{x}_{V^\perp}) \\ &= \boldsymbol{V}^\top \boldsymbol{x}_V + \boldsymbol{0} \\ &= \boldsymbol{V}^\top \boldsymbol{V} \boldsymbol{z} \\ &= \boldsymbol{I}_N \boldsymbol{z} \\ &= \boldsymbol{z}. \end{aligned} \tag{58}$$

Therefore, we have

$$\left\| \boldsymbol{V}^\top \boldsymbol{x} \right\|_2 \overset{(a)}{=} \|\boldsymbol{z}\|_2 \overset{(b)}{=} \|\boldsymbol{x}_V\|_2 \overset{(c)}{\leq} \|\boldsymbol{x}\|_2, \tag{59}$$

where (a) is by Equation (58), (b) is because $\|\boldsymbol{x}_V\|_2^2 = \left\| \sum_{i \in [N]} z_i \boldsymbol{v}_i \right\|_2^2 = \sum_{i \in [N]} z_i^2 = \|\boldsymbol{z}\|_2^2$, and (c) is because $\|\boldsymbol{x}\|_2^2 = \|\boldsymbol{x}_V\|_2^2 + \|\boldsymbol{x}_{V^\perp}\|_2^2$. Moreover, whenever $\boldsymbol{x} \in V$, then the equality holds for (c) of Equation (59). Therefore, $\left\| \boldsymbol{V}^\top \boldsymbol{x} \right\|_2 = \|\boldsymbol{x}\|_2$ for all $\boldsymbol{x} \in V$.

Since $\varphi$ is linear

$$\begin{aligned} \|\varphi(\boldsymbol{x}) - \varphi(\boldsymbol{x}')\|_2 &= \|\varphi(\boldsymbol{x} - \boldsymbol{x}')\|_2 \\ &= \left\| \boldsymbol{V}^\top (\boldsymbol{x} - \boldsymbol{x}') \right\|_2 \\ &\overset{(a)}{\leq} \|\boldsymbol{x} - \boldsymbol{x}'\|_2, \end{aligned}$$

where (a) is by Equation (59). This shows that $\varphi$ is 1-Lipschitz.

Next, for $i, j \in [N]$, we have

$$\begin{aligned} \|\varphi(\boldsymbol{x}_i) - \varphi(\boldsymbol{x}_j)\|_2 &= \|\varphi(\boldsymbol{x}_i - \boldsymbol{x}_j)\|_2 \\ &= \left\| \boldsymbol{V}^\top (\boldsymbol{x}_i - \boldsymbol{x}_j) \right\|_2 \\ &\overset{(a)}{=} \|\boldsymbol{x}_i - \boldsymbol{x}_j\|_2, \end{aligned}$$

where the last equality holds because $\boldsymbol{x}_i - \boldsymbol{x}_j \in \mathrm{Col}(\boldsymbol{X}) \subseteq V$.

This shows that

$$\begin{aligned} \epsilon_{\mathcal{D}'} &= \frac{1}{2} \min\{\|\varphi(\boldsymbol{x}_i) - \varphi(\boldsymbol{x}_j)\|_2 \mid y_i \neq y_j\} \\ &= \frac{1}{2} \min\{\|\boldsymbol{x}_i - \boldsymbol{x}_j\|_2 \mid y_i \neq y_j\} \\ &= \epsilon_{\mathcal{D}}. \end{aligned}$$

This shows that $\mathcal{D}'$ also has the desired property. $\square$

**Lemma B.22.** *For $t \geq e^4$, we have $t \geq 10 \log t$.*

*Proof.* Define $u(t) := t - 10 \log t$ on the domain $(0, \infty)$. Then, for all $t > 10$,

$$\frac{du}{dt} = 1 - \frac{10}{t} > 0,$$

so that $u$ is an increasing function on $(10, \infty)$. In particular,

$$u(e^4) = e^4 - 10 \log(e^4) = e^4 - 40 \geq 0$$

This concludes that $u(t) \geq 0$ for all $t \geq e^4$, or equivalently, $t \geq 10 \log t$ for all $t \geq e^4$. $\qquad\square$

### B.6 Lemmas for Bit Complexity

The following lemma bounds how much bit complexity is sufficient for implementing the parameters of the neural network in order to obtain the required precision of the output. Note that we do not require the network to output scala values. i.e. the following lemma also applies to neural networks that output vectors.

**Lemma B.23.** *Let $f$ be a neural network of $P$ parameters, depth $L$ and width $D$ in which the parameters have infinite precision. Let $R \geq 1$ be the radius of the domain in which we want to approximate $f$. If all the parameters of $f$ are bounded by some $M \geq 1$, then for any $0 < \nu < 1$, there exists $\bar{f}$, which is implemented with $P$ parameters, depth $L$, width $D$, and $\tilde{O}(L)$ bit complexity such that*

$$\max_{\|\boldsymbol{x}\|_2 \leq R} \left\|\bar{f} - f\right\|_2 \leq \nu,$$

*where $\tilde{O}(\cdot)$ hides a polylogarithmic dependency on $D, M, L, R$ and $1/\nu$.*

*Proof.* Let $f : \mathbb{R}^d \to \mathbb{R}$ be the neural network defined as

$$\begin{aligned}
\boldsymbol{a}_0 &= \boldsymbol{x} \\
\boldsymbol{a}_\ell &= \sigma(\boldsymbol{W}_\ell \boldsymbol{a}_{\ell-1}(\boldsymbol{x}) + \boldsymbol{b}_\ell) \text{ for } \ell = 1, 2, \cdots, L-1 \\
f(\boldsymbol{x}) &= \boldsymbol{W}_L(\boldsymbol{a}_{L-1}) + \boldsymbol{b}_L,
\end{aligned}$$

where $\boldsymbol{W}_\ell \in \mathbb{R}^{d_\ell \times d_{\ell-1}}, \boldsymbol{b}_\ell \in \mathbb{R}^{d_\ell}$ with $d_\ell \leq D$ for all $\ell \in [L]$. Although $\boldsymbol{a}_\ell$ depends on $\boldsymbol{x}$ for all $\ell = 0, \cdots, L-1$, we omit $\boldsymbol{x}$ in the notation. Note that $d_0 = d$. Given that every elements of $\boldsymbol{W}_\ell$ and $\boldsymbol{b}_\ell$ are bounded by $M$, for any $0 < \zeta \leq M$, there exists $\bar{\boldsymbol{W}}_\ell$ and $\bar{\boldsymbol{b}}_\ell$ that can be implemented with $\lceil \log_2(M/\zeta) \rceil$ bit complexity in which

$$\begin{aligned}
\left\|\bar{\boldsymbol{W}}_\ell - \boldsymbol{W}_\ell\right\|_\infty &\leq \zeta \\
\left\|\bar{\boldsymbol{b}}_\ell - \boldsymbol{b}_\ell\right\|_\infty &\leq \zeta.
\end{aligned}$$

Using the approximated parameters $\bar{\boldsymbol{W}}_\ell$ and $\bar{\boldsymbol{b}}_\ell$, we recursively define $\bar{f} : \mathbb{R}^d \to \mathbb{R}$, the finite-precision approximation of $f$.

$$\begin{aligned}
\bar{\boldsymbol{a}}_0 &= \boldsymbol{x} \\
\bar{\boldsymbol{a}}_\ell &= \sigma(\bar{\boldsymbol{W}}_\ell \bar{\boldsymbol{a}}_{\ell-1}(\boldsymbol{x}) + \bar{\boldsymbol{b}}_\ell) \text{ for } \ell = 1, 2, \cdots, L-1 \\
\bar{f}(\boldsymbol{x}) &= \bar{\boldsymbol{W}}_L(\bar{\boldsymbol{a}}_{L-1}) + \bar{\boldsymbol{b}}_L.
\end{aligned}$$

Similarly, although $\bar{\boldsymbol{a}}_\ell$ depends on $\boldsymbol{x}$ for all $\ell = 0, \cdots, L-1$, we omit $\boldsymbol{x}$ in the notation.

Let us denote the difference of parameters as $\Delta \boldsymbol{W}_\ell := \bar{\boldsymbol{W}}_\ell - \boldsymbol{W}_\ell, \Delta \boldsymbol{b}_\ell := \bar{\boldsymbol{b}}_\ell - \boldsymbol{b}_\ell$ for $\ell \in [L]$ and the difference of layer outputs $\Delta \boldsymbol{a}_\ell := \bar{\boldsymbol{a}}_\ell - \boldsymbol{a}_\ell$ for $\ell \in [L-1]$. It is straightforward to check

$$\begin{aligned}
\|\boldsymbol{W}_\ell\| &\leq \|\boldsymbol{W}_\ell\|_F \leq \sqrt{d_l d_{l-1}} M \leq DM \\
\|\boldsymbol{b}_\ell\| &\leq \sqrt{d_l} M \leq \sqrt{D} M \\
\|\Delta \boldsymbol{W}_\ell\| &\leq \|\Delta \boldsymbol{W}_\ell\|_F \leq \sqrt{d_l d_{l-1}} \zeta \leq D\zeta \\
\|\Delta \boldsymbol{b}_\ell\| &\leq \sqrt{d_l} \zeta \leq \sqrt{D} \zeta,
\end{aligned}$$

where the norm $\|\cdot\|$ and $\|\cdot\|_F$ for the matrix denote the spectral norm and the Frobenius norm, respectively.

We first claim that there exists a degree $2L+1$ polynomial $S$ on $D, M, L$ and $R$ such that $\|\boldsymbol{a}_\ell\|_2 \leq S$ for all $\ell \in [L-1]$.

$$\begin{aligned}
\|\boldsymbol{a}_\ell\|_2 &= \|\sigma(\boldsymbol{W}_\ell \boldsymbol{a}_{\ell-1} + \boldsymbol{b}_\ell)\|_2 \\
&\leq \|\boldsymbol{W}_\ell \boldsymbol{a}_{\ell-1} + \boldsymbol{b}_\ell\|_2 \\
&\leq \|\boldsymbol{W}_\ell\| \|\boldsymbol{a}_{\ell-1}\|_2 + \|\boldsymbol{b}_\ell\|_2 \\
&\leq DM \|\boldsymbol{a}_{\ell-1}\|_2 + \sqrt{D}M.
\end{aligned}$$

Thus for all $\ell \in [L-1]$,

$$\begin{aligned}
\|\boldsymbol{a}_\ell\|_2 &\leq \left( \sqrt{D}M \sum_{\ell' \in [\ell]} (DM)^{\ell'-1} \right) + (DM)^l \|\boldsymbol{a}_0\|_2 \\
&\leq \sqrt{D}ML(DM)^{L-1} + (DM)^{L-1}R \\
&\leq (DM)^{L-1}(DML + R) =: S
\end{aligned}$$

This proves the first claim. Moreover, $S$ is composed of two monomials whose coefficients are all 1.

We next claim that the error $\|\Delta \boldsymbol{a}_{L-1}\| \leq Q\zeta$ for some degree $4L+1$ polynomial $Q$ on $D, M, L$ and $R$. Consider the following recurrence

$$\begin{aligned}
\|\Delta \boldsymbol{a}_\ell\|_2 &= \|\bar{\boldsymbol{a}}_\ell - \boldsymbol{a}_\ell\|_2 \\
&= \left\| \sigma(\bar{\boldsymbol{W}}_\ell \bar{\boldsymbol{a}}_{\ell-1} + \bar{\boldsymbol{b}}_\ell) - \sigma(\boldsymbol{W}_\ell \boldsymbol{a}_{\ell-1} + \boldsymbol{b}_\ell) \right\|_2 \\
&\leq \left\| (\bar{\boldsymbol{W}}_\ell \bar{\boldsymbol{a}}_{\ell-1} + \bar{\boldsymbol{b}}_\ell) - (\boldsymbol{W}_\ell \boldsymbol{a}_{\ell-1} + \boldsymbol{b}_\ell) \right\|_2 \\
&= \left\| ((\boldsymbol{W}_\ell + \Delta \boldsymbol{W}_\ell)(\boldsymbol{a}_{\ell-1} + \Delta \boldsymbol{a}_\ell) + (\boldsymbol{b}_\ell + \Delta \boldsymbol{b}_\ell)) - (\boldsymbol{W}_\ell \boldsymbol{a}_{\ell-1} + \boldsymbol{b}_\ell) \right\|_2 \\
&= \|\Delta \boldsymbol{W}_\ell \boldsymbol{a}_{\ell-1} + \boldsymbol{W}_\ell \Delta \boldsymbol{a}_{\ell-1} + \Delta \boldsymbol{W}_\ell \Delta \boldsymbol{a}_{\ell-1} + \Delta \boldsymbol{b}_\ell\|_2 \\
&\leq \|\Delta \boldsymbol{W}_\ell\| \|\boldsymbol{a}_{\ell-1}\|_2 + \|\boldsymbol{W}_\ell\| \|\Delta \boldsymbol{a}_{\ell-1}\|_2 + \|\Delta \boldsymbol{W}_\ell\| \|\Delta \boldsymbol{a}_{\ell-1}\|_2 + \|\Delta \boldsymbol{b}_\ell\|_2 \\
&\leq D\zeta \|\boldsymbol{a}_{\ell-1}\|_2 + DM \|\Delta \boldsymbol{a}_{\ell-1}\|_2 + D\zeta \|\Delta \boldsymbol{a}_{\ell-1}\|_2 + \sqrt{D}\zeta \\
&= (DM + D\zeta) \|\Delta \boldsymbol{a}_{\ell-1}\|_2 + (D \|\boldsymbol{a}_{\ell-1}\|_2 + \sqrt{D})\zeta \\
&\leq (DM + D\zeta) \|\Delta \boldsymbol{a}_{\ell-1}\|_2 + (DS + \sqrt{D})\zeta.
\end{aligned}$$

Thus noting that $\Delta \boldsymbol{a}_0 = \boldsymbol{x} - \boldsymbol{x} = 0$ we have,

$$\begin{aligned}
\|\Delta \boldsymbol{a}_{L-1}\|_2 &\leq \left( DS + \sqrt{D} \right) \zeta \sum_{\ell \in [L-1]} (DM + D\zeta)^{\ell-1} \\
&\leq \left( DS + \sqrt{D} \right) \zeta \times L(DM + D\zeta)^{L-1} \\
&\leq DL(S+1)(DM + D\zeta)^{L-1}\zeta \\
&\leq DL(S+1)(2DM)^{L-1}\zeta,
\end{aligned}$$

where the last inequality follows from $\zeta \leq M$. Let $Q := DL(S+1)(2DM)^{L-1}$. Since $S$ is a degree $2L+1$ polynomial on $D, M, L$ and $R$, it follows that $Q$ is a degree $4L+1$ polynomial on $D, M, L$ and $R$. This proves the second claim. Moreover, $Q$ is composed of three monomials whose coefficients are at most $2^L$.

Thus,

$$\begin{aligned}
\|\bar{f}(\boldsymbol{x}) - f(\boldsymbol{x})\|_2 &= \|(\bar{\boldsymbol{W}}_L \bar{\boldsymbol{a}}_{L-1} + \bar{\boldsymbol{b}}_L) - \boldsymbol{W}_L \boldsymbol{a}_{L-1} + \boldsymbol{b}_L\|_2 \\
&\leq \|((\boldsymbol{W}_L + \Delta \boldsymbol{W}_L)(\boldsymbol{a}_{L-1} + \Delta \boldsymbol{a}_{L-1}) + (\boldsymbol{b}_L + \Delta \boldsymbol{b}_L)) - (\boldsymbol{W}_L \boldsymbol{a}_{L-1} + \boldsymbol{b}_L)\|_2 \\
&= \|\Delta \boldsymbol{W}_L \boldsymbol{a}_{L-1} + \boldsymbol{W}_L \Delta \boldsymbol{a}_{L-1} + \Delta \boldsymbol{W}_L \Delta \boldsymbol{a}_{L-1} + \Delta \boldsymbol{b}_L\|_2 \\
&\leq \|\Delta \boldsymbol{W}_L\| \|\boldsymbol{a}_{L-1}\|_2 + \|\boldsymbol{W}_L\| \|\Delta \boldsymbol{a}_{L-1}\|_2 + \|\Delta \boldsymbol{W}_L\| \|\Delta \boldsymbol{a}_{L-1}\|_2 + \|\Delta \boldsymbol{b}_L\|_2 \\
&\leq D\zeta \|\boldsymbol{a}_{L-1}\|_2 + DM \|\Delta \boldsymbol{a}_{L-1}\|_2 + D\zeta \|\Delta \boldsymbol{a}_{L-1}\|_2 + \sqrt{D}\zeta \\
&\leq D\zeta S + DMQ\zeta + D\zeta Q\zeta + D\zeta \\
&\leq (DS + DMQ + DMQ + D)\zeta,
\end{aligned}$$

where we use $\zeta \leq M$ in the last inequality. Now, by letting $\zeta := \frac{\nu}{DS+2DMQ+D}$, it follows that

$$\left\| \bar{f}(\boldsymbol{x}) - f(\boldsymbol{x}) \right\|_2 \leq \nu,$$

for all $\boldsymbol{x}$ with $\|\boldsymbol{x}\|_2 \leq R$. Thus, it suffices to have $\log_2(M/\zeta) = \log_2((DS + 2DMQ + D)M/\nu)$ bit complexity to attain an approximation of accuracy $\nu$ uniformly over the bounded domain with radius $R$. $(DS + 2DMQ + D)M$ is a degree $4L + 4$ polynomial on $D, M, L$ and $R$. Moreover, it is composed of $2 + 3 + 1 = 6$ monomials, whose coefficients are at most $2^{L+1}$. Hence, it follows that

$$
\begin{aligned}
\log_2(M/\zeta) &= \log_2((DS + 2DMQ + D)M/\nu) \\
&\leq \log_2(6 \times 2^{L+1} \times (DMLR)^{4L+4}) + \log(1/\nu) \\
&= O(L \log_2(2DMLR) + \log(1/\nu)) \\
&= \tilde{O}(L)
\end{aligned}
$$

bit complexity suffices. $\qquad\square$

# C  Extensions to $\ell_p$-norm

In this section, we extend the previous results on $\ell_2$-norm to arbitrary $p$-norm, where $p \in [1, \infty]$.

In the following, we use $\mathrm{dist}_p(\cdot, \cdot)$ to denote the $\ell_p$-norm distance between two points, a point and a set, or two sets. For the case $d = 1$, we omit the notation $p$ since every $\ell_p$-norm in 1-dimension denotes the absolute value.

We denote $\mathcal{B}_p(\boldsymbol{x}, \mu) = \{\boldsymbol{x}' \in \mathbb{R}^d \,|\, \|\boldsymbol{x}' - \boldsymbol{x}\|_p < \mu\}$ an open $\ell_p$-ball centered at $\boldsymbol{x}$ with a radius $\mu$.

**Definition C.1.** For $\mathcal{D} \in \boldsymbol{D}_{d,N,C}$, the separation constant $\epsilon_{\mathcal{D},p}$ under $\ell_p$-norm is defined as

$$\epsilon_{\mathcal{D},p} := \frac{1}{2} \min \left\{ \|\boldsymbol{x}_i - \boldsymbol{x}_j\|_p \,|\, (\boldsymbol{x}_i, y_i), (\boldsymbol{x}_j, y_j) \in \mathcal{D}, \ y_i \neq y_j \right\}.$$

As we consider $\mathcal{D}$ with $\boldsymbol{x}_i \neq \boldsymbol{x}_j$ for all $i \neq j$, we have $\epsilon_{\mathcal{D},p} > 0$. Next, we define robust memorization under $\ell_p$-norm.

**Definition C.2.** For $\mathcal{D} \in \boldsymbol{D}_{d,N,C}$, $p \in [1, \infty]$, and a given *robustness ratio* $\rho \in (0, 1)$, define the robustness radius as $\mu = \rho \epsilon_{\mathcal{D},p}$. We say that a function $f : \mathbb{R}^d \to \mathbb{R}$ $\rho$-*robustly memorizes* $\mathcal{D}$ under the $\ell_p$-norm if

$$f(\boldsymbol{x}') = y_i, \quad \text{for all } (\boldsymbol{x}_i, y_i) \in \mathcal{D} \text{ and } \boldsymbol{x}' \in \mathcal{B}_p(\boldsymbol{x}_i, \mu),$$

and $\mathcal{B}_p(\boldsymbol{x}_i, \mu)$ is referred as the *robustness ball* of $\boldsymbol{x}_i$.

Similarly, we extend the notion of $\rho$-robust memorization error to $\ell_p$-norm.

**Definition C.3.** Let $\mathcal{D} \in \boldsymbol{D}_{d,N,C}$ be a class(or point)-separated dataset. The $\rho$-robust error of a network $f : \mathbb{R}^d \to \mathbb{R}$ on $\mathcal{D}$ under the $\ell_p$-norm is defined as

$$\mathcal{L}_{\rho,p}(f, \mathcal{D}) = \max_{(\boldsymbol{x}_i, y_i) \in \mathcal{D}} \mathbb{P}_{\boldsymbol{x}' \sim \mathrm{Unif}(\mathcal{B}_p(\boldsymbol{x}_i, \mu))}[f(\boldsymbol{x}') \neq y_i], \quad \text{where } \mu = \rho \epsilon_{\mathcal{D},p} \text{ (or } \mu = \rho \epsilon'_{\mathcal{D},p}).$$

The following inclusion between $p$-norm balls with different $p$-values is well known.

**Lemma C.4** (Inclusion Between Balls). *Let $0 < p < q \leq \infty$. Then, for any $\boldsymbol{x} \in \mathbb{R}^d$ and $\mu > 0$,*

$$\mathcal{B}_p(\boldsymbol{x}, \mu) \subseteq \mathcal{B}_q(\boldsymbol{x}, \mu) \subseteq \mathcal{B}_p(\boldsymbol{x}, d^{\frac{1}{p} - \frac{1}{q}} \mu),$$

*or equivalently,*

$$\mathcal{B}_q(\boldsymbol{x}, d^{\frac{1}{q} - \frac{1}{p}} \mu) \subseteq \mathcal{B}_p(\boldsymbol{x}, \mu) \subseteq \mathcal{B}_q(\boldsymbol{x}, \mu).$$

For any $p \in [1, \infty]$, let us denote

$$\gamma_p(d) := d^{\left| \frac{1}{2} - \frac{1}{p} \right|}$$

throughout this section. For $0 < p < q \leq \infty$, we have

$$\epsilon_{\mathcal{D},q} \leq \epsilon_{\mathcal{D},p} \leq d^{\frac{1}{p} - \frac{1}{q}} \epsilon_{\mathcal{D},q}, \tag{60}$$

since $\|\boldsymbol{x}\|_q \leq \|\boldsymbol{x}\|_p \leq d^{\frac{1}{p} - \frac{1}{q}} \|\boldsymbol{x}\|_q$. In particular, we have

$$\epsilon_{\mathcal{D},p} \leq \epsilon_{\mathcal{D},2} \qquad\qquad \text{when } p \geq 2, \tag{61}$$
$$\epsilon_{\mathcal{D},p} \leq \gamma_p(d) \epsilon_{\mathcal{D},2} \qquad\qquad \text{when } p < 2. \tag{62}$$

## C.1  Extension of Necessity Condition to $\ell_p$-norm

**Theorem C.5.** *Let $\rho \in (0, 1)$. Suppose for any $\mathcal{D} \in \boldsymbol{D}_{d,N,2}$, there exists a neural network $f \in \mathcal{F}_{d,P}$ that can $\rho$-robustly memorize $\mathcal{D}$ under $\ell_p$-norm. Then, the number of parameters $P$ must satisfy*

- $P = \Omega\left( \left( \rho^2 \min\{N, d\} + 1 \right) d + \min\left\{ \frac{1}{\sqrt{1 - \rho^p}}, \sqrt{d} \right\} \sqrt{N} \right)$ *if $p \geq 2$.*

- $P = \Omega\left( \left( \left( \frac{\rho}{\gamma_p(d)} \right)^2 \min\{N, d\} + 1 \right) d + \min\left\{ \frac{1}{\sqrt{1 - \rho^p}}, \sqrt{d} \right\} \sqrt{N} \right)$ *if $1 \leq p < 2$.*

*Proof.* This follows by combining Proposition C.6 and Proposition C.8. □

**Proposition C.6.** *There exists $\mathcal{D} \in \boldsymbol{D}_{d,N,2}$ such that any neural network $f : \mathbb{R}^d \to \mathbb{R}$ that $\rho$-robustly memorizes $\mathcal{D}$ under $\ell_p$-norm must have the first hidden layer width at least*

- $\rho^2 \min\{N-1, d\}$ *if $p \geq 2$.*

- $\left(\frac{\rho}{\gamma_p(d)}\right)^2 \min\{N-1, d\}$ *if $1 \leq p < 2$.*

*Proof.* We take $\mathcal{D}$ the same dataset as in Proposition 3.2. Recall that in the proof of Proposition 3.2, we take the dataset $\mathcal{D} = \{\boldsymbol{e}_j, 2\}_{j \in [N-1]} \cup \{\boldsymbol{0}, 1\}$ when $N \leq d+1$, with additional data points $(2\boldsymbol{e}_1, 2), (3\boldsymbol{e}_1, 2), \cdots, ((N-d)\boldsymbol{e}_1, 2)$ when $N > d+1$. This has a separation $\epsilon_{\mathcal{D},p} = \frac{1}{2}$ under $\ell_p$-norm for all $p \geq 1$, on the both case $N \leq d+1$ and $N > d+1$. Let $f$ be a neural network that robustly memorizes $\mathcal{D}$ under $\ell_p$-norm. Since $\epsilon_{\mathcal{D},p} = \epsilon_{\mathcal{D},2}$, the robustness radius $\mu$ under $\ell_2$-norm satisfies $\mu = \rho\epsilon_{\mathcal{D},p} = \rho\epsilon_{\mathcal{D},2}$. With this in mind, we now prove the proposition. The statement of the proposition consists of two parts, $p \geq 2$ and $1 \leq p < 2$.

**Part I: $p \geq 2$.** First, we prove the result under $p \geq 2$ Robust memorization under $\ell_p$-norm implies

$$f(\boldsymbol{x}) = y_i \text{ for all } (\boldsymbol{x}_i, y_i) \in \mathcal{D} \text{ and } \boldsymbol{x} \in \mathcal{B}_p(\boldsymbol{x}_i, \mu),$$

where $\mu = \rho\epsilon_{\mathcal{D},p} = \rho\epsilon_{\mathcal{D},2}$. For $p \geq 2$, we have $\mathcal{B}_2(\boldsymbol{x}_i, \mu) \subseteq \mathcal{B}_p(\boldsymbol{x}_i, \mu)$ by Lemma C.4. Thus,

$$f(\boldsymbol{x}) = y_i \text{ for all } (\boldsymbol{x}_i, y_i) \in \mathcal{D} \text{ and } \boldsymbol{x} \in \mathcal{B}_2(\boldsymbol{x}_i, \mu).$$

Since $\mu = \rho\epsilon_{\mathcal{D},2}$ this implies that $f$ $\rho$-robustly memorize $\mathcal{D}$ under $\ell_2$-norm. By Proposition 3.2, $f$ should have the first hidden layer width at least $\rho^2 \min\{N-1, d\}$.

**Part II: $1 \leq p < 2$.** Next, we prove the result under $1 \leq p < 2$. Robust memorization under $\ell_p$-norm implies

$$f(\boldsymbol{x}) = y_i \text{ for all } (\boldsymbol{x}_i, y_i) \in \mathcal{D} \text{ and } \boldsymbol{x} \in \mathcal{B}_p(\boldsymbol{x}_i, \mu),$$

where $\mu = \rho\epsilon_{\mathcal{D},p} = \rho\epsilon_{\mathcal{D},2}$. For $1 \leq p < 2$, we have $\mathcal{B}_2(\boldsymbol{x}, d^{\frac{1}{2}-\frac{1}{p}}\mu) \subseteq \mathcal{B}_p(\boldsymbol{x}_i, \mu)$ by applying $p = p$ and $q = 2$ to Lemma C.4. Since $\gamma_p(d) = d^{\frac{1}{p}-\frac{1}{2}}$, we have $\mathcal{B}_2(\boldsymbol{x}_i, \mu/\gamma_p(d)) \subseteq \mathcal{B}_p(\boldsymbol{x}_i, \mu)$. In particular, $f$ memorize every $\mu/\gamma_p(d)$ neighbor around the data point under $\ell_2$-norm. Let

$$\rho' := \frac{\mu/\gamma_p(d)}{\epsilon_{\mathcal{D},2}} = \frac{\rho\epsilon_{\mathcal{D},2}/\gamma_p(d)}{\epsilon_{\mathcal{D},2}} = \frac{\rho}{\gamma_p(d)}$$

Then, $f$ memorize every $\mu/\gamma_p(d) = \rho'\epsilon_{\mathcal{D},2}$ radius neighbor around each data point under $\ell_2$-norm. In other words, $f$ $\rho'$-robustly memorize $\mathcal{D}$ under $\ell_2$-norm. By Proposition 3.2, $f$ should have the first hidden layer width at least $(\rho')^2 \min\{N-1, d\}$. Putting back $\rho' = \frac{\rho}{\gamma_p(d)}$ concludes the desired statement.

□

**Proposition C.7.** *There exists a point separated $\mathcal{D} \in \boldsymbol{D}_{d,N,2}$ such that any neural network that $\rho$-robustly memorizes $\mathcal{D}$ under $\ell_\infty$-norm must have the first hidden layer width at least*

- $\rho^2 \min\{d, N-1\}$ *if $\rho \in (0, \frac{1}{2}]$.*

- $\min\{d, N-1\}$ *if $\rho \in (\frac{1}{2}, 1)$.*

*Proof.* The first bullet is an immediate corollary of Proposition C.6, so we focus on the second bullet for $\rho \in (1/2, 1)$. To prove the second bullet, we consider two cases based on the relationship between $N-1$ and $d$. In the first case, where $N-1 \leq d$, establishing the proposition requires that the first hidden layer has width at least $N-1$. In the second case, where $N-1 > d$, the required width is at least $d$. For each case, we construct a dataset $\mathcal{D} \in \boldsymbol{D}_{d,N,2}$ such that any network that $\rho$-robustly memorizes $\mathcal{D}$ must have a first hidden layer of width no smaller than the corresponding bound.

**Case I : $N - 1 \le d$.** Let $\mathcal{D} = \{(\boldsymbol{e}_j, 2)\}_{j \in [N-1]} \cup \{(\boldsymbol{0}, 1)\}$. Then, $\mathcal{D}$ has a separation constant $\epsilon_{\mathcal{D}, \infty} = 1/2$ under $\ell_\infty$-norm. Let $f$ be a $\rho$-robust memorizer of $\mathcal{D}$ under $\ell_\infty$-norm whose first hidden layer width is $m$. Let $\boldsymbol{W} \in \mathbb{R}^{m \times d}$ denote the first hidden weight matrix. Suppose for a contradiction, $m < N - 1$.

Let $\mu = \rho\epsilon_{\mathcal{D}, \infty}$ denote the robustness radius. Then, $f$ has to distinguish every point in each $B_\mu(\boldsymbol{e}_j)$ from every point in $B_\mu(\boldsymbol{0})$ for all $j \in [N-1]$. Therefore, for $\boldsymbol{x} \in B_\infty(\boldsymbol{e}_j, \mu)$ and $\boldsymbol{x}' \in B_\infty(\boldsymbol{0}, \mu)$, we have

$$\boldsymbol{W}\boldsymbol{x} \neq \boldsymbol{W}\boldsymbol{x}',$$

or equivalently, $\boldsymbol{x} - \boldsymbol{x}' \notin \mathrm{Null}(\boldsymbol{W})$. Moreover

$$B_\infty(\boldsymbol{e}_j, \mu) - B_\infty(\boldsymbol{0}, \mu) := \{\boldsymbol{x} - \boldsymbol{x}' : \boldsymbol{x} \in B_\infty(\boldsymbol{e}_j, \mu) \text{ and } \boldsymbol{x}' \in B_\infty(\boldsymbol{0}, \mu)\} = B_\infty(\boldsymbol{e}_j, 2\mu).$$

Hence, it is necessary to have $B_\infty(\boldsymbol{e}_j, 2\mu) \cap \mathrm{Null}(\boldsymbol{W}) = \emptyset$ for all $j \in [N-1]$, or equivalently,

$$\mathrm{dist}_\infty(\boldsymbol{e}_j, \mathrm{Null}(\boldsymbol{W})) \ge 2\mu \tag{63}$$

for all $j \in [N-1]$.

Since $\dim \mathrm{Col}(W^\top) \le \dim \mathbb{R}^m = m$, we have $\dim \mathrm{Null}(W) \ge d - m$. Using Lemma C.10, we can upper bounds the maximum possible distance between $\{\boldsymbol{e}_j\}_{j \in [N-1]} \subseteq \mathbb{R}^d$ and arbitrary subspace of a fixed dimension.

Take $Z \subseteq \mathrm{Null}(W)$ such that $\dim Z = d - m$ and substitute $d = d$, $t = N - 1$, $k = d - m$ and $Z = Z$ into Lemma C.10. The assumptions $t \le d$ for the lemma are satisfied since $N - 1 \le d$. The additional assumption $k \ge d - t + 1$ is equivalent to $d - m \ge d - (N-1) + 1$ and is satisfied since $m < N - 1$. Therefore, we have

$$\min_{j \in [N-1]} \mathrm{dist}_\infty(\boldsymbol{e}_j, Z) \le \frac{1}{2}.$$

By combining the above inequality with Equation (63),

$$2\mu \le \min_{j \in [N-1]} \mathrm{dist}_\infty(\boldsymbol{e}_j, \mathrm{Null}(W)) \overset{(a)}{\le} \min_{j \in [N-1]} \mathrm{dist}_\infty(\boldsymbol{e}_j, Z) \le \frac{1}{2}, \tag{64}$$

where (a) is due to $Z \subseteq \mathrm{Null}(W)$. Since $\epsilon_{\mathcal{D}, \infty} = 1/2$, we have $2\mu = 2\rho\epsilon_{\mathcal{D}, \infty} = \rho$ so that Equation (64) becomes $\rho \le 1/2$. This contradicts our assumption $\rho \in (1/2, 1)$, and therefore the width requirement $m \ge N - 1$ is necessary. This concludes the proof for the case $N - 1 \le d$.

**Case II : $N - 1 > d$.** We construct the first $d + 1$ data points in the same manner as in Case I, using the construction for $N = d + 1$. For the remaining $N - d - 1$ data points, we set them sufficiently distant from the first $d + 1$ data points to keep $\epsilon_{\mathcal{D}, \infty} = 1/2$. In particular, we can set $\boldsymbol{x}_{d+2} = 2\boldsymbol{e}_1, \boldsymbol{x}_{d+3} = 3\boldsymbol{e}_1, \cdots, \boldsymbol{x}_N = (N - d)\boldsymbol{e}_1$ and $y_{d+2} = y_{d+3} = \cdots = y_N = 2$. Compared to the case $N = d + 1$, we have $\epsilon_{\mathcal{D}, \infty}$ unchanged while having more data points to memorize. By the necessity for the case $N = d + 1$, this dataset also requires the first hidden layer width at least $(d + 1) - 1 = d$. This concludes the statement for the case $N - 1 > d$.

Combining the result of the two cases $N - 1 \le d$ and $N - 1 > d$ concludes the proof of the theorem.

$\square$

**Proposition C.8.** *For $p \in [1, \infty)$, let $\rho \in \left(0, \left(1 - \frac{1}{d}\right)^{1/p}\right]$. Suppose for any $\mathcal{D} \in \boldsymbol{D}_{d,N,2}$ there exists $f \in \mathcal{F}_{d,P}$ that $\rho$-robustly memorizes $\mathcal{D}$ under $\ell_p$-norm. Then, the number of parameters $P$ must satisfy $P = \Omega(\sqrt{\frac{N}{1-\rho^p}})$.*

*Proof.* The main idea of the proof is the same as Proposition 3.3. We construct $\lfloor \frac{N}{2} \rfloor \times \lfloor \frac{1}{1-\rho^p} \rfloor$ number of data points that can be shattered by $\mathcal{F}_{d,P}$. This proves $\mathrm{VC\text{-}dim}(\mathcal{F}_{d,P}) \ge \lfloor \frac{N}{2} \rfloor \times \lfloor \frac{1}{1-\rho^p} \rfloor = \Omega(N/(1-\rho^p))$. Since $\mathrm{VC\text{-}dim}(\mathcal{F}_{d,P}) = O(P^2)$, this proves $P = \Omega(\sqrt{N/(1-\rho^p)})$.

For simplicity of the notation, let us denote $k := \lfloor \frac{1}{1-\rho^p} \rfloor$. To prove the lower bound on the VC-dimension, we construct $k \times \lfloor \frac{N}{2} \rfloor$ points in $\mathbb{R}^d$ that can be shattered by $\mathcal{F}_{d,P}$. As in the proof of Proposition 3.3, we define $\lfloor \frac{N}{2} \rfloor \times k$ number of points as $\lfloor \frac{N}{2} \rfloor$ groups, where each group consists of $k$ points.

We start by constructing the first group. Since $\rho \in (0, \left(\frac{d-1}{d}\right)^{1/p}]$, we have $k = \lfloor \frac{1}{1-\rho^p} \rfloor \in [1, d]$. The first group $\mathcal{X}_1 := \{e_j\}_{j=1}^k \subseteq \mathbb{R}^d$ is defined as the set of the first $k$ vectors in the standard basis of $\mathbb{R}^d$. The remaining $\lfloor \frac{N}{2} \rfloor - 1$ groups are simply constructed as a translation of $\mathcal{X}_1$. In particular, for $l \in [\lfloor \frac{N}{2} \rfloor]$, we define

$$\mathcal{X}_l := c_l + \mathcal{X}_1 = \{c_l + x \mid x \in \mathcal{X}_1\}$$

where $c_l := 2d^2(l-1) \times e_1$ ensures that each group is sufficiently far from one another. Note that $c_1 = 0$ ensures $\mathcal{X}_1$ also satisfies the consistency of the notation. Now, define $\mathcal{X} = \cup_{l \in [\lfloor N/2 \rfloor]} \mathcal{X}_l$, the union of all $\lfloor \frac{N}{2} \rfloor$ groups which consists of $k \times \lfloor \frac{N}{2} \rfloor$ points.

We claim that if for any $\mathcal{D} \in \mathbf{D}_{d,N,2}$, there exists $f \in \mathcal{F}_{d,P}$ that $\rho$-robustly memorizes $\mathcal{D}$ under $\ell_p$-norm, then $\mathcal{X}$ is shattered by $\mathcal{F}_{d,P}$. To prove the claim, suppose we are given arbitrary label $\mathcal{Y} = \{y_{l,j}\}_{l \in [\lfloor N/2 \rfloor], j \in [d]}$ of $\mathcal{X}$, where $y_{l,j} \in \{\pm 1\}$ denotes the label for $x_{l,j} := c_l + e_j \in \mathcal{X}$. Given the label $\mathcal{Y}$, we construct $\mathcal{D} \in \mathbf{D}_{d,N,2}$ such that whenever $f \in \mathcal{F}_{d,P}$ $\rho$-robustly memorize $\mathcal{D}$ under $\ell_p$-norm, then its affine translation $f' = 2f - 3 \in \mathcal{F}_{d,P}$ satisfies $f'(x_{l,j}) = y_{l,j}$ for all $x_{l,j} \in \mathcal{X}$.

For each $l \in [\lfloor N/2 \rfloor]$, let $J_l^+ = \{j \in [k] \mid y_{l,j} = +1\}$ and $J_l^- = \{j \in [k] \mid y_{l,j} = -1\}$. Define

$$x_{2l-1} = c_l + \sum_{j \in J_l^+} e_j - \sum_{j \in J_l^-} e_j$$

$$x_{2l} = c_l + \sum_{j \in J_l^-} e_j - \sum_{j \in J_l^+} e_j$$

Furthermore, define $y_{2l-1} = 2, y_{2l} = 1$ and let $\mathcal{D} = \{(x_i, y_i)\}_{i \in [N]} \in \mathbf{D}_{d,N,2}$. To consider the separation $\epsilon_{\mathcal{D},2}$, notice that

$$\|x_{2l-1} - x_{2l}\|_p = \left\| 2\left( \sum_{j \in J_l^+} e_j - \sum_{j \in J_l^-} e_j \right) \right\|_p \stackrel{(a)}{=} 2k^{1/p},$$

where (a) is due to $J_l^+ \cap J_l^- = \emptyset$ and $J_l^+ \cup J_l^- = [k]$. For $l \neq l'$,

$$d_p(x_{2l-1}, x_{2l'}) \stackrel{(a)}{\geq} d_p(c_l, c_{l'}) - d_p(c_l, x_{2l-1}) - d_p(c_{l'}, x_{2l'})$$

$$\stackrel{(b)}{\geq} 2d^2 - k^{1/p} - k^{1/p}$$

$$\stackrel{(c)}{\geq} 2d^2 - 2d^{1/p}$$

$$\stackrel{(d)}{\geq} 2d^{1/p}$$

$$\stackrel{(e)}{\geq} 2k^{1/p},$$

where (a) is by the triangle inequality under $\ell_p$-norm (namely, the Minkowski inequality), (b) uses $d_p(c_l, x_{2l-1}) = d_p(c_{l'}, x_{2l'}) = k^{1/p}$, (c),(e) is by $k \leq d$, and (d) holds for all $d \geq 2$ and $p \geq 1$. Thus, we have $\epsilon_{\mathcal{D},p} \geq k^{1/p}$.

Take $f \in \mathcal{F}_{d,P}$ that $\rho$-robustly memorize $\mathcal{D}$. We first lower bound the robustness radius $\mu$. Since $t \stackrel{\phi}{\mapsto} \sqrt[p]{\frac{t-1}{t}}$ is an strictly increasing function from $t \geq 1$ onto $[0, 1)$ [3], it has a well defined inverse mapping $\phi^{-1} : [0, 1) \to [1, \infty)$ defined as $\phi^{-1}(\rho) = \frac{1}{1-\rho^p}$. Therefore,

$$\rho = \phi(\phi^{-1}(\rho)) = \phi\left(\frac{1}{1-\rho^p}\right) \geq \phi\left(\lfloor \frac{1}{1-\rho^p} \rfloor\right) = \phi(k) = \sqrt[p]{\frac{k-1}{k}}.$$

---

[3] $\phi$ is a composition of two strictly increasing one-to-one corresponding functions $t \mapsto \frac{t-1}{t}$ from $[1, \infty)$ onto $[0, 1)$ and $u \mapsto \sqrt[p]{u}$ from $[0, 1)$ onto $[0, 1)$

Since $\epsilon_{\mathcal{D},p} \geq k^{1/p}$ and $\rho \geq (\frac{k-1}{k})^{1/p}$, we have $\mu = \rho\epsilon_{\mathcal{D},p} \geq \rho k^{1/p} \geq (k-1)^{1/p}$. Thus, every $f$ that $\rho$-robustly memorizes $\mathcal{D}$ must also memorize $(k-1)^{1/p}$ radius open $\ell_p$-ball around each point in $\mathcal{D}$ as the same label as the data point.

Moreover, for $\boldsymbol{x}_{l,j} \in \mathcal{X}$ with positive label $y_{l,j} = +1$, we have

$$\|\boldsymbol{x}_{l,j} - \boldsymbol{x}_{2l-1}\|_p = \left\| (\boldsymbol{c}_l + \boldsymbol{e}_j) - (\boldsymbol{c}_l + \sum_{j' \in J_l^+} \boldsymbol{e}_{j'} - \sum_{j' \in J_l^-} \boldsymbol{e}_{j'}) \right\|_p$$

$$= \left\| \sum_{\substack{j' \in J_l^+ \\ j' \neq j}} \boldsymbol{e}_{j'} - \sum_{j' \in J_l^-} \boldsymbol{e}_{j'} \right\|_p$$

$$= (k-1)^{1/p}.$$

Take a sequence of points $\{\boldsymbol{z}_n\}_{n \in \mathbb{N}}$ such that $\boldsymbol{z}_n \to \boldsymbol{x}_{l,j}$ as $n \to \infty$ [4] and

$$\|\boldsymbol{z}_n - \boldsymbol{x}_{2l-1}\|_p < (k-1)^{1/p},$$

for all $n \in \mathbb{N}$. In particular,

$$\boldsymbol{z}_n := \frac{n-1}{n}\boldsymbol{x}_{l,j} + \frac{1}{n}\boldsymbol{x}_{2l-1}$$

satisfies such properties. Then, we have $f(\boldsymbol{z}_n) = f(\boldsymbol{x}_{2l-1}) = 2$ for all $n \in \mathbb{N}$. Moreover, by the continuity of $f$ (under the usual topology),

$$f(\boldsymbol{x}_{l,j}) = f(\lim_{n\to\infty} \boldsymbol{z}_n) = \lim_{n\to\infty} f(\boldsymbol{z}_n) = \lim_{n\to\infty} 2 = 2.$$

Similarly, for $\boldsymbol{x}_{l,j}$ with negative label $y_{l,j} = -1$, we have $\|\boldsymbol{x}_{l,j} - \boldsymbol{x}_{2l}\|_p = (k-1)^{1/p}$, so that $f(\boldsymbol{x}_{l,j}) = 1$.

Since we can adjust the weight and the bias of the last hidden layer, $\mathcal{F}_{d,P}$ is closed under affine transformation; that is, $af + b \in \mathcal{F}_{d,P}$ whenever $f \in \mathcal{F}_{d,P}$. In particular, $f' := 2f - 3 \in \mathcal{F}_{d,P}$. This $f'$ satisfies $f'(\boldsymbol{x}_{l,j}) = 2f(\boldsymbol{x}_{l,j}) - 3 = 2 \cdot 2 - 3 = +1$ whenever $y_{l,j} = +1$ and $f'(\boldsymbol{x}_{l,j}) = 2f(\boldsymbol{x}_{l,j}) - 3 = 2 \cdot 1 - 3 = -1$ whenever $y_{l,j} = -1$. Thus, $\text{sign} \circ f'$ perfectly classify $\mathcal{X}$ with the label $\mathcal{Y}$. Since we can take such $f' \in \mathcal{F}_{d,P}$ given an arbitrary label $\mathcal{Y}$ of $\mathcal{X}$, it follows that $\mathcal{F}_{d,P}$ shatters $\mathcal{X}$, concluding the proof of the theorem. $\square$

### C.1.1 Lemmas for Appendix C.1

**Lemma C.9.** *Let $\{\boldsymbol{e}_j\}_{j \in [d]} \subseteq \mathbb{R}^d$ denote the standard basis in $\mathbb{R}^d$. Then, for any $k$-dimensional subspace $Z$ of $\mathbb{R}^d$ with $k \geq 1$ we have,*

$$\min_{j \in [d]} \text{dist}_\infty(\boldsymbol{e}_j, Z) \leq \frac{1}{2}.$$

*Proof.* For any subspace $Z'$ of $Z$, we have

$$\min_{j \in [d]} \text{dist}_\infty(\boldsymbol{e}_j, Z) \leq \min_{j \in [d]} \text{dist}_\infty(\boldsymbol{e}_j, Z').$$

As every $k$-dimensional subspace of $\mathbb{R}^d$ with $k \geq 1$ has a one-dimensional subspace, it suffices to prove the second statement for $k = 1$. i.e., for any one-dimensional subspace $Z$ of $\mathbb{R}^d$,

$$\min_{j \in [d]} \text{dist}_\infty(\boldsymbol{e}_j, Z) \leq \frac{1}{2}.$$

---

[4]We consider the convergence of the sequence on the usual topology induced by $\ell_2$-norm.

Let $Z = \text{Span}(\boldsymbol{z})$, where $\boldsymbol{z} = (z_1, \cdots, z_d) \neq \boldsymbol{0}$. Without loss of generality, let $\|\boldsymbol{z}\|_\infty = 1$ and take $j \in [d]$ such that $|z_j| = 1$. Let $\boldsymbol{z}' = \frac{z_j}{2}\boldsymbol{z} \in Z$. Then,

$$
\begin{aligned}
\|\boldsymbol{z}' - \boldsymbol{e}_j\|_\infty &= \left\|(\frac{z_j z_1}{2}, \cdots, \frac{z_j z_{j-1}}{2}, \frac{z_j z_j}{2} - 1, \frac{z_j z_{j+1}}{2}, \cdots, \frac{z_j z_d}{2})\right\|_\infty \\
&\overset{(a)}{=} \left\|(\frac{z_j z_1}{2}, \cdots, \frac{z_j z_{j-1}}{2}, -\frac{1}{2}, \frac{z_j z_{j+1}}{2}, \cdots, \frac{z_j z_d}{2})\right\|_\infty \\
&\overset{(b)}{\leq} \frac{1}{2},
\end{aligned}
$$

where (a) is by $|z_j| = 1$, and (b) is by $\|\boldsymbol{z}\|_\infty = 1$. Therefore,

$$
\min_{j' \in [d]} \text{dist}_\infty(\boldsymbol{e}_{j'}, Z) \leq \text{dist}_\infty(\boldsymbol{e}_j, Z) \leq \|\boldsymbol{z}' - \boldsymbol{e}_j\|_\infty \leq \frac{1}{2},
$$

concluding the statement. $\qquad\square$

The following lemma generalizes Lemma C.9 to the case where we consider only the distance to a subset of the standard basis, instead of the whole standard basis.

**Lemma C.10.** *For $1 \leq t \leq d$, let $\{\boldsymbol{e}_j\}_{j \in [t]} \subseteq \mathbb{R}^d$ denote the first $t$ vectors from the standard basis in $\mathbb{R}^d$. Then, for any $k$-dimensional subspace $Z$ of $\mathbb{R}^d$ with $k \geq d - t + 1$,*

$$
\min_{j \in [t]} \text{dist}_\infty(\boldsymbol{e}_j, Z) \leq \frac{1}{2}.
$$

*Proof.* Similar to Lemma A.2, we start by considering the dimension of the intersection between $Z$ and $\mathbb{R}^t$, both as a subspace of $\mathbb{R}^d$. Let $Q = [\boldsymbol{e}_1 \boldsymbol{e}_2 \cdots \boldsymbol{e}_t]^\top \in \mathbb{R}^{t \times d}$. Then,

$$
\mathbb{R}^d = \text{Col}(Q^\top) \oplus \text{Null}(Q) = (Z \cap \text{Col}(Q^\top)) \oplus (Z^\perp \cap \text{Col}(Q^\top)) \oplus \text{Null}(Q).
$$

By considering the dimension,

$$
\begin{aligned}
\dim(Z \cap \text{Col}(Q^\top)) &= \dim \mathbb{R}^d - \dim(Z^\perp \cap \text{Col}(Q^\top)) - \dim \text{Null}(Q) \\
&\geq \dim \mathbb{R}^d - \dim Z^\perp - \dim \text{Null}(Q) \\
&= d - (d - k) - (d - t) \\
&= k - (d - t).
\end{aligned}
$$

Under the assumption $k \geq d - t + 1$, we have

$$
\dim \phi(Z \cap \text{Col}(Q^\top)) = \dim(Z \cap \text{Col}(Q^\top)) \geq k - (d - t) \geq 1.
$$

Then,

$$
\begin{aligned}
\min_{j \in [t]} \text{dist}_\infty(\boldsymbol{e}_j, Z) &\leq \min_{j \in [t]} \text{dist}_\infty(\boldsymbol{e}_j, Z \cap \text{Col}(Q^\top)) \\
&= \min_{j \in [t]} \text{dist}_\infty(\phi(\boldsymbol{e}_j), \phi(Z \cap \text{Col}(Q^\top))) \\
&\overset{(b)}{\leq} \frac{1}{2},
\end{aligned}
$$

where (b) is by Lemma C.9. $\qquad\square$

## C.2 Extension of Sufficiency Condition to $\ell_p$-norm

**Theorem C.11.** *Let $p \in [1, \infty]$. For any dataset $\mathcal{D} \in \boldsymbol{D}_{d,N,C}$ and $\eta \in (0, 1)$, the following statements hold:*

(i) *If $\rho \in \left(0, \frac{1}{5N\sqrt{d}\gamma_p(d)}\right]$, there exists $f \in \mathcal{F}_{d,P}$ with $P = \tilde{O}(\sqrt{N})$ that $\rho$-robustly memorizes $\mathcal{D}$ under $\ell_p$-norm.*

(ii) If $\rho \in \left( \frac{1}{5N\sqrt{d}\gamma_p(d)}, \frac{1}{5\sqrt{d}\gamma_p(d)} \right]$, there exists $f \in \mathcal{F}_{d,P}$ with $P = \tilde{O}(Nd^{\frac{1}{4}}\rho^{\frac{1}{2}}\gamma_p(d)^{\frac{1}{2}})$ that $\rho$-robustly memorizes $\mathcal{D}$ under $\ell_p$-norm with error at most $\eta$.

(iii) If $\rho \in \left( \frac{1}{5\sqrt{d}\gamma_p(d)}, \frac{1}{\gamma_p(d)} \right)$, there exists $f \in \mathcal{F}_{d,P}$ with $P = \tilde{O}(Nd^2\rho^4\gamma_p(d)^4)$ that $\rho$-robustly memorizes $\mathcal{D}$ under $\ell_p$-norm.

To prove Theorem C.11, we decompose it into three theorems (Theorems C.12 to C.14), each corresponding to one of the cases in the statement. They are following.

**Theorem C.12.** *Let $\rho \in \left( 0, \frac{1}{5N\sqrt{d}\gamma_p(d)} \right]$ and $p \in [1, \infty]$. For any dataset $\mathcal{D} \in \boldsymbol{D}_{d,N,C}$, there exists $f \in \mathcal{F}_{d,P}$ with $P = \tilde{O}(\sqrt{N})$ that $\rho$-robustly memorizes $\mathcal{D}$ under $\ell_p$-norm.*

*Proof.* Let $\rho' = \gamma_p(d)\rho$. Then, we have $\rho' \in \left( 0, \frac{1}{5N\sqrt{d}} \right]$ from the condition of $\rho$. By Theorem 4.2(i), there exists $f \in \mathcal{F}_{d,P}$ with $P = \tilde{O}(\sqrt{N})$ that $\rho'$-robustly memorizes $\mathcal{D}$ under $\ell_2$-norm. In other words, it holds $f(\boldsymbol{x}') = y_i$, for all $(\boldsymbol{x}_i, y_i) \in \mathcal{D}$ and $\boldsymbol{x}' \in \mathcal{B}_2(\boldsymbol{x}_i, \rho'\epsilon_{\mathcal{D},2})$.

We consider two cases depending on whether $p \geq 2$ or $p < 2$, which affect the direction of inclusion between $\ell_p$ and $\ell_2$ balls.

**Case I : $p \geq 2$.** In this case, we have

$$\mathcal{B}_p(\boldsymbol{x}_i, \rho\epsilon_{\mathcal{D},p}) \overset{(a)}{\subseteq} \mathcal{B}_p(\boldsymbol{x}_i, \rho\epsilon_{\mathcal{D},2}) \overset{(b)}{\subseteq} \mathcal{B}_2(\boldsymbol{x}_i, \gamma_p(d)\rho\epsilon_{\mathcal{D},2}) = \mathcal{B}_2(\boldsymbol{x}_i, \rho'\epsilon_{\mathcal{D},2}),$$

where (a) holds by Equation (61) and (b) holds by Lemma C.4 applying $p = 2$ and $q = p$.

Thus, for all $(\boldsymbol{x}_i, y_i) \in \mathcal{D}$ and $\boldsymbol{x}' \in \mathcal{B}_p(\boldsymbol{x}_i, \rho\epsilon_{\mathcal{D},p})$, it also holds $f(\boldsymbol{x}') = y_i$. In other words, $f$ $\rho$-robustly memorizes $\mathcal{D}$ under $\ell_p$-norm with $\tilde{O}(\sqrt{N})$ parameters.

**Case II : $p < 2$.** In this case, we have

$$\mathcal{B}_p(\boldsymbol{x}_i, \rho\epsilon_{\mathcal{D},p}) \overset{(a)}{\subseteq} \mathcal{B}_p(\boldsymbol{x}_i, \gamma_p(d)\rho\epsilon_{\mathcal{D},2}) \overset{(b)}{\subseteq} \mathcal{B}_2(\boldsymbol{x}_i, \gamma_p(d)\rho\epsilon_{\mathcal{D},2}) = \mathcal{B}_2(\boldsymbol{x}_i, \rho'\epsilon_{\mathcal{D},2}),$$

where (a) holds by Equation (62) and (b) holds by Lemma C.4 applying $p = p$ and $q = 2$.

Thus, for all $(\boldsymbol{x}_i, y_i) \in \mathcal{D}$ and $\boldsymbol{x}' \in \mathcal{B}_p(\boldsymbol{x}_i, \rho\epsilon_{\mathcal{D},p})$, it also holds $f(\boldsymbol{x}') = y_i$. In other words, $f$ $\rho$-robustly memorizes $\mathcal{D}$ under $\ell_p$-norm with $\tilde{O}(\sqrt{N})$ parameters.

$\square$

**Theorem C.13.** *Let $\rho \in \left( \frac{1}{5N\sqrt{d}\gamma_p(d)}, \frac{1}{5\sqrt{d}\gamma_p(d)} \right]$ and $p \in [1, \infty]$. For any dataset $\mathcal{D} \in \boldsymbol{D}_{d,N,C}$, there exists $f \in \mathcal{F}_{d,P}$ with $P = \tilde{O}(Nd^{\frac{1}{4}}\rho^{\frac{1}{2}}\gamma_p(d)^{\frac{1}{2}})$ that $\rho$-robustly memorizes $\mathcal{D}$ under $\ell_p$-norm with error at most $\eta$.*

*Proof.* Let $\rho' = \gamma_p(d)\rho$. Then, we have $\rho' \in \left( \frac{1}{5N\sqrt{d}}, \frac{1}{5\sqrt{d}} \right)$ from the condition of $\rho$.

We consider two cases depending on whether $p \geq 2$ or $p < 2$, which affect the direction of inclusion between $\ell_p$ and $\ell_2$ balls.

**Case I : $p \geq 2$.** In this case, we have:

$$\mathcal{B}_p(\boldsymbol{x}_i, \rho\epsilon_{\mathcal{D},p}) \overset{(a)}{\subseteq} \mathcal{B}_p(\boldsymbol{x}_i, \rho\epsilon_{\mathcal{D},2}) \overset{(b)}{\subseteq} \mathcal{B}_2(\boldsymbol{x}_i, \gamma_p(d)\rho\epsilon_{\mathcal{D},2}) = \mathcal{B}_2(\boldsymbol{x}_i, \rho'\epsilon_{\mathcal{D},2}),$$

where (a) holds by Equation (61) and (b) holds by Lemma C.4 applying $p = 2$ and $q = p$.

**Case II : $p < 2$.** In this case, we have:

$$\mathcal{B}_p(\boldsymbol{x}_i, \rho\epsilon_{\mathcal{D},p}) \overset{(a)}{\subseteq} \mathcal{B}_p(\boldsymbol{x}_i, \gamma_p(d)\rho\epsilon_{\mathcal{D},2}) \overset{(b)}{\subseteq} \mathcal{B}_2(\boldsymbol{x}_i, \gamma_p(d)\rho\epsilon_{\mathcal{D},2}) = \mathcal{B}_2(\boldsymbol{x}_i, \rho'\epsilon_{\mathcal{D},2}),$$

where (a) holds by Equation (62) and (b) holds by Lemma C.4 applying $p = p$ and $q = 2$.

Thus, in both cases, it holds:

$$\mathcal{B}_p(\boldsymbol{x}_i, \rho\epsilon_{\mathcal{D},p}) \subseteq \mathcal{B}_2(\boldsymbol{x}_i, \rho'\epsilon_{\mathcal{D},2}). \tag{65}$$

We define $\eta' = \eta\frac{\text{Vol}(\mathcal{B}_p(\boldsymbol{x}_i,\rho\epsilon_{\mathcal{D},p}))}{\text{Vol}(\mathcal{B}_2(\boldsymbol{x}_i,\rho'\epsilon_{\mathcal{D},2}))}$. We apply Theorem 4.2(ii) with the robustness ratio $\rho'$ and the error rate $\eta'$, then we obtain $f \in \mathcal{F}_{d,P}$ with $P = \tilde{O}(Nd^{\frac{1}{4}}\rho'^{\frac{1}{2}}) = \tilde{O}(Nd^{\frac{1}{4}}\rho^{\frac{1}{2}}\gamma_p(d)^{\frac{1}{2}})$ that $\rho'$-robustly memorizes $\mathcal{D}$ with error at most $\eta'$ under $\ell_2$-norm. In other words, for all $(\boldsymbol{x}_i, y_i) \in \mathcal{D}$, it holds that

$$\mathbb{P}_{\boldsymbol{x}'\sim\text{Unif}(\mathcal{B}_2(\boldsymbol{x}_i,\rho'\epsilon_{\mathcal{D},2}))}[f(\boldsymbol{x}') \neq y_i] < \eta'. \tag{66}$$

For simplicity, we denote $E = \{\boldsymbol{x} \in \mathbb{R}^d \mid f(\boldsymbol{x}') \neq y_i\}$. Then, we have

$$\mathbb{P}_{\boldsymbol{x}'\sim\text{Unif}(\mathcal{B}_p(\boldsymbol{x}_i,\rho\epsilon_{\mathcal{D},p}))}[f(\boldsymbol{x}') \neq y_i]$$
$$=\mathbb{P}_{\boldsymbol{x}'\sim\text{Unif}(\mathcal{B}_p(\boldsymbol{x}_i,\rho\epsilon_{\mathcal{D},p}))}[\boldsymbol{x} \in E]$$
$$=\frac{\text{Vol}(E \cap \mathcal{B}_p(\boldsymbol{x}_i, \rho\epsilon_{\mathcal{D},p}))}{\text{Vol}(\mathcal{B}_p(\boldsymbol{x}_i, \rho\epsilon_{\mathcal{D},p}))}$$
$$\overset{(a)}{\leq}\frac{\text{Vol}(E \cap \mathcal{B}_2(\boldsymbol{x}_i, \rho'\epsilon_{\mathcal{D},2}))}{\text{Vol}(\mathcal{B}_p(\boldsymbol{x}_i, \rho\epsilon_{\mathcal{D},p}))}$$
$$=\frac{\text{Vol}(E \cap \mathcal{B}_2(\boldsymbol{x}_i, \rho'\epsilon_{\mathcal{D},2}))}{\text{Vol}(\mathcal{B}_2(\boldsymbol{x}_i, \rho'\epsilon_{\mathcal{D},2}))}\frac{\text{Vol}(\mathcal{B}_2(\boldsymbol{x}_i, \rho'\epsilon_{\mathcal{D},2}))}{\text{Vol}(\mathcal{B}_p(\boldsymbol{x}_i, \rho\epsilon_{\mathcal{D},p}))}$$
$$=\mathbb{P}_{\boldsymbol{x}'\sim\text{Unif}(\mathcal{B}_2(\boldsymbol{x}_i,\rho'\epsilon_{\mathcal{D},2}))}[\boldsymbol{x}' \in E] \cdot \frac{\text{Vol}(\mathcal{B}_2(\boldsymbol{x}_i, \rho'\epsilon_{\mathcal{D},2}))}{\text{Vol}(\mathcal{B}_p(\boldsymbol{x}_i, \rho\epsilon_{\mathcal{D},p}))}$$
$$=\mathbb{P}_{\boldsymbol{x}'\sim\text{Unif}(\mathcal{B}_2(\boldsymbol{x}_i,\rho'\epsilon_{\mathcal{D},2}))}[f(\boldsymbol{x}') \neq y_i] \cdot \frac{\text{Vol}(\mathcal{B}_2(\boldsymbol{x}_i, \rho'\epsilon_{\mathcal{D},2}))}{\text{Vol}(\mathcal{B}_p(\boldsymbol{x}_i, \rho\epsilon_{\mathcal{D},p}))}$$
$$\overset{(b)}{<}\eta'\frac{\text{Vol}(\mathcal{B}_2(\boldsymbol{x}_i, \rho'\epsilon_{\mathcal{D},2}))}{\text{Vol}(\mathcal{B}_p(\boldsymbol{x}_i, \rho\epsilon_{\mathcal{D},p}))}$$
$$\overset{(c)}{=}\eta,$$

where (a) holds by Equation (65), (b) holds by Equation (66), and (c) holds by the definition of $\eta'$.

Thus, for all $(\boldsymbol{x}_i, y_i) \in \mathcal{D}$, it holds:

$$\mathbb{P}_{\boldsymbol{x}'\sim\text{Unif}(\mathcal{B}_p(\boldsymbol{x}_i,\rho\epsilon_{\mathcal{D},p}))}[f(\boldsymbol{x}') \neq y_i] < \eta.$$

In other words, $f$ $\rho$-robustly memorizes $\mathcal{D}$ under $\ell_p$-norm with error at most $\eta$ and $\tilde{O}(Nd^{\frac{1}{4}}\rho^{\frac{1}{2}}\gamma_p(d)^{\frac{1}{2}})$ parameters.

$\square$

**Theorem C.14.** *Let $\rho \in \left(\frac{1}{5\sqrt{d}\gamma_p(d)}, \frac{1}{\gamma_p(d)}\right)$ and $p \in [1, \infty]$. For any dataset $\mathcal{D} \in \boldsymbol{D}_{d,N,C}$, there exists $f \in \mathcal{F}_{d,P}$ with $P = \tilde{O}(Nd^2\rho^4\gamma_p(d)^4)$ that $\rho$-robustly memorizes $\mathcal{D}$ under $\ell_p$-norm.*

*Proof.* Let $\rho' = \gamma_p(d)\rho$. Then, we have $\rho' \in \left(\frac{1}{5\sqrt{d}}, 1\right)$ from the condition of $\rho$. By Theorem 4.2(iii), there exists $f \in \mathcal{F}_{d,P}$ with $P = \tilde{O}(Nd^2\rho'^4) = \tilde{O}(Nd^2\rho^4\gamma_p(d)^4)$ that $\rho'$-robustly memorizes $\mathcal{D}$ under $\ell_2$-norm. In other words, it holds $f(\boldsymbol{x}') = y_i$, for all $(\boldsymbol{x}_i, y_i) \in \mathcal{D}$ and $\boldsymbol{x}' \in \mathcal{B}_2(\boldsymbol{x}_i, \rho'\epsilon_{\mathcal{D},2})$.

We consider two cases depending on whether $p \geq 2$ or $p < 2$, which affect the direction of inclusion between $\ell_p$ and $\ell_2$ balls.

**Case I :** $p \geq 2$.  In this case, we have:

$$\mathcal{B}_p(\boldsymbol{x}_i, \rho\epsilon_{\mathcal{D},p}) \overset{(a)}{\subseteq} \mathcal{B}_p(\boldsymbol{x}_i, \rho\epsilon_{\mathcal{D},2}) \overset{(b)}{\subseteq} \mathcal{B}_2(\boldsymbol{x}_i, \gamma_p(d)\rho\epsilon_{\mathcal{D},2}) = \mathcal{B}_2(\boldsymbol{x}_i, \rho'\epsilon_{\mathcal{D},2}),$$

where (a) holds by Equation (61) and (b) holds by Lemma C.4 applying $p = 2$ and $q = p$.

Thus, for all $(\boldsymbol{x}_i, y_i) \in \mathcal{D}$ and $\boldsymbol{x}' \in \mathcal{B}_p(\boldsymbol{x}_i, \rho\epsilon_{\mathcal{D},p})$, it also holds $f(\boldsymbol{x}') = y_i$. In other words, $f$ $\rho$-robustly memorizes $\mathcal{D}$ under $\ell_p$-norm with $\tilde{O}(Nd^2\rho^4\gamma_p(d)^4)$ parameters.

**Case II :** $p < 2$.  In this case, we have:

$$\mathcal{B}_p(\boldsymbol{x}_i, \rho\epsilon_{\mathcal{D},p}) \overset{(a)}{\subseteq} \mathcal{B}_p(\boldsymbol{x}_i, \gamma_p(d)\rho\epsilon_{\mathcal{D},2}) \overset{(b)}{\subseteq} \mathcal{B}_2(\boldsymbol{x}_i, \gamma_p(d)\rho\epsilon_{\mathcal{D},2}) = \mathcal{B}_2(\boldsymbol{x}_i, \rho'\epsilon_{\mathcal{D},2}),$$

where (a) holds by Equation (62) and (b) holds by Lemma C.4 applying $p = p$ and $q = 2$.

Thus, for all $(\boldsymbol{x}_i, y_i) \in \mathcal{D}$ and $\boldsymbol{x}' \in \mathcal{B}_p(\boldsymbol{x}_i, \rho\epsilon_{\mathcal{D},p})$, it also holds $f(\boldsymbol{x}') = y_i$. In other words, $f$ $\rho$-robustly memorizes $\mathcal{D}$ under $\ell_p$-norm with $\tilde{O}(Nd^2\rho^4\gamma_p(d)^4)$ parameters.

$\square$

# D Comparison to Existing Bounds

## D.1 Summary of Parameter Complexity across $\ell_p$-norms

Table 1: Summary of our results and a comparison with prior works. We omit the constants for the range of $\rho$. $\gamma_p(d) = 1$ under $p = 2$ reduces to the results in Sections 3 and 4.

| | $\ell_p$-norm | Robustness Ratio $\rho$ | Bound on Parameters |
|---|---|---|---|
| LB | $p > 2$ | $(0, 1)$ | $\Omega\left(\min\{N, d\}d\rho^2\right)$, Proposition C.6 |
| | $p \le 2$ | $(0, 1)$ | $\Omega\left(\min\{N, d\}d\left(\rho/\gamma_p(d)\right)^2\right)$, Proposition C.6 |
| | $p = \infty$ | $(1/2, 1)$ | $\Omega\left(\min\{N, d\}d\right)$, Proposition C.7 |
| | $p = \infty$ | $0.8$ | $\Omega\left(d^2\right)$, Yu et al. [2024] [1] |
| | $p < \infty$ | $\left(0, (1 - \frac{1}{d})^{1/p}\right]$ | $\Omega\left(\sqrt{\frac{N}{1-\rho^p}}\right)$, Proposition C.8 |
| | $p = 2$ | $\rho \to 1$ | $\Omega\left(\sqrt{Nd}\right)$, Li et al. [2022] [2] |
| UB | $p = p$ | $\left(0, \frac{1}{\gamma_p(d)N\sqrt{d}}\right)$ | $\tilde{O}\left(\sqrt{N}\right)$, Theorem C.12 |
| | | $\left(\frac{1}{\gamma_p(d)N\sqrt{d}}, \frac{1}{\gamma_p(d)\sqrt{d}}\right)$ | $\tilde{O}\left(Nd^{1/4}(\rho\gamma_p(d))^{1/2}\right)$, Theorem C.13 |
| | | $\left(0, \frac{1}{\gamma_p(d)\sqrt{d}}\right)$ | $\tilde{O}(N)$, Egosi et al. [2025] |
| | | $\left(\frac{1}{\gamma_p(d)\sqrt{d}}, \frac{1}{\gamma_p(d)}\right)$ | $\tilde{O}(Nd^2(\rho\gamma_p(d))^2)$, Theorem C.14 |
| | | | $\tilde{O}\left(Nd^3(\rho\gamma_p(d))^6\right)$, Egosi et al. [2025] |
| | | $(0, 1)$ | $\tilde{O}\left(Nd^2p^2\right)$, Yu et al. [2024] [3] |

[1] Requires $N > d$.
[2] The result only holds for $\rho$ sufficiently close to 1.
[3] Requires $p \in \mathbb{N}$.

## D.2 Parameter Complexity of the Construction by Yu et al. [2024]

We now analyze the number of parameters of the network construction proposed by Yu et al. [2024], which provides the upper bound not depending on $\rho$, but still applies to all $\rho \in (0, 1)$.

**Lemma D.1** (Theorem B.6, Yu et al. [2024]). *Let $p \in \mathbb{N}$. For any dataset $\mathcal{D} = \{(\boldsymbol{x}_i, y_i)\}_{i \in [N]} \in \boldsymbol{D}_{d,N,C}$, let $R > 1$ by any real value with $\|\boldsymbol{x}_i\|_\infty \le R$ for all $i \in [N]$. For $\rho \in (0, 1)$, define $\gamma := (1 - \rho)\epsilon_{\mathcal{D},p} > 0$. Then, there exists a network with width $O(d)$, and depth $O(Np(\log(\frac{d}{\gamma^p}) + p \log R + \log p))$ that $\rho$-robustly memorize $\mathcal{D}$ under $\ell_p$-norm.*

We note that in the Yu et al. [2024] uses the notation $\lambda_{\mathcal{D}}^p/2$ for $\epsilon_{\mathcal{D},p}$, and the radius $\lambda_{\mathcal{D}}^p/2 - \gamma$ in the original statement corresponds to the value $\mu := \rho\epsilon_{\mathcal{D},p}$ in our notation. We count parameters in their construction in the following lemma, specifically in the case $p = 2$. Although the original statement of Yu et al. [2024] includes a parameter count, they consider a different parameter counting strategy–by counting only the number of nonzero parameters. We therefore count the number of all parameters following Equation (3) in the subsequent lemma. Note that results and comparison under nonzero parameter counts are provided in Appendix E.

**Lemma D.2.** *For any $\mathcal{D} \in \boldsymbol{D}_{d,N,C}$ and $\rho \in (0, 1)$, define $\gamma := (1 - \rho)\epsilon_{\mathcal{D}} > 0$ and $R > 1$ with $\|\boldsymbol{x}_i\|_\infty \le R$ for all $i \in [N]$. Then, there exists a neural network $f$ such that $\rho$-robustly memorizes $\mathcal{D}$ using at most $O(Nd^2(\log(\frac{d}{\gamma^2}) + \log R))$ parameters. Moreover, the network has width $O(d)$ and depth $\tilde{O}(N)$.*

*Proof.* By applying Lemma D.1 with $p = 2$, we obtain a neural network $f$ that $\rho$-robustly memorizes $\mathcal{D}$ with width $O(d)$, and depth $L = O(N(\log(\frac{d}{\gamma^2} + \log R)))$. In their construction, $d_l = \Theta(d)$ through all $l$, as the input $x$ propagates over the layers using a width $d$. We count all parameters as defined in Equation (3), so we can upper bound the number of parameters used for the construction

of $f$ as follows:

$$\sum_{l=1}^{L}(d_{l-1}+1)\cdot d_l = \sum_{l=1}^{L}\Theta(d)\cdot\Theta(d)$$

$$= \Theta(N(\log(\frac{d}{\gamma^2})+\log R))\cdot\Theta(d^2)$$

$$= \Theta(Nd^2(\log(\frac{d}{\gamma^2})+\log R)).$$

$\square$

## D.3  Parameter Complexity of the Construction by Egosi et al. [2025]

We observe that although Egosi et al. [2025] do not explicitly quantify the total number of parameters in their construction, it implicitly yields a network with $O(Nd^3\rho^6)$ parameters. Specifically, we can establish the following:

For any $\mathcal{D} \in \boldsymbol{D}_{d,N,C}$ and $\rho \in (\frac{1}{\sqrt{d}}, 1)$, there exists a neural network $f$ that $\rho$-robustly memorizes $\mathcal{D}$ using $\tilde{O}(Nd^3\rho^6)$ parameters.

This result follows from the network constructed in Theorem 4.4 of Egosi et al. [2025]. The proof of Theorem 4.4 proceeds under the assumption that for $7 \leq k \leq d+5$, and $\rho \leq \frac{1}{4\sqrt{e}}\sqrt{\frac{k-6}{d}}N^{-\frac{2}{k-6}}$. Given this range, Theorem 4.2 of Egosi et al. [2025] is applied to construct a robust memorizer of the projected data from $\mathbb{R}^d$ to $\mathbb{R}^k$. Figures 4 and 5 in their paper illustrate this construction. In this construction, the projected point propagates through the network $\Theta(Nk)$ times. The width of the network scales with $k$, while the other component, that is not propagating the point remains constant in width. Thus, the number of parameters used for the construction is given by:

$$\sum_{l=1}^{L}(d_{l-1}+1)\cdot d_l = \sum_{l=1}^{\Theta(Nk)}\Theta(k^2) = \Theta(Nk\cdot k^2) = \Theta(Nk^3).$$

To translate this to a bound in terms of $\rho$, we analyze the relationship between $\rho$ and $k$. For $k \geq 4\log N + 6$, we verify the following inequality:

$$\frac{1}{4\sqrt{e}}\sqrt{\frac{k-6}{d}}N^{-\frac{2}{k-6}} \geq \frac{1}{4\sqrt{e}}\sqrt{\frac{k-6}{d}}N^{-\frac{1}{2\log N}} \overset{(a)}{=} \frac{1}{4e}\sqrt{\frac{k-6}{d}}$$

where (a) holds by $N = e^{\log N}$. Therefore, for $\rho = \frac{1}{4e}\sqrt{\frac{k-6}{d}}$, the network $\rho$-robustly memorizes $\mathcal{D}$ with $\Theta(Nk^3)$ parameters. From the relationship between $\rho$ and $k$, solving for $k$ in terms of $\rho$ yields $k = \Theta(d\rho^2)$. Since the minimum value of $k$ under the assumption is 7, the minimum achievable $\rho$ is $\frac{1}{4e}\frac{1}{\sqrt{d}}$.

Thus, for $\rho > \frac{1}{\sqrt{d}}$, the construction yields a network that $\rho$-robustly memorizes $\mathcal{D}$ with $\Theta(Nk^3) = \Theta(Nd^3\rho^6)$ parameters, as desired.

# E   Nonzero Parameter Counts

While our main parameter counting method follows the approach of counting all parameters, including zeros, as defined in Equation (3), some prior works on memorization and robust memorization adopt a different parameter counting strategy—counting only the nonzero parameters. We emphasize that counting all parameters, including zeros, better aligns with how the matrices are stored in practice. Nevertheless, we also present how our results extend to the case of counting only nonzero parameters, offering an alternative perspective for interpreting our findings and comparing them with prior work.

In contrast to Equation (4), let us define the set of neural networks with input dimension $d$ and at most $P$ nonzero parameters by

$$\bar{\mathcal{F}}_{d,P} = \big\{ f : \mathbb{R}^d \to \mathbb{R} \mid f \text{ is a neural network with at most } P \text{ nonzero parameters} \big\}. \tag{67}$$

## E.1   Nonzero Parameter Counts: An illustration.

We provide the corresponding illustration of Figure 1 under only nonzero parameter counting in Figure 8, combining Theorem E.1 and Theorem E.2.

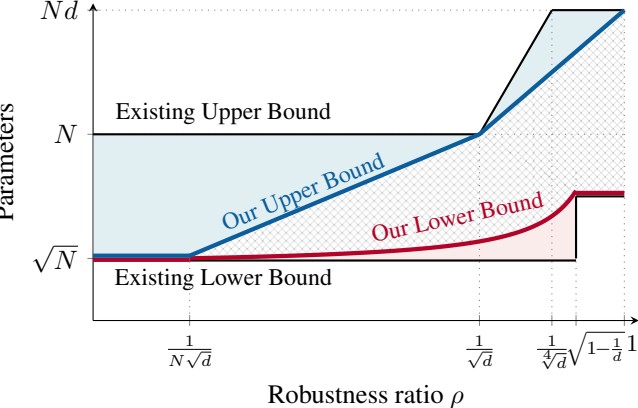

Figure 8: Summary of parameter bounds, counting only nonzero parameters on a log-log scale when $d = \Theta(\sqrt{N})$. We omit constant factors in both axes. Solid blue and red curves show the sufficient (Theorem E.2) and necessary (Theorem E.1) numbers of parameters, respectively; the solid black curve is the best prior bound. Light-blue shading highlights our improvement in the upper bound, and light-red shading highlights our improvement in the lower bound. The cross-hatched area marks the remaining gap.

## E.2   Nonzero Parameter Counts: Lower Bounds

The lower bound in Theorem 3.1 that counts all parameters consists of two terms: one based on the network width and another based on the VC-dimension. Although the lower bound by VC-dimension remains valid even when counting only nonzero parameters, the lower bound on the first hidden layer width can be translated into a lower bound on parameters only if we also include zero-valued parameters in the parameter counting convention. As a result, we obtain the following lower bound consisting of only the lower bound from the VC-dimension.

**Theorem E.1.** *Let $\rho \in (0,1)$. Suppose for any $\mathcal{D} \in \boldsymbol{D}_{d,N,2}$, there exists a neural network $f \in \bar{\mathcal{F}}_{d,P}$ that can $\rho$-robustly memorize $\mathcal{D}$. Then, the number of parameters $P$ must satisfy*

$$P = \Omega\left( \min\left\{ \frac{1}{\sqrt{1-\rho^2}}, \sqrt{d} \right\} \sqrt{N} \right).$$

The main reason why the VC-dimension lower bound remains valid even for the nonzero parameter count is because the key relation VC-dim$(\bar{\mathcal{F}}_{d,P}) = O(P^2)$ [Goldberg and Jerrum, 1995] holds even for the $\bar{\mathcal{F}}_{d,P}$ instead of $\mathcal{F}_{d,P}$. Below, we provide an explicit proof of the Theorem E.1.

*Proof.* Since $\mathcal{F}_{d,P} \subseteq \bar{\mathcal{F}}_{d,P}$, we have VC-dim$(\mathcal{F}_{d,P}) \leq$ VC-dim$(\bar{\mathcal{F}}_{d,P})$. In particular, by Equation (7), we have for $\rho \in \left(0, \sqrt{1 - \frac{1}{d}}\right]$ that

$$\text{VC-dim}(\bar{\mathcal{F}}_{d,P}) \geq \text{VC-dim}(\mathcal{F}_{d,P}) = \Omega\left(\frac{N}{1-\rho^2}\right).$$

By [Goldberg and Jerrum, 1995], we have VC-dim$(\bar{\mathcal{F}}_{d,P}) = O(P^2)$. Combining the two relations proves that for $\rho \in \left(0, \sqrt{1 - \frac{1}{d}}\right]$,

$$P = \Omega\left(\sqrt{\frac{N}{1-\rho^2}}\right).$$

Since $\frac{1}{\sqrt{1-\rho^2}} \leq \sqrt{d}$ for $\rho \in \left(0, \sqrt{1 - \frac{1}{d}}\right]$, the following relation holds:

$$\min\{\frac{1}{\sqrt{1-\rho^2}}, \sqrt{d}\}\sqrt{N} = \sqrt{\frac{N}{1-\rho^2}}.$$

For $\rho \in \left(\sqrt{1 - \frac{1}{d}}, 1\right)$, the lower bound $P = \Omega(\sqrt{Nd})$ obtained by the case $\rho = \sqrt{1 - \frac{1}{d}}$ also can be applied. Since $\frac{1}{\sqrt{1-\rho^2}} > \sqrt{d}$ for $\rho \in \left(\sqrt{1 - \frac{1}{d}}, 1\right)$, the following relation holds:

$$\min\{\frac{1}{\sqrt{1-\rho^2}}, \sqrt{d}\}\sqrt{N} = \sqrt{Nd}.$$

As a result, applying $P = \Omega\left(\sqrt{\frac{N}{1-\rho^2}}\right)$ for $\rho \in \left(0, \sqrt{1 - \frac{1}{d}}\right]$ and $\Omega(\sqrt{Nd})$ for $\rho \in \left(\sqrt{1 - \frac{1}{d}}, 1\right)$ results in

$$P = \Omega\left(\min\left\{\frac{1}{\sqrt{1-\rho^2}}, \sqrt{d}\right\}\sqrt{N}\right)$$

$\square$

### E.3 Nonzero Parameter Counts: Upper Bounds

While upper bounds on parameter counts of all parameters in Theorem 4.2 are naturally an upper bound for parameter counts of nonzero parameters, we provide a tighter upper bound regarding the nonzero parameters.

**Theorem E.2.** *For any dataset $\mathcal{D} \in \boldsymbol{D}_{d,N,C}$ and $\eta \in (0,1)$, the following statements hold:*

(i) *If $\rho \in \left(0, \frac{1}{5N\sqrt{d}}\right]$, there exists $f \in \bar{\mathcal{F}}_{d,P}$ with $P = \tilde{O}(\sqrt{N} + d)$ that $\rho$-robustly memorizes $\mathcal{D}$.*

(ii) *If $\rho \in \left(\frac{1}{5N\sqrt{d}}, \frac{1}{5\sqrt{d}}\right]$, there exists $f \in \bar{\mathcal{F}}_{d,P}$ with $P = \tilde{O}(Nd^{\frac{1}{4}}\rho^{\frac{1}{2}} + d)$ that $\rho$-robustly memorizes $\mathcal{D}$ with error at most $\eta$.*

(iii) *If $\rho \in \left(\frac{1}{5\sqrt{d}}, 1\right)$, there exists $f \in \bar{\mathcal{F}}_{d,P}$ with $P = \tilde{O}(Nd\rho^2 + d)$ that $\rho$-robustly memorizes $\mathcal{D}$.*

In comparison to the total parameter count as in Theorem 4.2, only Theorem E.2(iii) have a modified rate from $P = \tilde{O}(Nd^2\rho^4)$ to $P = \tilde{O}(Nd\rho^2)$. Below, we provide an explicit proof of Theorem E.2. The $d$ term in the parameter bounds of all three cases comes from the upper bound on the parameters of the first hidden layer.

*Proof.* Upper bounds on all parameter counts are natural upper bounds on the nonzero parameter counts. Since Theorem E.2(i) and Theorem E.2(ii) claims the same rate as Theorem 4.2(i) and

Theorem 4.2(ii) respectively, they trivially follows from Theorem 4.2. Another way of speaking, $\mathcal{F}_{d,P} \subseteq \bar{\mathcal{F}}_{d,P}$ and the first two cases directly follow from Theorem 4.2.

Now let us prove Theorem E.2(iii). Here, we mainly follow the proof of Theorem B.14, where instead of counting every parameter using Lemma D.2, we count only the nonzero parameters using Lemma E.3. We divide the cases into five, following Theorem B.14 as in Figure 6.

Let $\mathcal{D} = \{(\boldsymbol{x}_i, y_i)\}_{i \in [N]} \in \boldsymbol{D}_{d,N,C}$ be given. We divide the proof into five cases, the first case under $\rho \in [1/3, 1)$, the second case under $\rho \in (1/5\sqrt{d}, 1/3)$ and $d < 600 \log N$, the third case under $\rho \in (1/5\sqrt{d}, 1/3)$ and $N < 600 \log N \leq d$, the fourth case under $\rho \in (1/5\sqrt{d}, 1/3)$, $N \geq d \geq 600 \log N$, and finally the fifth case under $\rho \in (1/5\sqrt{d}, 1/3)$ and $d > N \geq 600 \log N$. To check that these cases cover all the cases, refer to Figure 6.

**Case I: $\rho \in [1/3, 1)$.** Let us denote $R := \max_{i \in [N]} \|\boldsymbol{x}_i\|_2$ and $\gamma := (1 - \rho)\epsilon_{\mathcal{D}}$. Note that $R \geq \|\boldsymbol{x}_i\|_\infty$ for all $i \in [N]$ as $\|\boldsymbol{x}\|_2 \geq \|\boldsymbol{x}\|_\infty$ for all $\boldsymbol{x} \in \mathbb{R}^d$. By applying Lemma E.3, there exists $f \in \bar{\mathcal{F}}_{d,P}$ with $P = O(Nd(\log(\frac{d}{\gamma^2}) + \log R))$ *nonzero* parameters that $\rho$-robustly memorize $\mathcal{D}$. The number of *nonzero* parameters can be further bounded as follows:

$$O(Nd(\log(\frac{d}{\gamma^2}) + \log R)) \overset{(a)}{=} O(Nd\rho^2 \cdot (\log(\frac{d}{\gamma^2}) + \log R)) \overset{(b)}{=} \tilde{O}(Nd\rho^2),$$

where (a) is due to $\rho = \Omega(1)$, (b) hides the logarithmic factors.

**Case II: $\rho \in (1/5\sqrt{d}, 1/3)$ and $d < 600 \log N$.** Let us denote $R := \max_{i \in [N]} \|\boldsymbol{x}_i\|_2$ and $\gamma := (1 - \rho)\epsilon_{\mathcal{D}}$. Note that $R \geq \|\boldsymbol{x}_i\|_\infty$ for all $i \in [N]$ as $\|\boldsymbol{x}\|_2 \geq \|\boldsymbol{x}\|_\infty$ for all $\boldsymbol{x} \in \mathbb{R}^d$. By Lemma E.3, there exists $f \in \bar{\mathcal{F}}_{d,P}$ with $P = O(Nd(\log(\frac{d}{\gamma^2}) + \log R))$ *nonzero* parameters that $\rho$-robustly memorize $\mathcal{D}$. The number of *nonzero* parameters can be further bounded as follows:

$$O(Nd(\log(\frac{d}{\gamma^2}) + \log R)) \overset{(a)}{=} O(N(\log N) \cdot (\log(\frac{d}{\gamma^2}) + \log R)) \overset{(b)}{=} \tilde{O}(N) \overset{(c)}{=} \tilde{O}(Nd\rho^2),$$

where (a) is due to $d \leq 600 \log N$, (b) hides the logarithmic factors, and (c) is because $N \leq 25Nd\rho^2$ for all $\rho \in \left(\frac{1}{5\sqrt{d}}, \frac{1}{3}\right)$.

**Case III: $\rho \in (1/5\sqrt{d}, 1/3)$ and $N < 600 \log N \leq d$.** We first apply Proposition B.21 to $\mathcal{D}$ to obtain 1-Lipschitz linear $\varphi : \mathbb{R}^d \to \mathbb{R}^N$ such that $\mathcal{D}' := \{(\varphi(\boldsymbol{x}_i), y_i)\}_{i \in [N]}$ has $\epsilon_{\mathcal{D}'} = \epsilon_{\mathcal{D}}$. This is possible as $d \geq N$.

Take $\boldsymbol{b} \in \mathbb{R}^N$ such that $\varphi(\boldsymbol{x}) - \boldsymbol{b} \geq \boldsymbol{0}$ for all $\boldsymbol{x} \in \mathcal{B}_2(\varphi(\boldsymbol{x}_i), \rho\epsilon_{\mathcal{D}'})$, ensuring that $\sigma$ does not affect the output of the first hidden layer. Let $\mathcal{D}'' = \{(\varphi(\boldsymbol{x}_i) - \boldsymbol{b}, y_i)\}_{i \in [N]}$. Then, $\epsilon_{\mathcal{D}} = \epsilon_{\mathcal{D}'} = \epsilon_{\mathcal{D}''}$. For simplicity of the notation, let us denote $\boldsymbol{z}_i := \varphi(\boldsymbol{x}_i) - \boldsymbol{b}$. Moreover, the first hidden layer is defined as $f_1(\boldsymbol{x}) = \varphi(\boldsymbol{x}) - \boldsymbol{b}$.

We apply Lemma E.3 to $\mathcal{D}''$. Let us denote $R := \max_{i \in [N]} \|\varphi(\boldsymbol{z}_i)\|_2$ and $\gamma := (1 - \rho)\epsilon_{\mathcal{D}''}$. Note that $R \geq \|\boldsymbol{z}_i\|_\infty$ for all $i \in [N]$ as $\|\boldsymbol{z}\|_2 \geq \|\boldsymbol{z}\|_\infty$ for all $\boldsymbol{z} \in \mathbb{R}^N$. By Lemma E.3, there exists $f_2 \in \bar{\mathcal{F}}_{N,P}$ with $P = O(N \cdot N(\log(\frac{N}{\gamma^2}) + \log R))$ *nonzero* parameters that $\rho$-robustly memorize $\mathcal{D}''$.

Let $f = f_2 \circ \sigma \circ f_1$. Since $f_1$ is 1-Lipschitz and $\epsilon_{\mathcal{D}''} = \epsilon_{\mathcal{D}}$, every robustness ball of $\mathcal{D}$ is mapped to the robustness ball of $\mathcal{D}''$ via $f_1$. Since the $\sigma$ does not affect the first hidden layer output of the robustness ball, and $f_2$ $\rho$-robustly memorizes $\mathcal{D}''$, the composed $f$ satisfies the desired property

The number of *nonzero* parameters can be further bounded as follows:

$$O(Nd + N \cdot N(\log(\frac{d}{\gamma^2}) + \log R)) \overset{(a)}{=} O(d \log N + (\log N)^2 \cdot (\log(\frac{d}{\gamma^2}) + \log R)) \overset{(b)}{=} \tilde{O}(d),$$

where (a) is due to $N \leq 600 \log N$, and (b) hides the logarithmic factors.

**Case IV: $\rho \in (1/5\sqrt{d}, 1/3)$, and $N \geq d \geq 600 \log N$.** We utilize the dimension reduction technique by Proposition B.19. We apply Proposition B.19 to $\mathcal{D}$ with $m =$

$\max\{\lceil 9d\rho^2 \rceil, \lceil 600 \log N \rceil, \lceil 10 \log d \rceil\}$ and $\alpha = 1/5$. Let us first check that the specified $m$ satisfies the condition $24\alpha^{-2} \log N \leq m \leq d$ for the proposition to be applied. $\alpha = 1/5$ and $m \geq 600 \log N$ ensure the first inequality $24\alpha^{-2} \log N \leq m$. The second inequality $m \leq d$ is decomposed into three parts. Since $\rho \leq \frac{1}{3}$, we have $9d\rho^2 \leq d$ so that

$$\lceil 9d\rho^2 \rceil \leq d. \tag{68}$$

Moreover, $600 \log N \leq d$ implies

$$\lceil 600 \log N \rceil \leq d. \tag{69}$$

Additionally, as $N \geq 2$, we have $d \geq 600 \log N \geq 600 \log 2 \geq 400$. By Lemma B.22, this implies $10 \log d \leq d$ and therefore

$$\lceil 10 \log d \rceil \leq d. \tag{70}$$

Gathering Equations (68) to (70) proves $m \leq d$.

By the Proposition B.19, there exists 1-Lipchitz linear mapping $\phi : \mathbb{R}^d \to \mathbb{R}^m$ and $\beta > 0$ such that $\mathcal{D}' := \{(\phi(\boldsymbol{x}_i), y_i)\}_{i \in [N]} \in \boldsymbol{D}_{m,N,C}$ satisfies

$$\epsilon_{\mathcal{D}'} \geq \frac{4}{5}\beta\epsilon_{\mathcal{D}}. \tag{71}$$

As $m \geq 10 \log d$, the inequality $\beta \geq \frac{1}{2}\sqrt{\frac{m}{d}}$ is also satisfied by Proposition B.19. Therefore, we have

$$\beta \geq \frac{1}{2}\sqrt{\frac{m}{d}} \overset{(a)}{\geq} \frac{1}{2}\sqrt{\frac{\lceil 9d\rho^2 \rceil}{d}} \geq \frac{1}{2}\sqrt{\frac{9d\rho^2}{d}} = \frac{3}{2}\rho, \tag{72}$$

where (a) is by the definition of $m$. Moreover, since $\phi$ is 1-Lipchitz linear,

$$\|\phi(\boldsymbol{x}_i)\|_2 = \|\phi(\boldsymbol{x}_i - \boldsymbol{0})\|_2 = \|\phi(\boldsymbol{x}_i) - \phi(\boldsymbol{0})\|_2 \leq \|\boldsymbol{x}_i - \boldsymbol{0}\|_2 = \|\boldsymbol{x}_i\|_2, \tag{73}$$

for all $i \in [N]$. Hence, by letting $R := \max_{i \in [N]}\{\|\boldsymbol{x}_i\|_2\}$, we have $\|\phi(\boldsymbol{x}_i)\|_2 \leq R$ for all $i \in [N]$.

Now, we set the first layer hidden matrix as the matrix $\boldsymbol{W} \in \mathbb{R}^{m \times d}$ corresponding to $\phi$ under the standard basis of $\mathbb{R}^d$ and $\mathbb{R}^m$. Moreover, set the first hidden layer bias as $\boldsymbol{b} := 2R\boldsymbol{1} = 2R(1, 1, \cdots, 1) \in \mathbb{R}^m$. Then, we have

$$\boldsymbol{W}\boldsymbol{x} + \boldsymbol{b} \geq \boldsymbol{0}, \tag{74}$$

for all $\boldsymbol{x} \in \mathcal{B}_2(\boldsymbol{x}_i, \epsilon_{\mathcal{D}})$ for all $i \in [N]$, where the comparison between two vectors are element-wise. This is because for all $i \in [N], j \in [m]$ and $\boldsymbol{x} \in \mathcal{B}_2(\boldsymbol{x}, \epsilon_{\mathcal{D}})$, we have

$$(\boldsymbol{W}\boldsymbol{x} + \boldsymbol{b})_j = (\boldsymbol{W}\boldsymbol{x})_j + 2R \geq 2R - \|\boldsymbol{W}\boldsymbol{x}\|_2 \overset{(a)}{\geq} 2R - \|\boldsymbol{x}\|_2 \overset{(b)}{\geq} 2R - (R + \epsilon_{\mathcal{D}}) \overset{(c)}{\geq} 0,$$

where (a) is by Equation (73), (b) is by the triangle inequality, and (c) is due to $R > \epsilon_{\mathcal{D}}$.

We construct the first layer of the neural network as $f_1(\boldsymbol{x}) := \sigma(\boldsymbol{W}\boldsymbol{x} + \boldsymbol{b})$ which includes the activation $\sigma$. Then, by above properties, $\mathcal{D}'' := \{(f_1(\boldsymbol{x}_i), y_i)\}_{i \in [N]}$ satisfies

$$\epsilon_{\mathcal{D}''} \geq \frac{6}{5}\rho\epsilon_{\mathcal{D}}. \tag{75}$$

This is because for $i \neq j$ with $y_i \neq y_j$ we have

$$\begin{aligned}
\|f_1(\boldsymbol{x}_i) - f_1(\boldsymbol{x}_j)\|_2 &= \|\sigma(\boldsymbol{W}\boldsymbol{x}_i + \boldsymbol{b}) - \sigma(\boldsymbol{W}\boldsymbol{x}_j + \boldsymbol{b})\|_2 \\
&\overset{(a)}{=} \|(\boldsymbol{W}\boldsymbol{x}_i + \boldsymbol{b}) - (\boldsymbol{W}\boldsymbol{x}_j + \boldsymbol{b})\|_2 \\
&= \|\phi(\boldsymbol{x}_i) - \phi(\boldsymbol{x}_j)\|_2 \\
&\overset{(b)}{\geq} 2\epsilon_{\mathcal{D}'} \\
&\overset{(c)}{\geq} 2 \times \frac{4}{5}\beta\epsilon_{\mathcal{D}} \\
&\overset{(d)}{\geq} 2 \times \frac{4}{5} \times \frac{3}{2}\rho\epsilon_{\mathcal{D}}
\end{aligned}$$

$$= \frac{12}{5}\rho\epsilon_{\mathcal{D}},$$

where (a) is by Equation (74), (b) is by the definition of the $\epsilon_{\mathcal{D}'}$, (c) is by Equation (71), and (d) is by Equation (72). By Lemma E.3 applied to $\mathcal{D}'' \in \boldsymbol{D}_{m,N,C}$, there exists $f_2 \in \mathcal{F}_{m,P}$ with $P = O(Nm(\log(\frac{m}{(\gamma'')^2}) + \log R''))$ *nonzero* number of parameters that $\frac{5}{6}$-robustly memorize $\mathcal{D}''$, where

$$\gamma'' := (1 - \frac{5}{6})\epsilon_{\mathcal{D}''} \stackrel{(a)}{\geq} \frac{1}{6} \times \frac{12}{5}\rho\epsilon_{\mathcal{D}} = \frac{2}{5}\rho\epsilon_{\mathcal{D}},$$

$$R'' := \max_{i \in [N]} \|f_1(\boldsymbol{x}_i)\|_2 = \max_{i \in [N]} \|\sigma(\boldsymbol{W}\boldsymbol{x}_i + \boldsymbol{b})\|_2 = \max_{i \in [N]} \|\boldsymbol{W}\boldsymbol{x}_i + \boldsymbol{b}\|_2$$

$$\leq \max_{i \in [N]} \|\boldsymbol{W}\boldsymbol{x}_i\|_2 + \|\boldsymbol{b}\|_2 \leq 3R.$$

Here (a) is by Equation (75).

Now, we claim that $f := f_2 \circ f_1$ $\rho$-robustly memorize $\mathcal{D}$. For any $i \in [N]$, take $\boldsymbol{x} \in \mathcal{B}_2(\boldsymbol{x}_i, \rho\epsilon_{\mathcal{D}})$. Then, by Equation (74), we have $f_1(\boldsymbol{x}) = \boldsymbol{W}\boldsymbol{x} + \boldsymbol{b}$ and $f_1(\boldsymbol{x}_i) = \boldsymbol{W}\boldsymbol{x}_i + \boldsymbol{b}$ so that

$$\|f_1(\boldsymbol{x}) - f_1(\boldsymbol{x}_i)\|_2 = \|\boldsymbol{W}\boldsymbol{x} - \boldsymbol{W}\boldsymbol{x}_i\|_2 \leq \|\boldsymbol{x} - \boldsymbol{x}_i\|_2 \leq \rho\epsilon_{\mathcal{D}}. \tag{76}$$

Moreover, combining Equations (75) and (76) results $\|f_1(\boldsymbol{x}) - f_1(\boldsymbol{x}_i)\|_2 \leq \frac{5}{6}\epsilon_{\mathcal{D}''}$. Since $f_2$ $\frac{5}{6}$-robustly memorize $\mathcal{D}''$, we have

$$f(\boldsymbol{x}) = f_2(f_1(\boldsymbol{x})) = f_2(f_1(\boldsymbol{x}_i)) = y_i.$$

In particular, $f(\boldsymbol{x}) = y_i$ for any $\boldsymbol{x} \in \mathcal{B}_2(\boldsymbol{x}_i, \rho\epsilon_{\mathcal{D}})$, concluding that $f$ is a $\rho$-robust memorizer $\mathcal{D}$. Regarding the number of parameters to construct $f$, notice that $f_1$ consists of $(d + 1)m = \tilde{O}(d^2\rho^2)$ parameters (and thus $\tilde{O}(d\rho^2)$ *nonzero* parameters) as $m = \tilde{O}(d\rho^2)$. $f_2$ consists of $\tilde{O}(Nm) = \tilde{O}(Nd\rho^2)$ *nonzero* parameters. Since the case IV assumes $N \geq d$, we have

$$d^2\rho^2 \leq Nd\rho^2$$

Therefore, $f$ in total consists of $\tilde{O}(d^2\rho^2 + Nd\rho^2) = \tilde{O}(Nd\rho^2)$ number of *nonzero* parameters. This proves the theorem for the fourth case.

**Case V:** $\rho \in (1/5\sqrt{d}, 1/3)$**, and** $d > N \geq 600 \log N$**.** The last case combines the two techniques used in Cases III and IV. We first apply Proposition B.21 to $\mathcal{D}$ to obtain 1-Lipschitz linear $\varphi : \mathbb{R}^d \to \mathbb{R}^N$ such that $\mathcal{D}' := \{(\varphi(\boldsymbol{x}_i), y_i)\}_{i \in [N]} \in \boldsymbol{D}_{N,N,C}$ has $\epsilon_{\mathcal{D}'} = \epsilon_{\mathcal{D}}$. Note that we can apply the proposition since $d \geq N$.

Next, we apply Proposition B.19 to $\mathcal{D}' \in \boldsymbol{D}_{N,N,C}$ with $m = \max\{\lceil 9N\rho^2 \rceil, \lceil 600 \log N \rceil\}$ and $\alpha = 1/5$. Let us first check that the specified $m$ satisfies the condition $24\alpha^{-2} \log N \leq m \leq N$ for the proposition to be applied. $\alpha = 1/5$ and $m \geq 600 \log N$ ensure the first inequality $24\alpha^{-2} \log N \leq m$. The second inequality $m \leq N$ is decomposed into two parts. Since $\rho \leq \frac{1}{3}$, we have $9N\rho^2 \leq N$ so that

$$\lceil 9N\rho^2 \rceil \leq N. \tag{77}$$

Moreover, $600 \log N \leq N$ implies

$$\lceil 600 \log N \rceil \leq N. \tag{78}$$

Gathering Equations (68) and (69) proves $m \leq N$. Additionally, as $N \geq 2$, we have $N \geq 600 \log N \geq 600 \log 2 \geq 400$. By Lemma B.22, this implies $10 \log N \leq N$.

By the Proposition B.19, there exists 1-Lipchitz linear mapping $\phi : \mathbb{R}^N \to \mathbb{R}^m$ and $\beta > 0$ such that $\mathcal{D}'' := \{(\phi(\varphi(\boldsymbol{x}_i)), y_i)\}_{i \in [N]} \in \boldsymbol{D}_{m,N,C}$ satisfies

$$\epsilon_{\mathcal{D}''} \geq \frac{4}{5}\beta\epsilon_{\mathcal{D}}'. \tag{79}$$

As $m \geq 600 \log N \geq 10 \log N$, the inequality $\beta \geq \frac{1}{2}\sqrt{\frac{m}{N}}$ is also satisfied by Proposition B.19. Therefore, we have

$$\beta \geq \frac{1}{2}\sqrt{\frac{m}{N}} \stackrel{(a)}{\geq} \frac{1}{2}\sqrt{\frac{\lceil 9N\rho^2 \rceil}{N}} \geq \frac{1}{2}\sqrt{\frac{9N\rho^2}{N}} = \frac{3}{2}\rho, \tag{80}$$

where (a) is by the definition of $m$. Moreover, since $\varphi$ and $\phi$ are both 1-Lipchitz linear, $\phi \circ \varphi : \mathbb{R}^d \to \mathbb{R}^m$ is also 1-Lipschitz linear. Therefore,

$$\|\phi(\varphi(\boldsymbol{x}_i))\|_2 = \|\phi(\varphi(\boldsymbol{x}_i - \boldsymbol{0}))\|_2 = \|\phi(\varphi(\boldsymbol{x}_i)) - \phi(\varphi(\boldsymbol{0}))\|_2 \le \|\boldsymbol{x}_i - \boldsymbol{0}\|_2 = \|\boldsymbol{x}_i\|_2 , \quad (81)$$

for all $i \in [N]$. Hence, by letting $R := \max_{i \in [N]}\{\|\boldsymbol{x}_i\|_2\}$, we have $\|\phi(\varphi(\boldsymbol{x}_i))\|_2 \le R$ for all $i \in [N]$.

Now, we set the first layer hidden matrix as the matrix $\boldsymbol{W} \in \mathbb{R}^{m \times d}$ corresponding to $\phi \circ \varphi$ under the standard basis of $\mathbb{R}^d$ and $\mathbb{R}^m$. Moreover, set the first hidden layer bias as $\boldsymbol{b} := 2R\boldsymbol{1} = 2R(1, 1, \cdots, 1) \in \mathbb{R}^m$. Then, we have

$$\boldsymbol{W}\boldsymbol{x} + \boldsymbol{b} \ge \boldsymbol{0}, \quad (82)$$

for all $\boldsymbol{x} \in \mathcal{B}_2(\boldsymbol{x}_i, \epsilon_\mathcal{D})$ for all $i \in [N]$, where the comparison between two vectors are element-wise. This is because for all $i \in [N], j \in [m]$ and $\boldsymbol{x} \in \mathcal{B}_2(\boldsymbol{x}, \epsilon_\mathcal{D})$, we have

$$(\boldsymbol{W}\boldsymbol{x} + \boldsymbol{b})_j = (\boldsymbol{W}\boldsymbol{x})_j + 2R \ge 2R - \|\boldsymbol{W}\boldsymbol{x}\|_2 \overset{(a)}{\ge} 2R - \|\boldsymbol{x}\|_2 \overset{(b)}{\ge} 2R - (R + \epsilon_\mathcal{D}) \overset{(c)}{\ge} 0,$$

where (a) is by Equation (81), (b) is by the triangle inequality, and (c) is due to $R \ge \epsilon_\mathcal{D}$.

We construct the first layer of the neural network as $f_1(\boldsymbol{x}) := \sigma(\boldsymbol{W}\boldsymbol{x} + \boldsymbol{b})$ which includes the activation $\sigma$. Next, we show that, $\mathcal{D}'' := \{(f_1(\boldsymbol{x}_i), y_i)\}_{i \in [N]}$ satisfies

$$\epsilon_{\mathcal{D}''} \ge \frac{6}{5}\rho\epsilon_\mathcal{D}, \quad (83)$$

by the above properties. This is because for $i \ne j$ with $y_i \ne y_j$ we have

$$\begin{aligned}
\|f_1(\boldsymbol{x}_i) - f_1(\boldsymbol{x}_j)\|_2 &= \|\sigma(\boldsymbol{W}\boldsymbol{x}_i + \boldsymbol{b}) - \sigma(\boldsymbol{W}\boldsymbol{x}_j + \boldsymbol{b})\|_2 \\
&\overset{(a)}{=} \|(\boldsymbol{W}\boldsymbol{x}_i + \boldsymbol{b}) - (\boldsymbol{W}\boldsymbol{x}_j + \boldsymbol{b})\|_2 \\
&= \|\phi(\varphi(\boldsymbol{x}_i)) - \phi(\varphi(\boldsymbol{x}_j))\|_2 \\
&\overset{(b)}{\ge} 2\epsilon_{\mathcal{D}''} \\
&\overset{(c)}{\ge} 2 \times \frac{4}{5}\beta\epsilon'_\mathcal{D} \\
&\overset{(d)}{\ge} 2 \times \frac{4}{5} \times \frac{3}{2}\rho\epsilon'_\mathcal{D} \\
&= \frac{12}{5}\rho\epsilon'_\mathcal{D} \\
&\overset{(e)}{=} \frac{12}{5}\rho\epsilon_\mathcal{D},
\end{aligned}$$

where (a) is by Equation (82), (b) is by the definition of the $\epsilon_{\mathcal{D}''}$, (c) is by Equation (79), (d) is by Equation (80), and (e) is because $\epsilon_{\mathcal{D}'} = \epsilon_\mathcal{D}$.

By Lemma E.3 applied to $\mathcal{D}'' \in \boldsymbol{D}_{m,N,C}$, there exists $f_2 \in \mathcal{F}_{m,P}$ with $P = O(Nm(\log(\frac{m}{(\gamma'')^2}) + \log R''))$ *nonzero* number of parameters that $\frac{5}{6}$-robustly memorize $\mathcal{D}''$, where

$$\begin{aligned}
\gamma'' &:= (1 - \frac{5}{6})\epsilon_{\mathcal{D}''} \overset{(a)}{\ge} \frac{1}{6} \times \frac{12}{5}\rho\epsilon_\mathcal{D} = \frac{2}{5}\rho\epsilon_\mathcal{D}, \\
R'' &:= \max_{i \in [N]} \|f_1(\boldsymbol{x}_i)\|_2 = \max_{i \in [N]} \|\sigma(\boldsymbol{W}\boldsymbol{x}_i + \boldsymbol{b})\|_2 = \max_{i \in [N]} \|\boldsymbol{W}\boldsymbol{x}_i + \boldsymbol{b}\|_2 \\
&\le \max_{i \in [N]} \|\boldsymbol{W}\boldsymbol{x}_i\|_2 + \|\boldsymbol{b}\|_2 \le 3R.
\end{aligned}$$

Here, (a) is by Equation (83).

Now, we claim that $f := f_2 \circ f_1$ $\rho$-robustly memorize $\mathcal{D}$. For any $i \in [N]$, take $\boldsymbol{x} \in \mathcal{B}_2(\boldsymbol{x}_i, \rho\epsilon_\mathcal{D})$. Then, by Equation (82), we have $f_1(\boldsymbol{x}) = \boldsymbol{W}\boldsymbol{x} + \boldsymbol{b}$ and $f_1(\boldsymbol{x}_i) = \boldsymbol{W}\boldsymbol{x}_i + \boldsymbol{b}$ so that

$$\|f_1(\boldsymbol{x}) - f_1(\boldsymbol{x}_i)\|_2 = \|\boldsymbol{W}\boldsymbol{x} - \boldsymbol{W}\boldsymbol{x}_i\|_2 \le \|\boldsymbol{x} - \boldsymbol{x}_i\|_2 \le \rho\epsilon_\mathcal{D}. \quad (84)$$

Moreover, putting Equation (83) to Equation (84) results $\|f_1(\boldsymbol{x}) - f_1(\boldsymbol{x}_i)\|_2 \leq \frac{5}{6}\epsilon_{\mathcal{D}''}$. Since $f_2$ $\frac{5}{6}$-robustly memorize $\mathcal{D}''$, we have

$$f(\boldsymbol{x}) = f_2(f_1(\boldsymbol{x})) = f_2(f_1(\boldsymbol{x}_i)) = y_i.$$

In particular, $f(\boldsymbol{x}) = y_i$ for any $\boldsymbol{x} \in \mathcal{B}_2(\boldsymbol{x}_i, \rho\epsilon_{\mathcal{D}})$, concluding that $f$ is a $\rho$-robust memorizer $\mathcal{D}$.

Regarding the number of *nonzero* parameters to construct $f$, notice that $f_1$ consists of $(d+1)m = \tilde{O}(Nd\rho^2)$ *nonzero* parameters as $m = \tilde{O}(N\rho^2)$. $f_2$ consists of $\tilde{O}(Nm) = \tilde{O}(N^2\rho^2)$ *nonzero* parameters. Since the case V assumes $N < d$, we have

$$N^2\rho^2 \leq Nd\rho^2.$$

Therefore, $f$ in total consists of $\tilde{O}(Nd\rho^2 + N^2\rho^2) = \tilde{O}(Nd\rho^2)$ number of *nonzero* parameters. This proves the theorem for the last case.

$\square$

**Nonzero Parameter Counts: Existing Upper Bounds.** In Section 1.1, the existing upper bound is stated by counting all parameters. When counting only the nonzero parameters, the corresponding existing upper bound takes a different form. Specifically, for any dataset $\mathcal{D}$ with input dimension $d$ and size $N$, there exist a neural network that achieves robust memorization on $\mathcal{D}$ with the robustness ratio $\rho$ under $\ell_2$-norm, with the number of parameters $P$ bounded as follows:

$$P = \begin{cases} \tilde{O}(N + d) & \text{if } \rho \in (0, 1/\sqrt{d}]. \\ \tilde{O}(Nd^2\rho^4 + d) & \text{if } \rho \in (1/\sqrt{d}, 1/\sqrt[4]{d}]. \\ \tilde{O}(Nd) & \text{if } \rho \in (1/\sqrt[4]{d}, 1). \end{cases} \tag{85}$$

This is the counterpart to Equation (2) that considers all parameter counts. As in the case of full parameter count, the first and the third case in Equation (85) directly follow from Yu et al. [2024] and Egosi et al. [2025] respectively. The work by Egosi et al. [2025] can be implicitly improved to the second case under the moderate $\rho$ condition, using the same translation technique provided in Appendix D.3.

### E.4 Lemmas for Nonzero Parameter Count

Here, we state Lemmas D.1 and D.2—that corresponds to Theorem B.6 of Yu et al. [2024]—to its original version that contains the **nonzero** parameter count with $\ell_2$-norm into the consideration.

**Lemma E.3** (Theorem B.6, Yu et al. [2024])**.** *For any $\mathcal{D} \in \boldsymbol{D}_{d,N,C}$ and $\rho \in (0, 1)$, define $\gamma := (1 - \rho)\epsilon_{\mathcal{D}} > 0$ and $R > 1$ with $\|\boldsymbol{x}_i\|_\infty \leq R$ for all $i \in [N]$. Then, there exists a neural network $f$ with width $O(d)$, depth $O(N(\log(\frac{d}{\gamma^2}) + \log R))$ that $\rho$-robustly memorizes $\mathcal{D}$ using at most $O(Nd(\log(\frac{d}{\gamma^2}) + \log R))$ **nonzero** parameters.*

