# OpenReview forum: "The Cost of Robustness: Tighter Bounds on Parameter Complexity for Robust Memorization in ReLU Nets"
_NeurIPS.cc/2025/Conference — NeurIPS 2025 poster_

### Official Review · Reviewer_YP8D · 2025-06-19

**Clarity:** 2
**Significance:** 3
**Originality:** 2
**Rating:** 5
**Confidence:** 4

**Summary:**

The paper studies the robust memoization problem for neural networks. In this problem, the input is a labeled dataset, and the goal is to return a memorizing neural network that is as small as possible and robust to small perturbations. In this paper, perturbations are measured in an l2 sense, so that the labels in a small Euclidean ball around any data point must remain constant, and the size of the network is measured in the number of parameters.
The paper derives several new upper and lower bounds for the network size, depending on the input dimension, dataset size, and radius of the robustness ball.

**Questions:**

1. I understand that some of the literature about robust memoization naturally deals with ReLU networks. However, since the labels are necessarily discrete, to me it seems natural to consider piecewise constant networks, rather than piecewise linear. I.e. replace ReLU by threshold/step. Would things simplify significantly in that case? Do the authors believe their approach could yield tight bounds for these networks?

2. For adversarial examples, the L_\infty norm is the most natural way to measure the size of perturbations. As far as I can see from Appendix C, not all results extend to this norm. Can the authors please comment on this point?

**Ethical Concerns:**

["NO or VERY MINOR ethics concerns only"]

**Final Justification:**

As indicated in my original report, I have a positive impression of the paper. I maintain that impression after reading the authors' response.

**Limitations:**

No societal impact.

**Quality:**

3

**Strengths And Weaknesses:**

Overall, this is an interesting paper. The problem of memoization in general, and in particular robust memoization, has garnered a lot of attention in recent years. The paper adds new results to this line of research and will probably draw the attention of some parts of the community.
It is particularly nice that the authors manage to derive tight bounds in the regime where the robustness radius goes to 0 with the other parameters. This is perhaps not too surprising, since they show that the robust memoization problem essentially reduces to the standard memoization problem. Still, it complements the existing literature in a natural way.

The techniques in the paper are not particularly novel. The upper bound relies on VC dimension consideration as well as an explicit natural construction. The lower bound follows from dimensionality reduction considerations, which have appeared before in one form or another. I will make the remark that even if the approach is based on existing techniques, it was not implemented it this exact way before, so I would not necessarily hold it against the paper.

In light of the above, it is my impression that the paper meets the bar for NeurIPS

I have gone over the proofs, some of which seem to be very technical, and I think I agree with everything. I have one comment about the current writing of the paper. The introduction seems to contain a lot of redundancies and many repetitions about the obtained bounds and existing bounds in the literature. In my opinion, it would be much better and improve readability to condense those repeated discussions into a single table to detail the new bounds and compare to previous results. This would leave more room to discuss the different aspects of the proofs, give more intuition for the technical parts, and explain the differences between the different regimes. Currently, Section 5 does a decent job of explaining some parts of the paper, but it would be of great benefit to expand this section.

---

> ### Author Rebuttal · Authors · 2025-07-31
>
> We thank the reviewer for the careful reading and constructive feedback. Below, we address the reviewer’s concerns and questions.
>
> **W1. Limited novelty in techniques for both lower and upper bounds**
>
> - We appreciate the reviewer’s recognition that, despite building on existing ideas, our implementation offers a novel contribution. As the reviewer noted, the lower bound based on VC-dimension and the use of dimensionality reduction in the upper bound can be seen as extensions of known ideas. Nevertheless, we would like to elaborate on parts of our approach that go beyond existing techniques and involve substantial technical challenges.
>     1. **Upper bound:** To the best of our knowledge, using lattice mapping to preserve the geometry of the ball is novel. In Theorem 4.2(i), we establish a condition ensuring that the robustness ball lies sufficiently far from the lattice grid (Lemma B.12), and we show that the discontinuous floor function can be approximated by a continuous ReLU network  (Lemma B.13).
>     In Theorem 4.2(ii), we compute the probability that a uniformly random point from the ball falls into the error zone under random projection (Lemma B.9). Another key technical challenge was ensuring correct classification even within regions where multiple robustness balls with the same label overlap. To address this, we devised disjoint, integer-aligned interval encodings and carefully bounded the propagation of error (Lemma B.10).
>     2. **Lower bound:** Prior work (Theorem 4.3 in Yu et al.) established a lower bound on the first hidden layer width under the $\ell_\infty$-norm, but not under the $\ell_2$-norm. In our work, we analyze the lower bound on width under the $\ell_2$ -norm using more sophisticated geometrical analysis—by computing the upper bound on the distance between the standard basis vectors and any subspace of given dimension (Lemma A.1). The $\ell_2$-norm requires a more intricate analysis since, unlike the $\ell_\infty$-norm, it does not allow coordinate-wise analysis.
>
>         Furthermore, to overcome the constraint $N \geq d$ in the prior lower bound and to cover even the case $N<d$, we derive a new bound on the distance between a **subset** of standard basis vectors and subspaces of given dimension (Lemma A.2), deriving the first lower bound on the first hidden layer width **without** such a restriction.
>
>
> **W2. Redundancy in the introduction**
>
> - We appreciate your thoughtful comments regarding the writing, particularly the redundancy in the introduction. We will revise the introduction to remove repetitions and present the key results more concisely.
>
> **Q1. Would the results simplify or yield tight bounds when using threshold networks instead of ReLU?**
>
> - We would like to discuss the potential relevance of piecewise constant networks, such as threshold networks, from two different perspectives—(1) using a combination of ReLU and threshold, and (2) using threshold alone.
> - Firstly, as we describe in lines 228-235, if we are allowed to use threshold activations on top of ReLU, we can construct a network where the upper bound in the moderate $\rho$ regime (Theorem 4.2(ii)) holds without any error.
>
>     Moreover, we agree with the reviewer’s view that the use of piecewise constant networks can be seen as natural for discrete label classification, especially at the last layer. In fact, our construction explicitly approximates threshold functions using ReLU in order to achieve robust memorization within robustness balls.
>
> - Secondly, consider the case where we use only threshold activations.
>     1. **Lower bound by the first hidden layer width** (Proposition 3.2): This result is independent of the activation function, so the same lower bound remains valid for threshold networks.
>     2. **Lower bound by VC-dimension** (Proposition 3.3): This result can in fact improve when using threshold activations. While the VC-dimension of a ReLU network with $P$ parameters is $O(P^2)$, the VC-dimension of a threshold network is only $\tilde{O}(P)$ (Baum and Haussler (1989) [1]).
>
>         Notably, our result that the VC-dimension of a robust memorizer is $\text{VC-dim}(\mathcal{F})=\Omega\left( \frac{N}{1-\rho^2} \right)$ for $\rho \in \left( 0, \sqrt{1-\frac{1}{d}} \right]$ still holds under threshold activation. This implies a lower bound on the number of parameters $P = \tilde{\Omega}\left( \frac{N}{1-\rho^2} \right)$ for $\rho \in \left( 0, \sqrt{1-\frac{1}{d}} \right]$ and $P=\tilde{\Omega}(Nd)$ for $\rho \in \left(\sqrt{1-\frac{1}{d}},1 \right)$. These bounds are stronger than those for ReLU networks, and they exceed $N$.
>
>     3. **Upper bound** (Theorem 4.2): The VC-dimension result implies that parameter complexity identical to the ReLU upper bounds in the small and moderate $\rho$  regimes (Theorem 4.2(i), (ii)) cannot be achieved by threshold-only networks, since those rates are sublinear in $N$. The threshold lower bounds by VC-dimension indicate that showing such upper bounds is impossible in general.
>
>         However, the adaptability of Theorem 4.2(iii) to threshold-only networks in the large $\rho$ regime remains an open question. Our current construction relies on the ability of ReLU to propagate input values as-is across layers, which is not directly achievable using threshold functions. As a result, it is nontrivial to determine whether the rate in Theorem 4.2(iii) can be matched using threshold networks. We believe this is an interesting direction for future research.
>
>         We also note that Rajput et al. (2021) [2] provide an upper bound $\tilde{O}(N+\frac{d}{\epsilon_\mathcal{D}})$ for non-robust memorization using threshold networks, which may offer a useful reference point for further developing analogous results in the robust setting.
>
>
>
> **Q2. Limited extension of results to the $\ell_\infty$-norm?**
>
> - We appreciate the reviewer’s question about the applicability of our results to the $\ell_\infty$-norm, which is indeed a natural choice.
> 1. **Lower bound by the first hidden layer width**:  Proposition C.7 shows that the same lower bound of width $\rho^2 \min\{N,d\}$ holds under the $\ell_\infty$-norm.
> 2. **Lower bound by VC-dimension**: While we do not provide an exact VC-dimension lower bound specifically for the $\ell_\infty$-norm, we do present a result under the general $\ell_p$-norm (Theorem C.6) for $p \in [1,\infty)$, where the VC-dimension is lower bounded by $\Omega \left( {\frac{N}{1-\rho^p}}\right)$. This implies that the number of parameter is lower bounded by $\Omega \left( \sqrt{\frac{N}{1-\rho^p}}\right)$. This bound also holds in $p \to \infty$, but in that case it simplifies to $\Omega(\sqrt{N})$, which is the trivial bound from the standard memorization.
> 3. **Upper bound:** We provide the result in Theorem C.10 with $\gamma_\infty(d)=\sqrt{d}$ .
>
>     (i) For $\rho \in(0,\frac{1}{5Nd})$, the same upper bound $\tilde{O}(\sqrt{N})$ holds as in the $\ell_2$ case.
>
>     (ii) For $\rho \in(\frac{1}{5Nd},\frac{1}{5d})$, we obtain an upper bound of $\tilde{O}(Nd^{1/2}\rho^{1/2})$ with arbitrarily small error.
>
>     (iii) For $\rho \in(\frac{1}{5d},\frac{1}{\sqrt{d}})$, we obtain an upper bound of  $\tilde{O}(Nd^4\rho^4)$.
>
>     We also note a minor mistake in the constants defining $\rho$ intervals in lines 1691 and 1693, where 2 should be corrected to 5.
>
>
> ---
>
> > **Reference**
> >
> > *[1] Eric B. Baum and David Haussler. What size net gives valid generalization? Computation, 1(1):151–160, 1989.*
> >
> > *[2] Shashank Rajput, Kartik Sreenivasan, Dimitris Papailiopoulos, and Amin Karbasi. An Exponential Improvement on the Memorization Capacity of Deep Threshold Networks. In Advances in Neural Information Processing Systems, volume 34, pp. 12674–12685. Curran Associates, Inc., 2021.*

---

### Official Review · Reviewer_nD17 · 2025-06-22

**Clarity:** 3
**Significance:** 3
**Originality:** 3
**Rating:** 5
**Confidence:** 3

**Summary:**

This paper analyzes the number of ReLU neural network parameters needed to achieve robust memorization, which requires that the network interpolate a discretely-labeled dataset while maintaining the same output value within $\mu$ distance of each training point. The parameter complexity is studied as a function of the robustness ratio $\rho = \mu/\epsilon$, where $2\epsilon$ is the minimum distance between any distinctly-labeled training points. The paper demonstrates novel and improved upper and lower bounds across all values of $\rho$, which are tight (unlike previous bounds) for small $\rho \in (0, 1/\sqrt{N/d})$ in the case $d = \Theta(\sqrt{N})$.

**Questions:**

- See my above question regarding Theorem 4.2 (ii) under "weaknesses."
- Comment: it would be helpful to illustrate the robustness radius $\mu$ of a simple low-dimensional dataset and various interpolating solutions with different robustness ratios.

**Ethical Concerns:**

["NO or VERY MINOR ethics concerns only"]

**Final Justification:**

I thank the authors for their feedback. I continue to recommend acceptance. I encourage them to consider adding some of what they included in their response to me regarding the $\eta > 0$ error in Theorem 4.2 (ii) to the paper itself (at least the appendix), as this may be a natural question that other readers could have, and to consider adding a figure illustrating interpolation with different robustness ratios in a simple scenario, as I believe this will help intuitively convey the concept.

**Limitations:**

Yes

**Quality:**

3

**Strengths And Weaknesses:**

Strengths:
- The paper is well-written and the organization is clear. Figure 1, which illustrates graphically how the authors' bounds compare to previous ones in different regimes, is particularly helpful.
- The idea of analyzing parameter complexity across the entire possible range of robustness ratio $\rho$ is interesting, as it allows for a more unified view of the complexity required for robust vs. non-robust memorization. The bounds demonstrated here are novel and strictly improve upon previous results in almost all cases.
- In the supplementary material, the authors correct an inconsistency in their original submission which led to overly tight bounds in the $d = \Theta(N)$ regime; their honesty and rigor in doing this is appreciated.

Weaknesses:
- There is still a substantial gap between the upper and lower bounds demonstrated in the paper, suggesting that they could be further improved (which the authors acknowledge). The claimed tightness in the $d = \Theta(N)$ regime in the initial submission is clarified to be incorrect in the supplementary material, although the revised bounds in this case still improve upon existing results.
- The upper bound in Theorem 4.2 (ii) only shows $\rho$-robust memorization up to an arbitrarily small error $\eta > 0$. The authors note that this arises from inability of ReLU networks to exactly represent discontinuous functions, but because it seems to arise from their specific coordinate map technique, it is not clear to me whether this error is inherent to the problem itself (i.e., memorization with $\eta = 0$ is actually not possible in this regime with $\tilde{O}(Nd^{1/4} \rho^{1/2})$ parameters) or whether it is just an artifact of the proof strategy. If the former, is there a simple example (say in $d = 1$) which illustrates why $\eta = 0$ error is not possible?

---

> ### Author Rebuttal · Authors · 2025-07-31
>
> We thank the reviewer for the positive and careful comments. Below, we address the reviewer’s concerns and comments.
>
> **W1. Substantial gap remains between upper and lower bounds**
>
> Yes, there does remain a gap between upper and lower bounds for certain ranges of $\rho$. Although we tried to further tighten this gap, we faced challenges on both sides.
>
> - On the upper bound side, devising a construction that can adapt to **arbitrary geometric configuration** of data points under a limited budget of parameters remains challenging.
> - On the lower bound side, our bound on the minimum width of the first hidden layer is already tight, so further improving the parameter lower bound would require more sophisticated techniques beyond characterizing the first hidden layer width. In the large $\rho$ regime, our bounds tightly characterize the necessary (Proposition 3.2) and sufficient (Proposition B.16 and the paragraph “Tight Bounds of Width with Large $\rho$” in Section 4) first hidden layer width. Additionally, for small and moderate $\rho$ regime, improving the lower bound on width is impossible, since it has been shown that logarithmic width suffices ([1] and our Theorem 4.2 (i), (ii)).
>
> Nevertheless, overcoming these challenges and closing the remaining gaps between the upper and lower bounds remains a substantial and promising direction for future work.
>
> **W2 & Q1. Unclear whether the small error $\eta$ in Theorem 4.2 (ii) is inherent or technical**
>
> The inevitability of $\eta$ error is **mainly due to our coordinate map technique** followed by the continuous nature of ReLU. Approximating the discontinuous coordinate map using a continuous function that is made of ReLU naturally requires **error-tolerant regions**—where the approximation can be inexact—around the discontinuous points of the coordinate map. These error-tolerant regions are illustrated as a purple region in Figure 4 in our construction and are the entire source of the error.
>
> We further conjecture that it is not inherent to the problem itself.
>
> - At least for the special case $d=1$ you mentioned, interestingly, a tight upper bound $\tilde{O}(\sqrt{N})$—matching the lower bound for memorization—can be achieved for all $\rho \in (0, 1)$ without error. The result is based on a careful modification of the (non-robust) memorization construction (Vardi et al. (2021)), which does not rely on our coordinate mapping techniques.
>
>     In the first step of their construction, they linearly project the inputs to 1-dimension. This projection step, when viewed in the context of robust memorization, significantly restricts the feasible $\rho$ at the cost of simplifying the geometry while keeping the projected balls separated. However, $d=1$ case already has the simplified geometry from the start, and so the feasible range of $\rho$ needs no restriction. A modification of the remaining steps, ensuring the predictions remain consistent within each robustness ball, proves the result. This suggests that the $\eta$ error is not inherent, at least in this special case.
>
> - As a side remark, threshold activations have inherent discontinuity unlike ReLU. Therefore, the additional use of threshold activation on top of ReLU is one possible remedy to overcome the error, although this makes the problem inherently different. Simply speaking, we can completely remove the error-tolerant regions in this case, because we can perfectly implement ideal building blocks for the floor function (introduced as $\Delta$ in line 1428) using both ReLU and threshold activations together. Thus, combining ReLU and threshold activations allows the robust memorization without an error using the same number of parameters as in Theorem 4.2(ii).
>
> **Q2. Illustration of robustness radius and interpolating solutions in a low-dimensional setting with varying robustness ratios?**
>
> We thank the reviewer for the suggestion on the illustration. We will consider adding an illustrative figure that depicts interpolating solutions corresponding to different robustness ratios.
>
> ---
> > **References**
> >
> > *[1] Amitsour Egosi, Gilad Yehudai, and Ohad Shamir. Logarithmic width suffices for robust memorization, 2025.*
> >
> > *[2] Gal Vardi, Gilad Yehudai, and Ohad Shamir. On the optimal memorization power of ReLU neural networks, 2021.*

---

> > ### Comment · Reviewer_nD17 · 2025-08-05
> >
> > I thank the authors for their feedback. I continue to recommend acceptance. I encourage them to consider adding some of what they included in their response to me regarding the $\eta > 0$ error in Theorem 4.2 (ii) to the paper itself (at least the appendix), as this may be a natural question that other readers could have, and to consider adding a figure illustrating interpolation with different robustness ratios in a simple scenario, as I believe this will help intuitively convey the concept.

---

> > > ### Author Response · Authors · 2025-08-05
> > >
> > > We thank the reviewer again for the positive and engaging feedback. The discussion above will be supplemented in the revision to further clarify the cause of the eta error in our construction. We will also contemplate adding an illustration that can help convey the concept.

---

### Official Review · Reviewer_LKzZ · 2025-06-30

**Clarity:** 3
**Significance:** 2
**Originality:** 2
**Rating:** 4
**Confidence:** 3

**Summary:**

The paper investigates the necessary and sufficient numbers of parameters needed for robust memorization
of a dataset of size N, i.e. the correct classification of data even with small perturbations in the input. They
establish improved or completely new lower and upper bounds for the parameter complexity for all reasonable
robustness ratios.

**Questions:**

The authors already realized that they made a mistake, being inconsistent in their notion of “number of
parameters”. While they considered only non-zero parameters in most of their results, one of their lower
bounds relies on counting all parameters. In a separate document, they give a detailed explanation of this and
their later choice to consider all parameters, also zero-valued ones. Further they provide the corrected results
and the differences to the ones in the original paper. However, these corrected results are not compared to
existing literature in as much detail and result in a significant difference to a very prominent part of the
proposed work, as the bound is no longer tight in the case of d= Θ(N). They should thoroughly revise the
submitted paper to include the changes and avoid any misinterpretation of their results. Additionally there
are some other corrections to make:

1-The authors do not explain why they add up the two lower bounds on the parameter complexity. This
seems counter intuitive and the sum is no longer a lower bound. Instead they should use the maximum
of both bounds.

2-In line 72, they wrote ρ ∈ (1/(5√d),1] as a half open interval including 1, while they consider ρ ∈
(1/(5√d),1) elsewhere in the paper and explicitly explain that ρ = 1 leads to a contradiction in the
robust memorization setting.

3-In Definition 2.1, they do not explicitly exclude the fact that all data points are classified equally
(yi = yj ∀i,j). In this case the minimum is taken over an empty seta and thus ϵD is not defined.

4-The authors write in line 230 that the robust memorization error mentioned in Theorem 4.2 ii) can be
arbitrarily small, but do not discuss why this is the case and how this can be done.

**Ethical Concerns:**

["NO or VERY MINOR ethics concerns only"]

**Final Justification:**

I have raised my score to 4 and vote for acceptance

**Quality:**

3

**Strengths And Weaknesses:**

The authors  give all necessary definitions and explain their choices and limitations (of which there are very few) in detail. They also give comprehensive sketches of proofs and visualized their results very well.

---

> ### Author Rebuttal · Authors · 2025-07-31
>
> We thank the reviewer for the constructive feedback. Below, we address the reviewer’s concerns.
>
> **Q. Insufficient comparison of revised results to prior work; bound not tight for $d=\Theta(N)$**
>
> - The corrected results under the new counting method are compared with prior works in Table 1 of Appendix D, and will be reflected in the main text. We will update the main text to reflect the new counting method as presented in the appendix and remove the tightness claim for $d=\Theta(N)$, which we acknowledged to be incorrect (line 768).
> - We nevertheless want to emphasize that the new counting method still preserves the originally claimed lower bounds. This is mainly because the lower bound on nonzero parameter complexity serves as a lower bound on total parameter complexity (including zeros). Hence, comparisons between our lower bounds and existing ones remain valid.
> - Even from the upper bound perspective, although the rates have changed in certain ranges of $\rho$, the important properties remain unchanged: tight complexity for small $\rho$, and a continuously increasing upper bound with respect to $\rho$.
> - Additionally, the original explanation and the comparison under the original counting method—including Figure 6—will be moved to an additional section in the appendix, to assist readers who are also interested in the results based on the non-zero parameter counts.
>
> **Q1. Justification for adding up the two lower bounds instead of taking the maximum?**
>
> - There are two lower bounds derived using different techniques: width and VC dimension. It is natural that the lower bound is given as the maximum of the two lower bounds. However, we can translate this into the addition of the two terms through the following inequality.
>
>     $\textrm{LB} \geq \max\{\textrm{LB by Width}, \textrm{LB by VC-Dim}\} \geq \frac{1}{2}(\textrm{LB by Width} + \textrm{LB by VC-Dim})$
>
>     Since $\Omega(\cdot)$ for the lower bound hides constant terms (e.g. $\frac{1}{2}$), it is reasonable to write the lower bound as the sum of the two lower bounds. We kindly refer the reviewer to lines 160-161 for a brief explanation on these, with further details provided in Appendix A.4.
>
>
> **Q2.** $\rho=1$ **included in line 72?**
>
> - Thank you for the careful reading. We will revise the typo by changing the half-open interval into an open interval, making it consistent with other parts of the writing.
>
> **Q3. Excluding trivial cases with single-class labels**
>
> - Yes, when all points in $\mathcal{D}$ have the same label, $\epsilon_\mathcal{D}$ using the minimum notation is not properly defined. However, if we interpret the minimum as an infimum, we get $\epsilon_\mathcal{D}=+\infty$ for being the infimum of an empty set. Consequently, $\mu=\rho\epsilon_\mathcal{D} = +\infty$ for any $\rho \in (0, 1)$. In this case, assigning the bias of the last layer to the common label and setting all other parameters to zero yields a constant output over the entire domain. Thus, this trivially satisfies $\rho$-robustly memorization for any $\rho$.
> - Since this case is mathematically trivial, we will revise our writing to **exclude** single-class label datasets to maintain rigor and clarity. Thank you for pointing this out.
>
> **Q4. Why can the error in Theorem 4.2 (ii) be made arbitrarily small?**
>
> - We provide a brief explanation of how an arbitrarily small error can be achieved in the last paragraph of Section 5.2 (proof sketch). To guide readers who may have similar concerns when first encountering Theorem 4.2, we will include a reference to Section 5.2 in the discussion following Theorem 4.2.
> - In addition, we elaborate below on how the construction ensures an arbitrarily small error in greater detail—this explanation will also be supplemented in the proof sketch:
>
>     The proof of Theorem 4.2(ii) partitions the $N$ points into multiple **groups** of approximately equal size. The robust memorizer is constructed by composing a set of subnetworks, each responsible for robustly memorizing one group.
>
>     To achieve this, each subnetwork creates its own grid based on its group's data points, ensuring that the robustness balls of the group’s points lie entirely within the grid. Using a coordinate mapping, the subnetwork is designed to output the correct labels within its associated grids and zero elsewhere (Lemma B.10).
>
>     However, since the coordinate map is discontinuous, its approximation using ReLU introduces an **error-tolerant regions**—areas where the output is not guaranteed. These regions are illustrated in purple in Figure 4(b).
>
>     Each subnetwork accurately labels its own group’s robustness balls by ensuring they are placed within the grid while avoiding its error-tolerant regions. However, robustness balls from other groups may intersect this error-tolerant region, resulting in errors. As shown in Lemma B.9, the volume of a ball intersecting the error-tolerant region is proportional to the region’s width. This width can be made arbitrarily small by increasing the slope of the ReLU network (Lemma B.15). Consequently, the probability that a point of the robustness ball falls into the error-tolerant region—i.e., the error of robust memorization—is inversely proportional to the ReLU slope and can be made arbitrarily small.

---

> > ### Comment · Reviewer_LKzZ · 2025-08-03
> >
> > Thank you to the authors for their responses. I have no further questions now and will raise my score accordingly.

---

> > > ### Author Response · Authors · 2025-08-05
> > >
> > > Thank you for your thoughtful reconsideration. We are pleased that our clarifications addressed your concerns.

---

> ### Author Response · Authors · 2025-08-09
>
> Dear Reviewer,
>
> Thank you again for your review and for expressing earlier that you intended to raise your score after the rebuttal. As the discussion period is coming to a close, we just wanted to kindly remind you of this in case you would like to update anything.
>
> Best,
> Authors

---

### Official Review · Reviewer_17QC · 2025-07-10

**Clarity:** 4
**Significance:** 2
**Originality:** 4
**Rating:** 5
**Confidence:** 4

**Summary:**

This work expands on existing work on memorization bounds for ReLU nets by expanding the problem definition to consider "robust memorization" and extends the bounds for robustness ratio $\rho \in (0, 1)$. They also show that their bounds are tight up to log factors.

**Questions:**

- Is there a reason the authors left out memorization results on threshold networks in their discussion on related work? I would strongly recommend including them as they are important contributions to the area. See for example Vershynin, Roman. "Memory capacity of neural networks with threshold and rectified linear unit activations." SIAM Journal on Mathematics of Data Science 2.4 (2020): 1004-1033.
 and Rajput, Shashank, et al. "An exponential improvement on the memorization capacity of deep threshold networks." Advances in Neural Information Processing Systems 34 (2021): 12674-12685.

**Ethical Concerns:**

["NO or VERY MINOR ethics concerns only"]

**Final Justification:**

The authors have clarified most of my concerns and I am continuing to recommend acceptance. I urge the authors to include their clarifications in their manuscript so that the readers may benefit from them.

**Limitations:**

yes

**Quality:**

4

**Strengths And Weaknesses:**

Strengths:
- The paper is well written and provides good intuition behind the proofs and constructions.
- The paper does a good job at positioning their results among existing results and comparing their settings.

Weaknesses:
- The authors updated the paper to change the parameter count definition in terms of all parameters, not just non-zero ones. However, they compare their results with other works - which I believe consider only non-zero parameters. I recommend sticking to a single standard.
- The main weakness I see in the paper is the importance of considering $\rho$ in the computations. The memorization problem inherently considers the challenges of memorizing arbitrarily labeled points with some separation assumptions. By enforcing $\rho \in (0, 1)$, we are limiting some of these configurations w.r.t how points can be labeled within $\ell_{p}$-balls around each of these points. I am not sure if this is significant - although it is clear that this is technically challenging.
- I would like to also see bit-complexity of the constructions factoring into the bounds since these constructions (including Baum's) typically rely on choosing intercepts very precisely using real numbers with extremely high bit complexity making these constructions unrealistic to implement.

---

> ### Author Rebuttal · Authors · 2025-07-31
>
> We thank the reviewer for the constructive feedback.
>
> **W1. Inconsistent parameter count standard across comparisons**
>
> - We thank the reviewer for pointing this out and apologize for any confusion caused by the change in the definition of parameter count. As described in the supplementary material, we chose to adopt the number of all parameters (including zero entries) as a unified standard throughout the paper.  We chose this because one of our lower bound results does not hold for the original standard and holds for the revised standard. We updated the upper bound results accordingly to ensure consistency. The lower bound results remain unchanged, and the updated upper bound results are provided on **page 20**.
> - We acknowledge that prior works we compared against report results in terms of the number of *non-zero* parameters. To ensure a fair comparison:
>     1. We compute the number of all parameters in their constructions and compare our results based on the total number of parameters. The comparison is presented in **Figure 5**, and details on the computation of their constructions are provided in **Appendix D.2 and D.3**.
>     2. Additionally, we provide a complementary analysis in **Figure 6** of the appendix, where we evaluate our construction under the non-zero parameter count and compare the prior work based on the number of non-zero parameters. This figure shows that our construction still improves upon the known upper bounds even under the non-zero parameter metric.
>
> **W2. Potential limitation on allowable labeling due to the assumption $\rho\in (0,1)$**
>
> - We appreciate the reviewer’s thoughtful comment regarding the role of $\rho$. We would like to clarify that **the dataset $\mathcal{D}$, consisting of arbitrarily labeled points, is fixed in advance.** Then, we define $\epsilon_\mathcal{D}$ as half of the minimum distance between points with different labels. The robustness ratio is set to $\mu=\rho \epsilon_\mathcal{D}$ for some $\rho \in (0,1)$, and the goal is to memorize $\mu$-balls centered at the data points.
> - Importantly, the choice of $\rho$ **does not constrain** the labeling of the dataset. Instead, it determines the relative size of the neighborhood around each point that must be robustly memorized. If $\rho>1$, then $\mu$-balls around differently labeled points necessarily overlap, making robust memorization impossible (see line 111). Moreover, in the case $\rho=1$, we show in lines 112-115 that robust memorization is impossible using continuous functions. In contrast, when $\rho \in (0,1)$, $2 \mu$ is strictly less than the separation $2\epsilon _{\mathcal{D}}$, which ensures that the $\mu$-balls around differently labeled points remain disjoint and can be robustly memorized.
> - Therefore, our choice to restrict to $\rho \in (0,1)$ is not a limitation on the dataset itself, but a natural condition that characterizes the feasible regime for robust memorization. This formulation is also **consistent with prior works** such as Yu et al. (2024) and Li et al. (2022), where robust memorization is similarly defined for neighborhoods of radius $\mu$ strictly smaller than the separation $\epsilon_\mathcal{D}$. Regarding the upper bound, Yu et al. (2024) consider the regime where $\rho = \frac{\mu}{\epsilon_\mathcal{D}}\in(0,1)$, while Li et al. (2022) analyze the regime where $\rho = \frac{\mu}{\epsilon_\mathcal{D}}\in(0, 1/2)$. However, their upper bounds do not depend on $\rho$, failing to capture the interpolation between non-robust ($\rho=0$) and fully-robust ($\rho=1$ or $1/2$) memorization.
>
> **W3. Potentially unrealistic precision requirements due to high bit complexity**
>
> - Under our construction, the bit complexity—defined as a bit needed **per parameter** under a fixed point precision—to implement our construction tightly matches the lower bound for the small and moderate $\rho$ regimes. Moreover, construction on large $\rho$ regimes also has a reasonable $\tilde{O}(N)$ bit complexity. Overall, our constructions **do not need unrealistically high bit complexity.**
> - Upper bounds
>
>    We first introduce a bit complexity upper bound of the constructions in Theorem 4.2(i) and (ii). In both cases, the subnetworks for operations such as projection, translation, or coordinate mapping do not contribute significantly to the bit complexity. Instead, the dominant cost comes from the (non-robust) memorization subnetwork used in the final step, extended from Vardi et al. (2021), whose bit complexity scales with the network depth. The resulting bounds are $\tilde{O}(\sqrt{N})$ for (i) and $\tilde{O}(d^{-1/4}\rho^{-1/2})$ for (ii).
>
>     For Theorem 4.2(iii), we provide a separate derivation, yielding an $\tilde{O}(N)$ upper bound on the bit complexity.
>
>     Let each weight $W_l, b_l$ be approximated by $\bar{W}_l, \bar{b}_l$ due to finite precision. Upon the recursive definition of a neural network on line 117, the limited-precision feedforward is:
>
>     $$
>     \bar{a}\_{l}(x)=\sigma(\bar{W}\_{l}\bar{a}\_{l-1}(x)+\bar{b}\_l).
>     $$
>
>     If we denote
>     - $\zeta$: the precision (element-wise upper bound on $\Delta W_l:=W_l-\bar{W}_l$ and $\Delta b_l:= b_l-\bar{b}_l$),
>     - $D:=\max_{0\leq l \leq L} d_l$ the maximum hidden layer width,
>     - $p$: an upper bound on the absolute value of every parameter,
>
>     We claim that the error $\Delta a_l:=a_l-\bar{a}_l$, can be bounded as $\|\|\Delta a_L\|\|\leq Q\zeta$ for some $O(L)$ degree polynomial $Q$ on problem parameters.
>
>     Consider the key recurrence:
>     $$
>     \begin{align*}
>     \|\|\Delta a_{l+1}\|\|&=\|\|a_{l+1}-\bar{a}_{l+1}\|\|\\\\
>     &= \|\|\sigma(W_la_l+b_l)-\sigma(\bar{W}_l \bar{a}_l+\bar{b}_l)\|\|\\\\
>     &\leq \|\|(W_l a_l+b_l)-(\bar{W}_l\bar{a}_l+\bar{b}_l)\|\|\\\\
>     &=\|\|(W_l a_l+b_l) - ((W_l +\Delta W_l)(a_l+\Delta a_l)+(b_l+\Delta b_l))\|\| \\\\
>     &\leq \|\|W_l\|\| \cdot \|\|\Delta a_l\|\|+\|\|\Delta W_l\|\| \cdot\|\|a_l\|\| + \|\|\Delta W_l\|\| \cdot \|\|\Delta a_l\|\| + \|\|\Delta b_l\|\|\\\\
>     &\leq (D p)\|\|\Delta a_l\|\| + (D \zeta) \|\|a_l\|\| + (D \zeta) \|\|\Delta a_l\|\| + \sqrt{D} \zeta\\\\
>     &= (Dp+D\zeta)\|\|\Delta a_l\|\|+ (D\|\|a_l\|\|+\sqrt{D}) \zeta.
>     \end{align*}
>     $$
>
>     By recursive definition of $a_l$, it is straightforward to check that $\|a_l\|$ is upper bounded in a scale of $S:=(\textrm{poly}(\cdot))^L$, where the polynomial is on problem parameters like $D, p$ and maximum radius $R$ of inputs. The recursive bound above therefore, concludes
>
>     $$
>     \|\Delta a_L\| \leq (DS+\sqrt{D})\sum_{l=0}^{L-1}(Dp+D\zeta)^l\zeta = Q\zeta,
>     $$
>     where $Q$ is some degree $O(L)$ polynomial.
>
>     Thus, once we have $\zeta \leq \nu/Q$ for some small $\nu$ (e.g. $\nu=0.1)$, we can guarantee $\|\Delta a_L\| \leq \nu$. Yet, the final output has a perturbation from the infinite precision network. One can resolve this perturbation by an additional “rounding” layer that maps $[c-\nu, c+\nu] \mapsto c$ for all $c \in [C]$.
>
>     To represent parameters bounded by $p$ with precision $\zeta$, we require $\log(p/\zeta) = O(\log(Qp/\nu))$ bits. As $Q$ is a degree $O(L)$ polynomial, this corresponds to $O(L(\log p + \log D + \log R))$ bit complexity. The construction of Theorem 4.2(iii) has $p=\textrm{poly}(\cdot)$ and $L = \tilde{O}(N)$. Hence, $\tilde{O}(N)$ bit complexity suffices. However, the framework introduced in (iii) for counting bit complexity works for a general neural network, leaving the possibility to reduce the complexity under construction-specific counting.
>
> - Lower bounds
>
>     Vardi et al. (2021) also provide a lower bound on bit complexity using upper and lower bounds on VC-dimension. When a network has bit-complexity $B$ and $P$ **nonzero** parameters, the VC-dimension is upper bounded as
>
>     $$
>     \textrm{VC-Dim}=O(PB+P\log P).
>     $$
>
>     As VC-dimension is lower bounded by $N$ by memorization, combining these two bounds suggests the necessary bit complexity required under our constructions in Theorem 4.2. For simplicity, assume the case where the omitted $d$ (as mentioned in line 210) is not dominant.
>
>     Recall that for small, moderate, and large $\rho$ regime, we use $\tilde{O}(\sqrt{N})$, $\tilde{O}(Nd^{1/4}\rho^{1/2})$, and $\tilde{O}(Nd\rho^2)$ nonzero parameters, respectively. Consequently, the bit complexity becomes $\tilde{\Omega}(\sqrt{N})$, $\tilde{\Omega}(\frac{1}{d^{1/4}\rho^{1/2}})$, and $\tilde{\Omega}(\frac{1}{d\rho^2})$. For the last case, the lower bound of bit complexity is upper bounded by $O(1)$ scale and becomes trivial.
>
> - As a result,
>     - For small $\rho$ regime (Theorem 4.2(i)), the bit complexity is $\tilde{\Theta}(\sqrt{N})$, tight up to the logarithmic factor.
>     - For moderate $\rho$ regime (Theorem 4.2(ii)), the bit complexity is $\tilde{\Theta}(\frac{1}{d^{1/4}\rho^{1/2}})$, tight up to the logarithmic factor.
>     - For large $\rho$ regime (Theorem 4.2(iii)), the bit complexity is $\tilde{O}(N)$.
>
> **Q1. Threshold network memorization results not mentioned in related work?**
>
> - Thank you for pointing out important contributions in the area of memorization, particularly the results on threshold networks. We included Baum et al. (1988), which is indeed an example of such work, showing that a 2-layer threshold network with $\lceil N/d\rceil$ neurons suffices to memorize an arbitrarily given $N$ data points in general position in $d$ dimensions.
> - However, we acknowledge that due to space constraints, we were not able to include all relevant work. We will incorporate additional works in the final version, including those you mentioned—such as Vershynin (2020), Rajput et al. (2021), as well as [1] and [2].
>
> ---
>
> > **References**
> >
> > [1] Eric B. Baum and David Haussler. What size net gives valid generalization? In Advances in Neural Information Processing Systems, 1989.
> >
> > [2] Wolfgang Maass. Bounds for the computational power and learning complexity of analog neural nets. SIAM Journal on Computing, 26(3):708–732, 1997.

---

> ### Comment · Reviewer_17QC · 2025-08-04
> **Thank you for the clarifications**
>
> I thank the authors for the detailed clarifications and adding the bit complexity bounds. I am glad to see that they are not unreasonable.
>
> I continue to recommend acceptance. I urge the authors to include their clarifications in their manuscript so that the readers may benefit from them.

---

> > ### Author Response · Authors · 2025-08-05
> >
> > We thank the reviewer again for raising the important question on the bit complexity. We will reflect those results in our revision.

---

### Decision · Program_Chairs · 2025-09-17

**Decision:**

Accept (poster)

**Comment:**

The paper considers the parameter-complexity robust memorization with ReLU networks: That is, given $N$ datapoints, how large should a ReLU network be to memorize this data such that it is robust to input perturbations. This work presents upper and lower bounds for the same. The reviewers raised some concerns such as:
- Inconsistent usage of non-zero parameter count and total parameter count
- Possible high bit complexity
- The gap between upper and lower bounds in terms of $\rho$

The rebuttal answered these concerns satisfactorily. This work pursues an important direction in understanding the parameter complexity of ReLU networks, even though it uses standard techniques. Thus, it would be a positive addition to the literature. Based on the positive reviewer recommendations, I recommend accepting the paper.